



# Dissolved iron in the North Atlantic Ocean and Labrador Sea along the GEOVIDE section (GEOTRACES section GA01)

Manon Tonnard[1,2,3], Hélène Planquette[1], Andrew R. Bowie[2,3], Pier van der Merwe[2], Morgane Gallinari[1], Floriane Desprez de Gésincourt[1], Yoan Germain[4], Arthur Gourain[5], Marion Benetti[6,7], Gilles Reverdin[7],
Paul Tréguer[1], Julia Boutorh[1], Marie Cheize[1], Jan-Lukas Menzel Barraqueta[8], Leonardo Pereira-Contreira[9], Rachel Shelley[1,10,11], Pascale Lherminier[12], Géraldine Sarthou[1]

[1]Laboratoire des sciences de l'Environnement MARin – CNRS UMR 6539 – Institut Universitaire Européen de la Mer, Plouzané, 29280, France
[2]Antarctic Climate and Ecosystems – Cooperative Research Centre, Hobart, TAS 7001, Australia
[3]Institute for Marine and Antarctic Studies, University of Tasmania, Hobart, TAS 7001, Australia
[4]Laboratoire Cycles Géochimiques et ressources – Ifremer, Plouzané, 29280, France
[5]Ocean Sciences Department, School of Environmental Sciences, University of Liverpool, L69 3GP, UK
[6]Institute of Earth Sciences, University of Iceland, Reykjavik, Iceland
[7]LOCEAN, Sorbonne Universités, UPMC/CNRS/IRD/MNHN, Paris, France
[8]GEOMAR Helmholtz-Zentrum für Ozeanforschung Kiel Wischhofstraße 1-3, Geb. 12 D-24148 Kiel, Germany
[9]Fundação Universidade Federal do Rio Grande (FURG), R. Luis Loréa, Rio Grande –RS, 96200-350, Brazil
[10]Dept. Earth, Ocean and Atmospheric Science, Florida State University, 117 N Woodward Ave, Tallahassee, Florida, 32301, USA
[11]School of Geography, Earth and Environmental Sciences, University of Plymouth, Drake Circus, Plymouth, PL4 8AA, UK
[12] Ifremer, Université de Bretagne Occidentale (UBO), CNRS, IRD, Laboratoire d'Océanographie Physique et Spatiale (LOPS), IUEM, F-29280, Plouzané, France

*Correspondence to*: Manon Tonnard (Manon.Tonnard@utas.edu.au)

**Abstract.** Dissolved Fe (DFe) samples from the GEOVIDE voyage (GEOTRACES GA01, May-June 2014) in the North Atlantic Ocean were analysed using a Sea*FAST*-pico™ coupled to an Element XR HR-ICP-MS and provided interesting insights on the Fe sources in this area. Overall, DFe concentrations ranged from $0.09 \pm 0.01$ nmol L$^{-1}$ to $7.8 \pm 0.5$ nmol L$^{-1}$. Elevated DFe concentrations were observed above the Iberian, Greenland and Newfoundland Margins likely due to riverine inputs from the Tagus River, meteoric water inputs and sedimentary inputs. Air-sea interactions were suspected to be

responsible for the increase in DFe concentrations within subsurface waters of the Irminger Sea due to deep convection occurring the previous winter, that provided iron-to-nitrate ratios sufficient to sustain phytoplankton growth. Increasing DFe concentrations along the flow path of the Labrador Sea Water were attributed to sedimentary inputs from the Newfoundland Margin. Bottom waters from the Irminger Sea displayed high DFe concentrations likely due to the dissolution of Fe-rich particles from the Denmark Strait Overflow Water and the Polar Intermediate Water. Finally, the nepheloid layers were found

to act as either a source or a sink of DFe depending on the nature of particles.



# 1 Introduction

The North Atlantic Ocean is a crucial area for Earth's climate as a result of deep water formation. The formation of the North Atlantic Deep Water (NADW) is essential to the Atlantic Meridional Overturning Circulation (AMOC), which is responsible for transporting large amounts of water, heat, salt, carbon, nutrients and other elements around the globe. The North Atlantic Ocean is also a region of high phytoplankton production, a key driver of the biological carbon pump through photosynthesis and transfer of energy to higher trophic levels (Henson et al., 2009). Hence, the North Atlantic Ocean stores huge amounts of anthropogenic $CO_2$ through both the physical and biological carbon pump, despite covering only 15% of the global ocean area (Humphreys et al., 2016; Sabine et al., 2004). Recent observations showed a reduction of oceanic heat loss to the atmosphere due to the slowdown of the AMOC resulting to the reduction of carbon uptake (Pérez et al., 2013).

In the North Atlantic Ocean, phytoplankton growth is largely light-limited at its northern boundaries (i.e. the Arctic Ocean, marginal ice zones, polynya areas and the Labrador Shelf) set primarily by freeze-thaw cycles of sea ice and the high-latitude extremes in the solar cycle (Longhurst, 2007). In contrast, at its more southerly boundaries (i.e. the subpolar and to a lesser extent the subtropical gyres), seasonal wind and thermal cycles that determine mixing patterns, have a greater influence on phytoplankton growth with the consequence that both light and nutrient inventories can limit productivity (Harrison et al., 2013). If light is thought to be the principal control on the timing of growth, nutrients play a significant role in the phytoplankton community structure. In particular, winter nutrient reserves in surface waters set a lower limit for biomass accumulation during the annual spring-to-summer bloom and will influence the duration of the bloom (Follows and Dutkiewicz, 2001; Henson et al., 2009; Moore et al., 2013; 2008). Hence, nutrient depletion due to biological consumption is considered as a major factor in the decline of blooms (Harrison et al., 2013). The North Atlantic Ocean is classically considered as a N-limited system (Moore et al., 2013). However, once the water column stratifies in spring and summer, and phytoplankton are released from the light limitation of winter, the North Atlantic Ocean becomes N and P-co-limited in the Subtropical gyre (e.g. Moore et al., 2008) and N and Fe-co-limited in the Subpolar gyre (e.g. Nielsdóttir et al., 2009; Painter et al., 2014; Sanders et al., 2005). The latter situation is typically observed in the Irminger Sea and Iceland Basin which may exhibit seasonal high nutrient low chlorophyll (HNLC) conditions, with macronutrient concentrations often well above limiting levels (Sanders et al., 2005).

Indeed, Fe is a key element for a number of metabolic processes (e.g. Morel et al., 2008) and within its physical speciation, its dissolved form (DFe) is considered to be the most available form for phytoplankton (Morel, 2008; Morel et al., 2008). In the North Atlantic Ocean, DFe is delivered through multiple pathways such as ice-melting (e.g. Klunder et al., 2012; Tovar-Sanchez et al., 2010), atmospheric inputs (Achterberg et al., 2018; Baker et al., 2013; Shelley et al., 2015; 2017), coastal runoff (Rijkenberg et al., 2014), sediment inputs (Hatta et al., 2015), hydrothermal inputs (Achterberg et al., 2018; Conway and John, 2014) and by water mass circulation (vertical and lateral advections, e.g. Laes et al., 2003). DFe can be regenerated through biological recycling (microbial loop, zooplankton grazing, e.g. Boyd et al., 2010; Sarthou et al., 2008). Iron is removed from the dissolved phase by biological uptake, export and scavenging along the water column and precipitation (itself a function of





salinity, pH of seawater and ligand concentrations). Despite these studies, uncertainties remain on the distribution of DFe in the North Atlantic Ocean and more specifically within the subpolar gyre where few studies have been undertaken, and even fewer in the Labrador Sea. In this biogeochemically important area, high-resolution studies are still lacking for understanding the processes influencing the cycle of DFe.

In this context, this paper elucidates the sources and sinks of DFe, its distribution regarding water masses and assesses the links with biological activity along the GEOVIDE (GEOTRACES-GA01) transect, which spanned several biogeochemical provinces including the West European Basin, the Iceland Basin, the Irminger and the Labrador Seas (Fig. 1).

## 2 Material and methods

### 2.1 Study area and sampling activities

Samples were collected during the GEOVIDE (GEOTRACES-GA01 section, Fig. 1) oceanographic voyage from 15 May 2014 (Lisbon, Portugal) to 30 June 2014 (St. John's, Newfoundland, Canada) aboard the N/O *Pourquoi Pas?*. The study was carried out along the OVIDE line (http://www.umr-lops.fr/Projets/Projets-actifs/OVIDE, previously referred to the WOCE A25 Greenland to Portugal section), which has been sampled every two years since 2002 in the North Atlantic (e.g. Mercier et al., 2015), and in the Labrador Sea (broadly corresponding to the WOCE A01 leg 3 Greenland to Newfoundland section). In total,

32 stations were occupied, and samples were usually collected at 22 depths, except at shallower stations close to the Iberian, Greenland and Canadian shelves (Fig. 1) where fewer samples (between 6 and 11) were collected. To avoid ship contamination of surface waters, the shallowest sampling depth was 15 m at all stations. These 15 m depth will be herein referred to as 'surface water samples'.

Samples were collected using the French-national ultra-clean sampling device. This consisted of a trace metal clean polyurethane powder-coated aluminium frame rosette (hereafter referred to as TMR) equipped with twenty-two 12L, externally closing, Teflon-lined, GO-FLO bottles (General Oceanics) and attached to a Kevlar® line. The cleaning protocols for sampling bottles and equipment followed the guidelines of the GEOTRACES Cookbook (www.geotraces.org, Cutter et al., 2017). After TMR recovery, GO-FLO bottles were transferred into a clean container equipped with a class 100 laminar flow hood. Samples

were either taken from the filtrate of particulate samples (collected on polyethersulfone filters, 0.45 µm supor®, see Gourain et al., this issue) or after filtration on 0.2 µm filter cartridges (Sartorius SARTOBRAN® 300) (Table 1). Seawater was collected in acid-cleaned 60 mL LDPE bottles, after rinsing 3 times with about 20 mL of seawater. Teflon® tubing used to connect the filter holders or cartridges to the GO-FLO bottles were washed in an acid-bath (10% v/v HCl, Suprapur®, Merck) for at least 12 h and rinsed three times with Ultra High Purity Water (UHPW > 18 MΩ.cm) prior to use. Samples were then acidified to

~ pH 1.7 with 2 ‰ (v/v) HCl (Ultrapur® Merck) under a class 100 laminar flow hood in the clean container. The sample bottles were then double bagged and stored at ambient temperature in the dark before shore-based analyses.



## 2.2 DFe analysis with Sea*FAST*-pico™

Seawater samples were introduced to a PFA-ST nebulizer and a cyclonic spray chamber via a Sea*FAST*-pico™ introduction system (Elemental Scientific Incorporated, Omaha, NE), following the protocol of Lagerström et al. (2013).

High-purity grade solutions and water (Milli-Q) were used on a daily basis to prepare the following reagents: the buffer was

made of 0.5 M acetic acid (Merck ultrapur) and 0.6 M ammonium hydroxide (Merck ultrapur) and was adjusted to pH 8.3. The elution acid was made of 1.6 M nitric acid (HNO$_3$, Merck Ultrapur) in Milli-Q water and spiked with 1 μg mL$^{-1}$ Indium (In, PlasmaCAL calibration standards) to allow for drift correction. Autosampler and column rinsing solutions were made of 0.012 M hydrochloric acid (HCl, Merck ultrapur) in Milli-Q water.

All reagents, standards, samples, and blanks were prepared in acid cleaned low density polyethylene (LDPE) or Teflon

fluorinated ethylene propylene (FEP) bottles. Bottles were cleaned following the GEOTRACES protocol.

Mixed element standard solution was prepared gravimetrically using high purity standards (Fe, Mn, Cd, Co, Zn, Cu, Pb; PlasmaCAL calibration standards) in 0.8 M HNO$_3$ (Merck Ultrapur). A six-point calibration curve was prepared by standard additions of the mixed element standard to our in-house standard (GEOVIDE filtered seawater, collected at station 69, 40m depth) and ran at the beginning, the middle and the end of each run. The distribution of the trace metals other than Fe will be

reported elsewhere (Planquette et al., in prep.). Final concentrations of samples and procedural blanks were calculated from In-normalized data. Precision was assessed through replicate samples (every 10th sample was a replicate) and accuracy was determined from analysis of consensus (SAFe S, GSP) and certified (NASS-7) seawaters. Values are reported in Table 2. Note that all the DFe values were generated in nmol kg$^{-1}$ using the Sea*FAST*-pico™ coupled to an Element XR HR-ICP-MS and were converted in nmol L$^{-1}$ (multiplied by a factor of 1.025 kg L$^{-1}$) to be directly comparable with literature.

## 2.3 Ancillary measurements and mixed layer depth determination

Potential temperature (θ), Salinity (S), dissolved oxygen (O$_2$) and beam attenuation data were retrieved from the CTD sensors (CTD SBE911 equipped with a SBE-43). Nutrient and pigment samples were obtained from the CTD casts and analysed according to Aminot and Kerouel (2007) and Ras et al. (2008), respectively. We used the data from the CTD casts that were deployed immediately before or after our TMR casts. All these data are/will be available on the LEFE/CYBER database

(http://www.obs-vlfr.fr/proof/php/geovide/geovide.php).

The mixed layer depth (Z$_m$) for each station was calculated using the function "calculate.mld" (part of the "rcalcofi" package, Ed Weber at NOAA SWFSC) created by Sam McClathie (NOAA Federal, 30$^{th}$ December 2013) for R software and where Z$_m$ is defined as an absolute change in $\Delta\sigma_t$ ($\geq 0.125$ kg m$^{-3}$) with respect to an approximately uniform region of density just below the ocean surface (Kara et al., 2000). In addition to the density criterion, the temperature and salinity profiles were inspected

at each station for uniformity within this layer. When they were not uniform, the depth of any perturbation in the profile was chosen as the base of the Z$_m$ (Table 1).

### 2.4 Statistical analysis

All statistical approaches were performed using the R statistical software (R development Core Team 2012). For all the results, p-values were measured against the threshold value alpha ($\alpha$), that we assigned at 0.05, corresponding to a 95% level of confidence. For all data sets, non-normal distributions were observed according to the Shapiro-Wilk test. Therefore, the

significance level was determined with a Wilcoxon test.

All sections and surface layer plots were prepared using Ocean Data View (Schlitzer, 2016).

### 2.5 Water mass determination and associated DFe concentrations

The water mass structure in the North Atlantic Ocean from the GEOVIDE voyage was quantitatively assessed by means of an extended Optimum Multi-Parameter (eOMP) analysis with 14 water masses (for details see García-Ibáñez et al., 2015; this

issue). From this water mass determination, we considered only a contribution higher than 60% of a specific water mass to the total water mass pool and calculated the average DFe concentrations within each water mass and considered them as representative of the DFe concentrations within these water masses, as identifying representative end-members was not found to be possible due to the non-conservative behaviour of DFe in seawater.

### 2.6 Database

The complete database of dissolved Fe is available in the electronic supplement www.biogeosciences.net. Overall, 540 data of dissolved Fe are reported, among which 511 values are used in this manuscript. The remaining 29 values (5.7% of the total dataset) are flagged as (suspect) outliers. These 29 outliers were not used in figures and in the interpretation of this manuscript. The criteria for rejection were based on the comparison with other parameters measured from the same GO-FLO sampler, and curve fitting versus samples collected above and below the suspect sample. The complete relational database will be available

in national and international databases (LEFE-CYBER, http://www.obs-vlfr.fr/proof/index2.php, and GEOTRACES http://www.bodc.ac.uk/geotraces/).

## 3 Results

### 3.1 Hydrography

The hydrology and circulation of the main water masses along the OVIDE section in the North Atlantic Subpolar Gyre and

their contribution to the Atlantic Meridional Overturning Circulation (AMOC) have been described using an eOMP analysis by García-Ibáñez et al., (2015; this issue) and Zunino et al. (2017). Hereafter we summarise the main features (Fig. 1 and 2). For a schematic of water masses, currents and pathways, see Daniault et al. (2016).





*Upper waters* - The cyclonic circulation of the Eastern North Atlantic Central Water (ENACW) ($12.3 < \theta < 16°C$, $35.66 < S < 36.2$, $241 < O_2 < 251$ µmol kg$^{-1}$) occupied the water column from 0 to ~ 800 m depth from stations 1 to 25 contributing to 60% of the water mass pool. The sharp Subarctic Front (between stations 26 and 29), caused by the northern branch of the North Atlantic Current (NAC) separated the cyclonic subpolar from the anticyclonic subtropical gyre domains at 50°N and 22.5°W. These Central Waters were also encountered to a lesser extent and only in surface waters (from 0 to ~ 100 m depth) between stations 29 and 34 (contributing to less than 40% of the water mass pool). West of the Subarctic Front, Iceland SubPolar Mode Waters (IcSPMW, $7.07 < \theta < 8°C$, $35.16 < S < 35.23$, $280 < O_2 < 289$ µmol kg$^{-1}$) and Irminger SubPolar Mode Waters (IrSPMW, $\theta \approx 5°C$, $S \approx 35.014$) were encountered from stations 34-40 (accounting for more than 45% of the water mass pool from 0 to ~ 800 m depth) and stations 42-60 (contribution to 40% of the water mass pool from 0 to ~ 250 m depth and until 1300 m depth for stations 49 and 60), respectively. The IcSPMW was also observed within the Subtropical gyre, subducted below the Central Waters until ~ 1000 m depth. Stations 63 (> ~ 200 m depth) and 64 (from surface down to ~ 500 m depth) exhibited a contribution of the IrSPMW higher than 45%. Stations 44, 49 and 60, from the Irminger Sea, and 63 from the Labrador Sea were characterised by lower sea-surface salinity ranges (S = [34.636, 34.903], stations 63 and 60, respectively). The Subarctic Intermediate Water (SAIW, $4.5 < \theta < 6.0°C$, $34.70 < S < 34.80$) contributed to more than 40% of the water mass pool in the Iceland Basin between the surface and ~ 400 m depth at stations 29 and 32 and throughout the water column of stations 53, 56 and 61 and from surface down to ~ 200 m depth at station 63. From stations 68 to 78 surface waters were characterized by a minimum of salinity and a maximum of oxygen (S = 34.91, $O_2$ = 285 µmol kg$^{-1}$, $\theta \approx 3°C$) and corresponded to the newly formed LSW. The LSW was also observed in surface waters of station 44 with a similar contribution than IrSPMW.

*Intermediate waters* - The Mediterranean Water (MW) flows westward from the Mediterranean Sea and is then transported northward by the Azores counter current. The Mediterranean Outflow Water (MOW), distinguishable from surrounding Atlantic Water by its high salinity tongue (up to 36.2), a minimum of oxygen ($O_2$ = 210 µmol kg$^{-1}$) and relatively high temperatures (up to 11.7°C) was observed from station 1 to 21 between 800 and 1400 m depth at a neutral density ranging from 27.544 to 27.751 kg m$^{-3}$ with the maximum contribution to the whole water mass pool seen at station 1 (64 ± 6%). Its main core was located at ~ 1200 m depth off the Iberian shelf from stations 1 to 11 and then gradually rising westward due to mixing with LSW within the North Atlantic subtropical gyre and a contribution of this water mass decreasing until station 21 down to 10-20%. The LSW (27.763 < neutral density < 27.724 kg m$^{-3}$) sourced from the SPMW after intense heat loss and led to its deep convection. During GA01, new LSW formed by deep convection the previous winter was found at several stations from the Labrador Sea (68, 69, 71 and 77). After convecting, some the LSW flows north-eastward in the Iceland Basin and Irminger Sea and then back to the Labrador Sea, while some flows along the Canadian coast, more specifically above the Newfoundland Margin as confirmed by hydrographic station 76 (52.5°W, 52.5°N) which exhibited the highest LSW fraction (up to 98% at the closest bottom sample, García-Ibáñez et al., this issue). On its way eastwards, the LSW splits in three main branches with two main cores separated by the Reykjanes Ridge, that corresponds to its different pathways (stations 1-32,



West European and Iceland Basins; stations 40-60, Irminger Sea). One of the branches flows eastward at intermediate depth, following the circulation of the warm surface NAC and enters the West European Basin and the Iceland Basin where it splits again into two branches (south-eastward and north-eastward flowing path, respectively).

*Overflows and Deep waters* - The North East Atlantic Deep Water (NEADW, $1.98 < \theta < 2.50°C$, $34.895 < S < 34.940$) was the dominant water mass in the West European Basin at stations 1-29 from 2000 m depth to the bottom and is characterized by high silicic acid, nitrate concentrations and low oxygen concentration ($O_2 \approx 252$ µmol kg$^{-1}$). The core of the NEADW (stations 1-13) was located near the seafloor and gradually decreased westward. The Polar Intermediate Water (PIW, $\theta \approx 0°C$, $S \approx 34.65$) is a dense shelf, low-salinity water intrusion in the deep overflows within the Irminger and Labrador Seas. PIW is

in contact with the atmosphere once a year during the time of winter convection (Strass et al., 1993) and hence is ventilated ($O_2 \approx 310$ µmol kg$^{-1}$). This water presented a low contribution to the entire water mass pool (up to 27%). The PIW was observed over the Greenland slope at stations 53 and 61 as well as in surface waters from station 63 (from 0 to ~ 200 m depth), in intermediate waters of stations 49, 60 and 63 (from ~ 500 to ~ 1500 m depth) and in bottom waters of stations 44, 68, 69, 71 and 77 with a contribution higher than 10%. The Iceland Scotland Overflow Water (ISOW, $\theta \approx 2.6°C$, $S \approx 34.98$) is partly

formed within the Arctic Ocean by convection of the modified Atlantic water. ISOW comes from the Iceland-Scotland sills and flows southward towards Charlie Gibbs Fracture Zone (CGFZ) and Bight Fracture Zone (BFZ) (stations 34 and 36) after which it reverses its flowing path northward and enters the Irminger Sea (stations 40 and 42) to finally reach the Labrador Sea close to the Greenland coast (station 49, station 44 being located in between this two opposite flow paths). In this study, ISOW properties are defined after the mixing of the overflow with Atlantic Central Water and Labrador Water downstream the sills.

Along the eastern (stations 26-36) and western (stations 40-44) flanks of the Reykjanes Ridge, ISOW had a contribution higher than 50% to the water mass pool. The ISOW was observed from 1500 m depth to the bottom of the entire Iceland Basin and from 1800 to 3000 m depth within the Irminger Sea (stations 40-60). The ISOW, despite having a fraction lower than 45% above the Reykjanes Ridge (station 38), was the main contributor to the water mass pool from 1300 m depth down to the bottom. The ISOW was also observed within the Labrador Sea from stations 68 to 77. The deepest part of the Irminger (stations

42 and 44) and Labrador (stations 68-71) Seas were occupied by the DSOW ($\theta \approx 1.30°C$, $S \approx 34.905$) that spills over the Greenland-Scotland ridge system and overflows south-westward into the deep North Atlantic basins. This water mass is formed partly in the Arctic Ocean by convection of the branch of the NAC that flows northward between Greenland, Iceland and Scotland. DSOW is a young water mass with a 3 to 4-year residence time north of the sill after surface contact.

### 3.2 Ancillary data

### 3.2.1 Nitrate

Surface nitrate (NO$_3^-$) concentrations (Sarthou et al., in prep.) ranged from 0.01 to 10.1 µmol L$^{-1}$ (stations 53 and 63, respectively). There was considerable spatial variability in NO$_3^-$ distributions with highest NO$_3^-$ concentrations found in the



Iceland Basin and Irminger Sea higher than 6 µmol L$^{-1}$ and to a lesser extent at stations 63 (10.1 µmol L$^{-1}$) and 64 (5.1 µmol L$^{-1}$) and the lowest concentrations observed in the Western European Basin, in the Labrador Sea and above continental margins. The low NO$_3^-$ concentrations in the Western European Basin extended from station 2 (closest station to continental land mass) to station 23 (most open ocean station) with concentrations ranging from 0.02 (station 11) to 1.7 (station 23) µmol L$^{-1}$. The

low nitrate concentrations in the Labrador Sea extended from station 68 to station 78 with concentrations ranging from 0.04 (station 68) to 1.8 (station 71) µmol L$^{-1}$. Lowest NO$_3^-$ concentrations (lower than 15.9 µmol L$^{-1}$) measured in surface and deep waters were found in the ENACW and DSOW, respectively. The highest NO$_3^-$ concentrations were measured within the NEADW (up to 23.5 µmol L$^{-1}$), and in the mesopelagic zone of the West European and Iceland Basins (higher than 18.4 µmol L$^{-1}$).

**3.2.2 Chlorophyll-*a***

Overall, total chlorophyll-*a* (TChl-*a*) concentrations were significantly correlated with the more extensive dataset from the CTD mounted fluorometer (R$^2$ = 0.76, n=162). TChl-*a* is the universal proxy for phytoplankton organisms. The maximum chlorophyll biomass ranged between 0.35 mg m$^{-3}$ (Station 19, 50 m depth) and 9.4 mg m$^{-3}$ (station 78, 30 m depth) highlighting the intense variability observed throughout this section (Fig. 3). Generally speaking, most of the phytoplankton biomass was

localised above 100 m depth. TChl-*a* concentrations were lower South of the Subarctic Front and higher at higher latitudes. Lowest TChl-*a* (<0.75 mg m$^{-3}$) were measured in the West European (stations 1-19) and Iceland (stations 34-38) Basins, while highest concentrations were measured at the Greenland (stations 53 and 61) and Newfoundland (station 78) margins (up to 4.9, 6.6 and 9.6 mg m$^{-3}$, respectively).

**3.3 Dissolved Fe concentrations**

The dataset is well distributed between upper, intermediate and deep ocean samples with 36% of samples collected at depths shallower than 200 m, 27% of samples between 200 and 1000 m depth and 37% of samples from depths deeper than 1000 m, including 11% of samples below 2500 m depth. Samples are distributed as follows between basins: 38% of samples were collected within the West European Basin (stations 1 and 11-26), 17% within the Iceland Basin (stations 29-36), 18% within the Irminger Sea (stations 38-49 and 60), 20% within the Labrador Sea (stations 63-77) and 7% above the continental margins

(stations 2, 4, 53, 56, 61, 78). The dataset is thus weighted towards the West European Basin. Dissolved Fe concentrations ranged from 0.09 ± 0.01 nmol L$^{-1}$ (station 19, 20 m depth) to 7.8 ± 0.5 nmol L$^{-1}$ (station 78, 371 m depth) (see Fig. 4).

**3.3.1 Margin stations**

Generally, vertical profiles of DFe for stations above the margins (2, 4, 53, 56, 61, and 78) showed an increase with depth, although sea-surface maxima were observed at stations 2, 4 and 56. For these margin stations, values were around 0.7-1.0

nmol L$^{-1}$ in the surface waters. Concentrations increased towards the bottom, with more than 7.8 nmol L$^{-1}$ measured at station 78, approximately 1-3 nmol L$^{-1}$ for stations 2, 4, 53, and 61, and just above 0.4 nmol L$^{-1}$ for station 56 (Fig. 5).



### 3.3.2 Open-ocean stations

For all regions, mean vertical profiles (Fig. 6) showed increasing DFe concentrations from the surface to 3000 m depth followed by decreasing DFe concentrations down to the bottom, with the lowest deep values in the West European Basin. The Irminger Sea displayed the highest DFe concentrations from 1000 m depth to the bottom relative to other basins at similar

depths (Figs. 4 and 6). In the Labrador Sea, DFe concentrations were low and relatively constant at about $0.87 \pm 0.06$ nmol L$^{-1}$ from 250 m to 3000 m depth (Fig. 6). Overall, surface DFe concentrations were higher in the North Atlantic Subpolar gyre (above 30°N) than in the North Atlantic Subtropical gyre. The upper surface DFe concentrations were generally smaller than 0.3 nmol L$^{-1}$, except for few stations in the Iceland Basin (stations 32 and 38), Irminger (stations 40 and 42) and Labrador (station 63) Seas, where values ranged between 0.4-0.5 nmol L$^{-1}$. Within the low-Fe West European Basin a few stations

exhibited slightly higher SML dissolved Fe concentrations relative to surrounding waters (stations 13 and 15, Table 1).

### 3.3.3. Fingerprinting water masses

In the West European Basin, the MOW was present but not characterized by particularly high or low DFe concentrations relative to the surrounding Atlantic waters (Fig. 7). The median DFe value in the MOW was very similar to the median value

when considering all the water masses (0.77 nmol L$^{-1}$, Figs. 4 and 7). The LSW and IcSPMW displayed slightly elevated DFe concentrations compared to the overall median with mean values of $0.82 \pm 0.08$ (n=28) and $0.80 \pm 0.04$ (n=8) nmol L$^{-1}$, respectively. The DFe concentrations in the NEADW were relatively low compared to the DFe median value of the GEOVIDE voyage (Fig. 4 and 7) with an average value of $0.74 \pm 0.16$ nmol L$^{-1}$ (n=18). DFe concentrations in the ENACW were the lowest of the whole section with an average value of $0.30 \pm 0.16$ nmol L$^{-1}$ (n=64).

In the Iceland Basin, the SAIW and the IcSPMW displayed similar DFe concentrations with averaged DFe concentrations of $0.67 \pm 0.30$ nmol L$^{-1}$ (n=7) and $0.55 \pm 0.34$ nmol L$^{-1}$ (n=22), respectively. The LSW exhibited higher DFe concentrations with an average value of $0.96 \pm 0.22$ nmol L$^{-1}$ (n=21). The ISOW had an averaged DFe concentration of $1.0 \pm 0.3$ nmol L$^{-1}$ (n=10) (Fig. 7).

In the Irminger Sea, surface waters were characterized by the SAIW ($0.56 \pm 0.24$ nmol L$^{-1}$, n=4) and the IrSPMW ($0.72 \pm 0.32$

nmol L$^{-1}$, n=34). The highest open-ocean DFe concentrations (up to $2.5 \pm 0.3$ nmol L$^{-1}$, station 44, 2600 m depth) were measured within this basin. In the upper intermediate water the LSW was identified only at stations 40 to 44, and had the highest DFe values with an average of $1.2 \pm 0.3$ nmol L$^{-1}$ (n=14). The ISOW showed higher DFe concentrations than in the Iceland Basin ($1.3 \pm 0.2$ nmol L$^{-1}$, n=4). At the bottom, the DSOW was mainly located at stations 42 and 44 and presented the highest average DFe values ($1.4 \pm 0.4$ nmol L$^{-1}$, n=5) as well as the highest variability from all the water masses presented in

this section (Fig. 7).

Finally, in the Labrador Sea, the IrSPMW exhibited an average DFe concentration of $0.61 \pm 0.21$ nmol L$^{-1}$ (n=14). DFe concentrations in the LSW were the lowest in this basin compared to the other ones, with an average value of $0.71 \pm 0.27$ nmol



$L^{-1}$ (n=53) (Fig. 7). Deeper, the ISOW displayed slightly higher DFe concentrations (0.82 ± 0.05 nmol $L^{-1}$, n=2). Finally, the DSOW had one of the lowest DFe values for intermediate and deep waters (0.68 ± 0.06 nmol $L^{-1}$, n=3, Fig. 7).

## 4 Discussion

In the following sections, we will discuss the relationship between water masses and the DFe concentrations (Section 4.1) in intermediate (Section 4.1.2 and 4.1.3) and deep (Section 4.1.4 and 4.1.5) waters. We will also discuss the role of wind (Section 4.1.1), rivers (Section 4.2.1), meteoric water and sea-ice processes (Section 4.2.2), atmospheric deposition (Section 4.2.3) and sediments (Section 4.3) in delivering DFe. Finally, we will discuss the potential Fe limitation using DFe:$NO_3^-$ ratios (Section 4.4).

Considering the entire section, only two stations (stations 1 and 17) showed high DFe concentrations (> 1 nmol $L^{-1}$) throughout the water column. The influence of water masses to explain these distributions was discarded as the observed high homogenized DFe concentrations were more likely to be impacted by vertical processes. Station 1, located at the continental shelf-break of the Iberian Margin, also showed enhanced PFe concentrations potentially suggesting a margin source (Gourain et al., in prep.). Conversely, no relationship was observed between DFe and PFe nor transmissometry for station 17. However, Ferron et al. (2016) reported a strong dissipation rate at the Azores-Biscay Rise (station 17) due to internal waves. The associated vertical energy fluxes could explain the homogenized profile of DFe at station 17, although such waves are not clearly evidenced in the velocity profiles. Consequently, the elevated DFe concentrations observed at station 17 remain unsolved. Another explanation which cannot be discarded would be a contamination issue, especially for station 1 as it was the first station sampled. With this in mind, one would expect a more random distribution of DFe concentrations and less consistence throughout the water column if there had been a problem with contamination.

### 4.1 DFe and hydrology keypoints

#### 4.1.1 Air-sea interactions in the Irminger Sea

Among the four distinct basins described in this paper, the Irminger Sea exhibited the highest DFe concentrations within the surface waters (ranging from 0.23 to 1.3 nmol $L^{-1}$ in open-ocean stations and from 0 to 250 m depth). Enhanced surface DFe concentrations in the Irminger Sea were previously reported by Rijkenberg et al. (2014) (April-May, 2010) and Achterberg et al. (2018) (April-May and July August, 2010). However, DFe concentrations measured during our study (May-June, 2014) in the Irminger Sea were higher than those reported before (Table 3). Differences might be due to the phytoplankton bloom advancement, the high remineralization rate observed (Lemaître et al., 2017) within the LSW in the Irminger Sea (see Section 4.1.3) and a deeper winter convection in early 2014.

In the North Atlantic Ocean, the warm and salty water masses of the upper limb of the MOC are progressively cooled and become denser, and subduct into the abyssal ocean. In the SubPolar North Atlantic, winter convective mixing represents the



dominant nutrient supply process (Louanchi and Najjar, 2001). Pickart et al. (2003) demonstrated that the conditions necessary for the development of deep convection are satisfied in the Irminger Sea by the presence of weakly stratified surface water, a close cyclonic circulation and intense winter air-sea buoyancy fluxes (Marshall and Schott, 1999). Since then, many studies (e.g. Bacon et al., 2003; de Jong et al., 2012) reported homogeneous density profiles from surface to 1000 m depth. Moore

(2003) and Piron et al. (2016) described Tip Jet events that are characterized by westerly flow over the Irminger Sea, centred northeast of Cape Farewell. It has also been shown that a positive NAO index is expected to favour the occurrence of such event as Greenland Tip Jets are more frequent during winters with positive NAO (Moore, 2003; Pickart et al., 2003) which was the case in the winter 2013-2014, preceding the GEOVIDE voyage (Lherminier, pers. comm.). Winter mixed layer depth prior to the cruise reached up to 1200 m depth in the Irminger Sea (Zunino et al., 2017) and were most likely attributed to a

final deepening due to wind forcing events (centred at station 44). This annual event recharges surface ocean nutrient inventories and subsequently fuels the spring phytoplankton bloom with DFe values close to the LSW ones. Therefore, the elevated DFe concentrations observed below 100 m depth from the central Irminger Sea were likely due to the wind forcing deepening of the mixed layer depth carrying upward LSW properties.

### 4.1.2 Lack of DFe signature in the MOW

The Mediterranean Sea on its northern shores is bordered by industrialized European countries, which act as a continuous source of anthropogenic derived constituents into the atmosphere, and on the southern shores by the arid and desert regions of north African and Arabian Desert belts, which act as sources of crustal material in the form of dust pulses (Chester et al., 1993; Guerzoni et al., 1999; Martin et al., 1989). During the stratification period, DFe concentrations in the SML can increase over the whole Mediterranean Sea by 1.6-5.3 nmol L$^{-1}$ in response to the accumulation of atmospheric Fe from both anthropogenic

and natural origins (Bonnet and Guieu, 2004; Guieu et al., 2010; Sarthou and Jeandel, 2001).

Despite these high atmospheric depositions into the Mediterranean Sea providing Fe enrichment of Mediterranean waters and the high Apparent Oxygen Utilization (AOU), during the GEOVIDE voyage, no particular DFe signature was associated with the MOW (Figs. 2 and 4), which was also the case in other studies in the same area (Hatta et al., 2015; Thuróczy et al., 2010). Conversely, Gerringa et al. (2017) and Sarthou et al. (2007) observed higher DFe concentrations in the MOW. During our

study the MOW signal was clearly observed on dissolved aluminium (DAl, Menzel Barraqueta et al., 2018) with the highest DAl values from the section (up to 38.7 nmol L$^{-1}$), suggesting intense dust deposition inputs to the Mediterranean Sea. A concurring phenomenon, suggested by Wagener et al. (2010), is likely to explain this absence of MOW DFe signature. They pointed to the fact that large dust deposition events can accelerate the export of Fe from the water column through scavenging. As a result, in seawater with high DFe and where high dust deposition occurs, a strong individual dust deposition event could

act as a sink for DFe. It thus becomes less evident to observe a systematic high DFe signature in the MOW despite dust inputs. The same explanation was reported by Gerringa et al. (2017) to account for the decrease in DFe concentrations between surface and deep waters from the Western Mediterranean Basin.



### 4.1.3 Labrador Sea Water (LSW) Fe enrichment

As described in Section 3.1, the LSW exhibited increasing DFe concentrations from its source area, the Labrador Sea, toward the other basins with the highest DFe concentrations observed within the Irminger Sea, suggesting that local sources of DFe may occur along its flow path (Fig. 7).

Lambelet et al. (2016) reported high dissolved neodymium (Nd) concentrations (up to 18.5 pmol.kg$^{-1}$) within the LSW at the edge of the Newfoundland Margin (45.73°W, 51.82°N) as well as slightly lower Nd isotopic ratio values relative to the one observed in the Irminger Sea. They suggested that this water mass had been in contact with sediments approximately within the last 30 years (Charette et al., 2015). Similarly, during GA03, Hatta et al. (2015) attributed the high DFe concentrations in the LSW to continental margin sediments. Consequently, it is also possible that the elevated DFe concentrations from the three

LSW branches which entered the West European and Iceland Basins and Irminger Sea was supplied through sediment dissolution (Measures et al., 2013) along the LSW pathway.

Lemaître et al. (2017) reported highest remineralization rates within the Labrador and Irminger Seas where high carbon production rates were observed earlier in the season and were attributed to diatoms (>50% of phytoplankton abundances, Tonnard et al., in prep.). In the Labrador Sea, Lemaître et al. (2017) hypothesized that the important convection of the LSW

enabled the deepening of the mesopelagic layer allowing these intense remineralization rates coinciding with the LSW in both basins. Gourain et al. (in prep.) measured relatively high PFe:PAl ratios (0.39 ± 0.08 mol mol$^{-1}$) in the LSW from the Irminger and Labrador Seas which could point to a more biogenic PFe contribution.

Remineralization occurred in both the Labrador and Irminger Seas with slightly higher rates in the former (Lemaître et al., 2017). Conspicuous DFe concentrations were, however, observed in the Irminger Sea. This could be explained by the reductive

dissolution of Newfoundland Margin sediments. Nevertheless, remineralization of organic matter likely plays an additional role in the observed high DFe concentrations from the Irminger Sea through bacteria-mediated ligand production (Boyd and Ellwood, 2010) helping the DFe supply from reductive dissolution of Newfoundland sediments to the LSW to remain in solution.

### 4.1.4 Enhanced Bottom Water Irminger Sea DFe concentrations

Bottom waters from the Irminger Sea exhibited the highest DFe concentrations from the whole section, excluding the stations above the margins. Such a feature could be due to i) sediment inputs, ii) intrusion of an Fe-rich water mass, iii) dissolution of Fe from particles. Here after, we discuss the plausibility of these three hypotheses to occur.

The GEOTRACES GA02 voyage (leg 1, 64PE319) which occurred in April-May 2010 from Iceland to Bermuda sampled two stations north and south of our station 44 (~ 38.95°W, 59.62°N): station 5 (~ 37.91°W, 60.43°N) and 6 (~ 39.71°W, 58.60°N),

respectively. High DFe concentrations in samples collected above the seafloor were also observed and attributed to sediment inputs highlighting boundary exchange between seawater and surface sediment (Lambelet et al., 2016; Rijkenberg et al., 2014). However, because a decrease in DFe concentrations was observed at our station 44 from 2500 m depth down to the bottom



(Fig. 4, Table 3), it appeared to be unlikely that these high DFe concentrations will be the result of sediment inputs, as no DFe gradient from the deepest samples to the above ones was observed.

Looking at salinity versus depth for these three stations, one can observe the intrusion of the Polar Intermediate Water (PIW) at station 44 from GA01 which was not observed during the GA02 voyage and which contributed to about 14% of the water

mass composition (García-Ibáñez et al., this issue) and might therefore be responsible for the high DFe concentrations (Fig. 8A). On the other hand, the PIW was also observed at station 49 (from 390 to 1240 m depth), 60 (from 440 to 1290 m depth), 63 (from 20 to 1540 m depth), 68 (3340 m depth), 69 (from 3200 to 3440 m depth), 71 (from 2950 to 3440 m depth) and 77 (60 and 2500 m depth) with similar or higher contributions of the PIW without such high DFe concentrations (maximum DFe = $1.3 \pm 0.1$ nmol $L^{-1}$, 1240 m depth at station 49).

Figure 8B) also shows the concentrations of both DFe and PFe for the mixing line between DSOW/PIW and ISOW at station 44 and considering 100% of contribution from these seawater at the deepest (2915 m depth) and the shallowest (2218 m depth) samples, respectively, as these were the main water masses. This figure shows an exponential decrease of PFe concentrations as the DSOW/PIW mixed with the ISOW (Gourain et al., in prep.). Concomitantly, DFe concentrations followed a polynomial distribution with a theoretical DFe maxima of 2.6 nmol $L^{-1}$ at 51.5% contribution of the ISOW (48.5% of the DSOW/PIW)

and corresponding to a theoretical PFe concentration of ~ 7.0 nmol $L^{-1}$. It is possible that as the ISOW mixed with the Fe-rich particles from the DSOW/PIW, DFe concentrations increased until PFe concentrations reached 7.0 nmol $L^{-1}$ after which the amounts of particles tended to scavenge Fe. Therefore, the high DFe concentrations observed might be inferred from local processes as ISOW mixes with both PIW and DSOW with a substantial load of dissolvable Fe-rich particles which might be sustained in solution by Fe-binding organic ligands.

**4.1.5 Reykjanes Ridge: Hydrothermal inputs or Fe-rich seawater?**

High DFe concentrations (up to $1.5 \pm 0.22$ nmol $L^{-1}$, station 36, 2200 m depth) were measured east of the Reykjanes Ridge (Fig. 4). Within the interridge database, based on a literature review on existing vent-field databases, and on unpublished sources, ~280 sites of active hydrothermal venting on spreading ridges, volcanic arcs and intraplate volcanoes were reported (http://www.interridge.org). In this database, the Reykjanes Ridge is reported to have inferred hydrothermal sites (Baker and

German, 2004). Several studies on hydrothermal activity at Reykjanes Ridge have been conducted (e.g. Chen, 2003; German et al., 1994; Sinha et al., 1997; Smallwood and White, 1998) but it remained unclear, as no high DFe concentration or temperature anomaly were observed above the ridge. According to the water mass circulation, if there was an active hydrothermal vent located on the Reykjanes Ridge, the only possibility to observe a signal would occur from a southward transfer of this signal through the ISOW in the Iceland Basin. Interestingly, previous studies (e.g. Fagel et al., 1996; Fagel et

al., 2001; Parra et al., 1985) reported marine sediment mineral clays in the Iceland Basin largely dominated by smectite (> 60%), a tracer of hydrothermal alteration of basaltic volcanic materials (Fagel et al., 2001; Tréguer and De La Rocha, 2013). Hence, the high DFe concentrations measured east of the Reykjanes Ridge could be due to a hydrothermal source as previously





suggested by Achterberg et al. (2018) who highlighted at ~60°N a lateral transport of the hydrothermal Fe plume of up to 250-300 km.

While hydrothermalism may explain the enrichment east of the Reykjanes Ridge, further downstream the enrichment may come from other sources. An intriguing phenomenon happens as the ISOW flows towards Charlie Gibbs Fracture Zone (CGFZ) and Bight Fracture Zone (BFZ). An enrichment in Fe relative to the ISOW mean DFe concentration observed in the Iceland Basin, was noticed in the Irminger Sea (Figs. 4 and 7). Hydrographic sections of the northern valley of the CGFZ showed that below 2000 m depth the passage through the Mid-Atlantic Ridge was filled mainly with the ISOW. Shor et al. (1980) highlighted a total westward transport across the sill, below 2000 m depth of about 2.4 x $10^6$ $m^3$ $s^{-1}$ with ISOW carrying a significant load of suspended sediment (25 µg $L^{-1}$), including a 100-m-thick bottom nepheloid layer. The seabed of this area has been identified as a depositional environment with patches of ripples and rock fragments surrounded by moat. Moreover, higher transmissometer values within the ISOW from the Iceland Basin relative to those in the Irminger Sea have been observed, highlighting a particle load of the ISOW in the Irminger Sea with however, lower median PFe concentrations in the Irminger Sea (2.2 nmol $L^{-1}$) than in the Iceland Basin (6.8 nmol $L^{-1}$) (Gourain et al., in prep.). Consequently, the increase in DFe within the ISOW more likely came from sediment resuspension and dissolution at the CGFZ and BFZ.

## 4.2 Main sources of DFe in surface waters

During GA01, enhanced DFe surface concentrations were observed at several stations (stations 1-4, 53, 61, 78) highlighting an external source of Fe to surface waters. The main sources able to deliver DFe to surface waters are the riverine inputs, glacial inputs and atmospheric deposition. In the following sections, these potential sources of DFe in surface will be discussed.

### 4.2.1 Tagus riverine inputs

Enhanced DFe surface concentrations (up to 1.07 ± 0.12 nmol $L^{-1}$) were measured over the Iberian Margin (stations 1-4) and coincided with low salinities (35.3) and enhanced DAl concentrations (up to 31.8 nmol $L^{-1}$, Menzel Barraqueta et al., 2018). DFe and DAl concentrations were both significantly negatively correlated with salinity ($R^2$ = ~1 and 0.94, respectively) from stations 1 to 13 (Fig. 9). Salinity profiles from station 1 to 4 showed evidence of a freshwater source with surface salinity ranging from 34.95 (station 1) to 35.03 (station 4). Within this area, only two freshwater sources were possible: 1) wet atmospheric deposition (4 rain events, Shelley, pers. comm.) and 2) the Tagus River, since the ship SADCP data revealed a northward circulation (Lherminier et al., pers. comm.). However, DFe:DAl ratios were very low (0.036 ± 0.004 mol $mol^{-1}$) from station 1 to 11 compared to surrounding waters (DFe:DAl ratios = 0.11 ± 0.04 mol $mol^{-1}$, stations 15 and 17) and Fe:Al ratios observed within aerosols (0.28 ± 0.04 mol $mol^{-1}$, Shelley et al., 2017b) which were indicative of a minor atmospheric source for those stations. The Tagus estuary is the largest in the western European coast and very industrialized (Canário et al., 2003; de Barros, 1986; Figueres et al., 1985; Gaudencio et al., 1991; Mil-Homens et al., 2009), extends through an area of 320 $km^2$ and is characterized by a large water flow of 15.5 $10^9$ $m^3$ $y^{-1}$ (Fiuza, 1984). Many types of industry (e.g. heavy





metallurgy, ore processing, chemical industry, petroleum refinery, shipbuilding, chlor-alkali industry, a smelter and a pyrite roasting plant) release heavy metals such as lead (Pb, Carvalho, 1995), mercury (Hg, Canario, 2000; Ferreira, 1997; Figueres et al., 1985), arsenic (As, Andreae et al., 1983), or other trace metals (Cd, Cu, Ni, Zn, Cotté-Krief et al., 2000) into the river which therefore result in high levels of these elements recorded in surface sediments, suspended particulate matter, water and

organisms in the lower estuary. Santos-Echeandia et al. (2010) reported that sediments, pore water and belowground biomass colonised by salt marsh plants from the Tagus estuary contained high concentrations of Fe, Mn, Zn, Cu, Pb and Cd which were exported to the water column as a result of tidal inundation. Similar discharge processes releasing large amount of DFe from creeks draining coastal wetland to the ocean have also been reported by Sanders et al. (2015). Consequently, the enhanced DFe concentrations observed above the Iberian Margin likely originated from the Tagus River, wet atmospheric deposition

playing a minor additional source.

### 4.2.2 High latitudes meteoric water and sea-ice processes

Potential sources of Fe at stations 53, 61 and 78 include meteoric water (MW, referring to precipitation, runoff and continental glacial melt), sea-ice melt (SIM), seawater interaction with shallow sediments and advection of water transported from the Arctic sourced by the Fe-rich TransPolar Drift (TPD, Klunder et al. (2012); Table 3). The vertical profiles of both potential

temperature and salinity in the Greenland and Newfoundland Margins (station 53, 61 and 78, Fig. 5 D), E) and F)) highlighted the influence of fresh and cold waters originating from the Arctic Ocean to separate surface and deeper samples at ~ 60 m (station 53) and ~ 40 m (stations 61 and 78) depth. The presence of this front suggests that sediment derived enrichment to these surface waters was unlikely. The most plausible sources would be glacial sources with multi-year sea-ice (which extended as close as 200 km from our Greenland stations) (http://nsidc.org/arcticseaicenews/) and land ice sheet.

SIM and MW contributions were determined for stations 53, 61 and 78, with mass balance calculations based on the end-members presented in Benetti et al. (2016). For stations over the Greenland Shelf, we assume that the Pacific Water (PW) contribution is negligible for the calculations, supported by the very low PW fractions found at Cape Farewell in May 2014 (see Figure B1 in Benetti et al., 2017). For station 78, located on the Newfoundland shelf, we used nutrient measurements to calculate the PW fractions, following the approach from Jones et al. (1998) (the data are published in Benetti et al., 2017).

*The Greenland shelf*

In Figure 5 D), E) and F), negative sea-ice fractions indicated a net brine release while positive sea-ice fractions indicated a net sea-ice melting. It appears that the highest brine release signal observed East of Greenland (station 53, 100 m depth) was associated with enhanced concentrations of both Fe fractions and with decreasing DAl concentrations, whilst West of

Greenland (station 61, 100 m depth) the brine signal was associated with decreasing concentrations of DFe, PFe and DAl. Considering the date of sampling at stations 53 (16 June 2014) and 61 (19 June 2014), sea-ice formation was unlikely as this period coincides with summer melting in both the Central Arctic and East Greenland (Markus et al., 2009). However, it is possible that the brine observed in our study could originate from sea-ice formation, which occurred during the previous




winter(s) at 66°N (and/or higher latitudes). The residence time can vary from days (von Appen et al., 2014, to 6-9 months (Sutherland et al., 2009). Due to our observed strong brine signal at station 61 we suggest that the residence time was potentially longer than average. Given that the brine signal was higher at station 61 than at station 53, (which was located upstream in the EGC), we suggest that station 53 was exhibiting a freshening as a result of the transition between the freezing period toward

the melting period.  This would result in a dilution of the brine signal at the upstream station. Consequently, the salinity of this brine signal may reflect sea ice formation versus melting which may have an effect on the trace metal concentration within this water (Hunke et al., 2011). The associated brine water at station 61 was slightly depleted in both DFe and PFe which may be attributed to sea ice formation processes and the associated uptake of these Fe fractions from the seawater. Conversely, the brine signal observed at station 53 showed slight enrichment in DFe which may be attributed to brine release during early sea

ice melting and the associated release DFe into the underlying seawater.

Surface waters (from 0 to ~ 50 m depth) from station 53 and 61 were characterized by high MW fractions (ranging from 8.3 to 7.4% and from 7.7 to 7.4% , respectively, from surface to ~50 m depth) with sea-ice melting contribution (1.5%, 4 m depth) for station 53 and low contribution (0.6%) from brine release at the surface, linearly increasing with depth (1.3% at 50 m depth and 2.2 % at 100 m depth). Station 53, exhibited enhanced PFe concentrations (19 nmol L$^{-1}$, 25 m depth) at the surface. The

corresponding DFe sample was lost and therefore no information was available. Conversely, both Fe fractions were lower at the sample closest to the surface, then reached a maximum at ~ 50 m depth and decreased at ~ 70 m depth, for station 61. The surface DFe depletion was likely explained by phytoplankton uptake, as indicated by the high TChl-*a* concentrations (up to 6.6 mg m$^{-3}$) measured from surface to about 40 m depth, drastically decreasing at ~ 50 m depth to 3.9 mg m$^{-3}$. Hence, it seemed that meteoric water inputs from the Greenland Margin likely fertilized surface waters with DFe, enabling the phytoplankton

bloom to subsist. The profile of PFe can be explained by two opposite plausible hypotheses: 1) MW inputs did not released PFe, as if it was the case, one should expect higher PFe concentrations at the surface (~25 m depth) than the one measured at 50 m depth due to both the release from MW and the assimilation of DFe by phytoplankton 2) MW inputs can release PFe in a form that is directly accessible to phytoplankton with subsequent export of PFe as phytoplankton died. The latter solution explains the PFe maximum measured at ~ 50 m depth and it thus the most plausible. These results are in agreement with the

capacity of MW from Greenland Ice Sheet and runoffs to deliver DFe and PFe to surrounding waters (Bhatia et al., 2013; Hawkings et al., 2014; Schroth et al., 2014; Statham et al., 2008). These results are also in line with previous observations which highlighted strong inputs of DFe from a meteoric water melting source in Antarctica (Annett et al., 2015, with DFe:DAl ratios up to 1.6 mol mol$^{-1}$; Hawkings et al., 2014), noting that our DFe:DAl ratios from a MW source were 3 to 5 times lower than the one observed by Annett et al. (2015). These differences were likely explained by the sampling period and the stage of

phytoplankton bloom development in our study.

*The Newfoundland shelf*

Newfoundland shelf waters (station 78) were characterized by high MW fractions (up to 7%), decreasing from surface to depth. These waters were associated with a net sea-ice melting signal from the near surface to ~10 m depth followed by a brine release





signal down to 200 m depth with the maximum contribution measured at ~30 m depth. Within the surface waters (above 20 m depth), no elevation in DFe, DAl nor PFe was noticed despite the low measured TChl-*a* concentrations (TChl-*a* ~ 0.20 mg m$^{-3}$). This suggests that none of these inputs (sea-ice melting and meteoric water) were able to deliver DFe or that these inputs were minor compared to sediment inputs from the Newfoundland Margin. Surprisingly, the highest TChl-*a* biomass (TChl-*a*

> 9 mg m$^{-3}$) from the whole section was measured at 30 m depth corresponding to the strongest brine release signal. This suggests that the brine likely contained important amounts of Fe (dissolved and/or particulate Fe) that were readily available for phytoplankton and consumed at the sampling period by potentially sea-ice algae themselves (Riebesell et al., 1991). If such was the case, then a PFe maximum should be noticed at the same depth. However, it should be noted that TChl-*a* and δ$^{18}$O samples were collected about four hours prior to sampling for DFe and PFe. Therefore, it is more likely that by the time DFe

and PFe samples were collected, the PFe was exported deeper in the water column. Indeed, Krembs et al. (2002) highlighted the presence of exopolymeric substances (EPS) in sea ice. Such compounds were reported to undergo fast aggregation (minutes to hours) from the colloidal to the particulate phase (i.e. Transparent Exopolymer Particles, TEP) (e.g. Baalousha et al., 2006; Verdugo et al., 2004) taking in-depth other particulate material as they sank.

### 4.2.3 Atmospheric deposition

On a regional scale, the North Atlantic basin receives the largest amount of atmospheric inputs due to its proximity to the Saharan Desert (Jickells et al., 2005), yet even in this region of high atmospheric deposition, inputs are not evenly distributed. Indeed in the North Atlantic, two areas can be separated: from ~ 5 to 30°N which receives the vast majority of mineral dust from the Saharan outflow, and north of ~ 30°N, which is frequently (but not always) outside of the influence of the atmospheric transport of dust from North African dust source regions and where atmospheric inputs are more likely to contain a higher

proportion of anthropogenic aerosols (from Europe and North America) and high latitude sources in Iceland and Greenland, and occasionally volcanic sources (Achterberg et al., 2013; Bullard et al., 2016; Jickells and Moore, 2015). This division in atmospheric inputs was reflected in the aerosol Fe loading observed during GA01 (Shelley et al., 2017b) which were much lower (up to four orders of magnitude) than those measured during studies from lower latitudes in the North Atlantic (e.g. Baker et al., 2013; Buck et al., 2010; and for GA03, Shelley et al., 2015), but atmospheric inputs could still be an important

source of Fe in areas far from land.

During GA01, 18 aerosol samples were collected roughly every 48 h (Shelley et al., 2017a; 2017b). Shelley et al. (2017a; 2017b) used a combination of air mass back trajectories (five-day simulation period), aerosol trace element concentrations, elemental ratios and multivariate statistics to broadly group the aerosol samples by their dominant, but not mutually-exclusive, source(s). Using this approach, the GA01 transect can be split into four sections. These are those predominantly influenced

by: (1) atmospheric inputs from sources in Iceland and Greenland which likely include proglacial till (stations 11-29), (2) an anthropogenic source coming from UK/Ireland (stations 32-40), (3) a remote marine source influenced by sea salt aerosols and shipping emissions (stations 42-69) and (4) a North American source (stations 71 and 77). Total trace and major elements were determined from strong acid digestions, and fractional solubility was determined following a two-stage sequential leach





(firstly, a weak leach using ultra-high purity water, followed by a stronger leach using 25% acetic acid (HAc), Berger et al., 2008; Buck et al., 2006, respectively). Shelley et al. (2017a; 2017b) reported aerosol fractional solubility for Fe ranging from 7-56%, and for Al from 7-59% following a two-stage sequential leach (an ultra-high purity water leach then a 25% HAc leach). The fractional solubility of aerosol trace metals is dependent on a number of factors, such as aerosol type (mineral dust,

industrial emissions, sea salt, volcanic), and particle size (finer particles are generally more soluble than coarse ones) (Baker and Croot, 2010). Once in the water, Meskhidze et al. (2017) have recently argued that the presence/absence of atmospheric organic acids might be the key parameter that controls the binding of soluble aerosol Fe with organic ligands in seawater, with over 95% of the soluble Fe potentially able to bind to marine organic ligands, compared to less than 20% in the absence of atmospheric organic acids, the rest precipitating out. On the other hand, many studies (Bressac and Guieu, 2013; Bressac et

al., 2014; Desboeufs et al., 2014) argued that the aerosol trace metal fractional solubility is driven by the amount of Dissolved Organic Matter (DOM) in seawater. High and fresh DOM conditions induce a negative feedback on DFe concentrations through rapid formation of aggregates, whereas low DOM conditions allow a transient increase in DFe concentrations before being removed by adsorption onto settling particles. OM (in aerosols and seawater) is thus undeniably a key parameter for controlling the amount of soluble Fe from aerosols which will eventually be available for phytoplankton prior to scavenging.

However, as the organic composition of aerosols was not determined during our study we cannot comment further here.

With this in mind, it seems that DFe:DAl ratios are more or less comparable to soluble aerosol Fe:Al ratios, rather than total aerosol Fe:Al ratios (Fig. 10). This approach allowed us to directly compare the concentrations measured in the seawater with those of aerosols. If atmospheric deposition was the dominant source of Fe and Al, the DFe:DAl ratios would be expected to be close to the soluble aerosol Fe:Al ratios if we assume no involvement in biological cycles, or lower if biological demand

for Fe is higher than Al. In an attempt to estimate whether there was enough atmospheric input to sustain the SML DFe concentrations, we compare the integrated SML DFe concentrations for each station to the corresponding basin (west European and Iceland Basins, Irminger or Labrador Seas) in which soluble aerosol Fe concentrations were averaged (Fig. 10). To do so, we made the following assumptions: 1) the aerosol concentrations are a snapshot in time but are representative of the study region, 2) the aerosol solubility estimates based on two sequential leaches are an upper limit of the aerosol Fe in seawater and

3) the water column stratified just before the deposition of atmospheric inputs, so MLD DFe:DAl will reflect inputs from above. Few stations (26, 38, 40, 63, 68 and 71) exhibited similar DFe:DAl and soluble aerosol Fe:Al ratios, and only station 26 also exhibited a PFe:PAl ratio similar to total Fe:Al aerosols (Fig. 10). This could suggest recent atmospheric deposition at these stations, and particularly at station 26. Indeed, Desboeufs et al. (2005) and Mackey et al. (2015) showed that the dissolution rate of Fe from atmospheric particles is rapid and that the vast majority of dissolvable Fe is mobilised within the

first few minutes after contact with seawater, after which time, Fe is more prone to be scavenged onto particles.

In addition to the evidence from the Fe:Al ratios (Fig. 10), the lowest aerosol Fe loading during GA01 came from samples from the Irminger and Labrador Seas (stations 42-69) which provides further evidence that atmospheric deposition might be unlikely to supply Fe in sufficient quantity to be the main source of DFe (see Section 4.2.2). However, atmospheric deposition



could potentially have been the main source of Fe for stations 26 and 38 as atmospheric Fe loading was up to two orders of magnitude higher here than in the Irminger and Labrador Seas.

### 4.3 Sediment input

*Margins:*

DFe concentration profiles from all coastal stations (stations 2, 4, 53, 56, 61 and 78) are reported on Figure 5. To avoid surface processes, only depths below 100 m depth will be considered in the following discussion. Stations where DFe and PFe followed a similar pattern are stations 2, 53, 56, and 78, suggesting that either the sources of Fe supplied both Fe fractions (dissolved and particulate) or that PFe dissolution from sediments supplied DFe. Conversely, stations 4 and 61 exhibited a decrease in DFe concentrations at the closest samples to the seafloor whereas PFe increased. DFe:PFe ratios ranged from 0.01 (station 2,

bottom sample) to 0.27 (station 4, ~ 400 m depth) mol mol$^{-1}$ with an average value of $0.11 \pm 0.07$ mol mol$^{-1}$ (n = 23), highlighting a different behaviour between DFe and PFe and suggesting different composition of sediments between different margins. Based on particulate and dissolved Fe and dissolved Al data (Gourain et al., in prep.; Menzel Barraqueta et al., 2018, Table 4), three main different types of margins were reported (Gourain et al., in prep.) with the highest lithogenic contribution observed at the Iberian Margin (stations 2 and 4) and the highest biogenic contribution at the Newfoundland Margin (station

78). The East (stations 53 and 56) and West (station 61) Greenland Margins displaying intermediate behaviour. These observations are consistent with the high TChl-*a* concentrations measured at the Newfoundland Margin and to a lesser extent at the Greenland Margin and the predominance of diatoms (Tonnard et al., in prep.). To sum up, biogenic sediments (Newfoundland Margin) were able to mobilise more Fe in the dissolved phase than lithogenic sediments (Iberian Margin), in agreement with Boyd et al. (2010) who reported greater remineralization of PFe from biogenic PFe than from lithogenic PFe

based on field experiment and modelling simulations.

*Nepheloid:*

Samples associated with high levels of particles (transmissometer < 99%) displayed a huge variability in DFe concentrations. To determine which parameter was susceptible to explain the variation in DFe concentrations in these nepheloid layers, a

Principal Component Analysis (PCA) was performed on samples which exhibited a transmissometry lower than 99% and below 500 m depth to avoid surface processes. From the entire dataset, 66 samples (~13% of the entire dataset) respected this criterion with 3 samples from the Iberian Margin (station 4), 14 samples from the West European Basin (station 1), 4 samples from the Iceland Basin (stations 29, 32, 36 and 38), 43 samples from the Irminger Sea (stations 40, 42, 44, 49 and 60) and 2 samples from the Labrador Sea (station 69). The particulate Fe, Al, and particulate manganese oxides (MnO$_2$) (Gourain et al.,

in prep.), the DAl (Menzel Barraqueta et al., 2018) and the Apparent Oxygen Utilization (AOU) were the input variables of the PCA and explained ~93% of the subset variance (Fig. 11). The first dimension of the PCA was represented by the PAl, PFe and MnO$_2$ concentrations and explained 59.5% of the variance, while the second dimension was represented by the DAl





and the AOU concentrations, explaining 33.2% of the variance. The two sets of variables were nearly at right angle from each other, indicating no correlation between them.

The variations in DFe concentrations measured in bottom samples from stations 32, 36 (Iceland Basin), 42 and 44 (Irminger Sea) and 69 (Labrador Sea) were mainly explained by the first dimension of the PCA (Fig. 11). Therefore, samples

characterized by the lowest DFe concentrations (stations 32 and 69) were driven by particulate Al and $MnO_2$ concentrations and resulted in an enrichment of Fe within particles. These results are in agreement with previous studies showing that the presence of Mn oxide particles accelerates the formation of Fe-Mn oxides, contributing to the removal of Fe and Mn from the water column (Kan et al., 2012; Teng et al., 2001).

Low DFe concentrations (bottom samples from stations 42 and 1) were linked to DAl inputs and associated with lower $O_2$

concentrations. The release of Al has previously been observed from Fe and Mn oxide coatings on resuspended sediments under mildly reducing conditions (Van Beusekom, 1988). Conversely, higher DFe concentrations were observed for stations 44 and 49 and to a lesser extent station 60 coinciding with low DAl inputs and higher oxygen levels. This observation challenges the traditional view of Fe oxidation with oxygen, either abiotically or microbially induced which will inevitably leads to rapid Fe removal through precipitation of nanoparticulate or colloidal Fe (oxyhydr)oxides, followed by aggregation

or scavenging by larger particles (Boyd and Ellwood, 2010; Lohan and Bruland, 2008). It is only when sufficient organic matter and more specifically organic ligands are present in solution, that these sediment-derived DFe can remain in solution in excess of its solubility through complexation (Kondo and Moffett, 2015; Noble et al., 2012) or in suspension as colloids or nanoparticles (Raiswell and Canfield, 2012). These higher oxygenated samples were located within the DSOW which has been reported by Tanhua et al. (2005) to mainly originate (75% of the overflow) from the Nordic Seas and the Arctic Ocean. Klunder

et al. (2012) noticed that the ultimate source of Fe to the Arctic Ocean is coming from Eurasian river waters and Slagter et al. (2017) reported that the TPD and therefore the Arctic major rivers are a source of Fe-binding organic ligands. Consequently, these Fe-binding organic ligands are likely transported to the deep ocean as the DSOW formed enabling higher DFe concentrations in seawater.

In summary, the occurrence of particulate $MnO_2$ and the amount of organic matter are the main drivers explaining the DFe

distributions within the benthic nepheloid layers.

### 4.4 DFe and biological activity

Overall, almost all the stations from the GA01 voyage displayed DFe minima in surface water associated with maxima of TChl-*a* (Fig. 3). Consideration of the relationship between DFe and biological uptake are specifically discussed in Tonnard et al. (in prep.), while following discussion specifically addresses the question "Did DFe concentrations potentially limit

phytoplankton growth?".

The $DFe:NO_3^-$ ratios in surface waters varied from 0.02 (station 36) to 37.68 (station 61) mmol mol$^{-1}$. Values were typically equal or lower than 0.27 mmol mol$^{-1}$ in all basins except at the margins and at stations 11, 13, 68, 69 and 77. The low nitrate



concentrations observed at the Iberian, East Greenland, West Greenland and Newfoundland Margins reflected both a strong phytoplankton bloom which had reduced the concentrations and the influence of the Tagus River (Stations 1, 2 and 4), the southward flowing EGC (station 53), the northward flowing WGC (station 61) and the southward flowing LC (station 78). The high $DFe:NO_3^-$ ratios determined at those stations suggested that waters from these areas, despite having the lowest $NO_3^-$

concentrations, were relatively enriched in DFe compared to waters from Iceland Basin and Irminger Sea. In our study, $DFe:NO_3^-$ ratios displayed a gradient from the West European Basin to Greenland. This trend only reverses when the influence of Greenland was encountered, as also observed by Painter et al. (2014). Phytoplankton cellular Fe:N ratios have been found to range from 0.05 to 0.9 mmol mol$^{-1}$ under Fe replete conditions (Ho et al., 2003; Sunda and Huntsman, 1995; Twining et al., 2004). Consequently, $DFe:NO_3^-$ ratios lower than 0.05 mmol mol$^{-1}$ suggested Fe-limitation of phytoplankton growth which

was the case for most of the stations within the Iceland Basin and Irminger Sea. In figure 12, vertical profiles of $DFe:NO_3^-$ ratios within the upper 200 m of the water column at all stations clearly showed that this trend was not only restricted to the surface. Indeed, below 50 m depth, all the stations with the exception of margin stations displayed $DFe:NO_3^-$ ratios lower than the phytoplankton cellular Fe:N ratios, as previously reported by Nielsdóttir et al. (2009) and Painter et al. (2014). The largest drawdown in $DFe:NO_3^-$ ratios was observed between stations 34 and 38 and was likely due to the intrusion of the IcSPMW,

this water mass exhibiting low DFe and high in $NO_3^-$ (from 7 to 8 µmol L$^{-1}$) concentrations. Similarly, the SAIW exhibited high $NO_3^-$ concentrations. Both the IcSPMW and the SAIW sourced from the NAC. The NAC as it flows along the coast of North America receives atmospheric depositions from anthropogenic sources (Shelley et al., 2017b; 2015) which deliver high N relative to Fe (Jickells and Moore, 2015) and might be responsible for the observed disequilibrium.

To assess where DFe concentrations potentially limit phytoplankton growth we subtracted the contribution of organic matter remineralization to the dissolved Fe pool using the tracer Fe$^*$, as defined by Rijkenberg et al. (2014) and Parekh et al. (2005) for $PO_4^{3-}$, and modified here for $NO_3^-$ as follow:

$$Fe^* = [DFe] - R_{Fe:N} \times [NO_3^-] \quad \text{(eq. 1)}$$

where $R_{Fe:N}$ refers to the average biological uptake ratio Fe over nitrogen, and [$NO_3^-$] refers to nitrate concentrations in

seawater. In the following, we use the two end-member ratios $R_{Fe:N}$ ratios which represented the lowest and highest Fe:N uptake found in literature ($R_{Fe:N}$ = 0.05 and 0.9 mmol mol$^{-1}$, respectively). Negative values of Fe* indicate a deficit in DFe concentrations whereas positive values are pointing out to a source of DFe relative to the uptake of $NO_3^-$ (Fig. 13). Consequently, figure 13 shows that phytoplankton communities with very high Fe requirements relative to $NO_3^-$ ($R_{Fe:N}$ = 0.9) will only be able to grow above continental shelves where there is a high supply of DFe. All these results are corroborating the

importance of the Tagus River (Iberian Margin, see section 4.2.1), glacial inputs in the Greenland and Newfoundland Margins (see section 4.2.2) and to a lesser extent atmospheric inputs (see section 4.2.3) in supplying Fe with Fe:N ratios higher than the average biological uptake/demand ratio.

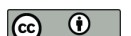



Figures 12 and 14 also highlight the Fe limitation for the low–Fe requirement phytoplankton class ($R_{Fe:N} = 0.05$, Fig. 13) within the Iceland Basin, Irminger and Labrador Seas. The Fe deficiency from the Iceland Basin and Labrador Sea might be explained by low atmospheric deposition for the IcSPMW and the LSW (Shelley et al., 2017b). Low atmospheric Fe supply and sub-optimal Fe:N ratios in winter overturned deep water could facilitate the formation of the High-Nutrient, Low-Chlorophyll

(HNLC) conditions, representing the inefficiency of the biological carbon pump as little or no carbon is transferred below 1000 m depth (Lemaître et al., 2017; Nielsdóttir et al., 2009). Consequently, the low $DFe:NO_3^-$ ratios observed above 100 m depth were probably due to the phytoplankton bloom advancement, coinciding with high remineralized carbon fluxes in this area (station 44; Lemaître et al., 2017). The West European Basin, despite exhibiting some of the highest $DFe:NO_3^-$ ratios within surface waters (Fig. 12), displayed the strongest Fe-depletion from 50 m depth down to the bottom, suggesting that the

main source of Fe was coming from dust deposition. In our study, Shelley et al. (2017b) reported low aerosol Fe loading compared to other studies in the North Atlantic (e.g. Shelley et al., 2015 (GA03)). However, atmospheric loading (and especially Fe) was higher within the subtropical gyre than elsewhere in the GEOVIDE section mainly due to the proximity to mineral dust source (i.e. the Sahara Desert).

## 5 Conclusion

The objectives of the present paper were to describe and discuss the DFe distributions over the whole water column along the 5000 km long transect in the North Atlantic Ocean and the Labrador Sea.

The most striking feature observed during the GEOVIDE voyage was the increasing DFe concentrations inherent to the LSW along its flow path which were likely explained by two processes: i) dissolution of Newfoundland sediments, and ii) potential bacteria-mediated Fe-binding organic ligand production as indicated by intense remineralization rates in the Irminger Sea. This

observation has a broader implication in terms of primary production. Indeed, the intense wind-forcing of deep convection occurring in the Irminger Sea enables the LSW with its enhanced DFe concentrations to reach surface waters, thus initially sustaining intense phytoplankton growth during spring, but which will potentially limit the biological activity later on in the season due to its relative depletion in $NO_3^-$ as indicated by Fe*.

The distribution of DFe along the section also revealed the influence of external sources such as meteoric water melting in the

subpolar gyre close to margins and the input of DFe from the Tagus river above the Iberian Margin. The latter source appears to have less impact as DFe is scavenged onto particles which will inevitably remove it from the mixed layer and entrained it to deep ocean. Dust deposition appears to have been only a minor source of DFe into surface waters, except in the subtropical gyre closer to the African continent.

If the partition between dissolved and particulate forms of Fe is still not well understood in deep ocean, it is clear that it is

mainly dependant of the nature of the sediments and not a direct function of the hydrographic characteristics. Indeed, different processes occurring within the DSOW in the Irminger and Labrador Seas result in DFe sometimes being scavenged onto particles due to Mn-oxide-sediment composition, yet at other times being released from the sediment. We have no clear



explanation regarding the unusually high DFe concentrations (for a non-hydrothermal source) measured between 2000 and 3000 m depth in the Irminger Sea, except from dissolution of Fe-rich particles within DSOW and PIW as they mixed with ISOW.

**Acknowledgements**

We are greatly indebted to the master, Gilles Ferrand, the officers and crew from the N/O *Pourquoi Pas?* for their logistic support during the GEOVIDE voyage. We would like to give a special thanks to Pierre Branellec, Michel Hamon, Catherine Kermabon, Philippe Le Bot, Stéphane Leizour, Olivier Ménage (Laboratoire d'Océanographie Physique et Spatiale), Fabien Pérault and Emmanuel de Saint Léger (Division Technique de l'INSU, Plouzané, France) for their technical expertise during clean CTD deployments as well as Emilie Grosteffan and Manon Le Goff for the analysis of nutrients. We also wanted to

thank the Pôle Spectrométrie Océan (PSO, Plouzané, France) for letting us use the Element XR HR-ICP-MS. Greg Cutter is also strongly acknowledged for his help in setting up the new French clean sampling system. Catherine Schmechtig is thanked for the LEFE-CYBER database management. This work was funded by the French National Research Agency ANR GEOVIDE (ANR-13-BS06-0014) and RPDOC BITMAP (ANR-12-PDOC-0025-01), the French National Center for Scientific Research (CNRS-LEFE-CYBER), the LabexMER (ANR-10-LABX-19) and Ifremer and was supported for the

logistic by DT-INSU and GENAVIR. Manon Tonnard was supported by a cotutelle joint PhD scholarship from the Université de Bretagne Occidentale (UBO-IUEM) and the University of Tasmania (UTAS-IMAS).

All dissolved iron (DFe) data are available in the supplementary material S1.

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





**Figure 1: Map of the GEOTRACES GA01 voyage plotted on bathymetry as well as the major topographical features and main basins. Crossover station with GEOTRACES voyage (GA03) is shown as a red star. (Ocean Data View (ODV) software, version 4.7.6, R. Schlitzer, http://odv.awi.de, 2016).**





**Figure 2: Parameters measured from the regular CTD cast represented as a function of depth for GA01 section for (A) Dissolved Oxygen (O₂, µmol kg⁻¹), (B) Salinity and (C) Temperature (°C). The contour lines represent isopycnals (neutral density, $\gamma^n$, in units of kg m⁻³).**





**Figure 3: Section plot of Total Chlorophyll-a (TChl-*a*) concentrations (mg m⁻³) measured for the GA01 voyage. The black contour lines highlight the TChl-*a* concentrations and the white contour lines highlight the dissolved iron (DFe) concentrations. The red dashed line indicates the depth of the Surface Mixed Layer (SML) (see text for details). (Ocean Data View (ODV) software, version 4.7.6, R. Schlitzer, http://odv.awi.de, 2016).**

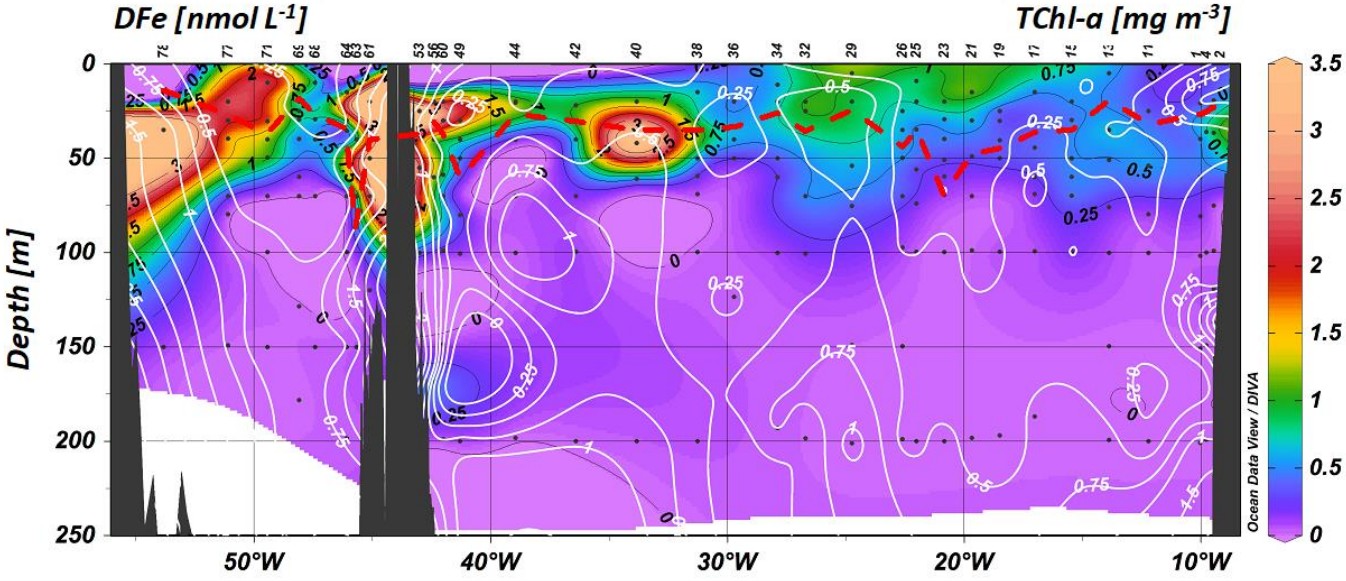





**Figure 4: Contour plot of the distribution of dissolved iron (DFe) concentrations in nmol L⁻¹ along the GA01 voyage transect: upper 1000 m (top) and full depth range (bottom). The red dashed line indicates the depth of the Surface Mixed Layer (SML). Small black dots represent collected water samples at each sampling station. (Ocean Data View (ODV) software, version 4.7.6, R. Schlitzer, http://odv.awi.de, 2016).**





**Figure 5: Vertical profiles of dissolved iron (DFe, black dots, solid line), particulate iron (PFe, black open dots, dashed line, Gourain et al., in prep.) and dissolved aluminium (DAl, grey dots, Menzel Barraqueta et al., 2018) at Stations 2 (A), and 4 (B) located above the Iberian shelf, Station 56 (C), Stations 53 (D) 53 and Station 61 (E) located above the Greenland shelf and Station 78 (F) located above the Newfoundland shelf. Note that for stations 53, 61 and 78, plots of the percentage of meteoric water (open dots) and sea-ice (black dots and dashed line) (Benetti et al., see text for details), Total Chlorophyll-*a* (TChl-*a*, green), temperature (blue) and salinity (black) are also displayed as a function of depth.**

Vertical profiles at Stations 2 (A), 4 (B), 56 (C) along the Iberian Margin; Stations 53 (D), 61 (E) along the West and East Greenland Margin; and Station 78 (F) along the Newfoundland Margin.



**Figure 6: Mean profiles of dissolved iron (Fe) along the North Atlantic section in the West European Basin (purple), Iceland Basin (blue), Irminger Sea (green) and Labrador Sea (red) over the depth intervals: 0-100 m, 100-250 m, 250-500 m, 500-1000 m, 1000-1500 m, 1500-2000 m, 2000-3000 m, 3000-4000 m, 4000-5500 m without considering stations located above the continental plateau.**

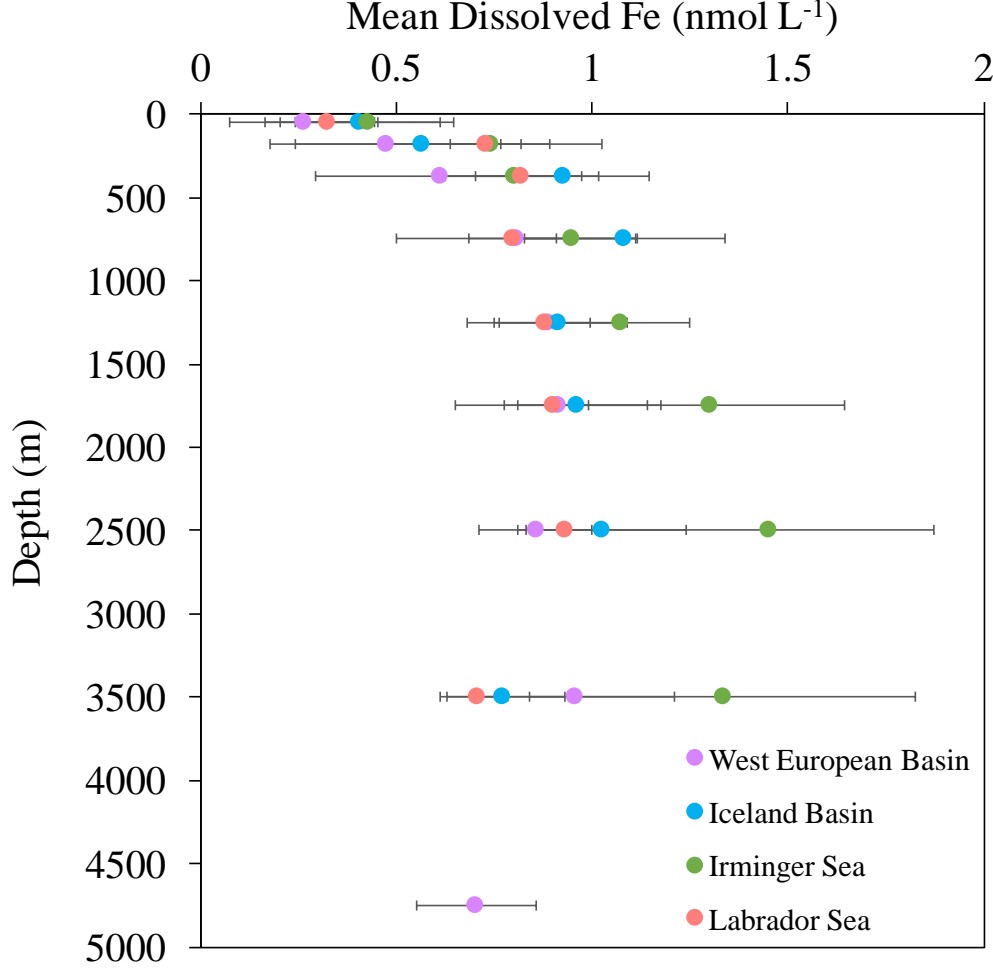

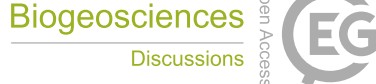

**Figure 7: Box and whisker plot of dissolved iron (DFe) in nmol L$^{-1}$ per water mass and basin. Color coding representing from West to East: the Labrador Sea (red), the Irminger Sea (green), the Iceland Basin (blue) and the West European Basin (purple). Note that stations 1 and 17 were not considered in this plot. SAIW: Sub-Arctic Intermediate Water, ENACW: East North Atlantic Central Water, IrSPMW: Irminger Sub-Polar Mode Water, IcSPMW: Iceland Sub-Polar Mode Water, MOW: Mediterranean Overflow Water, LSW: Labrador Sea Water, ISOW: Iceland-Scotland Overflow Water, DSOW: Denmark Strait Overflow Water, NEADW: North East Atlantic Deep Water.**





**Figure 8: A) Plot of dissolved iron (DFe) concentrations as a function of the percentage of Polar Intermediate Water (PIW) contribution for open-ocean stations (stations 44, 49, 60, 63, 68, 69, 71 and 77). Station 44 highlighted in green and dashed-line representing the linear regression line between DFe concentrations and percentage of PIW contribution for all stations except station 44. B) Plot of dissolved (DFe, black dots) and particulate iron (PFe, open dots, Gourain et al., in prep.) for station 44 (from 2220 m depth to the bottom) as a function of the percentage of mixing between Iceland-Scotland Overflow Water (ISOW) as opposed to Polar Intermediate Water (PIW) and Denmark Strait Overflow Water (DSOW) (Garcia-Ibanez et al., 2015) with polynomial (DFe) and exponential (PFe) regression equations.**

A)

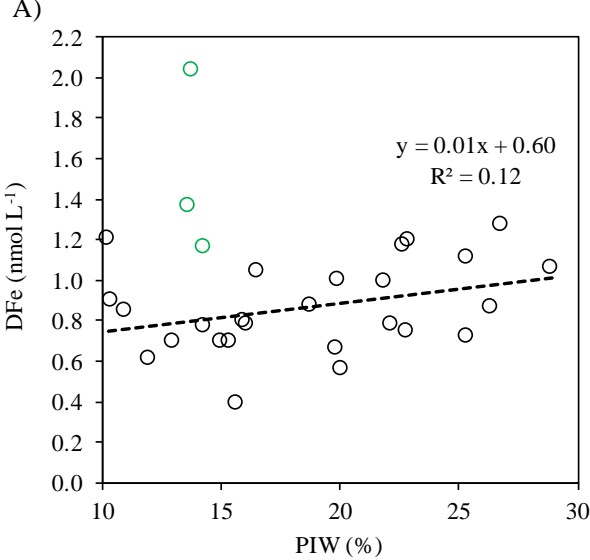

B)

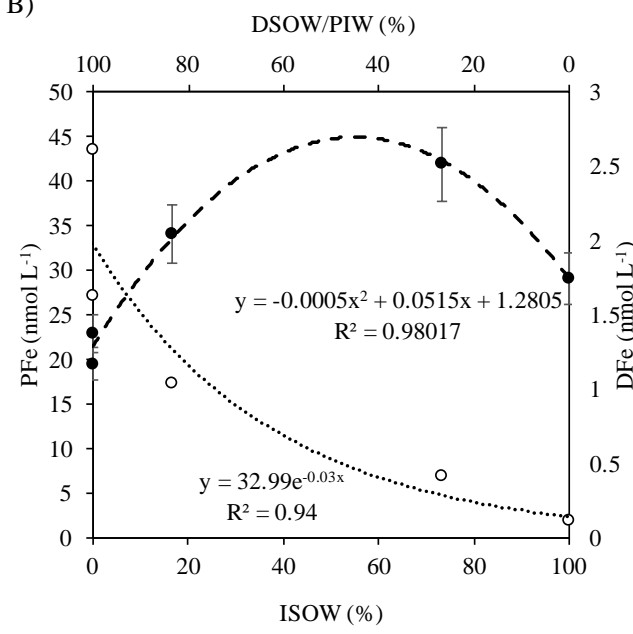





**Figure 9: Plot of dissolved iron (DFe, black circles) and dissolved aluminium (DAl, white circles, Menzel Barraqueta et al., 2018) along the salinity gradient between stations 1, 2, 4, 11 and 13 with linear regression equations. Numbers close to sample points representing station numbers.**

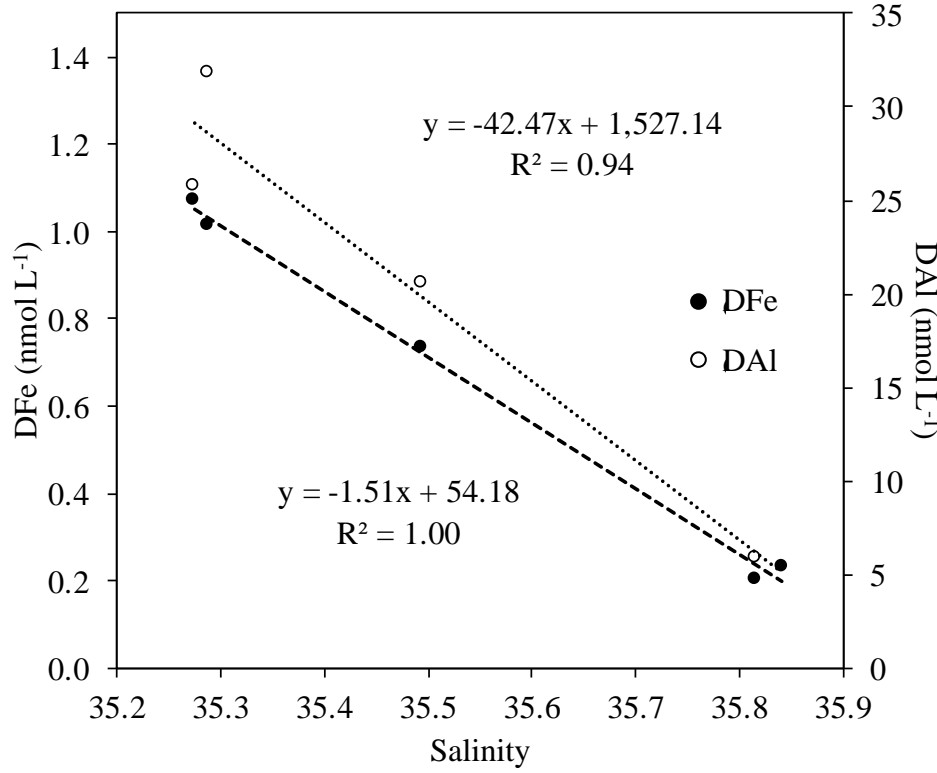



**Figure 10: Graph of iron:aluminium (Fe:Al) ratios for: dissolved (DFe:DAl, black dots, Menzel Barraqueta et al., 2018), particulate (PFe:PAl, open dots, Gourain et al., in prep.), total aerosol (Fe:Al aerosols total, red continuous line) and soluble aerosol (Fe:Al aerosols soluble, black continuous line and dashed lines as standard deviation (SD)) estimated from a two-stage sequential leach (UHP water, then 25% HAc, Shelley et al., this issue). Note that for total and soluble aerosols the Fe:Al ratios are averages per**
5 **section of transect grouped by aerosol source region (Europe, N. America, High latitude, Marine; Shelley et al., this issue). Note that numbers on top of points represent station numbers.**

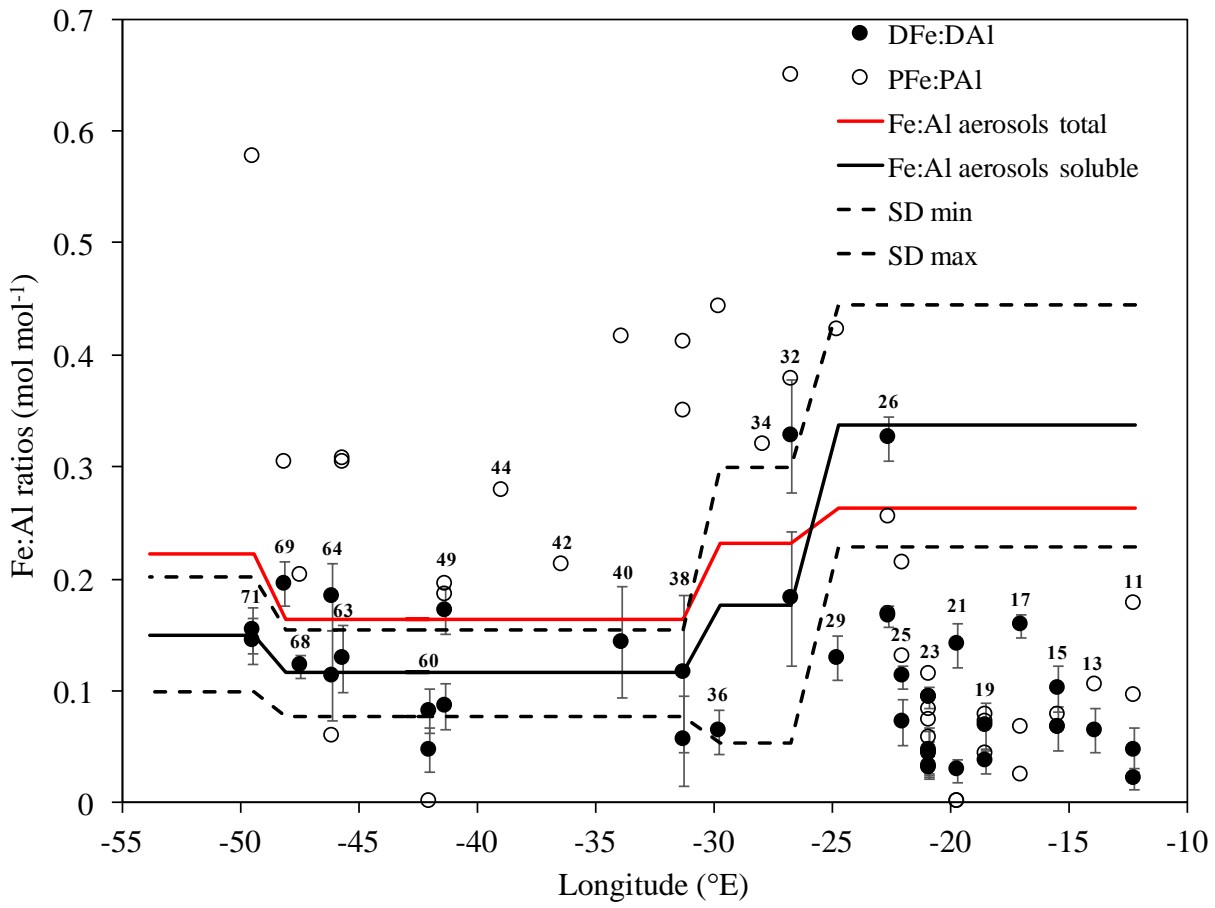





**Figure 11: Plots of the first two dimensions of a Principal Component Analysis (PCA) performed on A) the following variables: Apparent Oxygen Utilization (AOU), dissolved aluminium (DAl, Menzel Barraqueta et al., 2018), particulate iron, aluminum and manganese oxides (PFe, PAl and MnO$_2$, Gourain et al., in prep.) and B) for samples which presented a transmissometry lower than 99% and below 500 m depth to avoid surface processes. Note that the color coding corresponds to different water masses (contribution >60% of the whole water mass pool) with the Denmark Strait Overflow Water (DSOW) in grey, the East North Atlantic Central Water (ENACW) in yellow, the Irminger Sub-Polar Mode Water (IrSPMW) in blue, the Iceland-Scotland Overflow Water (ISOW) in green, the Labrador Sea Water (LSW) in red, the Mediterranean Overflow Water (MOW) in orange, the North East Atlantic Deep Water (NEADW) in pink and mixing of multiple water masses (NA) in white. Plots of dissolved iron (DFe) plotted as a function of distance height above the seafloor for C) the first dimension of the PCA and D) the second dimension of the PCA. Note that positive and negative values are represented in blue and red, respectively and that dot size are function of the particulate iron and manganese oxide ratios (PFe:MnO$_2$, mol mol$^{-1}$).**

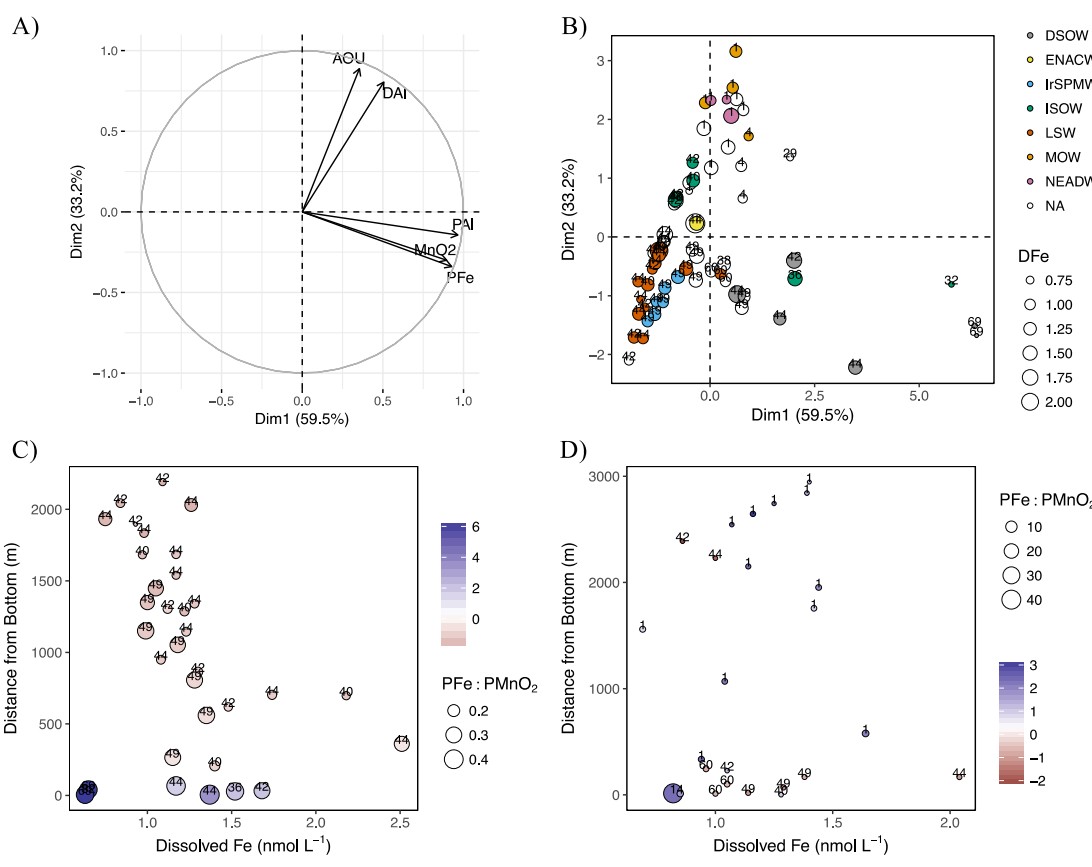





**Figure 12: Vertical profiles of the DFe:NO₃⁻ ratio over the upper 200 m of the water column along the GEOVIDE section. Profiles from the West European Basin are plotted in black, from the Iceland Basin in grey, from the Irminger Sea in green and from the Labrador Sea in red. Stations located above the continental Plateau (stations 1, 2 and 4 from the Iberian Margin; stations 53 and 61 from the Greenland shelf; station 78 from the Newfoundland Margin) are represented with dotted lines. The vertical dashed lines (light blue) indicate lower and upper limits of phytoplankton cellular DFe:NO₃⁻ ratios under Fe replete conditions Ho et al., 2003; Sunda and Huntsman, 1995; Twining et al., 2004.**





**Figure 13: Section plot of the Fe\* tracer in the North Atlantic Ocean with a remineralization rate ($R_{Fe:N}$) of 0.05 mmol mol$^{-1}$ from 100 m depth to bottom waters. A contour line of 0 separates areas of negative Fe\* from areas with positive Fe\*. Positive values of Fe\* imply there is enough iron to support complete consumption of $NO_3^-$ when this water is brought to surface, and negative Fe\* imply a deficit. See text for details.**

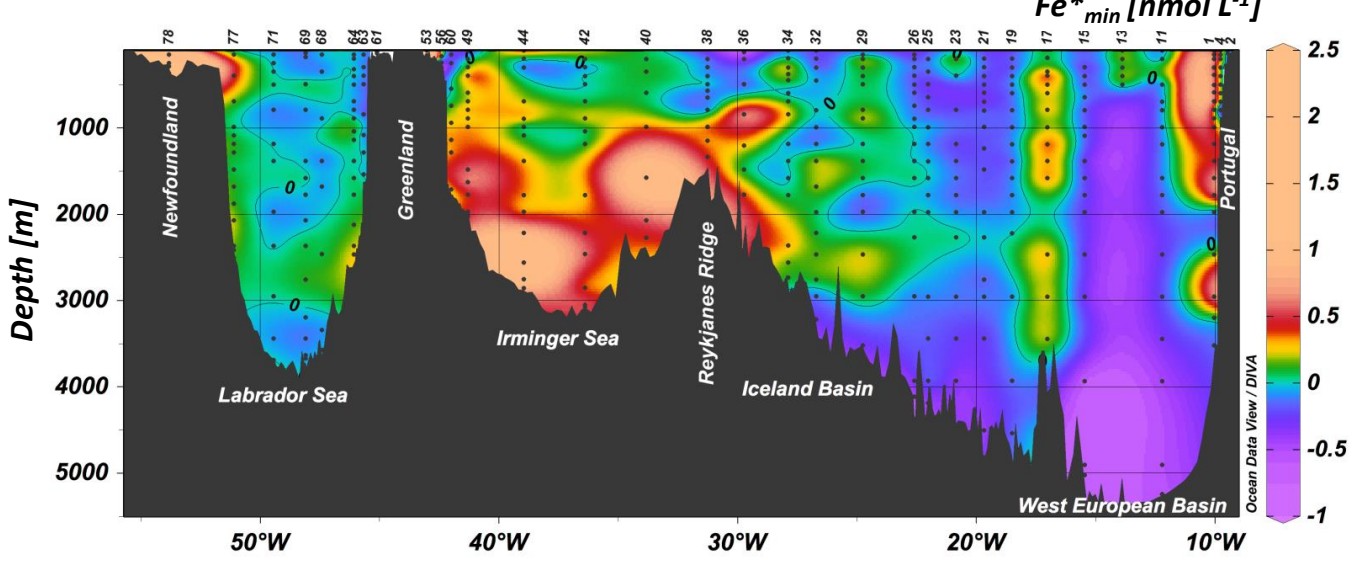





**Figure 14: Box and whisker plot of Fe\* in units of nmol L⁻¹ as determined per water mass and basin with a Fe:N uptake rate of 0.05. Color coding representing from West to East, the Labrador Sea (red), the Irminger Sea (green), the Iceland Basin (blue) and the West European Basin (purple). Abbreviation referring to SAIW: Sub-Arctic Intermediate Water, ENACW: East North Atlantic Central Water, IrSPMW: Irminger Sub-Polar Mode Water, IcSPMW: Iceland Sub-Polar Mode Water, MOW: Mediterranean Overflow Water, LSW: Labrador Sea Water, ISOW: Iceland-Scotland**
5  **Overflow Water, DSOW: Denmark Strait Overflow Water, NEADW: North East Atlantic Deep Water.**

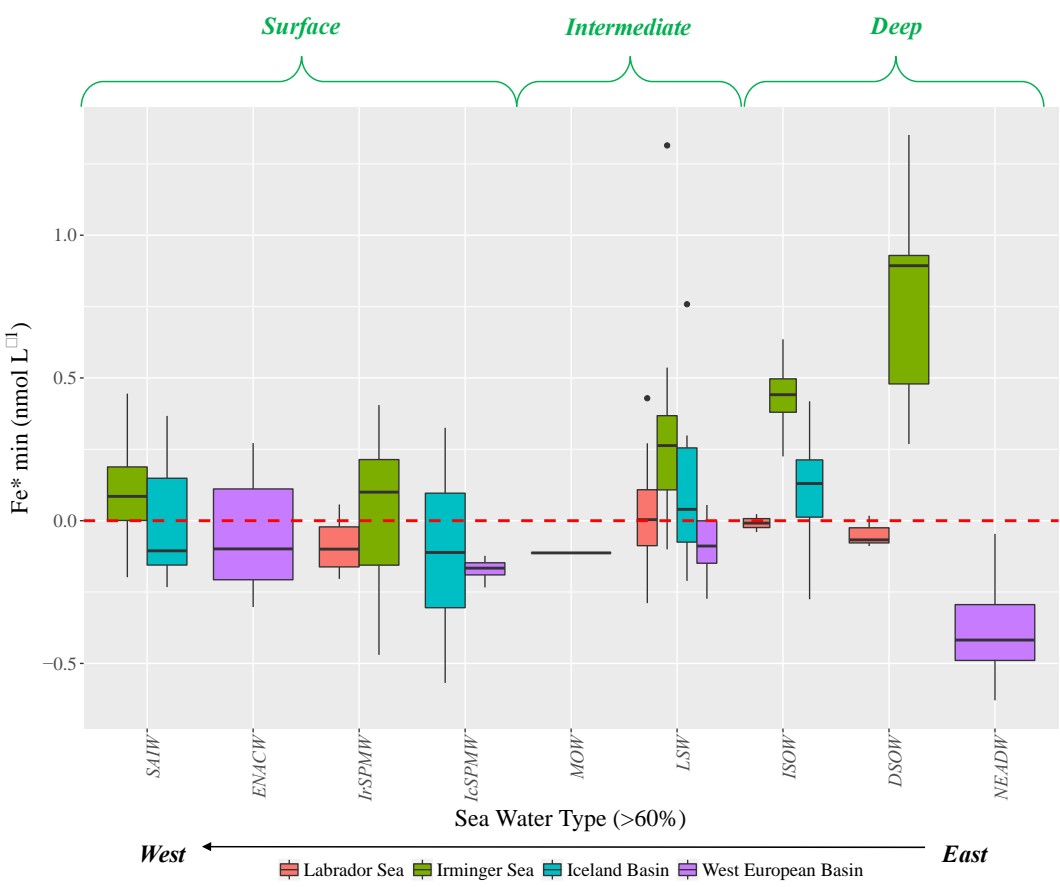




| Station | Date sampling | filtration | Latitude | Longitude | $Z_m$ | DFe (nmol L$^{-1}$) | | |
|---|---|---|---|---|---|---|---|---|
| | DD/MM/YYYY | µm | °N | °E | m | average | SD | n |
| 1 | 19/05/2014 | 0.2 | 40.33 | -10.04 | 25.8 | 1.07 ± | 0.12 | 1 |
| 2 | 21/05/2014 | 0.2 | 40.33 | -9.46 | 22.5 | 1.01 ± | 0.04 | 1 |
| 4 | 21/05/2014 | 0.2 | 40.33 | -9.77 | 24.2 | 0.73 ± | 0.03 | 1 |
| 11 | 23/05/2014 | 0.2 | 40.33 | -12.22 | 31.3 | 0.20 ± | 0.11 | 2 |
| 13 | 24/05/2014 | 0.45 | 41.38 | -13.89 | 18.8 | 0.23 ± | 0.02 | 1 |
| 15 | 28/05/2014 | 0.2 | 42.58 | -15.46 | 34.2 | 0.22 ± | 0.03 | 2 |
| 17 | 29/05/2014 | 0.2 | 43.78 | -17.03 | 36.2 | 0.17 ± | 0.01 | 1 |
| 19 | 30/05/2014 | 0.45 | 45.05 | -18.51 | 44.0 | 0.13 ± | 0.05 | 2 |
| 21 | 31/05/2014 | 0.2 | 46.54 | -19.67 | 47.4 | 0.23 ± | 0.08 | 2 |
| 23 | 02/06/2014 | 0.2 | 48.04 | -20.85 | 69.5 | 0.21 ± | 0.05 | 6 |
| 25 | 03/06/2014 | 0.2 | 49.53 | -22.02 | 34.3 | 0.17 ± | 0.04 | 2 |
| 26 | 04/06/2014 | 0.45 | 50.28 | -22.60 | 43.8 | 0.17 ± | 0.03 | 2 |
| 29 | 06/06/2014 | 0.45 | 53.02 | -24.75 | 23.8 | 0.17 ± | 0.02 | 1 |
| 32 | 07/06/2014 | 0.2 | 55.51 | -26.71 | 34.8 | 0.59 ± | 0.08 | 2 |
| 34 | 09/06/2014 | 0.45 | 57.00 | -27.88 | 25.6 | NA ± | | 0 |
| 36 | 10/06/2014 | 0.45 | 58.21 | -29.72 | 33.0 | 0.12 ± | 0.02 | 1 |
| 38 | 10/06/2014 | 0.45 | 58.84 | -31.27 | 34.5 | 0.36 ± | 0.16 | 2 |
| 40 | 12/06/2014 | 0.45 | 59.10 | -33.83 | 34.3 | 0.39 ± | 0.05 | 1 |
| 42 | 12/06/2014 | 0.45 | 59.36 | -36.40 | 29.6 | 0.36 ± | 0.05 | 1 |
| 44 | 13/06/2014 | 0.2 | 59.62 | -38.95 | 25.8 | NA ± | | 0 |
| 49 | 15/06/2014 | 0.45 | 59.77 | -41.30 | 60.3 | 0.30 ± | 0.05 | 2 |
| 53 | 17/06/2014 | 0.45 | 59.90 | -43.00 | 36.4 | NA ± | | 0 |
| 56 | 17/06/2014 | 0.45 | 59.82 | -42.40 | 30.0 | 0.87 ± | 0.06 | 1 |
| 60 | 17/06/2014 | 0.45 | 59.80 | -42.00 | 36.6 | 0.24 ± | 0.02 | 2 |
| 61 | 19/06/2014 | 0.45 | 59.75 | -45.11 | 39.8 | 0.79 ± | 0.12 | 1 |
| 63 | 19/06/2014 | 0.45 | 59.43 | -45.67 | 86.7 | 0.40 ± | 0.03 | 1 |
| 64 | 20/06/2014 | 0.45 | 59.07 | -46.09 | 33.9 | 0.27 ± | 0.06 | 2 |
| 68 | 21/06/2014 | 0.45 | 56.91 | -47.42 | 26.3 | 0.22 ± | 0.01 | 1 |
| 69 | 22/06/2014 | 0.45 | 55.84 | -48.09 | 17.5 | 0.24 ± | 0.02 | 1 |
| 71 | 24/06/2014 | 0.45 | 53.69 | -49.43 | 36.7 | 0.32 ± | 0.04 | 2 |
| 77 | 26/06/2014 | 0.45 | 53.00 | -51.10 | 26.1 | NA ± | | 0 |
| 78 | 27/06/2014 | 0.45 | 51.99 | -53.82 | 13.4 | 0.79 ± | 0.05 | 1 |

Table 1: Station number, date of sampling (in the DD/MM/YYYY format), size pore used for filtration (µm), station location, mixed layer depth (m) and associated average dissolved iron (DFe) concentrations, standard deviation and number of samples during the GEOTRACES GA01 transect.





| Seawater used for calibration | SeaFAST-pico™ DFe values (nmol L⁻¹) | | | reference or certified DFe values (nmol L⁻¹) | | |
|---|---|---|---|---|---|---|
| | Average | | SD | n | Average | | SD |
| SAFe S | 0.100 | ± | 0.006 | 2 | 0.095 | ± | 0.008 |
| GSP | 0.16 | ± | 0.04 | 15 | NA | ± | NA |
| NASS-7 | 6.7 | ± | 1.7 | 12 | 6.3 | ± | 0.5 |

**Table 2: SAFe S, GSP and NASS-7 dissolved iron concentrations (DFe, nmol L⁻¹) determined by the Sea*FAST*-pico™ and their consensus (SAFe S, GSP; https://websites.pmc.ucsc.edu/~kbruland/GeotracesSaFe/kwbGeotracesSaFe.html) and certified (NASS-7; https://www.nrc-cnrc.gc.ca/eng/solutions/advisory/crm/certificates/nass_7.html) DFe concentrations. Note that yet no consensual value is reported for the GSP seawater.**





| Area | Time period | | DFe (nmol L$^{-1}$) | | Filtration | Cruise | Reference |
|---|---|---|---|---|---|---|---|
| | Months | Year | range | median | µm | Name | |
| **West European Basin** | | | | | | | |
| *surface (<200 m)* | | | | **0.69** | | | |
| | May-June | 2014 | 0.09 - 3.0 | 0.26 | 0.2/0.45 | GEOVIDE | *this study* |
| | May | 2013 | 0.01 - 0.45 | 0.05 | 0.2 | GA04 | *Gerringa et al., 2017* |
| | October-November | 2010 | 0.06 - 0.98 | 0.49 | 0.2 | GA03 | *Hatta et al., 2015* |
| | June | 2005 | 0.35 - 0.76 | 0.57 | 0.2 | AMT16 | *Ussher et al., 2013* |
| | September | 2004 | 0.33 - 2.6 | 0.49 | 0.2 | AMT15 | *Ussher et al., 2013* |
| | June-August | 2003 | 0.02 - 0.25 | 0.08 | 0.4 | CLIVAR-CO2 | *Measures et al., 2008* |
| | July | 2003 | 0.05 - 5.4 | 0.7 | 0.4 | JR98 | *Nédélec et al., 2007* |
| | October | 2002 | 0.07 - 7.0 | 0.35 | 0.2 | IRONAGE III | *Sarthou et al., 2007* |
| | March | 2002 | 0.23 - 0.47 | 0.34 | 0.2 | IRONAGES | *Laës et al., 2003* |
| | February-March | 2001 | 0.22 - 0.64 | 0.4 | 0.2/0.45 | POMME | *Blain et al., 2004* |
| | June | 1998 | 0.10 - 1.5 | 0.71 | unfiltered | AMT6 | *Bowie et al., 2002* |
| | March | 1998 | 0.48 - 1.6 | 0.82 | 0.2 | MERLIM | *Boyé et al., 2006; 2003* |
| | March | 1998 | 0.34 - 5.9 | 1 | 0.2 | 64PE114 | *de Jong et al., 2007* |
| | May | 1989 | 0.08 - 0.27 | 0.19 | 0.4 | Atlantis II | *Martin et al., 1993* |
| *intermediate (200-1000 m)* | | | | **0.81** | | | |
| | May-June | 2014 | 0.24 - 1.4 | 0.72 | 0.2/0.45 | GEOVIDE | *this study* |
| | May | 2013 | 0.10 - 0.72 | 0.42 | 0.2 | GA04 | *Gerringa et al., 2017* |
| | October-November | 2010 | 0.38 - 1.0 | 0.61 | 0.2 | GA03 | *Hatta et al., 2015* |
| | July | 2003 | 0.35 - 2.2 | 1.2 | 0.4 | JR98 | *Nédélec et al., 2007* |
| | March | 2002 | 0.57 - 0.86 | 0.64 | 0.2 | IRONAGES | *Laës et al., 2003* |
| | June | 1998 | 0.72 - 0.83 | 0.81 | 0.2 | AMT6 | *Bowie et al., 2002* |
| | March | 1998 | 1.1 - 1.6 | 1.4 | 0.2 | MERLIM | *Boyé et al., 2006; 2003* |





| | | | | | | |
|---|---|---|---|---|---|---|
| March | 1998 | 1.3 - 1.9 | 1.6 | 0.2 | 64PE114 | *de Jong et al., 2007* |
| May | 1989 | 0.26 - 0.57 | 0.35 | 0.4 | Atlantis II | *Martin et al., 1993* |
| *deep (>1000 m)* | | | **0.82** | | | |
| May-June | 2014 | 0.54 - 1.6 | 0.81 | 0.2/0.45 | GEOVIDE | *this study* |
| May | 2013 | 0.44 - 0.87 | 0.61 | 0.2 | GA04 | *Gerringa et al., 2017* |
| October-November | 2010 | 0.46 - 1.1 | 0.75 | 0.2 | GA03 | *Hatta et al., 2015* |
| July | 2003 | 1.2 - 4.4 | 1.6 | 0.4 | JR98 | *Nédélec et al., 2007* |
| March | 2002 | 0.67 - 1.2 | 0.82 | 0.2 | IRONAGES | *Laës et al., 2003* |
| June | 1998 | 0.57 - 0.94 | 0.76 | 0.2 | AMT6 | *Bowie et al., 2002* |
| March | 1998 | 1.3 - 1.6 | 1.4 | 0.2 | MERLIM | *Boyé et al., 2006; 2003* |
| March | 1998 | 1.3 - 2.0 | 1.7 | 0.2 | 64PE114 | *de Jong et al., 2007* |
| May | 1989 | 0.54 - 0.66 | 0.6 | 0.4 | Atlantis II | *Martin et al., 1993* |
| **Iceland Basin** | | | | | | |
| *surface (<200 m)* | | | **0.16** | | | |
| May-June | 2014 | 0.09 – 0.77 | 0.35 | 0.2/0.45 | GEOVIDE | *this study* |
| June | 2009 | 0.08 - 0.87 | 0.24 | 0.2 | D340 | *Mohamed et al., 2011* |
| August-September | 2007 | 0.04 - 0.34 | 0.14 | 0.2 | D321 | *Mohamed et al., 2011* |
| July-September | 2007 | 0.02 - 0.41 | 0.06 | 0.2 | | *Nielsdóttir et al., 2009* |
| June-August | 2003 | 0.02 - 0.30 | 0.10 | 0.4 | CLIVAR-CO2 | *Measures et al., 2008* |
| May | 1989 | 0.06 - 0.23 | 0.12 | 0.4 | Atlantis II | *Martin et al., 1993* |
| *intermediate (200-1000 m)* | | | **0.63** | | | |
| May-June | 2014 | 0.29 - 1.7 | 0.91 | 0.2/0.45 | GEOVIDE | *this study* |
| June | 2009 | 0.24 - 2.23 | 0.63 | 0.2 | D340 | *Mohamed et al., 2011* |
| August-September | 2007 | 0.2 - 0.85 | 0.46 | 0.2 | D321 | *Mohamed et al., 2011* |
| July-September | 2007 | 0.07 - 0.80 | 0.40 | 0.2 | | *Nielsdóttir et al., 2009* |
| May | 1989 | 0.17 - 0.54 | 0.37 | 0.4 | Atlantis II | *Martin et al., 1993* |
| *deep (>1000 m)* | | | **0.88** | | | |





| | | | | | | | |
|---|---|---|---|---|---|---|---|
| | May-June | 2014 | 0.65 - 1.5 | 0.94 | 0.2/0.45 | GEOVIDE | *this study* |
| | June | 2009 | 1.42 - 2.6 | 1.5 | 0.2 | D340 | *Mohamed et al., 2011* |
| | August-September | 2007 | 0.08 - 1.5 | 0.71 | 0.2 | D321 | *Mohamed et al., 2011* |
| | May | 1989 | 0.53 - 0.79 | 0.59 | 0.4 | Atlantis II | *Martin et al., 1993* |
| **Irminger Sea** | | | | | | | |
| *surface (<200 m)* | | | | **0.38** | | | |
| | May-June | 2014 | 0.23 – 3.1 | 0.51 | 0.2/0.45 | GEOVIDE | *this study* |
| | April-May | 2010 | 0.08 - 0.55 | 0.15 | 0.2 | GA02 | *Rijkenberg et al., 2014* |
| *intermediate (200-1000 m)* | | | | **0.61** | | | |
| | May-June | 2014 | 0.34 - 1.3 | 0.88 | 0.2/0.45 | GEOVIDE | *this study* |
| | April-May | 2010 | 0.28 - 0.69 | 0.48 | 0.2 | GA02 | *Rijkenberg et al., 2014* |
| *deep (>1000 m)* | | | | **0.88** | | | |
| | May-June | 2014 | 0.75 - 2.5 | 1.2 | 0.2/0.45 | GEOVIDE | *this study* |
| | April-May | 2010 | 0.65 - 0.99 | 0.75 | 0.2 | GA02 | *Rijkenberg et al., 2014* |
| **Labrador Sea** | | | | | | | |
| *surface (<200 m)* | | | | **0.31** | | | |
| | May-June | 2014 | 0.11 - 2.4 | 0.54 | 0.2/0.45 | GEOVIDE | *this study* |
| | April-May | 2010 | 0.05 - 0.58 | 0.17 | 0.2 | GA02 | *Rijkenberg et al., 2014* |
| *intermediate (200-1000 m)* | | | | **0.67** | | | |
| | May-June | 2014 | 0.63 - 7.8 | 0.81 | 0.2/0.45 | GEOVIDE | *this study* |
| | April-May | 2010 | 0.35 - 0.87 | 0.55 | 0.2 | GA02 | *Rijkenberg et al., 2014* |
| *deep (>1000 m)* | | | | **0.70** | | | |
| | May-June | 2014 | 0.62 - 1.1 | 0.87 | 0.2/0.45 | GEOVIDE | *this study* |
| | April-May | 2010 | 0.47 - 0.66 | 0.59 | 0.2 | GA02 | *Rijkenberg et al., 2014* |
| **Arctic Ocean** | | | | | | | |
| *surface (<200 m)* | July | 2008 | 2.1 - 16 | | 0.22 | | *Nishimura et al., 2012* |





| | September | 2008 | 0.5 - 3.2 | | 0.22 | MR 08-04 | *Nakayama et al., 2011* |
|---|---|---|---|---|---|---|---|
| | August - September | 2007 | 0.10 - > 10 | 0.6 | 0.2 | ARK XXII/2 | *Klunder et al., 2012* |
| | July | 2007 | 5.7 - 23 | | unfiltered | ATOS-Arctic | *Tovar-Sanchez et al., 2010* |
| | April-May | 2007 | 0.8 - 3.1 | 1.5 | 0.4 | | *Aguilar-Islas et al., 2008* |
| *intermediate (200-1000 m)* | August - September | 2007 | 0.20 - 1.4 | 0.5 | 0.2 | ARK XXII/2 | *Klunder et al., 2012* |
| *deep (>1000 m)* | August - September | 2007 | 0.18 - 1.7 | 0.56 | 0.2 | ARK XXII/2 | *Klunder et al., 2012* |

20 **Table 3: Compilation of median dissolved iron (DFe) concentrations (min, max) for Surface (> 200 m depth), Intermediate (from 200 to 1000 m depth) and Deep (>1000 m depth) Waters in the four distinct basins of the GA01 transect and in the Arctic Ocean (data from: the British Oceanographic Data Center website http://www.amt-uk.org/Data, PANGAEA website http://doi.pangaea.de/10.1594/PANGAEA.609968, Clivar & Carbon Hydrographic Data Office website https://cchdo.ucsd.edu/search?bbox=-75,-60,20,65 and GEOTRACES intermediate data product 2017 www.bodc.ac.uk/geotraces/data/idp2017/ and https://webodv.awi.de/geotraces). Bold values indicated for each depth range represent the median DFe concentrations all studies considered per basin.**

| Margins | Stations | DFe:DAl (mol mol⁻¹) | | | PFe:PAl (mol mol⁻¹) | | | n |
|---|---|---|---|---|---|---|---|---|
| | # | average | | SD | average | | SD | |
| *Iberian Margin* | 2 and 4 | 0.07 | ± | 0.03 | 0.20 | ± | 0.01 | 10 |
| *East Greenland Margin* | 56 and 53 | 0.21 | ± | 0.09 | 0.30 | ± | 0.01 | 6 |
| *West Greenland Margin* | 61 | 0.18 | ± | 0.02 | 0.32 | ± | 0.01 | 3 |
| *Newfoundland Margin* | 78 | 1.1 | ± | 0.41 | 0.31 | ± | 0.01 | 4 |

**Table 4: Averaged DFe:DAl (Menzel Barraqueta et al., 2018) and PFe:PAl (Gourain et al., in prep.) ratios reported per margins. Note that to avoid phytoplankton uptake, only depth below 100 m depth are considered.**