# Peer review of "Dissolved iron in the North Atlantic Ocean and Labrador"

_Biogeosciences, 2018_

## Referee Comment (RC1) · CS Schlosser (Referee) · 16 May 2018

The paper submitted discusses the distribution and sources of DFe along the GA01 transact in the North Atlantic. The included DFe data looks great and is of big interest for the entire GEOTRACES community. Thus the manuscript is suitable for Biogeosciences. However, apart from the introduction and MM section, large parts of the result, discussion and conclusion section need substantial overhaul before the article can be published. I recommend major revision.

One of the biggest difficulties for me was to follow their argumentation in paragraphs. The authors did a great job to include large amounts of ideas and literature findings

in each paragraph to explain their DFe distribution. However, in most cases the final outcome drowns by too much detail and unnecessary sentences that do not contribute to the finding.

Another problem I had, some discussions were performed superficial. When the authors discuss the aerosol distribution and DFe, for instance, they focus on elemental ratios and argue then, that not enough soluble Fe from aerosol particles was introduced. You may be able to get some insight about DFe and dust, by comparing fluxes and residence times, but not with ratios. You can use ratios to pinpoint sources, but quantitative assumptions are highly uncertain.

My recommendations are listed below!

However, I see the great potential of the paper, which will help to understand the DFe cycle in the high latitudinal ocean, between the artic and subtropics. In addition, reduces blind spots in the GEOTRACES map (IDP). Anyway, I am happy to review a revised version of the manuscript!

With best regards,

Christian Schlosser

Abstract

Line 29ff: Air-sea interactions responsible for deep winter convection – Did you mean special cooling! Introduction

Page 2 Line 4: I would also include oxygen, the whole ventilation and redox state of the deep ocean depends on deep water formation in the North Atlantic and Weddell Sea

Page 2 Line 6: "stores" is maybe the wrong term; I would rather go with "accumulates"

MM

Page 3 Line 11: Remove "the" from "...aboard the N/O..."

Page 3 Line 24ff: Two different filtration techniques were applied, 0.2 and 0.45 um. Did you test that both approaches deliver the same result? I know water is restricted and sometimes sampling techniques need to be changed, however, please indicate why you did this and that swapping between both filtration techniques did not cause problems (offset, etc.)!

Page 3 Line 26: exchange "on" by "using". By the way, did you apply pressured air to the Go-Flo's to filter your samples. If so, please state that!

Page Line 30: You did you use 0.2% HCl to acidify your water, or? It reads like that! I assume you used concentrated HCl and the dilution with the seawater was than 0.2%.

Page 4 Line 2: The first sentence does not fit here; first you preconcentrated your sample using a SeaFAST system. Than the eluent was introduced via a PFA nebulizer and cyclonic spray chamber into your instrument (please indicate what kind of instrument you used, Element?). Please clarify!

Page 4 Line 11: gravimetrically is perhaps not the right word, you used a balance, right!

Page 4 Line 13ff: please include "..in-house standard seawater..", was this seawater acidified in the same way? And how many samples did you run normally and how much samples were between each calibration curve?

Page 4 Line 16ff: Please include the analytical precision, the blank, detection limit of the analytical method. Please also include, how you calculated your errors, standard deviation of the three slopes? Or just the s.d. of the Element?

Page 4 Line 21: The CTD sensors were deployed on a stainless steel rosette. Correct? Please indicate and correct throw-out the rest of the text.

Page 4 Line 28: Name the parameter $\Delta\sigma$t

Page 4 Line 30: What do you mean with perturbation, at which depth, please indicate in Table 1, for which station this was the case.

Page 5 Line 2ff: Please indicate for which data you applied statists on?

Page 5 Line 3: You did not measure the p-value, you maybe determined or calculated the value.

Page 5 Line 15: Include "...540 data points...

Page 5 Line 19: Exchange "The complete relational database..." by " The complete data set..." Results

Page 5 Line 27: I would swap the two sentences "For a schematic of water masses, currents and pathways, see Daniault et al. (2016)." and "Hereafter we summarise the main features (Fig. 1 and 2)."

Page 6 Line 1: Give a depth range of the "Upper waters (0 – 800 m)" or so! Please also include this to the Intermediate and Deep waters.

Page 6 Line 5: Did you mean with central water the Subarctic intermediate water (SAIW). Please clarify! Please also increase the letter size in Fig. 2. It is really hard to see on a normal A4 print out! There are no currents in Fig. 2, either you somehow include them or remove the caption.

Page 6 Line 18: Please rewrite "..Labrador Sea Water (LSW).

Page 6 Line 29ff: I do not understand the sentence, starting with "During GA01,..."

Page 6 Line 30ff, I am not sure about, explaining the different flow paths, It is really hard to follow without any drawing. Other question, is it really important, since you are just interested in water masses and their DFe signal, and not about currents! I would remove that!

Page 7 Line 8: I do not see any silicic acid and nitrate data, please indicate concentrations and where they can be found.

Page 7 Line 9ff: It is hard to understand what you mean with "PIW is in contact with the

atmosphere once a year (?) during the time of winter convection.." All together there is a lot of water mass information, that can be found elsewhere in the special issue, I would rather shorten that part of the result section.

Page 7 Line 30: Cannot check if this is correct! No nitrate data available.

Page 8 Line 11-12: This is school book knowledge, that is why we are using sensors! Remove the two sentences! However this entire senction 3.2.2 needs an overhaul.

Page 8 Line 20ff: You can delete the first three sentences, they do not contain any important data!

Page 8 Line 29: Also station 61 and 78 are high, at least this is shown by your plot! And replace "...were around..." by "...ranged from..."

Generally, I would merge section 3.3, 3.3.1 and 3.3.2. Figure 4: Are you sure that single elevated values at site 40 (1500m) and at site 44 (500m) are correct. They just seam like outliers to me! Do we really need Fig.5 and 6, we see everything already in Fig. 4.

Page 9 Line 1ff: rewrite sentence, hard to read!

Page 9 Line 6ff: Please provide numbers for surface waters.

Page Line 9ff: But also at station 21 the DFe value is high. I do not think they are significantelly different from the others, s.d. is $\pm$ 20% and higher.

Page 9 Line 17: NEADW was very similar to the median GEOVIDE voyage but compared to test of deep waters lower, please rewrite! But the DSOW in the Labrador Sea was similar.

I am not sure Fig. 7 is really required. It just comprises what we already sea in Fig. 4. And apart from some outliers (hydrothermal? Any Mn data), surface waters, NADW and waters from the Labrador Sea, concentrations are around 1nM. And as numerous times shown, it is impossible to fingerprint water masses with DFe.

Page 10 Line 9: Others showed also elevated concentrations, for instance, station 44. However I understand why the authors decided to explain both station! For myself station 1 is not a problem, it is very close to the continental margin and influenced by lateral water mass transport than the other stations farther off-shore. However, site 17 is a bit more tricky. Did you reanalyze that station, that would confirm that the analysis was alright and you do not face just a strange offset. Anyway, I would discuss station 17, but please rephrase some sentences, it was really hard to grasp the issue you wanted to bring across. From the first sentence it should be clear what the issue is, than explain (eg. Concentrations are irregularly high).

Page 10 Line 23ff: Please provide the numbers from the other studies. Would it be possible to plot the surafe DFe concentration and put the graph in the sup material. Than you can relate to that!

Page 10 Line 29: Please include an opening sentence, what you think is the reason (something similar to the last sentence). It is quite a step from Fe distribution to the original of water mass mixing.

Page 11 Line 5: Explain what tip jets are!

Page 11 Line 10ff: This process is called winter entrainment (Tagliabue et al. 2014). Rephrase sentence and delete the last one (You just repeat yourself.

Page 11 Line 16: Also contaminated waters are introduced!

Page 11 Line 18: What is a stratification period? Be preciss!

Page 11 Line 218ff: You can not compare the Mediterranean surface waters with MOW. Rewrite! DAl and DFe behave entirely different in the water column (residence time, organic complexation, concentrations , etc.), but both of them are likely to be scavenged from particles. So when a dust storm hits, both elements should decrease, do they actualy do this in the water column of the Mediterranean sea. However, I am not too much surprised to see no DFe signal in the MOW. However, I suggest you have a

look for DFe literature values from deep Mediterranean waters (GA04 is not available, a pitty).

Page 12 Line 1: The entire section 4.1.3 is highly speculative. I agree elevated DFe in the Irminger Basin needs to come from somewhere, however, just looking at your Chl a data it is a very productive site, so presumably PFe concentrations are elevated as well, if so you should mention that, than it is just elevated remineralization and intense deep mixing during winter time that is responsible. However, you need to rewrite that section, to make it less speculative, look for existing data!

Page 12 Line 25: the elevated concentration on station 44, is not this just a single point?

Page 12 Line 26ff: Replace "above" by "at", and what are i) sediment inputs (these are particles), and ii) intrusion of an Fe-rich water mass, please be more specific!

Page 12 Line 33: How often have you analyzed the samples below 2.500 m at site 44. For me this is just one outlier, the two other samples from cast 44 in Fig 8A are not that out of the range.

Page 13 Line 10ff: Your argument is based on four data points, I could aslo put a straight line through, with a similar R2. However this entire paragraph is highly speculative! In an earlier paragraph you mention that DFe do not fingerprint different water masses, and now they do? You should remove this section!

Page 13 Line 22: unpublished sources? You need to explain that! Did you look through your Mn and Pb data, when they are also high, we talk about a hydrothermal input of trace metals.

Page 14 Line 3ff: Theer are no elevated DFe values farther east from the ridge! Where is the CGFC and BFC. Questions over questions!

Page 14 Line 13ff: I am confused. Do we talk about station 40 and 1.75nM at 1500 m, this is a single high value for me, and not located in ISOW waters.

Page 14 Line 26ff: The DFe/DAl ratio in seawater can not compared with the Fe/Al ratio of dust particles. Both elements have different fractional solubility's. So the ratio is always different! Remove!

Page 15 Line 1: Remove most of them does not add to the story!

Page 15 Line 5: What do you mean with "..below ground biomass.." In general I do not understand, why you excluded sediments, that could be an additional source.

Page 15 Line 14ff: Fronts refer to temperature and salinity changes in surface waters, such as the Polar Front, not in the water column. Call it different; just use the term "fresh water lens". Why multi-year-sea ice?

Page 15 Line 18ff: But glacial sources and land ice sheet is the same, just call, it "
. . .freshwater induced by meteoric water and sea-ice melt." Than all is clear.

Page 15 Line 27: Where do get the sea-ice fractions from, and explain how it works, include references! And what have brines to do with it, either ice forms or not! Brines are not part of your story, so far I can tell. Brines always from when sea-ice is formed, or in the desert by evaporation. And in line 31 you switch back to sea-ice formation, please stay with that term.

Page 15 Line 33: But brines usually sink, because they are heavier than the surrounding water!!! It is really hard to follow your argumentation here.

Page 16 Line 11: You have to explain how you produced these numbers, a citation in an earlier paragraph is not enough!

Page 16 Line 15: How do you lose a sample! Generally fist you talk about the contribution of MW and then you switch to biological uptake of DFe, that in the same paragraph? You lose the reader here; this entire section needs an overhaul.

Page 16 Line 32: ".. decreasing from surface to depth." Which depth, down to the bottom in 400 m depth? Be precise

Page 16 Line 15-25: What has the tropical and subtropical North Atlantic to do with your work! I assume very little, please delete or at least reduce the text.

Page 17 Line 30: I would rather suggest to say: "Shelley et al. concluded that..." because without any trajectories here I can check, and more or less all this work was already published.

Page 18 Line 13: Do you mean DOM? Or organic material OM. However, you talk about DOM for 7 lines, and then you don't have the data. Once sentence should be enough to point out the importance of DOM.

Page 18 Line 16: This entire paragraph is very poor! It is interesting to compare elemental ratios of seawater with the soluble fraction of dust. But the reasoning here "...whether there was enough atmospheric input to sustain the SML DFe concentrations..." without any flux numbers, residence times is unscientific. Even more strange, at the end of the paragraph you don t even say, whether there is enough or not. Similar to the above, this needs serious work to make it worthwhile reading. There is too much hand waving, and too few data, sorry! I suggest you look up the actual flux numbers and then compare them with your data.

Page 19 Line 5ff: replace "on" by "in". And which similar pattern followed the station. Be precise! Sentence stating in Line 6 makes no sense, please rewrite!

Page 19 Line 11: What has the composition of sediments to do with your PFe value? Nothing...

Page 19 Line 15: "Intermediate behavior" of what? And then Chla? This paragraph is very hard to follow, what is the message you want to bring across, I can't tell!

Page 19 Line 26: "respect"? I respect you as a person, but samples usually don't respect anything?

Page 19 Line 30ff: How do you know its manganese oxide, just use particulate Mn. And why you do not include the transmissometer data. That is what you wanted to

show, or not that resuspended sediments control you particulate fraction.

Page 20 Line 3ff: You did not do a PCA for dFe, so how can you be sure that the dim1 controls DFe? I cannot follow.

Page 20 Line 24: You did not show any evident information that would suggest that DFe is controlled by OM (you did not even show any data) and PMn. Like the others this paragraph needs more work!

Page 20 Line 27ff: Include "some" in front of "maxima, Please tell me the difference between "the relationship between DFe and biological uptake" and "Did DFe concentrations potentially limit phytoplankton growth?" This sounds to me very connected with each other! Why not discussing that in the follow up paper?

Page 20 Line 31: Include mean or average Plus standard deviation

Page 21 Line 4: Please include numbers! Following the text in the paragraph, it is very hard to follow, you jump between F:N ratios, water masses and Chl a. Try to keep it short and weed out unnecessary details. Otherwise you will lose the reader!

Page 21 Line 20ff: Can you explain to me why you calculate Fe* for the entire water body (Fig. 13) and explain DFe limitation of the phytoplankton community. They live in the first 100-200 m. Same fro Fig.14. Reading the last sentence of the section on page 22 "However, atmospheric loading (and especially Fe) was higher within the subtropical gyre than elsewhere in the GEOVIDE section mainly due to the proximity to mineral dust source (i.e. the Sahara Desert)." I feel I am still stuck in the Atmospheric chapter. Please shorten the paragraph and just say what you can prove with data.

Page 22 Line 14ff: The entire conclusion needs an overhaul!

Figures: Figure 1: great;

Figure 2: increase letter size, it is hard to read;

Figure 3: I am not sure that white contour lines for DFe help to understand Chl a. I

would remove DFe and include this figure in the supplementary material.

Figure 4, 5 and 6(?): great

Figure 7, 8, 9, 10: can go in the supplementary material, maybe Fig. 9 you can keep

Figure 11 -14: in the sup mat., Maybe Figure 13 for the first 200 m can stay!

Table 3 and 4: belongs into the sup. material

---

## Referee Comment (RC2) · Anonymous Referee #2 · 26 Jun 2018

The authors present a beautiful set of high-resolution, full-depth dissolved iron data in the North Atlantic Ocean, which contributes to our understanding of the biogeochemical cycling of Fe in the ocean. I highly recommend publication of this article in the journal Biogeosciences since it is of great interest to the scientific and wider communities.

I am aware how much effort has gone into obtaining these data (sampling and analyses). That stage is already a major achievement! However, the text still needs to be considerably improved before publication. The discussion is very hard to follow, due to a lack of flow of thoughts and a lack of clear messages in subsections and para-

graphs. Very often information appears out of context. In general, it is important that the authors rethink what message they want to transmit, organise this information and rewrite the discussion. A similar mistake is repeated, where the authors give almost review type information of published work and then try to find the same in their work. The authors should state what they find, with support of figures, correlations, calculations and ultimately compare their conclusions with the available literature (and not the other way around)! The conclusions section is inherently very poor and needs a revisit. Therefore I suggest major revisions of the discussions section.

Specific comments are listed below:

Line 3: I can spot at least two English native authors and would have therefore expected a better written text.

Page 1, Line 35: "in the Denmark Straight. . ."

Page 1, Line 35: explain what types of particles you are talking about and briefly explain the differences observed (which ones scavenge and which ones release dFe)

Page2, Line 4: The reasoning is not flowing properly here. You need to say that (1) high productivity leads to high atmospheric carbon capture and that (2) Deep water formation leads to sequestration of this carbon into deeper waters, where carbon is stored for longer. The last sentence comes a little out of the blue, needs to be better linked – instead close the paragraph highlighting why it is important to study trace metals in this area.

Page 2, Line10: I can not follow the reasoning in this paragraph. A little bit of a muddle of all the phytoplankton limiting factors (light, nutrients, wind, temp) without a clear insight what factor limits where. Needs to be better explained

Page 2, Line 15: the connection between light limitation and nutrient limitation is not clearly explained

Page 2, Line26: what about soluble Fe? Is this not considered the most bioavailable

form?

Page 3, Line5: Aims of this paper are a bit poor. Add better understanding of the biogeochemical cycling of dFe in the oceans - inform biogeochemical models – and why this is important, what you expect to achieve….

Page 3, Line 20: remove "national"

Page 3, line 30: do you mean concentrated HCl?

Page 3, Line25: why different filtering methods? Have you compared the Fe concentrations in those fractions? i.e., have you collected the same sample with both filtration cut-offs and checked there is no significant difference?

Page 4, Line 4: remove "daily basis"

Page 4, Line 14: replace "run" with "analytical session"

Page 4, line 19: replace "in nmol L-1" with "to nmol L-1"

Page 4, Line 19: would it not be more correct to multiply by the specific density of each sample? If you do not have this data (normally this is a standard parameter obtained from temperature and salinity.

Page 4, Line 19: Please show a comparison of dFe data at the crossover-station with GA02, as an intercalibration exercise.

Page 4, Line25: Why did you not use the CTD data from the trace metal casts? Please explain

Page 5, Line 12: awkward sentence, difficult to follow, please rewrite!

Page 6, Line 5: which central waters? Names?

Page 6, Line 9: to keep consistency, keep "stations 49 and 60" out of the parenthesis. Rephrase the end of the sentence to do so.

Page 6, Line10: specify which stations

Page 6, Line 14: remove "The" from start of sentence

Page 6, Line 19: what is that contribution? 40 %? Please specify!

Page 6, Line 28: ... "was" sourced from...

Page 6, Line 30: and "in the" Labrador Sea

Page 6, Line 34: I am getting a little lost with all those branches, not sure when you're talking about the same one and when you change talking about another one. Not clear, please rephrase this section.

Page 7, Line 5: delete "The" from beginning of sentence. In this entire section please remove "the" in front of water masses. The text should be revised by one of the English speaking co-authors before submission.

Page 7, Line7: "lower" oxygen...

Page 7, Line 4: what do you mean by dense shelf? Do you mean the water masses have higher density? Please rephrase

Page 7, Line 16: "the" Charlie Gibbs...

Page 7, Line 19: mixing with... remove "of the overflow"

Page 7, Line 21: Which stations?

Page 7, Line 30: At least put a nitrate section figure in the supplementary file; otherwise text hard to follow. Data not yet available on the site you referenced

Page 8, Line 4: how do you define the "most open ocean station" for the transect? Deepest? Furthest away from land masses? Are you sure this is st 23?

Page 8, Line 4: it is unclear when you switch to talk about non-surface nitrate concentrations. Please rephrase this section to make this clearer.

Page 8, Line 6: which depths are you talking about?

Page 8, Line 12: isn't the fluorometer calibrated with the Chl-a measurements?

Page 8, Line 13: Specify which depth range you are considering for looking at min/max Chl-a concentrations. Evidently, minimum Chl-a concentrations are found in the deep ocean

Page 8, Line 19: remind the reader here here that all the dFe data can be found in the supplementary file.

Page 9, Line 13: when describing the regions, go in same order as in Figure, otherwise confusing (start Labrador Sea and end WEB)

Page 9, Line 13: remove "the" before MOW and in front of LSW (line 15), SAIW (line 24) and IrSPMW (line 24).

Page 9, Line 20: this is confusing, specify that you mean similar averages and not ranges (the range is larger for IcSPMW)

Page 9, Line 22: LSW and ISOW averages are also similar, combine sentences.

Page 9: Line 24: "composed of" instead of "characterised by"

Page 9, Line 32: Delete "compared to other ones"

Page 10, Line 2: "lowest average dFe value" (DSOW also shows the highest deep water dFe concentrations)

Page 10, Line 9: it is a little odd to start explaining what cannot be included in any of your sub-sections. You start introducing the general structure of your discussion and then you go into much detail explaining dFe trends all the sudden. This is totally out of place here. You should add this paragraph to the end of the discussion or in a new section.

Page 10, Line 20: I don't understand the aim of discussing dFe with water masses? I'd

rather focus on the sources and sinks of dFe along the section. Discuss then the role of water masses in distributing the dFe signals. This means completely changing the focus of this section

Page 11, Line 2: Flow of thoughts not clear, reasons of deep winter mixing scattered; bits of information thrown in a little randomly. You should start off by saying that deep winter mixing is an important mechanism supplying nutrients to the surface ocean in the North Atlantic Ocean; then say how this deep winter mixing is produced (from what I can understand in your text are you trying to say this is due to the effects of wind + convective mixing+ subduction/upwelling; am I right? This was not clear); then say what the specific conditions were in the year you sampled. I am still giving you corrections on the section, which you can incorporate in your rewritten discussion if fitting. I think you can recycle some parts of your discussion.

Page 11, Line 7: "events" and "with a positive NAO"

Page 11, Line 8: "The winter mixed layer depth"

Page 11, Line 9: instead of " and were" use "which was"

Page 11, Line 11: "close to those found in LSW"

Page 11, Line 13: sentence incomprehensive, please rephrase

Page 11, Line 24: need to improve a little the flow of thoughts in this paragraph. Start by saying why you see no MOW dFe signal, then support/contradict that argument(s) by what has been seen in other studies.

Page 12, Line 3: "suggesting that the water mass is enriched in dFe during its flow path"

Page 12, Line 4: start by saying what those sources are and then support with the available literature.

Page 12, Line 6: change "the ones" for "those"

Page 12, Line 12: Provide a brief description of how the remineralisation rates were measured...

Page 12, Line 18: You are repeating yourself!

Page 12, Line 19: conspitious? clearly visible? change this word

Page 12, Line 21: confusing between remineralisation and bacteria mediated ligand production. Please be clear.

Page 12, Line 27: "Hereafter"

Page 13, Line 2: you also need to consider vertical/lateral inputs (think in 3D), so sedimentary Fe could be coming from further north, for example, within nepheloid layers. You can't discard sedimentary inputs just by looking at vertical gradients.

Page 13, Line 3: remove "the" before "Polar Intermediate Water" and "PIW" (line 6)

Page 13, Line 9: instead of thinking that one water mass carries a certain dFe concentration (think of the short Fe residence times), this water mass might have "picked up" some dFe from, e.g., the sediments, on its pathway or It might have picked up particles in suspension which dissolve over time, etc!

Page 13, Line 11: instead of "seawater" use "water masses"

Page 13, Line 11: specify what you mean; 100 % contribution of which water mass at which depth

Page 13, Line 18: watch out, this leads to miss-understanding as the reader thinks you are saying that particulate Fe is sustained by organic ligands. I don't think that is what you want to say.

Page 13, Line 19: restructure this paragraph, say what you think the reasons are behind the high dFe concentrations, then support that by correlations, graphs etc and then compare to literature to confirm or dispute your theory. end the section with a

stronger statement of what you think is happening in these deep water masses

Page 13, Line 26: your explanation is a little long winded. Say that Mid Ocean Ridges can be a source of dFe but that this has not been found in the Reykjanes Ridge so far

Page 13, Line 30: you are repeating you pattern again; please first explain the signals you see and then compare to the literature

Page 14, Line 4-6: are these located on your transect? If not, you need to give their locations (coordinates) and explain where they are relative to your section. Are these GEOVIDE hydrographic sections (I don't think so) or from other cruises? Specify which ones. Cite appropriate Figures, and supporting literature.

Page 14, Line 11: complete this section with more recent references. There have been previous studies in this area

Page 14, Line 20: restructure, again start by saying what you see, explaining the reason of these signals showing correlations, and then supporting literature

Page 14, Line 26: You can't say "et al." in a personal communication. All people should be mentioned. Also State first name and affiliation in a personal communication.

Page 14, Line 26: Very difficult to follow; why "however"? Start by explaining the typical ratios of different sources before you give the observed ratios in your study. Then discuss the What dFe:dAl ratio is expected from a river source? And from an atmospheric source?

Page 14, Line 29: Difficult to understand; you need to present each option that could lead to the enhanced dFe signal, discuss and then accept or discard

Page 15, Line 18: I don't understand this "extended as close as 200 km from our Greenland stations". Please rephrase. Also avoid parenthesis next to parenthesis, and explain clearly what can be found in the link and what info you got from there.

Page 15, Line 20: how were these calculated? give equations

Page 15, Line 27: I suggest instead of describing the Figures, you should use them to support your discussion. So try to avoid starting sentences and paragraphs describing Figures! You do this very often

Page 15, Line 27: how is sea-ice fraction calculated? please provide equations (if you calculated it yourself) or references where this data is published. Provide clear info so the reader can understand

Page 15, Line 30: explain why you compare to dAl in these profiles.

Page 15, Line 33: "originates"

Page 16, Line 7: I can't believe that sea-ice can "uptake" Fe! The concentration of Fe in the newly formed ice and in the remaining water should stay the same. You need to find another explanation!

Page 16, Line 10: you mean sea-ice formation? Hence release of brine? Also explain that the brine sinks because it is denser (this is why it is observed below the surface)

Page 16, Line 10: "release of dFe"

Page 16, Line 10: "underlying water column"

Page 16, Line 11: split this sentence, it is too long

Page 16, Line 15: instead of describing the data (this is more appropriate for results section) you should say what correlates with what (e.g., low Fe with low MW). difficult to follow flow of thoughts

Page 16, Line 17: remind the reader in which figure information can be found (Chl-a, ect)

Page 16, Line 24: results in agreement with the capacity of. . .? weird sentence, please change

Page 16, Line 27: so far you have not talked about dFe:dAl ratio in meteoric water.

Explain if this ratio is used to trace MW inputs and what you see in your profiles. should explain at the start.

Page 16, Line 28: change "noting that" for "although"

Page 17, Line 6: or maybe the brine conditions (pH, salinity etc) make the Fe more bioavailable; or maybe this peak is not Fe related, but related to the release of other TM, or a phytoplankton group that thrives in brine and does not require much Fe? Since this shelf is further south, the environmental conditions may be more favourable for phytoplankton to grow and hence consuming all the dFe more rapidly. You should explore all the possibilities

Page 17, Line 19: You repeat the word "dust" too much

Page 17, Line 20, "proportions"

Page 18, Line 3: you are going into too much detail here about aerosols, which is part of the Shelley papers

Page 18, Line 6: Meskhidze et al. (2017) is not in your reference list

Page 18, Line 10: information is a little randomly thrown in... what is your point? This is not a review paper

Page 18, Line 13: What does OM mean? write abbreviations out first time. Do you mean organic matter?

Page 18, Line 15: So all this background information to lastly say that you don't comment on this? Rephrase, make some assumptions or delete some detail

Page 18, Line 26: of those stations, station 40 is most similar to total aerosol dFe:dAl ratios. station 26 is closer to the soluble than the total composition, so I don't understand what you are saying

Page 18, Line 28: And what about all the other stations where the ratios are different?

I would say this is more of a "coincidence" that these data fall onto the black line. The multiple reactions occurring as Fe enters the ocean change this Fe:Al ratio rapidly

Page 18, Line 30: remove "time"

Page 19, Line 2: I don't understand.... these station points on your figure 10 are similar to those of other stations. What makes you think they have a higher atmospheric influence??

Page 19, Line 4: in this section you should mention that bottom water dFe concentrations were significantly higher on the Newfoundland margin than on the Greenland margins

Page 19, Line 10: "mol:mol" to stay consistent

Page 19, Line 10: what is the average useful for? Show a plot with dFe:pFe ratios or a table

Page 19, Line 11: as well as different sediment compositions, this could be also due to different supply mechanisms? Different sediment conditions (redox, organic content, temp, etc)

Page 19, Line 17: Where is the predominance of diatoms?

Page 19, Line 21: I think this section is great, but you need to organise the ideas clearly. Now difficult to follow. Also name this section "nepheloid layers"

Page 19, Line 26: explain the criterion first. Explain briefly the PCA and the results you show in figure 11. Information is thrown in a little randomly. Please organise you paragraphs

Page 20, Line 1: I would not call it AOU "concentrations" find another way to express this

Page 20, Line 13: you should look into non-reducing dissolution of lithogenic material.

You are missing out on a big topic! Radic et al., 2011; Labatut et al., 2014; Abadie et al., 2017

Page 20, Line 14: "lead"

Page 20, Line 16: instead of "these" use "this sediment-derived..."

Page 20, Line 17: do not state this as facts... these are assumptions

Page 20, Line 25: very poor sum-up, please improve

Page 20, Line 26: You should also compare surface dFe data to AOU to look at dFe released from remineralisation. You have done this for >500 m depth, but it will be worth looking at this more closely below the surface mixed layer, where remineralisation occurs (below 100 m depth).

Page 20, Line 30: change the ending of this sentence, not properly expressed

Page 20, Line 31: First explain why you talk about Fe:nitrate ratios. THis comes out of the blue! Also cite Fig 12.

Page 21, Line 3: what do you mean by influence of the river, and the currents? specify what you mean

Page 21, Line 6: Can you provide a different kind of plot to help visualise this gradient you are talking about. In figure 12 this is impossible

Page 21, Line 18: "disequilibrium" sounds a bit odd, better use the word "ranges"

Page 21, Line 23: do you assume that all the nitrate in seawater comes from remineralisation? Better explain what assumptions this equation relies on

Page 21, Line 26: Rather, negative values of Fe* indicate the removal of dFe that is faster than the input through remineralisation or external sources and positive values suggest input of dFe from external sources

Page 21, Line 27: remove "out"

[Figure]

Page 21, Line 28: you talk about surface waters here but the calculations are done below 100 m depth. I would keep discussion on the external sources of dFe and then link to inputs of dFe rich water masses to surface waters above

Page 22, Line 5: what has the low Fe supply to do with the "inefficient" carbon pump? If you want to talk about the carbon pump, and its inefficiency, you need to support with adequate statements/findings. I do not think this is a finding of your study

Page 22, Line 7: this comes a little out of the blue. Explain a little more the high remineralised carbon fluxes and how they were measured

Page 22, Line 12: poor ending. What about other sources? Margins? Rivers?

Page 22, Line 15: first sentence superfluous

Page 22, Line 23: depletion of nitrate? That doesn't make sense

Page 22, Line 27: "entrained it to the deep..."

Page 22, Line 29: "in the deep ocean"

Page 22, Line 30: do you mean particles? Sediments are on the seafloor. Same for line 32

Page 23, Line 3: conclusions need to be rewritten after the discussion is reworked.

Page 33, Figure 1: explain why you show two dotted lines for the subarctic front in our figure caption.

Page 34, Figure 2: for consistency, cite here also ODV

Page 37, Figure 5, Line 2: "coastal stations 2 (A), and..." Remove "note that" from mine 4. In the figure itself, make dFe scale from 0 to 3 nM in all plots except Stn 78 for better comparison. Similar for all other parameters, make scales the same where possible to better compare stations.

Page 38, Figure 6: if the averaged values are not from exactly the same depths, you

need to give depth error bars too.

Page 39, Figure 7: Explain what the black bar represents in each colour box, and what the red dotted line represents. Remove "note that". vertical lines should delimit the different water masses. As of now it is difficult to tell where each water mass starts and ends.

Page 40, Figure 8: which depth range is used for the plot in A? in the text you say mixing between ISOW/PIW and DSOW; why do you say "opposed" here? Confusing

Page 41, Figure 9: you need to add the station numbers to the dots and depth info in the caption.

Page 42, Figure 10: specify which depth the dissolved and particulate Fe and Al data are from. Why is soluble aerosol data higher than total aerosol? This makes no sense. Remove "note that"

Page 43, Figure 11: you don't need to give all the colour coding if you already have this in the caption. What are the units of the colour bar in plots C and D?

Page 44, Figure 12: this figure needs a legend instead of the long-winded explanation in the caption

Page 45, Figure 13: the contour line needs to be made more visible. be careful how you use ODV integration! The aim is not to fill in "empty spaces", but to reproduce something realistic. This goes for the previous section plots too

Page 45, Figure 14: combine this figure with figure 13 (i.e., A and B) since it is about the same topic

Page 47, Table 1: on the last column put "SML dFe" or "Zm dFe"; "surface mixed layer (Zm)" . . . "and number of samples in the SML (n)"

Page 48, Table 2: "As of yet, no consensus value is. . ."

---

## Author Comment (AC1) · 17 Aug 2018

**Christian Schlosser reviews**

Dear Christian Schlosser and Editor,

The reviewers are thanked for their insightful comments; these have helped to improve the manuscript considerably. Please see our detailed answers to the referees' comments below. Line numbers refer to the new version.

All the answers are attached as a supplementary file.

Best regards,

Tonnard et al.,

**Abstract**

Line 29ff: Air-sea interactions responsible for deep winter convection – Did you mean special cooling!

→ We have changed the sentence "Air-sea interactions were suspected to be responsible for the increase in DFe concentrations within subsurface waters of the Irminger Sea due to deep convection occurring the previous winter…," by "Enhanced air-sea interactions were suspected to be responsible for the increase in DFe concentrations within subsurface waters of the Irminger Sea due to deep convection occurring the previous winter…"

**Introduction**

Page 2 Line 4: I would also include oxygen, the whole ventilation and redox state of the deep ocean depends on deep water formation in the North Atlantic and Weddell Sea

→ We have added "oxygen"

Page 2 Line 6: "stores" is maybe the wrong term; I would rather go with "accumulates"

→ We have modified this part based on your comment and Reviewer#2 comment.

Page 2 Lines 2-33 Pgae 3 Lines 1-20: The North Atlantic Ocean is known for its pronounced spring phytoplankton blooms (Henson et al., 2009; Longhurst, 2007) that induce the capture of atmospheric carbon through photosynthesis, which allows its conversion into particulate organic carbon (POC). This POC is then exported into deeper waters through the production of biogenic particles () and through the Atlantic Meridional Overturning Circulation (AMOC), which is responsible for transporting large amounts of water, heat, salt, carbon, nutrients, oxygen and other elements around the globe. Hence, the North Atlantic Ocean shows the largest oceanic storage rate of anthropogenic $CO_2$ (Pérez et al., 2013) through both the physical and biological carbon pumps, despite covering only 15% of the global ocean area (Humphreys et al., 2016; Sabine et al., 2004) and is therefore crucial for Earth's climate. However, the rapid attenuation of light with depth restricts the growth of phytoplankton organisms to the surface layer, as it is the principal control on phytoplankton growth timing. The extensive studies conducted in the North Atlantic Ocean through the Continuous Plankton Recorder (CPR) have nevertheless highlighted the relationship between the strength of the westerlies and the displacement of the subarctic front (SAF), which corresponds to the North Atlantic Oscillation (NAO) index (Bersch et al., 2007), and the phytoplankton dynamics of

[revised manuscript text omitted]

masses that constitutes it, namely, Iceland Scotland Overflow Water (ISOW), Denmark Strait Overflow Water (DSOW) and Labrador Sea Water (LSW). This will thus better constrain our understanding of the biogeochemical cycling of DFe and how its cycling is linked to wider biogeochemical cycles (i.e. carbon and macronutrients) in the oceans and implement biogeochemical models,

**MM**

Page 3 Line 11: Remove "the" from ". . .aboard the N/O. . ."

➔ We have modified this part as suggested

Page 3 Line 24ff: Two different filtration techniques were applied, 0.2 and 0.45 um. Did you test that both approaches deliver the same result? I know water is restricted and sometimes sampling techniques need to be changed, however, please indicate why you did this and that swapping between both filtration techniques did not cause problems (offset, etc.)!

➔ We have added precision as suggested. Please note that there was no station where both filtration techniques were used.

Page 4 Lines 5-11: Samples were either taken from the filtrate of particulate samples (collected on polyethersulfone filters, 0.45 µm supor°, see Gourain et al., this issue) or after filtration using 0.2 µm filter cartridges (Sartorius SARTOBRAN° 300) due to water restriction (Table 1). No significant difference was observed between DFe values filtered through 0.2 µm and 0.45 µm (p-value > 0.2, Wilcoxon test), neither between stations (i.e. stations 17, 19, 21, 25, 26, 29, 32, 34, 42, 44, 49) while swapping between both filtration techniques (p-values > 0.05, Wilcoxon tests paired by depth and against the sign of the alternative hypothesis depending on the filtration technique used), except between station 11 and 13 and 13 and 15.

Page 3 Line 26: exchange "on" by "using". By the way, did you apply pressured air to the Go-Flo's to filter your samples. If so, please state that!

➔ We have added precision as suggested

Page 3 Line 30: You did you use 0.2% HCl to acidify your water, or? It reads like that! I assume you used concentrated HCl and the dilution with the seawater was than 0.2%.

➔ We have modified the text for clarification

Page 4 Lines 14-15: Samples were then acidified to ~ pH 1.7 with HCl (Ultrapur° Merck, 2 ‰ v/v) under a class 100 laminar flow hood inside the clean container.

Page 4 Line 2: The first sentence does not fit here; first you preconcentrated your sample using a SeaFAST system. Than the eluent was introduced via a PFA nebulizer and cyclonic spray chamber into your instrument (please indicate what kind of instrument you used, Element?). Please clarify!

➔ We have modified this part as suggested

Page 4 Lines 24-27: Seawater samples were preconcentrated using a Sea*FAST*-pico™ (ESI, Elemental Scientific, USA) and the eluent was directly introduced via a PFA-ST nebulizer and a cyclonic spray chamber in an Element XR Sector Field Inductively Coupled Plasma Mass Spectrometry (Element XR

SF-ICP-MS, Thermo Fisher Scientific Inc., Omaha, NE), following the protocol of Lagerström et al. (2013).

**Note that we have also changed part of the reagent text, as we found out there were mistakes.**

Page 4 Lines 28-32, Page 5 Lines 1-2: High-purity grade solutions and water (Milli-Q) were used to prepare the following reagents: the acetic acid-ammonium acetate buffer ($CH_3COO^-$ and $NH_4^+$) was made of 140 mL acetic acid (> 99% NORMATOM® - VWR chemicals) and ammonium hydroxide (25%, Merck Suprapur®) in 500 mL PTFE bottles and was adjusted to pH 6.0 ± 0.2 for the on-line pH adjustment of the samples. The elution acid was made of 1.4 M nitric acid ($HNO_3$, Merck Ultrapur®) in Milli-Q water by a 10-fold dilution and spiked with 1 µg $L^{-1}$ $^{115}$In (SCP Science calibration standards) to allow for drift correction. Autosampler and column rinsing solutions were made of $HNO_3$ 2.5% (v/v) (Merck Suprapur®) in Milli-Q water. The carrier solution driven by the syringe pumps to move the sample and buffer through the flow injection system was made in the same way.

Page 4 Line 11: gravimetrically is perhaps not the right word, you used a balance, right!

➔ We did not changed our sentence here since gravimetrically means by weighting the standards.

Page 4 Line 13ff: please include "..in-house standard seawater..", was this seawater acidified in the same way?

➔ We have modified this part as suggested and added at the end of section 2.1 the precision on the sampling and acidification methods of the in-house standard seawater.

In section 2.1:

Page 4 lines 17-22: Large volumes of seawater sample (referred hereafter as the in-house standard seawater) were also collected using a towed fish at around 2-3 m deep and filtered in-line inside a clean container through a 0.2 µm pore size filter capsule (Sartorius SARTOBRAN® 300) and was stored unacidified in 20-30 L LDPE carboys (Nalgene™). All the carboys were cleaned following the guidelines of the GEOTRACES Cookbook (Cutter et al., 2017). This in-house standard seawater was used for calibration on the Sea*FAST*-pico™ - SF-ICP-MS (see Section 2.2) and was acidified to ~ pH 1.7 with HCl (Ultrapur® Merck, 2 ‰ v/v) at least 24h prior to analysis.

Page 4 Line 16ff: Please include the analytical precision, the blank, detection limit of the analytical method. And how many samples did you run normally and how much samples were between each calibration curve? Please also include, how you calculated your errors, standard deviation of the three slopes? Or just the s.d. of the Element?

➔ We have added precision as suggested.

Page 5 Lines 9-18: Data were blank-corrected by subtracting an average acidified Milli-Q blank that was pre-concentrated on the Sea*FAST*-pico™ in the same way as the samples and seawater standards. Each analytical session consisted of about fifty samples and two calibrations, one at the beginning and another one at the end of each analytical session. The errors associated to each sample were calculated as the standard deviation of the two calibration slopes. The mean Milli-Q blank was equal to 0.08 ± 0.09 nmol $L^{-1}$ (n = 17). The detection limit, calculated for a given run as 3 times the standard deviation of the Milli-Q blanks, was on average 0.05 ± 0.05 nmol $L^{-1}$ (n = 17). Reproducibility was assessed through the standard deviation of replicate samples (every 10th sample was a replicate) and the average of the in-house standard seawater, and was equal to 17% (n

= 84). Accuracy was determined from the analysis of consensus (SAFe S, GSP) and certified (NASS-7) seawater matrices (see Table 2).

Page 4 Line 21: The CTD sensors were deployed on a stainless steel rosette. Correct? Please indicate and correct throw-out the rest of the text.

➔ We have modified this part as suggested

Page 6 Lines 9-13: Potential temperature (θ), Salinity (S), dissolved oxygen ($O_2$) and beam attenuation data were retrieved from the CTD sensors (CTD SBE911 equipped with a SBE-43) that were deployed on a stainless steel rosette. Nutrient and pigment samples were obtained from the stainless steel rosette casts and analysed according to Aminot and Kerouel (2007) and Ras et al. (2008), respectively. We used the data from the stainless steel rosette casts that were deployed immediately before or after our TMR casts. All these data are/will be available on the LEFE/CYBER database (http://www.obs-vlfr.fr/proof/php/geovide/geovide.php).

Page 4 Line 28: Name the parameter Δσt

▪ We have added this precision (Page 6 Line 16): "…where $Z_m$ is defined as an absolute change in **the density of seawater at a given temperature** ($\Delta\sigma_\theta \geq 0.125$ kg m$^{-3}$)…"

Page 4 Line 30: What do you mean with perturbation, at which depth, please indicate in Table 1, for which station this was the case.

➔ We have changed the word "perturbation" by "disturbance" for clarification. In addition, we have reported in Table 1 the precision on whether temperature and salinity profiles were uniform or disturbed with an asterisk symbol next to stations where profiles were not uniform and we added the following sentence in the legend of Table 1: "Note that the asterisk next to station numbers refers to disturbed temperature and salinity profiles as opposed to uniform profiles."

Page 5 Line 2ff: Please indicate for which data you applied statists on?

➔ We have modified the text as suggested.

Page 6 Lines (21-22): All statistical approaches, namely the comparison between the pore size used for filtration, correlations and Principal Component Analysis (PCA), were performed using the R statistical software (R development Core Team 2012).

Page 5 Line 3: You did not measure the p-value, you maybe determined or calculated the value.

➔ We have modified this part as suggested (Page 6 Line 23)

Page 5 Line 15: Include ". . .540 data points. . .

➔ We have modified this part as suggested (Page 7 Lines 5-6)

Page 5 Line 19: Exchange "The complete relational database. . ." by " The complete data set. . ."

➔ We have modified this part as suggested (Page 7 Line 9)

**Results**

Page 5 Line 27: I would swap the two sentences "For a schematic of water masses, currents and pathways, see Daniault et al. (2016)." and "Hereafter we summarise the main features (Fig. 1 and 2)."

➔ We have modified this part as suggested (Page 7 Lines 14-17)

Page 6 Line 1: Give a depth range of the "Upper waters (0 – 800 m)" or so! Please also include this to the Intermediate and Deep waters.

➔ We have added precisions as suggested

Page 6 Line 5: Did you mean with central water the Subarctic intermediate water (SAIW). Please clarify! Please also increase the letter size in Fig. 2. It is really hard to see on a normal A4 print out! There are no currents in Fig. 2, either you somehow include them or remove the caption.

➔ By central waters, we meant ENACW as defined in the first sentence of the paragraph and therefore changed "Cnetral Waters" by "ENACW" . We have removed the currents from the figure caption in Fig. 2 and we have increased the font size.

Page 6 Line 18: Please rewrite "..Labrador Sea Water (LSW).

➔ We have modified this part as suggested (Page 8 Line 5)

Page 6 Line 29ff: I do not understand the sentence, starting with "During GA01,. . ."

➔ We have rewritten this part

Page 8 Lines 15-16: During GA01, LSW formed by deep convection the previous winter was found at several stations from the Labrador Sea (68, 69, 71 and 77).

Page 6 Line 30ff, I am not sure about, explaining the different flow paths, It is really hard to follow without any drawing. Other question, is it really important, since you are just interested in water masses and their DFe signal, and not about currents! I would remove that!

➔ We have modified this part to make it shorter

Page 8 Lines 15-18: After convecting, the LSW splits into three main branches with two main cores separated by the Reykjanes Ridge (stations 1-32, West European and Iceland Basins; stations 40-60, Irminger Sea), and the last one entering the West European Basin (Daniault et al., 2016).

Page 7 Line 8: I do not see any silicic acid and nitrate data, please indicate concentrations and where they can be found.

➔ We have added the averages and SD for silicic acid and nitrate concentrations (Page 8, Line 22) and the reference where data can be found.

Page 8 Lines 20-23: North East Atlantic Deep Water (NEADW, $1.98 < \theta < 2.50°C$, $34.895 < S < 34.940$) was the dominant water mass in the West European Basin at stations 1-29 from 2000 m depth to the bottom and is characterized by high silicic acid ($42 \pm 4$ $\mu mol\ L^{-1}$), nitrate ($21.9 \pm 1.5$ $\mu mol\ L^{-1}$) concentrations and lower oxygen concentration ($O_2 \approx 252$ $\mu mol\ kg^{-1}$) (see Sarthou et al., subm.).

Page 7 Line 9ff: It is hard to understand what you mean with "PIW is in contact with the atmosphere once a year (?) during the time of winter convection.." All together there is a lot of water mass

information, that can be found elsewhere in the special issue, I would rather shorten that part of the result section.

➔ We have modified this part to make it shorter

Page 8 Lines 24-28: Polar Intermediate Water (PIW, θ ≈ 0°C, S ≈ 34.65) is a ventilated, dense, low-salinity water intrusion to the deep overflows within the Irminger and Labrador Seas that is formed at the Greenland shelf. PIW represents only a small contribution to the whole water mass pool (up to 27%) and was observed over the Greenland slope at stations 53 and 61 as well as in surface waters from station 63 (from 0 to ~ 200 m depth), in intermediate waters of stations 49, 60 and 63 (from ~ 500 to ~ 1500 m depth) and in bottom waters of stations 44, 68, 69, 71 and 77 with a contribution higher than 10%.

Page 7 Line 30: Cannot check if this is correct! No nitrate data available.

➔ The location of these data was already precised "Sarthou et al. (this issue)" . However, we have changed this reference by the accurate one and added the reference of the SEANOE data base and associated paper: García-Ibáñez et al., 2018; Pérez et al., 2018; Sarthou et al., subm. Please note that in this manuscript, Nitrate data are changed for RFe/N data, therefore we did not added the nitrate data.

Page 8 Line 11-12: This is school book knowledge, that is why we are using sensors! Remove the two sentences! However this entire section 3.2.2 needs an overhaul.

➔ We have removed this part as suggested.

Page 9 Lines 20-24: Overall, most of the phytoplankton biomass was localised above 100 m depth with lower total chlorophyll-$a$ (TChl-$a$) concentrations South of the Subarctic Front and higher at higher latitudes (Fig. 3). While comparing TChl-$a$ maxima considering all stations, the lowest value (0.35 mg m$^{-3}$) was measured within the West European Basin (station 19, 50 m depth) while the highest values were measured at the Greenland (up to 4.9 mg m$^{-3}$, 30 m depth, station 53 and up to 6.6 mg m$^{-3}$, 23 m depth, station 61) and Newfoundland (up to 9.6 mg m$^{-3}$, 30 m depth, station 78) margins.

Page 8 Line 20ff: You can delete the first three sentences, they do not contain any important data!

➔ We have removed this part as suggested

Page 8 Line 29: Also station 61 and 78 are high, at least this is shown by your plot!

➔ We were talking about enhanced DFe at the surface compared to deeper DFe values, and this is only the case for stations 2, 4 and 56.

And replace ". . .were around. . ." by ". . .ranged from. . ."

➔ We have modified the text as suggested.

Generally, I would merge section 3.3, 3.3.1 and 3.3.2.

➔ We have modified the text as suggested.

Figure 4: Are you sure that single elevated values at site 40 (1500m) and at site 44 (500m) are correct. They just seem like outliers to me! Do we really need Fig.5 and 6, we see everything already in Fig. 4.

➔ These enhanced DFe concentrations are consistent with high ligand concentrations measured during this study. In addition, these samples were analysed during 3 separated analytical sessions on the seaFAST SF-ICP-MS. Regarding Figs. 5 and 6, we agree that Fig. 6 is repetitive and therefore we removed it from the ms and included it into the supplementary material. However, we think Fig. 5 is helpful to understand the section 4.2.2 on high latitudes meteoric water and sea-ice processes.

Page 9 Line 1ff: rewrite sentence, hard to read!

➔ We have rewritten the sentence

Page 10 Lines 1-3: Considering the four oceanic basins, mean vertical profiles (supplementary material SX) showed increasing DFe concentrations down to 3000 m depth followed by decreasing DFe concentrations down to the bottom. Among deep-water masses, the lowest DFe concentrations were measured in the West European Basin.

Page 9 Line 6ff: Please provide numbers for surface waters.

➔ We have added this precision.

Page 10 Lines 6-7: Overall, surface DFe concentrations were higher ($0.36 \pm 0.18$ nmol L$^{-1}$) in the North Atlantic Subpolar gyre (above 52°N) than in the North Atlantic Subtropical gyre ($0.17 \pm 0.05$ nmol L$^{-1}$).

Page Line 9ff: But also at station 21 the DFe value is high. I do not think they are significantelly different from the others, s.d. is $\pm$ 20% and higher.

➔ We agree with you and removed this sentence.

Page 9 Line 17: NEADW was very similar to the median GEOVIDE voyage but compared to test of deep waters lower, please rewrite! But the DSOW in the Labrador Sea was similar.

➔ We have modified the text accordingly

Page 10 Lines 30-32: The DFe concentrations in the NEADW were relatively similar to the DFe median value of the GEOVIDE voyage (median DFe = 0.75 nmol L$^{-1}$, Figs. 4 and 7) with an average value of $0.74 \pm 0.16$ nmol L$^{-1}$ (n=18) and presented relatively low median DFe concentrations (median DFe = 0.71 nmol L$^{-1}$) compared to other deep water masses.

I am not sure Fig. 7 is really required. It just comprises what we already sea in Fig. 4. And apart from some outliers (hydrothermal? Any Mn data), surface waters, NADW and waters from the Labrador Sea, concentrations are around 1nM. And as numerous times shown, it is impossible to fingerprint water masses with DFe.

➔ We agree with this suggestion and Figure 7 was removed from the MS and added to supplementary material.

Page 10 Line 9: Others showed also elevated concentrations, for instance, station 44. However I understand why the authors decided to explain both station! For myself station 1 is not a problem, it is very close to the continental margin and influenced by lateral water mass transport than the other stations farther off-shore. However, site 17 is a bit more tricky. Did you reanalyze that station, that would confirm that the analysis was alright and you do not face just a strange offset. Anyway, I would discuss station 17, but please rephrase some sentences, it was really hard to grasp the issue

you wanted to bring across. From the first sentence it should be clear what the issue is, than explain (eg. Concentrations are irregularly high).

➔ We have modified the text as suggested

Page 11 Lines 9-17: Considering the entire section, two stations (stations 1 and 17) showed irregularly high DFe concentrations (> 1 nmol $L^{-1}$) throughout the water column, thus suggesting analytical issues. However, these two stations were analysed twice and provided similar results, therefore discarding any analytical issues. This means that these high values originated either from genuine processes or from contamination issues. If there had been contamination issues, one would expect a more random distribution of DFe concentrations and less consistence throughout the water column. It thus appears that contamination issues were unlikely to happen. Similarly, the influence of water masses to explain these distributions was discarded as the observed high homogenized DFe concentrations were restricted to these two stations. Station 1, located at the continental shelf-break of the Iberian Margin, also showed enhanced PFe concentrations from lithogenic origin suggesting a margin source (Gourain et al., subm.).

Page 10 Line 23ff: Please provide the numbers from the other studies. Would it be possible to plot the surface DFe concentration and put the graph in the sup material. Than you can relate to that!

➔ We have provided the numbers from other studies, updated Table 3 with the DFe values from Achterberg et al., 2018 and we added the following plot to the supplementary material:

[Figure]

Figure SX: Surface layer of DFe concentrations, new measurements are shown in red dots (GEOVIDE voyage), while previous studies are displayed in black . (Achterberg et al., 2018; Bergquist et al., 2007; Blain et al., 2004; Boye et al., 2006, 2003; de Jong et al., 2007; Gledhill et al., 1998; Hatta et al., 2015; Klunder et al., 2012; Laës et al., 2003; Martin et al., 1993; Measures et al., 2008; Mills et al., 2008; Mohamed et al., 2011; Nédélec et al., 2007; Nielsdóttir et al., 2009; Pohl et al., 2011; Rijkenberg et al., 2014; Sarthou et al., 2007, 2003; Sedwick et al., 2005; Ussher et al., 2013; Witter and Luther III, 1998; Wu and Boyle, 2002; Wu and Luther III, 1996, 1994; Wu et al., 2001).

➔ We also changed the text as we made a mistake in this section. Indeed, low DFe concentrations were previously measured in the central Irminger Sea. When we first wrote this part we considered stations that were closed to land likely impacted by sea-ice melting.

Page 11 Lines 23-37: Among the four distinct basins described in this paper, the Irminger Sea exhibited the highest DFe concentrations within the surface waters (from 0 to 250 m depth) with values ranging from 0.23 to 1.3 nmol L$^{-1}$ for open-ocean stations. Conversely, low DFe concentrations were previously reported in the central Irminger Sea by Rijkenberg et al. (2014) (April-May, 2010) and Achterberg et al. (2018) (April-May and July-August, 2010) with DFe concentrations ranging from 0.11 to 0.15 and from ~ 0 to 0.14 nmol L$^{-1}$, respectively (see supplementary material SX and Table 3).

Page 10 Line 29: Please include an opening sentence, what you think is the reason (something similar to the last sentence). It is quite a step from Fe distribution to the original of water mass mixing.

➔ We have added an opening sentence

Page 11 Lines 29-31: Indeed, enhanced surface DFe concentrations measured during GEOVIDE in the Irminger Sea could be due to intense wind forcing events that would deepen the winter $Z_m$ down to the core of the Fe-rich LSW.

Page 11 Line 5: Explain what tip jets are!

➔ We have added a definition

Page 12 Lines 8-10: Moore (2003) and Piron et al. (2016) described low-level westerly jets centred northeast of Cape Farewell, over the Irminger Sea, known as tip jet events, whose structure depends upon the splitting occurring as the flow encounter the orographic features from Cape Farewell.

Page 11 Line 10ff: This process is called winter entrainment (Tagliabue et al. 2014). Rephrase sentence and delete the last one (You just repeat yourself.

➔ We have modified the text as suggested

Page 12 Lines 15-17: Such winter entrainment was likely the process involved in the vertical supply of DFe within surface waters fuelling the spring phytoplankton bloom with DFe values close to those found in LSW.

Page 11 Line 16: Also contaminated waters are introduced!

➔ Yes, we completely agree. However, since this section is dedicated to atmospheric deposition we did not specify this.

Page 11 Line 18: What is a stratification period? Be preciss!

➔ We have changed the text

Page 12 Line 22: "During the summer, when thermal stratification occurs, …"

Page 11 Line 218ff: You can not compare the Mediterranean surface waters with MOW. Rewrite! DAl and DFe behave entirely different in the water column (residence time, organic complexation, concentrations , etc.), but both of them are likely to be scavenged from particles. So when a dust storm hits, both elements should decrease, do they actualy do this in the water column of the

Mediterranean sea. However, I am not too much surprised to see no DFe signal in the MOW. However, I suggest you have a look for DFe literature values from deep Mediterranean waters (GA04 is not available, a pitty).

→ Yes, we agree that DAl and DFe behave differently in the water column depending on organic complexation and that they are both likely to be scavenged from particles. However, DAl and DFe originating from dust deposition are not scavenged by the same type of particles. Indeed, Wuttig et al. (2013) reported that after a single dust deposition event DAl loss rates was highly affected by the concentrations of biogenic particles while DFe was removed by sinking dust particles. The same authors highlighted that the following dust deposition event were likely inducing the dissolution of Fe from dust particles depending on the amounts of Fe-binding organic ligands. Therefore, both elements should not necessarily decrease.

→ We have changed the text as suggested

Page 12 Lines 24-33 and Page 13 Lines 1-3: After atmospheric deposition, the fate of Fe will depend on the nature of aerosols, vertical mixing, biological uptake and scavenging processes (Bonnet and Guieu, 2006; Wuttig et al., 2013). During GEOVIDE, MOW was observed from stations 1 to 29 between 1000 and 1200 m depth and associated with high dissolved aluminium (DAl, Menzel Barraqueta et al., 2018) concentrations (up to 38.7 nmol L$^{-1}$), confirming the high atmospheric deposition in the Mediterranean region. In contrast to Al, no DFe signature was associated with MOW (Figs. 2 and 4). This feature was also reported in some studies (Hatta et al., 2015; Thuróczy et al., 2010), while others measured higher DFe concentrations in MOW (Gerringa et al., 2017; Sarthou et al., 2007). However, MOW coincides with the maximum Apparent Oxygen Utilization (AOU) and it is not possible to distinguish the MOW signal from the remineralisation one (Sarthou et al., 2007). On the other hand, differences between studies are likely originating from the intensity of atmospheric deposition and the nature of aerosols. Indeed, Wagener et al. (2010) highlighted that large dust deposition events can accelerate the export of Fe from the water column through scavenging. As a result, in seawater with high DFe concentrations and where high dust deposition occurs, a strong individual dust deposition event could act as a sink for DFe. It thus becomes less evident to observe a systematic high DFe signature in MOW despite dust inputs.

Page 12 Line 1: The entire section 4.1.3 is highly speculative. I agree elevated DFe in the Irminger Basin needs to come from somewhere, however, just looking at your Chl a data it is a very productive site, so presumably PFe concentrations are elevated as well, if so you should mention that, than it is just elevated remineralization and intense deep mixing during winter time that is responsible. However, you need to rewrite that section, to make it less speculative, look for existing data!

→ We have changed the text as suggested.

Page 13 Lines 5-32 and Page 14 Lines 1-8: As described in Section 3.1, the LSW exhibited increasing DFe concentrations from its source area, the Labrador Sea, toward the other basins with the highest DFe concentrations observed within the Irminger Sea, suggesting that the water mass was enriched in DFe either locally in each basin or during its flow path (Fig. 7). These DFe sources could originate from a combination of high export of PFe and its remineralisation in the mesopelagic area and/or the dissolution of sediment.

The Irminger and Labrador Seas exhibited the highest averaged integrated TChl-a concentrations (98 ± 32 mg m$^{-2}$ and 59 ± 42 mg m$^{-2}$) compared to the West European and Iceland Basins (39 ± 10 mg m$^{-2}$

and 53 ± 16 mg m$^{-2}$), when the influence of margins was discarded. Stations located in the Irminger (stations 40-56) and Labrador (stations 63-77) Seas, were largely dominated by diatoms (>50% of phytoplankton abundances) and displayed the highest chlorophillid-*a* concentrations, a tracer of senescent diatom cells, likely reflecting post-bloom condition (Tonnard et al., in prep.). This is in line with the highest POC export data reported by Lemaitre et al. (subm.) in these two oceanic basins. This likely suggests that biogenic PFe export was also higher in the Labrador and Irminger Seas than in the West European and Iceland Basins. Although, Gourain et al. (subm.) highlighted a higher biogenic contribution for particles located in the Irminger and Labrador Seas with relatively high PFe:PAl ratios (0.44 ± 0.12 mol:mol and 0.38 ± 0.10 mol:mol, respectively) compared to particles from the West European and Iceland Basins (0.22 ± 0.10 and 0.38 ± 0.14 mol:mol, respectively, see Fig. 6 in Gourain et al., subm.), they reported no difference in PFe concentrations between the four oceanic basins (see Fig. 12A in Gourain et al., subm.) when the influence of margins was discarded, which likely highlighted the remineralisation of PFe within the Irminger and Labrador Seas. Indeed, Lemaître et al. (2017) reported higher remineralisation rates within the Labrador (up to 13 mmol C m$^{-2}$ d$^{-1}$) and Irminger Seas (up to 10 mmol C m$^{-2}$ d$^{-1}$) using the excess barium proxy (Dehairs et al., 1997), compared to the West European and Iceland Basins (ranging from 4 to 6 mmol C m$^{-2}$ d$^{-1}$). Therefore, the intense remineralisation rates measured in the Irminger and Labrador Seas likely resulted in enhanced DFe concentrations within LSW.

Higher DFe concentrations were, however, measured in the Irminger Sea compared to the Labrador Sea and coincided with lower transmissometer values (i.e. 98.0-98.5% vs. >99%), thus suggesting a particle load of the LSW. This could be explained by the reductive dissolution of Newfoundland Margin sediments. Indeed, Lambelet et al. (2016) reported high dissolved neodymium (Nd) concentrations (up to 18.5 pmol.kg$^{-1}$) within the LSW at the edge of the Newfoundland Margin (45.73°W, 51.82°N) as well as slightly lower Nd isotopic ratio values relative to those observed in the Irminger Sea. They suggested that this water mass had been in contact with sediments approximately within the last 30 years (Charette et al., 2015). Similarly, during GA03, Hatta et al. (2015) attributed the high DFe concentrations in the LSW to continental margin sediments. Consequently, it is also possible that the elevated DFe concentrations from the three LSW branches which entered the West European and Iceland Basins and Irminger Sea was supplied through sediment dissolution (Measures et al., 2013) along the LSW pathway.

The enhanced DFe concentrations measured in the Irminger Sea and within the LSW were thus likely attributed to the combination of higher productivity, POC export and remineralisation as well as a DFe supply from reductive dissolution of Newfoundland sediments to the LSW along its flow path. As reported by Boyd and Ellwood (2010), when bacteria dissolve PFe they also release Fe-binding organic ligands especially from biogenic PFe than from lithogenic PFe (Boyd et al., 2010). It is thus possible that bacteria-mediated ligand production helped the DFe supply from Newfoundland sediments to remain in solution within the Irminger Sea LSW.

Page 12 Line 25: the elevated concentration on station 44, is not this just a single point?

➔ No, the elevated DFe concentrations at station 44 concerned three data points.

Page 12 Line 26ff: Replace "above" by "at", and what are i) sediment inputs (these are particles), and ii) intrusion of an Fe-rich water mass, please be more specific!

➔ We have changed the sentence as suggested and added precision.

Page 14 Lines 11-12: "… i) vertical diffusion from local sediment, ii) lateral advection of a water mass displaying enhanced DFe concentrations, and iii) local dissolution of Fe from particles."

Page 12 Line 33: How often have you analyzed the samples below 2.500 m at site 44. For me this is just one outlier, the two other samples from cast 44 in Fig 8A are not that out of the range.

➔ The full station 44 was analysed during two separated analytical sessions on the seaFAST SF-ICP-MS from different sampling bottles with a good agreement between results. Therefore, we fo not think that this data point is an outlier.

Page 13 Line 10ff: Your argument is based on four data points, I could aslo put a straight line through, with a similar R2. However this entire paragraph is highly speculative! In an earlier paragraph you mention that DFe do not fingerprint different water masses, and now they do? You should remove this section!

➔ We agree on the fact that the polynomial fitting could also be a linear fitting. However, with either a polynomial or a linear fitting on the 5 data points, the conclusion would be the same with apparently, the dissolution of Fe-rich particles.
➔ We reformulated this section as we considered it too speculative.

[revised manuscript text omitted]

➔ Note that for Pb, no particular hydrothermal signal was observed during GEOVIDE (Zurbrick et al., 2018). For Mn, data are analysed but not yet processed.

Page 14 Line 3ff: Theer are no elevated DFe values farther east from the ridge!

➔ We have changed the text for clarification (see above). The DFe enrichment east of Reykjanes Ridge corresponded to the section on top of this sentence while further downstream corresponded to west of the Reykjanes Ridge.

Where is the CGFC and BFC. Questions over questions!

➔ These two features are now added to Fig. 1

[Figure]

**Figure 1: Map of the GEOTRACES GA01 voyage plotted on bathymetry as well as the major topographical features and main basins. Crossover station with GEOTRACES voyage (GA03) is shown as a red star. (Ocean Data View (ODV) software, version 4.7.6, R. Schlitzer, http://odv.awi.de, 2016). BFZ: Bight Fracture Zone, CGFZ: Charlie-Gibbs Fracture Zone.**

Page 14 Line 13ff: I am confused. Do we talk about station 40 and 1.75nM at 1500 m, this is a single high value for me, and not located in ISOW waters.

➔ Yes, we talk about station 40 (one point) and station 42 (three points) that are all located in the ISOW (see Fig. 7).

Page 14 Line 26ff: The DFe/DAl ratio in seawater can not compared with the Fe/Al ratio of dust particles. Both elements have different fractional solubility's. So the ratio is always different! Remove!

➔ We agree with the reviewer and have changed the text as suggested and have added some information

Page 16 lines 15-17: Our SML DFe inventories were higher at station 1 than those calculated during the GA03 voyage (~ 1nmol L$^{-1}$, station 1) during which atmospheric deposition were about one order

of magnitude higher (Shelley et al., 2017a; Shelley et al., 2015), suggesting that the atmospheric source was not significant.

Page 15 Line 1: Remove most of them does not add to the story!

➔ We agree and have removed most of them.

Page 16 Lines 22-24: Many types of industry (e.g. heavy metallurgy, ore processing, chemical industry) release metals including Fe, which therefore result in high levels recorded in surface sediments, suspended particulate matter, water and organisms in the lower estuary (Santos-Echeandia et al., 2010).

Page 15 Line 5: What do you mean with "..below ground biomass.." In general I do not understand, why you excluded sediments, that could be an additional source.

➔ We did not intend to exclude the sediment source and have change the text for clarification.

Page 16 lines 18-19: Consequently, the Tagus River appears to be the most likely source responsible for these enhanced DFe concentrations, either as direct input of DFe or indirectly through Fe-rich sediment carried by the Tagus River and their subsequent dissolution.

Page 15 Line 14ff: Fronts refer to temperature and salinity changes in surface waters, such as the Polar Front, not in the water column. Call it different; just use the term "fresh water lens". Why multi-year-sea ice?

➔ We have changed the text as suggested

Page 16 Line 31 Page 17 Line 1: The presence of this freshwater lens suggests that sediment derived enrichment to these surface waters was unlikely.

➔ We talked about multiyear sea ice because of drainage processes and the release of brines (see below)

Page 15 Line 18ff: But glacial sources and land ice sheet is the same, just call, it " . . .freshwater induced by meteoric water and sea-ice melt." Than all is clear.

➔ We have made the correction as suggested (Page Line)

Page 15 Line 27: Where do get the sea-ice fractions from, and explain how it works, include references! And what have brines to do with it, either ice forms or not! Brines are not part of your story, so far I can tell. Brines always from when sea-ice is formed, or in the desert by evaporation. And in line 31 you switch back to sea-ice formation, please stay with that term.

➔ We have included a section in the method on how these fractions were calculated.

Page 5 Lines 22-32, Page 6 Lines 1-7: We separated the mass contributions to samples from stations 53, 61 and 78 in Sea-Ice Melt (SIM) Meteoric Water (MW) and saline seawater inputs using the procedure and mass balance calculations that are fully described in Benetti et al. (2016) (Fig. 5D), E) and F)). Hereafter, we describe briefly the principle. We considered two types of seawater, namely the Atlantic Water (AW) and the Pacific Water (PW). After estimating the relative proportions of AW ($f_{AW}$) and PW ($f_{PW}$) and their respective salinity and $\delta^{18}$O affecting each samples, the contribution of SIM and MW can be determined using measured salinity ($S_m$) and $\delta^{18}$O ($\delta O_m^{18}$). The mass balance calculations are presented below:

$$f_{AW} + f_{PW} + f_{MW} + f_{SIM} = 1 \text{ (eq.1)}$$

$$f_{AW} \cdot S_{AW} + f_{PW} \cdot S_{PW} + f_{MW} \cdot S_{MW} + f_{SIM} \cdot S_{SIM} = S_m \text{ (eq.2)}$$

$$f_{AW} \cdot \delta O_{AW}^{18} + f_{PW} \cdot \delta O_{PW}^{18} + f_{MW} \cdot \delta O_{MW}^{18} + f_{SIM} \cdot \delta O_{SIM}^{18} = \delta O_m^{18} \text{ (eq.3)}$$

where $f_{AW}$, $f_{PW}$, $f_{MW}$, $f_{SIM}$ are the relative fraction of AW, PW, MW, and SIM. To calculate the relative fractions of AW, PW, MW and SIM we used the following end-members: $S_{AW}$ = 35, $\delta O_{AW}^{18}$ = +0.18‰ (Benetti et al., 2016); $S_{PW}$ = 32.5, $\delta O_{PW}^{18}$ = -1‰ (Cooper et al., 1997; Woodgate and Aagaard, 2005); $S_{MW}$ = 0, $\delta O_{MW}^{18}$ = -18.4‰ (Cooper et al., 2008); $S_{SIM}$ = 4, $\delta O_{SIM}^{18}$ = +0.5‰ (Melling and Moore, 1995).

In Figure 5 D), E) and F), negative sea-ice fractions indicated a net brine release while positive sea-ice fractions indicated a net sea-ice melting. Note that for stations over the Greenland Shelf, we assumed that the Pacific Water (PW) contribution was negligible for the calculations, supported by the very low PW fractions found at Cape Farewell in May 2014 (see Figure B1 in Benetti et al., 2017), while for station 78, located on the Newfoundland shelf, we used nutrient measurements to calculate the PW fractions, following the approach from Jones et al. (1998) (the data are published in Benetti et al., 2017).

→ Regarding brines, they can originate from two different processes: either as a result of multiyear sea-ice melting or during sea-ice formation. Indeed, during the early melting season, multiyear sea-ice has a higher porosity and gravitational drainage of brine occur. These two processes of brine release might lead to different TM signatures in brine originating from sea-ice formation and brine originating from early melting of multiyear sea-ice (Petrich and Eicken, 2010; Wadhams, 2000).

Page 15 Line 33: But brines usually sink, because they are heavier than the surrounding water!!! It is really hard to follow your argumentation here.

We agree with the reviewer in the fact that brines sink due to higher density. However, after reaching neutral buoyancy, they will stop sinking.

Page 16 Line 11: You have to explain how you produced these numbers, a citation in an earlier paragraph is not enough!

→ We have included a section in the method on how these fractions were calculated (see above)

Page 16 Line 15: How do you lose a sample! Generally fist you talk about the contribution of MW and then you switch to biological uptake of DFe, that in the same paragraph? You lose the reader here; this entire section needs an overhaul.

→ We have reorganised this section.

Page 17 Lines 26-34, Page 18 Lines 1-10: Surface waters (from 0 to ~ 100 m depth) from station 53 and 61 were characterized by high MW fractions (ranging from 8.3 to 7.4% and from 7.7 to 7.3%, respectively, from surface to ~100 m depth, Figs. 5D and E). Within these surface waters, station 53 exhibited substantial sea-ice melting contribution (1.5%, 4 m depth, Fig. 5D), while station 61 exhibited low contribution (0.6%, Fig. 5E) from brine release that was linearly increasing with depth (1.3% at 50 m depth and 2.2 % at 100 m depth, Fig. 5E). These high MW fractions were both enriched in PFe and DFe (except station 53 for which no data was available) compared to seawater located below 50 m depth, thus suggesting a MW source. These results are in line with previous observations, which highlighted strong inputs of DFe from a meteoric water melting source in Antarctica (Annett et al., 2015). Although the ability of MW from Greenland Ice Sheet and runoffs to deliver DFe and PFe to surrounding waters have previously been demonstrated (Bhatia et al., 2013; Hawkings et al., 2014; Schroth et al., 2014; Statham et al., 2008), both Fe fractions were lower at the

sample closest to the surface, then reached a maximum at ~ 50 m depth and decreased at ~ 70 m depth, for station 61 (Fig. 5D). The surface DFe depletion was likely explained by phytoplankton uptake, as indicated by the high TChl-*a* concentrations (up to 6.6 mg m$^{-3}$) measured from surface to about 40 m depth, drastically decreasing at ~ 50 m depth to 3.9 mg m$^{-3}$ (Fig. 5D). Hence, it seemed that meteoric water inputs from the Greenland Margin likely fertilized surface waters with DFe, enabling the phytoplankton bloom to subsist. The profile of PFe can be explained by two opposite plausible hypotheses: 1) MW inputs did not released PFe, as if it was the case, one should expect higher PFe concentrations at the surface (~25 m depth) than the one measured at 50 m depth due to both the release from MW and the assimilation of DFe by phytoplankton 2) MW inputs can release PFe in a form that is directly accessible to phytoplankton with subsequent export of PFe as phytoplankton died. The latter solution explains the PFe maximum measured at ~ 50 m depth and is thus the most plausible.

Page 16 Line 32: ".. decreasing from surface to depth." Which depth, down to the bottom in 400 m depth? Be precise

➔ We have added this precision.

Page 18 Lines 13-14: Newfoundland shelf waters (station 78) were characterized by high MW fractions (up to 7%), decreasing from surface to 200 m depth (~2%).

Page 17 Line 15-25: What has the tropical and subtropical North Atlantic to do with your work! I assume very little, please delete or at least reduce the text.

➔ We agree with the reviewer and removed the part on tropical North Atlantic.

Page 19 Lines 2-7: On a regional scale, the North Atlantic basin receives the largest amount of atmospheric inputs due to its proximity to the Saharan Desert (Jickells et al., 2005), yet even in this region of high atmospheric deposition, inputs are not evenly distributed. Indeed, aerosol Fe loading measured during GEOVIDE (Shelley et al., 2017b) were much lower (up to four orders of magnitude) than those measured during studies from lower latitudes in the North Atlantic (e.g. Baker et al., 2013; Buck et al., 2010; and for GA03, Shelley et al., 2015), but atmospheric inputs could still be an important source of Fe in areas far from land.

Page 17 Line 30: I would rather suggest to say: "Shelley et al. concluded that. . ." because without any trajectories here I can check, and more or less all this work was already published.

➔ We have changed the text as suggested.

Page 19 Lines 11-14: Using this approach, Shelley et al. (2017b) concluded that the GEOVIDE transect could be split into four areas predominantly influenced by: (1) atmospheric inputs from sources in Iceland and Greenland which likely include proglacial till (stations 11-29), (2) an anthropogenic source coming from UK/Ireland (stations 32-40), (3) a remote marine source influenced by sea salt aerosols and shipping emissions (stations 42-69) and (4) a North American source (stations 71 and 77).

Page 18 Line 13: Do you mean DOM? Or organic material OM. However, you talk about DOM for 7 lines, and then you don't have the data. Once sentence should be enough to point out the importance of DOM.

➔ We have changed the text for clarification and as suggested.

Page 19 Lines 20-23: Once in the water, many studies (Bressac and Guieu, 2013; Bressac et al., 2014; Desboeufs et al., 2014; Meskhidze et al., 2017) argued that the aerosol trace metal fractional solubility is driven by the amount of organic matter in seawater and aerosols. However, as the organic composition of aerosols was not determined during our study we cannot comment further here.

Page 18 Line 16: This entire paragraph is very poor! It is interesting to compare elemental ratios of seawater with the soluble fraction of dust. But the reasoning here ". . .whether there was enough atmospheric input to sustain the SML DFe concentrations. . ." without any flux numbers, residence times is unscientific. Even more strange, at the end of the paragraph you don t even say, whether there is enough or not. Similar to the above, this needs serious work to make it worthwhile reading. There is too much hand waving, and too few data, sorry! I suggest you look up the actual flux numbers and then compare them with your data.

➔ We agree with the reviewer and removed this section to replace it by Turnover Times relative to Atmospheric Deposition (TTADs) as defined in Guieu et al. (2014).

Page 19 Lines 24-33 Page 20 Lines 1-5: With this in mind, it seems that DFe concentrations are more or less comparable to soluble Fe inventories from aerosols, rather than total Fe in aerosols. In an attempt to estimate whether there was enough atmospheric input to sustain the SML DFe concentrations, we calculated Turnover Times relative to Atmospheric Deposition (TTADs, Guieu et al., 2014). To do so, we made the following assumptions: 1) the aerosol concentrations are a snapshot in time but are representative of the study region, 2) the aerosol solubility estimates based on two sequential leaches are an upper limit of the aerosol Fe in seawater and 3) the water column stratified just before the deposition of atmospheric inputs, so MLD DFe will reflect inputs from above. Thus, the TTADs were defined as the integrated DFe concentrations in the SML for each station divided by the contribution of soluble Fe contained in aerosols averaged per basin to the water volume of the SML. Although, TTADs were lower in the West European and Iceland Basins with an average of ~ 9 ± 3 months compared to other basins (7 ± 2 years and 5 ± 2 years for the Irminger and Labrador Seas, respectively) (Fig. 10) they were about three times higher than those reported for areas impacted by Saharan dust inputs (~ 3 months, Guieu et al., 2014). Therefore, the high TTADs measured in the Irminger and Labrador Seas and ranging from 2 to 15 years provided further evidence that atmospheric deposition were unlikely to supply Fe in sufficient quantity to be the main source of DFe (see Sections 4.2.1 and 4.3.2) while in the West European and Iceland Basins they played an additional source, perhaps the main source of Fe especially at station 36 which displayed TTAD of 3 months.

[Figure]

Figure 10: Plot of dissolved Fe (DFe) Turnover Times relative to Atmospheric Deposition (TTADs) calculated from soluble Fe contained in aerosols estimated from a two-stage sequential leach (UHP water, then 25% HAc, Shelley et al., this issue). Note that numbers on top of data points represent station numbers and that the colour coding refers to different region with in yellow, margin stations; in purple, the West European Basin; in blue, the Iceland Basin; in green, the Irminger Sea and in red, the Labrador Sea. The numbers on top of the plot represent TTADs averaged for each oceanic basin and their standard deviation.

Page 19 Line 5ff: replace "on" by "in". And which similar pattern followed the station. Be precise! Sentence stating in Line 6 makes no sense, please rewrite!

➔ We have corrected the text and rephrased the next sentence as suggested.

Page 20 Lines 8-11: DFe concentration profiles from all coastal stations (stations 2, 4, 53, 56, 61 and 78) are reported in Figure 5. To avoid surface processes, only depths below 100 m depth will be considered in the following discussion. DFe and PFe followed a similar pattern at stations 2, 53, 56, and 78 with increasing concentrations towards the sediment, suggesting that either the sources of Fe supplied both Fe fractions (dissolved and particulate) or that PFe dissolution from sediments supplied DFe.

Page 19 Line 11: What has the composition of sediments to do with your PFe value? Nothing. . .

➔ We have changed the text for clarification.

Page 20 Lines 13-16: DFe:PFe ratios ranged from 0.01 (station 2, bottom sample) to 0.27 (station 4, ~ 400 m depth) mol:mol with an average value of 0.11 $\pm$ 0.07 mol:mol (n = 23, Table 4), highlighting a different behaviour of Fe between margins. This could be explained by the different nature of the sediments and/or different sediment conditions (e.g. redox, organic content).

Page 19 Line 15: "Intermediate behavior" of what? And then Chla? This paragraph is very hard to follow, what is the message you want to bring across, I can't tell!

➔ We removed this sentence for clarification.

Page 20 Lines 16-21: Based on particulate and dissolved Fe and dissolved Al data (Gourain et al., subm.; Menzel Barraqueta et al., 2018, Table 4), three main different types of margins were reported (Gourain et al., subm.) with the highest lithogenic contribution observed at the Iberian Margin (stations 2 and 4) and the highest biogenic contribution at the Newfoundland Margin (station 78). These observations are consistent with higher TChl-*a* concentrations measured at the Newfoundland Margin and to a lesser extent at the Greenland Margin and the predominance of diatoms relative to other functional phytoplankton classes at both margins (Tonnard et al., in prep.).

Page 19 Line 26: "respect"? I respect you as a person, but samples usually don't respect anything?

➔ We have changed the word "respected" by "followed".

Page 19 Line 30ff: How do you know its manganese oxide, just use particulate Mn. And why you do not include the transmissometer data. That is what you wanted to show, or not that resuspended sediments control you particulate fraction.

➔ We agree with the reviewer in the way that we are not sure these are manganese oxides as they were estimated as the fraction from the PMn that was not originating from a lithogenic fraction using Mn:Ti UCC ratio. Therefore, a biological source or a co-precipitation source without oxidation were not considered. We thus agree with the reviewer and we have changed the MnOx data towards PMn data in the PCA.
➔ We did not include the transmissometer data as we do not have true values for all samples and used the interpolated data.

Page 20 Line 3ff: You did not do a PCA for dFe, so how can you be sure that the dim1 controls DFe? I cannot follow.

➔ Before performing the PCA, a huge number of variables were considered and we only kept the one that were correlated to DFe to build the PCA.
➔ We have changed the text for clarification.

[revised manuscript text omitted]

Page 20 Line 27ff: Include "some" in front of "maxima, Please tell me the difference between "the relationship between DFe and biological uptake" and "Did DFe concentrations potentially limit phytoplankton growth?" This sounds to me very connected with each other! Why not discussing that in the follow up paper?

We have corrected the first sentence as suggested and re-wrote the end of the paragraph for clarification. Note that we wanted to keep this discussion in this paper as it summarises the different processes discussed in this MS.

Page 22 Lines 2-5: Overall, almost all the stations from the GA01 voyage displayed DFe minima in surface water associated with some maxima of TChl-*a* (Fig. 3). In the following section, we specifically address the question of whether DFe concentrations potentially limit phytoplankton growth. Note that macronutrients and DFe limitations relative to phytoplankton functional classes are dealt in Tonnard et al. (in prep.).

Page 20 Line 31: Include mean or average Plus standard deviation

➔ We have included the average and SD and corrected few mistakes.

Page 22 Lines 7-8: The DFe:$NO_3^-$ ratios in surface waters varied from 0.02 (station 36) to 38.62 (station 61) mmol:mol with an average of 5 ± 10 mmol:mol (Fig. 12).

Page 21 Line 4: Please include numbers! Following the text in the paragraph, it is very hard to follow, you jump between F:N ratios, water masses and Chl a. Try to keep it short and weed out unnecessary details. Otherwise you will lose the reader!

➔ We have changed the text accordingly and added a surface map of DFe:NO3- ratios.

[Figure]

*DFe:NO3 [mmol:mol] @ depth=first*

[revised manuscript text omitted]

Page 21 Line 20ff: Can you explain to me why you calculate Fe* for the entire water body (Fig. 13) and explain DFe limitation of the phytoplankton community. They live in the first 100-200 m. Same fro Fig.14.

→ We calculated the Fe* for the entire water body as water mass circulation and/or processes such as deep convection/upwelling, … can homogenized deep water masses with surface water masses. Thus, looking at DFe:NO3- ratios in these water masses appeared for us to be as important as just looking at the surface where phytoplankton live.

→ However, we understand the reviewer's opinion and decided to restrict this section to the top 250 meters. Consequently, we did a new plot for Fig. 13 with only the upper water column and removed Fig. 14.

[Figure]

Reading the last sentence of the section on page 22 "However, atmospheric loading (and especially Fe) was higher within the subtropical gyre than elsewhere in the GEOVIDE section mainly due to the proximity to mineral dust source (i.e. the Sahara Desert)." I feel I am still stuck in the Atmospheric chapter. Please shorten the paragraph and just say what you can prove with data.

→ We have removed the last two sentences to shorten the paragraph.

Page 22 Line 14ff: The entire conclusion needs an overhaul!

→ We have modified the conclusion to be more specific.

Page 24 Lines 2-30: The DFe concentrations measured during this study were in good agreement with previous studies that spanned the West European Basin. However, within the Irminger Basin the DFe concentrations measured during this study were up to 3 times higher than the ones measured by Rijkenberg et al. (2014) in deep waters (> 1000 m depth) that was likely explained by

the different water masses encountered (i.e. the Polar Intermediate Water, ~ 2800 m depth) and by a stronger signal of the Iceland Scotland Overflow Water (ISOW) from 1200 to 2300 m depth. This corresponded to the most striking feature of the whole section with DFe concentrations reaching up to 2.5 nmol L$^{-1}$ within the ISOW, Denmark Strait Overflow Water (DSOW) and Labrador Sea Water (LSW), three water masses that are part of the Deep Western Boundary Current and was likely the result of a lateral advection of particles in the Irminger  However, as these water masses reached the Labrador Sea, lower DFe levels were measured. These differences could be explained by different processes occurring within the benthic nepheloid layers, where DFe was sometimes trapped onto particles due to Mn-sediment within the Labrador Sea (Gourain et al., subm.) and sometimes released from the sediment potentially as a result of interactions with dissolved organic matter. Such Fe-binding organic ligands could have also be produced locally due to the intense remineralisation rate reported by Lemaître et al. (2017) of biogenic particles (Boyd et al., 2010; Gourain et al., subm.). The LSW exhibited increasing DFe concentrations along its flow path, likely resulting from sediment inputs at the Newfoundland Margin. Although DFe inputs through hydrothermal activity were expected at the slow spreading Reykjanes Ridge (Baker and German, 2004b; German et al., 1994), our data did not evidence this specific source as previously pointed by Achterberg et al. (2018) further north (~60°N) from our section.

In surface waters several sources of DFe were highlighted especially close to lands, with riverine inputs from the Tagus River at the Iberian margin (Menzel Barraqueta et al., 2018) and meteoric inputs (including coastal runoff and glacial meltwater) at the Newfoundland and Greenland margins (Benetti et al., 2016). Substantial sediment inputs were observed at all margins but with different intensity. The highest DFe sediment input was located at the Newfoundland margin, while the lowest was observed at the eastern Greenland margin. These differences could be explained by the different nature of particles with the most lithogenic located at the Iberian margin and the most biogenic, at the Newfoundland margin (Gourain et al., subm.). Although previous studies (e.g. Jickells et al., 2005; Shelley et al., 2015) reported that atmospheric inputs substantially fertilized surface waters from the West European Basin, in our study only stations located in the West European and Iceland Basins exhibited enhanced SML DFe inventories with lower TTADs. However, these TTADs were about three times higher that those reported for Saharan dust inputs and thus atmospheric deposition appeared to be a minor source of Fe at the sampling period. Finally, there was evidence of convective inputs of the LSW to surface seawater caused by long tip jet event (Piron et al., 2016) that deepened the winter mixed layer down to ~ 1200 m depth (Zunino et al., 2017), in which Fe was in excess of nitrate and where thus Fe was not limiting at the sampling period.

**Figures:**

Figure 1: great;

Figure 2: increase letter size, it is hard to read;

Figure 3: I am not sure that white contour lines for DFe help to understand Chl a. I would remove DFe and include this figure in the supplementary material.

Figure 4, 5 and 6(?): great

Figure 7, 8, 9, 10: can go in the supplementary material, maybe Fig. 9 you can keep

Figure 11 -14: in the sup mat.,

Maybe Figure 13 for the first 200 m can stay!

Table 3 and 4: belongs into the sup. Material

➔ As you suggested we only kept Figs 1, 2, 4, 5, 9, 10 (the new one) and 13 (with your suggestions), all other Figures are now in the supplementary material. Tables 3 and 4 are also in the supplementary material.

[revised manuscript text omitted]

---

## Author Comment (AC2) · 17 Aug 2018

**Anonymous Referee#2 reviews**

Dear Referee#2 and  Editor,

The reviewers are thanked for their insightful comments; these have helped to improve the manuscript considerably. Please see our detailed answers to the referees' comments below. Line numbers refer to the new version.

All the answers are attached as a supplementary file.

Best regards,

Tonnard et al.,

Page 1, Line 35: "in the Denmark Straight. . ."

➔ We have changed the text accordingly.

Page 1, Line 35: explain what types of particles you are talking about and briefly explain the differences observed (which ones scavenge and which ones release dFe)

➔ We have added precision

Page 1 Lines 34-36: Finally, the nepheloid layers located in the different basins and at the Iberian Margin were found to act as either a source or a sink of DFe depending on the nature of particles with organic particles likely releasing DFe and Fe-Mn oxides scavenging DFe.

Page2, Line 4: The reasoning is not flowing properly here. You need to say that (1) high productivity leads to high atmospheric carbon capture and that (2) Deep water formation leads to sequestration of this carbon into deeper waters, where carbon is stored for longer. The last sentence comes a little out of the blue, needs to be better linked – instead close the paragraph highlighting why it is important to study trace metals in this area.

➔ We have reorganised the full introduction as suggested.

Page 2 Lines 1-33, Page 3 Lines 1-20:

**1 Introduction**

The North Atlantic Ocean is known for its pronounced spring phytoplankton blooms (Henson et al., 2009; Longhurst, 2007),  that induce the capture of atmospheric carbon through photosynthesis, which allows its conversion into particulate organic carbon (POC). This POC is then exported into deeper waters through the production of biogenic particles () and through the Atlantic Meridional Overturning Circulation (AMOC), which is responsible for transporting large amounts of water, heat, salt, carbon, nutrients, oxygen and other elements around the globe. Hence, the North Atlantic Ocean shows the largest oceanic storage rate of anthropogenic $CO_2$ (Pérez et al., 2013) through both the physical and biological carbon pumps, despite covering only 15% of the global ocean area (Humphreys et al., 2016; Sabine et al., 2004) and is therefore crucial for Earth's climate.  However, the rapid attenuation of light with depth restricts the growth of phytoplankton organisms to the surface layer as it is the principal control on phytoplankton growth timing. The extensive studies conducted in the North Atlantic Ocean through the Continuous Plankton Recorder (CPR) have nevertheless highlighted the relationship between the strength of the westerlies and the

[revised manuscript text omitted]

potential long-range transport of DFe through the Deep Western Boundary Current (DWBC) via the investigation of the local processes affecting the DFe concentrations within the three main water masses that constitutes it, namely, Iceland Scotland Overflow Water (ISOW), Denmark Strait Overflow Water (DSOW) and Labrador Sea Water (LSW). This will thus better constrain our understanding of the biogeochemical cycling of DFe and how its cycling is linked to wider biogeochemical cycles (i.e. carbon and macronutrients) in the oceans and implement biogeochemical models,

Page 2, Line10: I can not follow the reasoning in this paragraph. A little bit of a muddle of all the phytoplankton limiting factors (light, nutrients, wind, temp) without a clear insight what factor limits where. Needs to be better explained

➔ We have reorganised the full introduction as suggested (see above).

Page 2, Line 15: the connection between light limitation and nutrient limitation is not clearly explained

➔ We have reorganised the full introduction as suggested (see above).

Page 2, Line26: what about soluble Fe? Is this not considered the most bioavailable?

➔ Yes, we absolutely agree. Therefore to avoid confusion we have decided to remove the following sentence "…and within its physical speciation, its dissolved form (DFe) is considered to be the most available form for phytoplankton (Morel, 2008; Morel et al., 2008).".

Page 3, Line5: Aims of this paper are a bit poor. Add better understanding of the biogeochemical cycling of dFe in the oceans - inform biogeochemical models – and why this is important, what you expect to achieve. . ..

➔ We have reorganised the full introduction as suggested (see above).

Page 3, Line 20: remove "national"

➔ Removed

Page 3, line 30: do you mean concentrated HCl?

➔ We have changed the text for clarification

Page 4 Lines 1-2: Samples were then acidified to ~ pH 1.7 with HCl (Ultrapur® Merck, 2 ‰ v/v) under a class 100 laminar flow hood inside the clean container.

Page 3, Line25: why different filtering methods? Have you compared the Fe concentrations in those fractions? i.e., have you collected the same sample with both filtration cut-offs and checked there is no significant difference?

➔ We added precision for clarification. Note that we did not collected the same samples with the two different techniques.

Page 3 Lines 24-30: Samples were either taken from the filtrate of particulate samples (collected on polyethersulfone filters, 0.45 µm supor®, see Gourain et al., this issue) or after filtration using 0.2 µm filter cartridges (Sartorius SARTOBRAN® 300) due to water restriction (Table 1). No significant difference was observed between DFe values filtered through 0.2 µm and 0.45 µm (p-value > 0.2, Wilcoxon test), neither between stations (i.e. stations 17, 19, 21, 25, 26, 29, 32, 34, 42, 44, 49) while swapping between both filtration techniques (p-values > 0.05, Wilcoxon tests paired by depth and

against the sign of the alternative hypothesis depending on the filtration technique used), except between station 11 and 13 and 13 and 15.

Page 4, Line 4: remove "daily basis"

➔ We have changed the text "High-purity grade solutions and water (Milli-Q) were used on a daily basis to prepare the following reagents:" by "High-purity grade solutions and water (Milli-Q) were used to prepare the following reagents each day:" (Page 4 Line 13)

Page 4, Line 14: replace "run" with "analytical session"

➔ We have changed the text as suggested (Page 4 Line 25)

Page 4, line 19: replace "in nmol L-1" with "to nmol L-1"

➔ We have changed the text as suggested (Page 5 Line 2)

Page 4, Line 19: would it not be more correct to multiply by the specific density of each sample? If you do not have this data (normally this is a standard parameter obtained from temperature and salinity.

➔ Yes, we agree with the reviewer. However, the converting factor used for consensus materials has always been 1.025 kg L$^{-1}$, which is why we used this converting factor.

Page 4, Line 19: Please show a comparison of dFe data at the crossover-station with GA02, as an intercalibration exercise.

➔ We did not include an intercalibration exercise with the GA02 voyage as DFe was determined on board by FIA-CL during GA02, which thus means that some refractory DFe was not measured on board with only a short time of acidification (Chever et al., 2010).

➔ Just for your information, here after you will find the plot comparing both voyages:

[Figure]

Page 4, Line25: Why did you not use the CTD data from the trace metal casts? Please explain

➔ We did not use the CTD data from the trace metal casts as the O2 data could not be calibrated. We therefore decided to use all the parameters from the same rosette, i.e. the stainless steel rosette.

Page 5, Line 12: awkward sentence, difficult to follow, please rewrite!

➔ We have changed the text for clarification

Page 5 Lines 25-28: Using this water mass determination, DFe concentrations were considered as representative of a specific water mass only when the contribution of this specific water mass was higher than 60% of the total water mass pool.

Page 6, Line 5: which central waters? Names?

➔ The central waters are Eastern North Atlantic Central Waters (ENACW). The acronym was added in the text there page 6 line 2.

Page 6, Line 9: to keep consistency, keep "stations 49 and 60" out of the parenthesis. Rephrase the end of the sentence to do so.

➔ We have modified the text as suggested

Page 6 Lines 21-25: West of the Subarctic Front, Iceland SubPolar Mode Waters (IcSPMW, $7.07 < \theta < 8°C$, $35.16 < S < 35.23$, $280 < O_2 < 289$ µmol kg$^{-1}$) was encountered from stations 34-40 (accounting for more than 45% of the water mass pool from 0 to ~ 800 m depth) and Irminger SubPolar Mode Waters (IrSPMW, $\theta \approx 5°C$, $S \approx 35.014$) from stations 42-44 (contributing to 40% of the water mass pool from 0 to ~ 250 m depth) and stations 49 and 60 (accounting for 40% of the water mass pool down to 1300 m depth).

Page 6, Line10: specify which stations

➔ We have changed the text as suggested.

Page 6 Lines 25-26: The IcSPMW was also observed within the Subtropical gyre (stations 11-26), subducted below ENACW until ~ 1000 m depth.

Page 6, Line 14: remove "The" from start of sentence

➔ Removed (Page 6 Line 29)

Page 6, Line 19: what is that contribution? 40 %? Please specify!

➔ We have changed the text as suggested.

Page 7 Lines 2-3: The LSW was also observed in surface waters of station 44 with a similar contribution than IrSPMW (~ 40%).

Page 6, Line 28: ... "was" sourced from...

➔ We have corrected the text accordingly (Page 7 Line 11)

Page 6, Line 30: and "in the" Labrador Sea

➔ We have corrected the text accordingly (Page 7 Line 12)

Page 6, Line 34: I am getting a little lost with all those branches, not sure when you're talking about the same one and when you change talking about another one. Not clear, please rephrase this section.

➔ We have changed the text for clarification.

Page 7 Lines 13-14: After convecting, LSW splits into three main branches with two main cores separated by the Reykjanes Ridge (stations 1-32, West European and Iceland Basins; stations 40-60, Irminger Sea), and the last one entering the West European Basin (Zunino et al., 2017).

Page 7, Line 5: delete "The" from beginning of sentence. In this entire section please remove "the" in front of water masses. The text should be revised by one of the English speaking co-authors before submission.

➔ We have changed the text accordingly

Page 7, Line7: "lower" oxygen. . .

➔ We have changed the text accordingly (Page 7 Line 18)

Page 7, Line 4: what do you mean by dense shelf? Do you mean the water masses have higher density? Please rephrase

➔ We have changed the text for clarification

Page 7 Lines 20-24: Polar Intermediate Water (PIW, $\theta \approx 0°C$, $S \approx 34.65$) is a ventilated, dense, low-salinity water intrusion to the deep overflows within the Irminger and Labrador Seas that is formed at the Greenland shelf. PIW represents only a small contribution to the whole water mass pool (up to 27%) and was observed over the Greenland slope at stations 53 and 61 as well as in surface waters from station 63 (from 0 to ~ 200 m depth), in intermediate waters of stations 49, 60 and 63 (from ~ 500 to ~ 1500 m depth) and in bottom waters of stations 44, 68, 69, 71 and 77 with a contribution higher than 10%.

Page 7, Line 16: "the" Charlie Gibbs. . .

➔ We have changed the text as suggested (Page 7 Line 26)

Page 7, Line 19: mixing with... remove "of the overflow"

➔ These lines have been removed according to Referee #1 comments

Page 7, Line 21: Which stations?

➔ The stations numbers were added for the Iceland Basin.

Page 7 Line 30-32: ISOW was observed from 1500 m depth to the bottom of the entire Iceland Basin (stations 29-38) and from 1800 to 3000 m depth within the Irminger Sea (stations 40-60)

Page 7, Line 30: At least put a nitrate section figure in the supplementary file; otherwise text hard to follow. Data not yet available on the site you referenced

➔ Data are now available on the Sarthou et al., 2018 paper. However, we have changed this reference by the accurate one and added the reference of the SEANOE data base and associated paper: García-Ibáñez et al., 2018; Pérez et al., 2018; Sarthou et al., subm. Please note that in this manuscript, Nitrate data are changed for RFe/N data, therefore we did not added the nitrate data.

Page 8, Line 4: how do you define the "most open ocean station" for the transect? Deepest? Furthest away from land masses? Are you sure this is st 23?

➔ We have change the text for clarification and corrected the station number.

Page 8 Lines 8-11: The low surface $NO_3^-$ concentrations (lower than 6 µmol $L^{-1}$) in the West European Basin extended from station 2 (closest station to continental land mass) to station 25 (open ocean station) with concentrations ranging from 0.02 (station 11) to 3.9 (station 25) µmol $L^{-1}$.

Page 8, Line 4: it is unclear when you switch to talk about non-surface nitrate concentrations. Please rephrase this section to make this clearer.

➔ We have changed the section for clarification as suggested

Page 8 Lines 5-17: Surface nitrate (NO$_3^-$) concentrations (García-Ibáñez et al., 2018; Pérez et al., 2018; Sarthou et al., subm.) ranged from 0.01 to 10.1 µmol L$^{-1}$ (stations 53 and 63, respectively). There was considerable spatial variability in NO$_3^-$ surface distributions with high concentrations found in the Iceland Basin and Irminger Sea (higher than 6 µmol L$^{-1}$), as well as at stations 63 (10.1 µmol L$^{-1}$) and 64 (5.1 µmol L$^{-1}$), and low concentrations observed in the West European Basin, in the Labrador Sea and above continental margins. The low surface concentrations in the West European Basin ranged from 0.02 (station 11) to 3.9 (station 25) µmol L$^{-1}$. Station 26 delineating the extreme western boundary of the West European Basin exhibited enhanced NO$_3^-$ concentrations as a result of mixing between ENACW and IcSPMW, although these surface waters were dominated by ENACW. In the Labrador Sea (stations 68-78) low surface concentrations were observed with values ranging from 0.04 (station 68) to 1.8 (station 71) µmol L$^{-1}$. At depth, the lowest concentrations (lower than 15.9 µmol L$^{-1}$) were measured in ENACW (~ 0 - 800 m depth) and DSOW (> 1400 m depth), while the highest concentrations were measured within NEADW (up to 23.5 µmol L$^{-1}$), and in the mesopelagic zone of the West European and Iceland Basins (higher than 18.4 µmol L$^{-1}$).

Page 8, Line 6: which depths are you talking about?

➔ We added precision as suggested (see above)

Page 8, Line 12: isn't the fluorometer calibrated with the Chl-a measurements?

➔ We removed this part as suggested by Referee#1. However, the fluorometer was calibrated separately from the Chl-*a* measurements. To do so, 6 stations (including 6 different water masses) at 6 depths (surface, chlorophyll-max down to the base of the euphotic zone) were sampled. These samples included early morning, late evening and daytime samples to account for non-photochemical quenching, which causes a decrease of fluorescence signal at the surface during day-time. Therefore, all profiles were calibrated using night-time dependency and corrected day-time surface data for non-photochemical quenching.

Page 8, Line 13: Specify which depth range you are considering for looking at min/max Chl-a concentrations. Evidently, minimum Chl-a concentrations are found in the deep ocean

➔ As you said, the minimum Chl-*a* concentrations are found in the deep ocean and therefore we only reported min and max values for the maximum Chl-*a* concentrations considering all stations. We have changed the text for clarification

Page 8 Lines 17-21: Overall, most of the phytoplankton biomass was localised above 100 m depth with lower total chlorophyll-*a* (TChl-*a*) concentrations South of the Subarctic Front and higher at higher latitudes (Fig. 3). While comparing TChl-*a* maxima considering all stations, the lowest value (0.35 mg m$^{-3}$) was measured within the West European Basin (station 19, 50 m depth) while the highest values were measured at the Greenland (up to 4.9 mg m$^{-3}$, 30 m depth, station 53 and up to 6.6 mg m$^{-3}$, 23 m depth, station 61) and Newfoundland (up to 9.6 mg m$^{-3}$, 30 m depth, station 78) margins.

Page 8, Line 19: remind the reader here here that all the dFe data can be found in the supplementary file.

➔ We have added this precision as suggested. Note that Referee#1 suggested to gather subsections 3.3.1 and 3.3.2, to remove the first lines from section 3.3 (Page 8 Lines 20-25) and to change section 3.3.3 for section 3.4, which is what we did.

Page 8 Lines 26-27: Dissolved Fe concentrations (see supplementary material) ranged from 0.09 ± 0.01 nmol L$^{-1}$ (station 19, 20 m depth) to 7.8 ± 0.5 nmol L$^{-1}$ (station 78, 371 m depth) (see Fig. 4).

Page 9, Line 13: when describing the regions, go in same order as in Figure, otherwise confusing (start Labrador Sea and end WEB)

➔ We have made the corrections accordingly.

Page 8 Lines 26-30 and Page 9 Lines 1-10:

Page 9, Line 13: remove "the" before MOW and in front of LSW (line 15), SAIW (line 24) and IrSPMW (line 24).

➔ We have removed "the" in front of each water masses throughout the section

Page 9, Line 20: this is confusing, specify that you mean similar averages and not ranges (the range is larger for IcSPMW)

➔ We have added this precision as suggested

Page 9 Lines 23-24: In the Iceland Basin, SAIW and IcSPMW displayed similar averaged DFe concentrations (0.67 ± 0.30 nmol L$^{-1}$, n=7 and 0.55 ± 0.34 nmol L$^{-1}$, n=22, respectively).

Page 9, Line 22: LSW and ISOW averages are also similar, combine sentences.

➔ We have changed the text accordingly.

Page 9 Lines 24-26: Averaged DFe concentrations were similar in both LSW and ISOW, and higher than in SAIW and IcSPMW (0.96 ± 0.22 nmol L$^{-1}$, n=21 and 1.0 ± 0.3 nmol L$^{-1}$, n=10, respectively, Fig. 7).

Page 9: Line 24: "composed of" instead of "characterised by"

➔ We have changed the text as suggested (Page 9 Line 17).

Page 9, Line 32: Delete "compared to other ones"

➔ We have changed as suggested (Page 9 Line 16)

Page 10, Line 2: "lowest average dFe value" (DSOW also shows the highest deep water dFe concentrations)

➔ We have changed the text as suggested (Page 9 Lines 15-16). However, within the Labrador Sea DSOW presented the lowest DFe concentration. The highest deep water DFe concentrations are, we agree, found in DSOW but within the Irminger Sea.

Page 10, Line 9: it is a little odd to start explaining what cannot be included in any of your sub-sections. You start introducing the general structure of your discussion and then you go into much detail explaining dFe trends all the sudden. This is totally out of place here. You should add this paragraph to the end of the discussion or in a new section.

➔ We have added a new section for this paragraph but we kept this new section there in the MS, as we think it would be odd to let the reader know at the end of the MS what we did not include.

Page 10 Lines 1-24:

**4 Discussion**

In the following sections, we will first discuss the high DFe concentrations observed throughout the water column of stations 1 and 17 located in the West European Basin (Section 4.1), then, the relationship between water masses and the DFe concentrations (Section 4.2) in intermediate (Section 4.2.2 and 4.2.3) and deep (Section 4.2.4 and 4.2.5) waters. We will also discuss the role of wind (Section 4.2.1), rivers (Section 4.3.1), meteoric water and sea-ice processes (Section 4.3.2), atmospheric deposition (Section 4.3.3) and sediments (Section 4.4) in delivering DFe. Finally, we will discuss the potential Fe limitation using $DFe:NO_3^-$ ratios (Section 4.5).

**4.1 High DFe concentrations at station 1 and 17**

Considering the entire section, two stations (stations 1 and 17) showed irregularly high DFe concentrations (> 1 nmol $L^{-1}$) throughout the water column, thus suggesting analytical issues. However, these two stations were analysed twice and provided similar results, therefore discarding any analytical issues. This means that these high values originated either from genuine processes or from contamination issues. If there had been contamination issues, one would expect a more random distribution of DFe concentrations and less consistence throughout the water column. It thus appears that contamination issues were unlikely to happen. Similarly, the influence of water masses to explain these distributions was discarded as the observed high homogenized DFe concentrations were restricted to these two stations. Station 1, located at the continental shelf-break of the Iberian Margin, also showed enhanced PFe concentrations from lithogenic origin suggesting a margin source (Gourain et al., subm.). Conversely, no relationship was observed between DFe and PFe nor transmissometry for station 17. However, Ferron et al. (2016) reported a strong dissipation rate at the Azores-Biscay Rise (station 17) due to internal waves. The associated vertical energy fluxes could explain the homogenized profile of DFe at station 17, although such waves are not clearly evidenced in the velocity profiles. Consequently, the elevated DFe concentrations observed at station 17 remain unsolved.

Page 10, Line 20: I don't understand the aim of discussing dFe with water masses? I'd rather focus on the sources and sinks of dFe along the section. Discuss then the role of water masses in distributing the dFe signals. This means completely changing the focus of this section

➔ We decided to discuss DFe with the main water masses, i.e. LSW, ISOW and DSOW as they all constitute the Deep Western Boundary Current (DWBC).

Page 11, Line 2: Flow of thoughts not clear, reasons of deep winter mixing scattered; bits of information thrown in a little randomly. You should start off by saying that deep winter mixing is an important mechanism supplying nutrients to the surface ocean in the North Atlantic Ocean; then say how this deep winter mixing is produced (from what I can understand in your text are you trying to say this is due to the effects of wind + convective mixing+ subduction/upwelling; am I right? This was not clear); then say what the specific conditions were in the year you sampled. I am still giving you corrections on the section, which you can incorporate in your rewritten discussion if fitting. I think you can recycle some parts of your discussion.

➔ We have modified the text as suggested for clarification.

Page 11 Lines 1-11: In the North Atlantic Ocean, the warm and salty water masses of the upper limb of the MOC are progressively cooled and become denser, and subduct into the abyssal ocean. In some areas of the SubPolar North Atlantic, deep convective winter mixing provides a rare connection between surface and deep waters of the MOC thus constituting an important mechanism in supplying nutrients to the surface ocean (de Jong et al., 2012; Louanchi and Najjar, 2001). Deep convective winter mixing is triggered by the effect of wind and a pre-conditioning of the

ocean in such a way that the inherent stability of the ocean is minimal. Pickart et al. (2003) demonstrated that these conditions are satisfied in the Irminger Sea with the presence of weakly stratified surface water, a close cyclonic circulation, which leads to the shoaling of the thermocline and intense winter air-sea buoyancy fluxes (Marshall and Schott, 1999). Moore (2003) and Piron et al. (2016) described low-level westerly jets centred northeast of Cape Farewell, over the Irminger Sea, known as tip jet events, whose structure depends upon the splitting occurring as the flow encounter the orographic features from Cape Farewell, and that are strong enough to induce deep convective mixing (Bacon et al., 2003; Pickart et al., 2003).

Page 11, Line 7: "events" and "with a positive NAO"

➔ We have corrected as suggested.

Page 11, Line 8: "The winter mixed layer depth"

➔ We have corrected as suggested.

Page 11, Line 9: instead of " and were" use "which was"

➔ We have corrected as suggested.

Page 11, Line 11: "close to those found in LSW"

➔ We have corrected as suggested.

Page 11, Line 13: sentence incomprehensive, please rephrase

➔ This sentence has been removed according to Referee#1 comments and gather with previous sentence.

Page 11 Lines 16-18: Such winter entrainment was likely the process involved in the vertical supply of DFe within surface waters fuelling the spring phytoplankton bloom with DFe values close to those found in LSW.

Page 11, Line 24: need to improve a little the flow of thoughts in this paragraph. Start by saying why you see no MOW dFe signal, then support/contradict that argument(s) by what has been seen in other studies.

➔ We have modified this section according to your comments and Referee#1 comments.

Page 11 Lines 21-33 and Page 12 Lines 1-9: The Mediterranean Sea on its northern shores is bordered by industrialized European countries, which act as a continuous source of anthropogenic derived constituents into the atmosphere, and on the southern shores by the arid and desert regions of north African and Arabian Desert belts, which act as sources of crustal material in the form of dust pulses (Chester et al., 1993; Guerzoni et al., 1999; Martin et al., 1989). During the summer, when thermal stratification occurs, DFe concentrations in the SML can increase over the whole Mediterranean Sea by 1.6-5.3 nmol L$^{-1}$ in response to the accumulation of atmospheric Fe from both anthropogenic and natural origins (Bonnet and Guieu, 2004; Guieu et al., 2010; Sarthou and Jeandel, 2001). After atmospheric deposition, the fate of Fe will depend on the nature of aerosols, vertical mixing, biological uptake and scavenging processes (Bonnet and Guieu, 2006; Wuttig et al., 2013). During GEOVIDE, MOW was observed from stations 1 to 29 between 1000 and 1200 m depth and associated with high dissolved aluminium (DAl, Menzel Barraqueta et al., 2018) concentrations (up to 38.7 nmol L$^{-1}$), confirming the high atmospheric deposition in the Mediterranean region. In contrast to Al, no DFe signature was associated with MOW (Figs. 2 and 4). This feature was also reported in some studies (Hatta et al., 2015; Thuróczy et al., 2010), while others measured higher

DFe concentrations in MOW (Gerringa et al., 2017; Sarthou et al., 2007). However, MOW coincides with the maximum Apparent Oxygen Utilization (AOU) and it is not possible to distinguish the MOW signal from the remineralisation one (Sarthou et al., 2007). On the other hand, differences between studies are likely originating from the intensity of atmospheric deposition and the nature of aerosols. Indeed, Wagener et al. (2010) highlighted that large dust deposition events can accelerate the export of Fe from the water column through scavenging. As a result, in seawater with high DFe concentrations and where high dust deposition occurs, a strong individual dust deposition event could act as a sink for DFe. It thus becomes less evident to observe a systematic high DFe signature in MOW despite dust inputs.

Page 12, Line 3: "suggesting that the water mass is enriched in dFe during its flow path"

➔ We have changed the text as suggested taking into account Referee#1 comments.

Page 12 Lines 11-14: As described in Section 3.1, the LSW exhibited increasing DFe concentrations from its source area, the Labrador Sea, toward the other basins with the highest DFe concentrations observed within the Irminger Sea, suggesting that the water mass was enriched in DFe either locally in each basin or during its flow path (Fig. 7).

Page 12, Line 4: start by saying what those sources are and then support with the available literature.

➔ We have reorganised this section based on your comments and Reviewer#1 comments.

Page 12 Lines 5-32, Page 13 Lines 1-8: As described in Section 3.1, the LSW exhibited increasing DFe concentrations from its source area, the Labrador Sea, toward the other basins with the highest DFe concentrations observed within the Irminger Sea, suggesting that the water mass was enriched in DFe either locally in each basin or during its flow path (Fig. 7). These DFe sources could originate from a combination of high export of PFe and its remineralisation in the mesopelagic area and/or the dissolution of sediment.

The Irminger and Labrador Seas exhibited the highest averaged integrated TChl-a concentrations (98 $\pm$ 32 mg m$^{-2}$ and 59 $\pm$ 42 mg m$^{-2}$) compared to the West European and Iceland Basins (39 $\pm$ 10 mg m$^{-2}$ and 53 $\pm$ 16 mg m$^{-2}$), when the influence of margins was discarded. Stations located in the Irminger (stations 40-56) and Labrador (stations 63-77) Seas, were largely dominated by diatoms (>50% of phytoplankton abundances) and displayed the highest chlorophillid-$a$ concentrations, a tracer of senescent diatom cells, likely reflecting post-bloom condition (Tonnard et al., in prep.). This is in line with the highest POC export data reported by Lemaitre et al. (subm.) in these two oceanic basins. This likely suggests that biogenic PFe export was also higher in the Labrador and Irminger Seas than in the West European and Iceland Basins. Although, Gourain et al. (subm.) highlighted a higher biogenic contribution for particles located in the Irminger and Labrador Seas with relatively high PFe:PAl ratios (0.44 $\pm$ 0.12 mol:mol and 0.38 $\pm$ 0.10 mol:mol, respectively) compared to particles from the West European and Iceland Basins (0.22 $\pm$ 0.10 and 0.38 $\pm$ 0.14 mol:mol, respectively, see Fig. 6 in Gourain et al., subm.), they reported no difference in PFe concentrations between the four oceanic basins (see Fig. 12A in Gourain et al., subm.) when the influence of margins was discarded, which likely highlighted the remineralisation of PFe within the Irminger and Labrador Seas. Indeed, Lemaître et al. (2017) reported higher remineralisation rates within the Labrador (up to 13 mmol C m$^{-2}$ d$^{-1}$) and Irminger Seas (up to 10 mmol C m$^{-2}$ d$^{-1}$) using the excess barium proxy (Dehairs et al., 1997), compared to the West European and Iceland Basins (ranging from 4 to 6 mmol C m$^{-2}$ d$^{-1}$). Therefore, the intense remineralisation rates measured in the Irminger and Labrador Seas likely resulted in enhanced DFe concentrations within LSW.

Higher DFe concentrations were, however, measured in the Irminger Sea compared to the Labrador Sea and coincided with lower transmissometer values (i.e. 98.0-98.5% vs. >99%), thus suggesting a particle load of the LSW. This could be explained by the reductive dissolution of Newfoundland Margin sediments. Indeed, Lambelet et al. (2016) reported high dissolved neodymium (Nd) concentrations (up to 18.5 pmol.kg$^{-1}$) within the LSW at the edge of the Newfoundland Margin (45.73°W, 51.82°N) as well as slightly lower Nd isotopic ratio values relative to those observed in the Irminger Sea. They suggested that this water mass had been in contact with sediments approximately within the last 30 years (Charette et al., 2015). Similarly, during GA03, Hatta et al. (2015) attributed the high DFe concentrations in the LSW to continental margin sediments. Consequently, it is also possible that the elevated DFe concentrations from the three LSW branches which entered the West European and Iceland Basins and Irminger Sea was supplied through sediment dissolution (Measures et al., 2013) along the LSW pathway.

The enhanced DFe concentrations measured in the Irminger Sea and within the LSW were thus likely attributed to the combination of higher productivity, POC export and remineralisation as well as a DFe supply from reductive dissolution of Newfoundland sediments to the LSW along its flow path. As reported by Boyd and Ellwood (2010), when bacteria dissolve PFe they also release Fe-binding organic ligands especially from biogenic PFe than from lithogenic PFe (Boyd et al., 2010). It is thus possible that bacteria-mediated ligand production helped the DFe supply from Newfoundland sediments to remain in solution within the Irminger Sea LSW.

Page 12, Line 6: change "the ones" for "those"

➔ We have corrected as suggested (see above).

Page 12, Line 12: Provide a brief description of how the remineralisation rates were measured...

➔ We have added this precision (see above).

Page 12, Line 18: You are repeating yourself!

➔ We removed this sentence.

Page 12, Line 19: conspitious? clearly visible? change this word

➔ We have changed the text for clarification (see above).

Page 12, Line 21: confusing between remineralisation and bacteria mediated ligand production. Please be clear.

➔ We have reformulated this sentence (see above).

Page 12, Line 27: "Hereafter"

➔ We have corrected the text.

Page 13, Line 2: you also need to consider vertical/lateral inputs (think in 3D), so sedimentary Fe could be coming from further north, for example, within nepheloid layers. You can't discard sedimentary inputs just by looking at vertical gradients.

➔ We have reorganised the idea of this paragraph and wehave considered your suggestions (see below).

Page 13, Line 3: remove "the" before "Polar Intermediate Water" and "PIW" (line 6)

➔ We have corrected the text accordingly.

Page 13, Line 9: instead of thinking that one water mass carries a certain dFe concentration (think of the short Fe residence times), this water mass might have "picked up" some dFe from, e.g., the sediments, on its pathway or It might have picked up particles in suspension which dissolve over time, etc!

→ We have reorganised the idea of this paragraph and we have considered your suggestions.

Page 13 Lines 27-33 and Page 14 Lines 1-11: However, considering the short residence time of DFe and the circulation of water masses in the Irminger Sea, it is possible that instead of being attributed to one specific water mass, these enhanced DFe concentrations resulted from lateral advection of the deep waters. Figure 8B) shows the concentrations of both DFe and PFe for the mixing line between DSOW/PIW and ISOW at station 44 and considering 100% contribution of ISOW for the shallowest sample (2218 m depth) and of DSOW/PIW for the deepest (2915 m depth), as these were the main water masses. This figure shows increasing DFe concentrations as DSOW/PIW mixed with ISOW. In addition, Le Roy et al. (2018) reported for the GEOVIDE voyage at station 44 a deviation from the conservative behaviour of $^{226}$Ra reflecting an input of this tracer centred at 2500 m depth, likely highlighting diffusion from deep-sea sediments and coinciding with the highest DFe concentrations measured at this station. Although the transmissometer values were lower at the sediment interface than at 2500 m depth, Deng et al. (subm.) reported a stronger scavenged component of the $^{230}$Th at the same depth range, likely suggesting that the mixture of water masses were in contact with highly reactive particles. If there is evidence that the enhanced DFe concentrations observed at station 44 coincided with lateral advection of water masses that were in contact with particles, the difference of behaviour between DFe and $^{230}$Th remains unsolved. The only parameter that would explain without any ambiguity such differences of behaviour between DFe and $^{230}$Th  would be the amounts of Fe-binding organic ligands for these samples. Indeed, although PFe concentrations decreased from the seafloor to the above seawater, this trend would likely be explained by a strong vertical diffusion alone and not necessarily from the dissolution of particles that were laterally advected.

Therefore, the high DFe concentrations observed might be inferred from local processes as ISOW mixes with both PIW and DSOW with a substantial load of Fe-rich particles that might have dissolved in solution due to Fe-binding organic ligands.

Page 13, Line 11: instead of "seawater" use "water masses"

→ We have reformulated this sentence (see above)

Page 13, Line 11: specify what you mean; 100 % contribution of which water mass at which depth

→ We have changed the text as suggested (see above).

Page 13, Line 18: watch out, this leads to miss-understanding as the reader thinks you are saying that particulate Fe is sustained by organic ligands. I don't think that is what you want to say.

→ We have changed the text as suggested (see above).

Page 13, Line 19: restructure this paragraph, say what you think the reasons are behind the high dFe concentrations, then support that by correlations, graphs etc and then compare to literature to confirm or dispute your theory. end the section with a stronger statement of what you think is happening in these deep water masses

→ We have changed the text as suggested (see above).

Page 13, Line 26: your explanation is a little long winded. Say that Mid Ocean Ridges can be a source of dFe but that this has not been found in the Reykjanes Ridge so far

➔ We have changed the text according to your comments and referee#1 comments.

Page 13 Lines 31-21 and Page 14 lines 1-5: Hydrothermal activity was assessed over the Mid Atlantic Ridge, namely the Reykjanes Ridge, from stations 36 to 42. Indeed, within the interridge database (http://www.interridge.org), the Reykjanes Ridge is reported to have active hydrothermal sites that were either confirmed (Baker and German, 2004a; German et al., 1994; Olaffson et al., 1991; Palmer et al., 1995) close to Iceland or inferred (e.g. Chen, 2003; Crane et al., 1997; German et al., 1994; Sinha et al., 1997; Smallwood and White, 1998) closer to the GEOVIDE section as no plume was detected but a high backscatter was reported potentially corresponding to a lava flow. Therefore, hydrothermal activity at the sampling sites remains unclear with no elevated DFe concentrations or temperature anomaly above the ridge (station 38).

Page 13, Line 30: you are repeating you pattern again; please first explain the signals you see and then compare to the literature

➔ We have changed the text according to your comments and Referee#1 comments.

Page 14 Lines 5-15: However, enhanced DFe concentrations (up to $1.5 \pm 0.22$ nmol L$^{-1}$, station 36, 2200 m depth) were measured east of the Reykjanes Ridge (Fig. 4). This could be due to hydrothermal activity and resuspension of sunken particles at sites located North of the section and transported through the ISOW towards the section (Fig. 7). Indeed, Achterberg et al. (2018) highlighted at ~60°N and over the Reykjanes Ridge a southward lateral transport of an Fe plume of up to 250-300 km. In agreement with these observations, previous studies (e.g. Fagel et al., 1996; Fagel et al., 2001; Lackschewitz et al., 1996; Parra et al., 1985) reported marine sediment mineral clays in the Iceland Basin largely dominated by smectite (> 60%), a tracer of hydrothermal alteration of basaltic volcanic materials (Fagel et al., 2001; Tréguer and De La Rocha, 2013). Hence, the high DFe concentrations measured east of the Reykjanes Ridge could be due to a hydrothermal source and/or the resuspension of particles and their subsequent dissolution.

Page 14, Line 4-6: are these located on your transect? If not, you need to give their locations (coordinates) and explain where they are relative to your section. Are these GEOVIDE hydrographic sections (I don't think so) or from other cruises? Specify which ones. Cite appropriate Figures, and supporting literature.

➔ We have changed the text as suggested. Note that we also added the location of CGFZ and BFZ on Figure 1.

Page 14 Lines 17-19: West of the Reykjanes Ridge, a DFe-enrichment was also observed in ISOW within the Irminger Sea (Figs. 4 and 7). The low transmissometer values within ISOW in the Irminger Sea compared to the Iceland Basin suggest a particle load. These particles could come from the Charlie Gibbs Fracture Zone (CGFZ, 52.67°N and 34.61°W) and potentially Bight Fracture Zone (BFZ, 56.91°N and 32.74°W) (Fig. 1) (Lackschewitz et al., 1996; Zou et al., 2017). Indeed, hydrographic sections of the northern valley of the CGFZ showed that below 2000 m depth the passage through the Mid-Atlantic Ridge was mainly filled with the ISOW (Kissel et al., 2009; Shor et al., 1980). Shor et al. (1980) highlighted a total westward transport across the sill, below 2000 m depth of about 2.4 x $10^6$ m$^3$ s$^{-1}$ with ISOW carrying a significant load of suspended sediment (25 µg L$^{-1}$), including a 100-m-thick benthic nepheloid layer. It thus appears that the increase in DFe within ISOW likely came from sediment resuspension and dissolution as the ISOW flows across CGFZ and BFZ.

Page 14, Line 11: complete this section with more recent references. There have been previous studies in this area

➜ We have removed the following sentence "The seabed of this area 10 has been identified as a depositional environment with patches of ripples and rock fragments surrounded by moat." Based on Referee#1 comments and have changed the text (see above).

Page 14, Line 20: restructure, again start by saying what you see, explaining the reason of these signals showing correlations, and then supporting literature

➜ We have changed the text as suggested.

Page 15 Lines 9-24: Enhanced DFe surface concentrations (up to $1.07 \pm 0.12$ nmol $L^{-1}$) were measured over the Iberian Margin (stations 1-4) and coincided with salinity minima (~ <35) and enhanced DAl concentrations (up to 31.8 nmol $L^{-1}$, Menzel Barraqueta et al., 2018). DFe and DAl concentrations were both significantly negatively correlated with salinity ($R^2$ = ~1 and 0.94, respectively) from stations 1 to 13 (Fig. 9). Salinity profiles from station 1 to 4 showed evidence of a freshwater source with surface salinity ranging from 34.95 (station 1) to 35.03 (station 4). Within this area, only two freshwater sources were possible: 1) wet atmospheric deposition (4 rain events, Shelley, pers. comm.) and 2) the Tagus River, since the ship SADCP data revealed a northward circulation (P. Lherminier and P. Zunino, Ifremer Brest, pers. comm.). Our SML DFe inventories were higher at station 1 than those calculated during the GA03 voyage (~ 1nmol $L^{-1}$, station 1) during which atmospheric deposition were about one order of magnitude higher (Shelley et al., 2017a; Shelley et al., 2015), suggesting that the atmospheric source was not significant. Consequently, the Tagus River appears to be the most likely source responsible for these enhanced DFe concentrations, either as direct input of DFe or indirectly through Fe-rich sediment carried by the Tagus River and their subsequent dissolution. The Tagus estuary is the largest in the western European coast and is very industrialized (Canário et al., 2003; de Barros, 1986; Figueres et al., 1985; Gaudencio et al., 1991; Mil-Homens et al., 2009). It extends through an area of 320 $km^2$ and is characterized by a large water flow of 15.5 $10^9$ $m^3$ $y^{-1}$ (Fiuza, 1984). Many types of industry (e.g. heavy metallurgy, ore processing, chemical industry) release metals including Fe, which therefore result in high levels recorded in surface sediments, suspended particulate matter, water and organisms in the lower estuary (Santos-Echeandia et al., 2010).

Page 14, Line 26: You can't say "et al." in a personal communication. All people should be mentioned. Also State first name and affiliation in a personal communication.

➜ We have modified the text to fulfil this comment.

Page 15 Line 8: (P. Lherminier and P. Zunino, Ifremer Brest, pers. comm.)

Page 14, Line 26: Very difficult to follow; why "however"? Start by explaining the typical ratios of different sources before you give the observed ratios in your study. Then discuss the What dFe:dAl ratio is expected from a river source? And from an atmospheric source?

➜ We have removed this sentence according to your comment and Referee#1 comment and we have changed the text for clarification (see above).

Page 14, Line 29: Difficult to understand; you need to present each option that could lead to the enhanced dFe signal, discuss and then accept or discard

➜ We have changed the text as suggested (see above).

Page 15, Line 18: I don't understand this "extended as close as 200 km from our Greenland stations". Please rephrase. Also avoid parenthesis next to parenthesis, and explain clearly what can be found in the link and what info you got from there.

➔ We reword the sentence for clarification.

Page 17 Lines 1-6: The most plausible sources would be freshwater induced by meteoric water and sea-ice melt. Conversely, deeper in the water column, brine signals were calculated at stations 53 (100 m depth, Fig. 5D) 61 (100 m depth, Fig. 5E) and 78 (30 m depth, Fig. 5F). The release of brines could originate from two different processes: the sea-ice formation or the early melting of multiyear sea ice due to gravitational drainage and subsequent brine release (Petrich and Eicken, 2010; Wadhams, 2000). Indeed, during the winter preceding the GEOVIDE voyage, multiyear sea ice extended 200 km far from our Greenland stations (http://nsidc.org/arcticseaicenews/).

Page 15, Line 20: how were these calculated? give equations

➔ We have included a section in the method on how these fractions were calculated.

Page 5 Lines 7-24: We separated the mass contributions to samples from stations 53, 61 and 78 in Sea-Ice Melt (SIM) Meteoric Water (MW) and saline seawater inputs using the procedure and mass balance calculations that are fully described in Benetti et al. (2016) (Fig. 5D), E) and F)). Hereafter, we describe briefly the principle. We considered two types of seawater, namely the Atlantic Water (AW) and the Pacific Water (PW). After estimating the relative proportions of AW ($f_{AW}$) and PW ($f_{PW}$) and their respective salinity and δ18O affecting each samples, the contribution of SIM and MW can be determined using measured salinity ($S_m$) and δ18O ($\delta O_m^{18}$). The mass balance calculations are presented below:

$$f_{AW} + f_{PW} + f_{MW} + f_{SIM} = 1 \text{ (eq.1)}$$

$$f_{AW}.S_{AW} + f_{PW}.S_{PW} + f_{MW}.S_{MW} + f_{SIM}.S_{SIM} = S_m \text{ (eq.2)}$$

$$f_{AW}.\delta O_{AW}^{18} + f_{PW}.\delta O_{PW}^{18} + f_{MW}.\delta O_{MW}^{18} + f_{SIM}.\delta O_{SIM}^{18} = \delta O_m^{18} \text{ (eq.3)}$$

where $f_{AW}$, $f_{PW}$, $f_{MW}$, $f_{SIM}$ are the relative fraction of AW, PW, MW, and SIM. To calculate the relative fractions of AW, PW, MW and SIM we used the following end-members: $S_{AW}$ = 35, $\delta O_{AW}^{18}$ = +0.18‰ (Benetti et al., 2016); $S_{PW}$ = 32.5, $\delta O_{PW}^{18}$ = -1‰ (Cooper et al., 1997; Woodgate and Aagaard, 2005); $S_{MW}$ = 0, $\delta O_{MW}^{18}$ = -18.4‰ (Cooper et al., 2008); $S_{SIM}$ = 4, $\delta O_{SIM}^{18}$ = +0.5‰ (Melling and Moore, 1995).

In Figure 5 D), E) and F), negative sea-ice fractions indicated a net brine release while positive sea-ice fractions indicated a net sea-ice melting. Note that for stations over the Greenland Shelf, we assumed that the Pacific Water (PW) contribution was negligible for the calculations, supported by the very low PW fractions found at Cape Farewell in May 2014 (see Figure B1 in Benetti et al., 2017), while for station 78, located on the Newfoundland shelf, we used nutrient measurements to calculate the PW fractions, following the approach from Jones et al. (1998) (the data are published in Benetti et al., 2017).

Page 15, Line 27: I suggest instead of describing the Figures, you should use them to support your discussion. So try to avoid starting sentences and paragraphs describing Figures! You do this very often

➔ We have changed the text as suggested.

Page 17 Lines 11-26: Considering the sampling period at stations 53 (16 June 2014) and 61 (19 June 2014), sea-ice formation unlikely explained the brine signals calculated as this period coincides with summer melting in both the Central Arctic and East Greenland (Markus et al., 2009). However, it is

possible that the brines observed in our study could originates from sea-ice formation, which occurred during the previous winter(s) at 66°N (and/or higher latitudes). The residence time can vary from days (von Appen et al., 2014) to 6-9 months (Sutherland et al., 2009). Due to our observed strong brine signal at station 61 we suggest that the residence time was potentially longer than average. Given that the brine signal was higher at station 61 than at station 53 (which was located upstream in the EGC), we suggest that station 53 was exhibiting a freshening as a result of the transition between the freezing period toward the melting period. This would result in a dilution of the brine signal at the upstream station. Consequently, the salinity of this brine signal may reflect sea ice formation versus melting which may have an effect on the trace metal concentration within this water (Hunke et al., 2011). The associated brine water at station 61 was slightly depleted in both DFe and PFe, which may be attributed to sea ice formation processes. Indeed, Janssens et al. (2016) highlighted that as soon as sea ice forms, sea salts are efficiently flushed out of the ice while PFe is trapped within the crystal matrix and DFe accumulates, leading to an enrichment factor of these two Fe fractions compared to underlying seawater. Conversely, the brine signal observed at station 53 (100 m depth) showed slight enrichment in DFe, which may be attributed to brine release during early sea ice melting and the associated release of DFe into the underlying water column as the brine sinks until reaching neutral buoyancy due to higher density.

Page 15, Line 27: how is sea-ice fraction calculated? please provide equations (if you calculated it yourself) or references where this data is published. Provide clear info so the reader can understand

➔ We have included a section in the method on how these fractions were calculated (see above)

Page 15, Line 30: explain why you compare to dAl in these profiles.

➔ We have removed DAl distribution.

Page 15, Line 33: "originates"

➔ We have corrected accordingly.

Page 16, Line 7: I can't believe that sea-ice can "uptake" Fe! The concentration of Fe in the newly formed ice and in the remaining water should stay the same. You need to find another explanation!

➔ We reword the sentence and add a reference for clarification.

Page 16 Lines 23-26: The associated brine water at station 61 was slightly depleted in both DFe and PFe which may be attributed to sea ice formation processes. Indeed, Janssens et al. (2016) highlighted that as soon as sea ice forms, sea salts are efficiently flushed out of the ice while PFe is trapped within the crystal matrix and DFe accumulates, leading to an enrichment factor of these two Fe fractions compared to underlying seawater.

Page 16, Line 10: you mean sea-ice formation? Hence release of brine? Also explain that the brine sinks because it is denser (this is why it is observed below the surface)

➔ We have changed the text as suggested.

Page 16 Lines 27-29: Conversely, the brine signal observed at station 53 (100 m depth) showed slight enrichment in DFe, which may be attributed to brine release during early multiyear sea-ice melting and the associated release of DFe into the underlying water column as the brine sinks due to higher density until reaching neutral buoyancy.

➔ Regarding brines, they can originate from two different processes: either as a result of multiyear sea-ice melting or during sea-ice formation. Indeed, during the early melting season,

multiyear sea-ice has a higher porosity and gravitational drainage of brine occur. These two processes of brine release might lead to different TM signatures in brine originating from sea-ice formation and brine originating from early melting of multiyear sea-ice (Petrich and Eicken, 2010; Wadhams, 2000).

Page 16, Line 10: "release of dFe"

➔ We have corrected accordingly.

Page 16, Line 10: "underlying water column"

➔ We have corrected accordingly.

Page 16, Line 11: split this sentence, it is too long

➔ We have changed the text as suggested.

Page 16 Lines 29-32: Surface waters (from 0 to ~ 50 m depth) from station 53 and 61 were characterized by high MW fractions (ranging from 8.3 to 7.4% and from 7.7 to 7.4% , respectively, from surface to ~50 m depth). Within these surface waters, station 53 exhibited substantial sea-ice melting contribution (1.5%, 4 m depth) while station 61 exhibited low contribution (0.6%) from brine release that was linearly increasing with depth (1.3% at 50 m depth and 2.2 % at 100 m depth).

Page 16, Line 15: instead of describing the data (this is more appropriate for results section) you should say what correlates with what (e.g., low Fe with low MW). difficult to follow flow of thoughts

➔ We have changed the text as suggested.

Page 17 Lines 27-34, Page 18 Lines 1-11: Surface waters (from 0 to ~ 100 m depth) from station 53 and 61 were characterized by high MW fractions (ranging from 8.3 to 7.4% and from 7.7 to 7.3% , respectively, from surface to ~100 m depth, Figs. 5D and E). Within these surface waters, station 53 exhibited substantial sea-ice melting contribution (1.5%, 4 m depth, Fig. 5D), while station 61 exhibited low contribution (0.6%, Fig. 5E) from brine release that was linearly increasing with depth (1.3% at 50 m depth and 2.2 % at 100 m depth, Fig. 5E). These high MW fractions were both enriched in PFe and DFe (except station 53 for which no data was available) compared to seawater located below 50 m depth, thus suggesting a MW source. These results are in line with previous observations, which highlighted strong inputs of DFe from a meteoric water melting source in Antarctica (Annett et al., 2015). Although the ability of MW from Greenland Ice Sheet and runoffs to deliver DFe and PFe to surrounding waters have previously been demonstrated (Bhatia et al., 2013; Hawkings et al., 2014; Schroth et al., 2014; Statham et al., 2008), both Fe fractions were lower at the sample closest to the surface, then reached a maximum at ~ 50 m depth and decreased at ~ 70 m depth, for station 61 (Fig. 5D). The surface DFe depletion was likely explained by phytoplankton uptake, as indicated by the high TChl-$a$ concentrations (up to 6.6 mg m$^{-3}$) measured from surface to about 40 m depth, drastically decreasing at ~ 50 m depth to 3.9 mg m$^{-3}$ (Fig. 5D). Hence, it seemed that meteoric water inputs from the Greenland Margin likely fertilized surface waters with DFe, enabling the phytoplankton bloom to subsist. The profile of PFe can be explained by two opposite plausible hypotheses: 1) MW inputs did not released PFe, as if it was the case, one should expect higher PFe concentrations at the surface (~25 m depth) than the one measured at 50 m depth due to both the release from MW and the assimilation of DFe by phytoplankton 2) MW inputs can release PFe in a form that is directly accessible to phytoplankton with subsequent export of PFe as phytoplankton died. The latter solution explains the PFe maximum measured at ~ 50 m depth and is thus the most plausible.

Page 16, Line 17: remind the reader in which figure information can be found (Chl-a, ect)

➔ We added these precisions throughout this section.

Page 16, Line 24: results in agreement with the capacity of. . .? weird sentence, please change

➔ We have changed the text as suggested.

Page 17 Lines 10-12: Indeed, the ability of MW from Greenland Ice Sheet and runoffs to deliver DFe and PFe to surrounding waters have previously been demonstrated (Bhatia et al., 2013; Hawkings et al., 2014; Schroth et al., 2014; Statham et al., 2008).

Page 16, Line 27: so far you have not talked about dFe:dAl ratio in meteoric water. Explain if this ratio is used to trace MW inputs and what you see in your profiles. should explain at the start.

➔ We have added a sentence at the beginning of the section to fulfil this comment and removed the DAl data.

Page 17 Lines 1-3: In the following sections, we discuss the potential for meteoric water supply, sea-ice formation and sea-ice melting to affect DFe distribution.

Page 16, Line 28: change "noting that" for "although"

➔ We changed the text as suggested (Page 17 Line 14)

Page 17, Line 6: or maybe the brine conditions (pH, salinity etc) make the Fe more bioavailable; or maybe this peak is not Fe related, but related to the release of other TM, or a phytoplankton group that thrives in brine and does not require much Fe? Since this shelf is further south, the environmental conditions may be more favourable for phytoplankton to grow and hence consuming all the dFe more rapidly. You should explore all the possibilities

➔ We have changed the text as suggested.

Page 18 Lines 20-31: This either suggests that the brine likely contained important amounts of Fe (dissolved and/or particulate Fe) that were readily available for phytoplankton and consumed at the sampling period by potentially sea-ice algae themselves (Riebesell et al., 1991) or that another nutrient was triggering the phytoplankton bloom. Since these waters were dominated by microphytoplankton with up to 98% of diatoms (Tonnard et al., in prep.) and that none of the macronutrients were limiting at the sampling period (Sarthou et al., subm.; Tonnard et al., in prep.), the main phytoplankton functional class was expected to have high DFe requirements and thus a more bioavailable Fe associated to the brine signal seemed to be more plausible. If such was the case, then a PFe maximum should be noticed at the same depth. However, it should be noted that TChl-$a$ and $\delta^{18}O$ samples were collected about four hours prior to sampling for DFe and PFe. Therefore, it is more likely that by the time DFe and PFe samples were collected, the PFe was exported deeper in the water column. Indeed, Krembs et al. (2002) highlighted the presence of exopolymeric substances (EPS) in sea ice. Such compounds were reported to undergo fast aggregation (minutes to hours) from the colloidal to the particulate phase (i.e. Transparent Exopolymer Particles, TEP) (e.g. Baalousha et al., 2006; Verdugo et al., 2004) taking in-depth other particulate material as they sank

Page 17, Line 19: You repeat the word "dust" too much

➔ We have changed the text according to your comment and Referee#1 comment.

Page 19 Lines 2-7: On a regional scale, the North Atlantic basin receives the largest amount of atmospheric inputs due to its proximity to the Saharan Desert (Jickells et al., 2005), yet even in this region of high atmospheric deposition, inputs are not evenly distributed. Indeed, aerosol Fe loading measured during GEOVIDE (Shelley et al., 2017b) were much lower (up to four orders of magnitude) than those measured during studies from lower latitudes in the North Atlantic (e.g.

Baker et al., 2013; Buck et al., 2010; and for GA03, Shelley et al., 2015), but atmospheric inputs could still be an important source of Fe in areas far from land.

Page 17, Line 20, "proportions"

➔ We have changed the text (see above).

Page 18, Line 3: you are going into too much detail here about aerosols, which is part of the Shelley papers

➔ We shorten the text.

Page 19 Lines 17-18: Shelley et al. (2017a; 2017b) reported aerosol fractional solubility for Fe ranging from 7-56% following a two-stage sequential leach.

Page 18, Line 6: Meskhidze et al. (2017) is not in your reference list

➔ We added the reference to the reference list.

Page 18, Line 10: information is a little randomly thrown in... what is your point? This is not a review paper

➔ We have shorten the text as suugested.

Page 19 Lines 20-23: Once in the water, many studies (Bressac and Guieu, 2013; Bressac et al., 2014; Desboeufs et al., 2014; Meskhidze et al., 2017) argued that the aerosol trace metal fractional solubility is driven by the amount of organic matter in seawater and aerosols. However, as the organic composition of aerosols was not determined during our study we cannot comment further here.

Page 18, Line 13: What does OM mean? write abbreviations out first time. Do you mean organic matter?

➔ Yes, we changed OM for organic matter (see above).

Page 18, Line 15: So all this background information to lastly say that you don't comment on this? Rephrase, make some assumptions or delete some detail

➔ We shorten the text (see above).

Page 18, Line 26: of those stations, station 40 is most similar to total aerosol dFe:dAl ratios. station 26 is closer to the soluble than the total composition, so I don't understand what you are saying

➔ We removed all this part and changed it for turnover times relative to atmospheric deposition as defined in Guieu et al., 2014.

Page 19 Lines 24-33, Page 20 Lines 1-5: With this in mind, it seems that DFe concentrations are more or less comparable to soluble Fe inventories from aerosols, rather than total Fe in aerosols. In an attempt to estimate whether there was enough atmospheric input to sustain the SML DFe concentrations, we calculated Turnover Times relative to Atmospheric Deposition (TTADs, Guieu et al., 2014). To do so, we made the following assumptions: 1) the aerosol concentrations are a snapshot in time but are representative of the study region, 2) the aerosol solubility estimates based on two sequential leaches are an upper limit of the aerosol Fe in seawater and 3) the water column stratified just before the deposition of atmospheric inputs, so MLD DFe will reflect inputs from above. Thus, the TTADs were defined as the integrated DFe concentrations in the SML for each station divided by the contribution of soluble Fe contained in aerosols averaged per basin to the

water volume of the SML. Although, TTADs were lower in the West European and Iceland Basins with an average of ~ 9 ± 3 months compared to other basins (7 ± 2 years and 5 ± 2 years for the Irminger and Labrador Seas, respectively) (Fig. 10) they were about three times higher than those reported for areas impacted by Saharan dust inputs (~ 3 months, Guieu et al., 2014). Therefore, the high TTADs measured in the Irminger and Labrador Seas and ranging from 2 to 15 years provided further evidence that atmospheric deposition were unlikely to supply Fe in sufficient quantity to be the main source of DFe (see Sections 4.2.1 and 4.3.2) while in the West European and Iceland Basins they played an additional source, perhaps the main source of Fe especially at station 36 which displayed TTAD of 3 months.

[Figure]

Figure 10: Plot of dissolved Fe (DFe) Turnover Times relative to Atmospheric Deposition (TTADs) calculated from soluble Fe contained in aerosols estimated from a two-stage sequential leach (UHP water, then 25% HAc, Shelley et al., this issue). Note that numbers on top of data points represent station numbers and that the colour coding refers to different region with in yellow, margin stations; in purple, the West European Basin; in blue, the Iceland Basin; in green, the Irminger Sea and in red, the Labrador Sea. The numbers on top of the plot represent TTADs averaged for each oceanic basin and their standard deviation.

Page 18, Line 28: And what about all the other stations where the ratios are different? I would say this is more of a "coincidence" that these data fall onto the black line. The multiple reactions occurring as Fe enters the ocean change this Fe:Al ratio rapidly

➔ Same as above, this part has been removed.

Page 18, Line 30: remove "time"

→ This part has been removed (see above)

Page 19, Line 2: I don't understand.... these station points on your figure 10 are similar to those of other stations. What makes you think they have a higher atmospheric influence??

→ This part has also been removed (see above)

Page 19, Line 4: in this section you should mention that bottom water dFe concentrations were significantly higher on the Newfoundland margin than on the Greenland margins

→ We have included this precision.

Page 19 Lines 23-27: DFe concentration profiles from all coastal stations (stations 2, 4, 53, 56, 61 and 78) are reported on Figure 5. To avoid surface processes, only depths below 100 m depth will be considered in the following discussion. Stations where DFe and PFe followed a similar pattern are stations 2, 53, 56, and 78, suggesting that either the sources of Fe supplied both Fe fractions (dissolved and particulate) or that PFe dissolution from sediments supplied DFe. Among the different margins, the Newfoundland Margin exhibited the highest deep-water DFe concentrations.

Page 19, Line 10: "mol:mol" to stay consistent

→ We have changed the text as suggested. (Page 19, Line 28).

Page 19, Line 10: what is the average useful for? Show a plot with dFe:pFe ratios or a table

→ Because the SD is relatively high thus conferring distinct signature throughout the water column (below 100 m depth) of each station. This allows the comparison station by station instead of comparing each samples. These ratios were added to Table 4.

Page 19, Line 11: as well as different sediment compositions, this could be also due to different supply mechanisms? Different sediment conditions (redox, organic content, temp, etc)

→ We added this precision in the text.

Page 19 Lines 28-31: DFe:PFe ratios ranged from 0.01 (station 2, bottom sample) to 0.27 (station 4, ~ 400 m depth) mol:mol with an average value of $0.11 \pm 0.07$ mol:mol (n = 23, Table 4), highlighting a different behaviour of Fe between margins. This could be explained by the different nature of the sediments and/or different sediment conditions (e.g. redox, organic content).

Page 19, Line 17: Where is the predominance of diatoms?

→ We changed the sentence for clarification.

Page 19 Line 33, Page 20 Lines 1-3: These observations are consistent with the high TChl-*a* concentrations measured at the Newfoundland Margin and to a lesser extent at the Greenland Margin and the predominance of diatoms relative to other functional phytoplankton classes at both margins (Tonnard et al., in prep.).

Page 19, Line 21: I think this section is great, but you need to organise the ideas clearly. Now difficult to follow. Also name this section "nepheloid layers"

→ We changed the section title as suggested and reorganised the flow of ideas.

Page 19, Line 26: explain the criterion first. Explain briefly the PCA and the results you show in figure 11. Information is thrown in a little randomly. Please organise you paragraphs

→ We reorganised the paragraph as suggested.

Page 20 Lines 27-33, Page 21 Lines 1-4: Samples associated with high levels of particles (transmissometer < 99%) and below 500 m depth displayed a huge variability in DFe concentrations. From the entire dataset, 66 samples (~13% of the entire dataset) followed this criterion with 3 samples from the Iberian Margin (station 4), 14 samples from the West European Basin (station 1), 4 samples from the Iceland Basin (stations 29, 32, 36 and 38), 43 samples from the Irminger Sea (stations 40, 42, 44, 49 and 60) and 2 samples from the Labrador Sea (station 69). To determine which parameter was susceptible to explain the variation in DFe concentrations in these nepheloid layers, a Principal Component Analysis (PCA) on these samples. The input variables of the PCA were the particulate Fe, Al, and particulate manganese (PMn) (Gourain et al., subm.), the DAl (Menzel Barraqueta et al., 2018) and the Apparent Oxygen Utilization (AOU) and were all correlated to DFe concentrations explaining all together 93% of the subset variance (Fig. 11). The first dimension of the PCA was represented by the PAl, PFe and PMn concentrations and explained 59.5% of the variance, while the second dimension was represented by the DAl and the AOU parameters, explaining 33.2% of the variance. The two sets of variables were nearly at right angle from each other, indicating no correlation between them.

Page 20, Line 1: I would not call it AOU "concentrations" find another way to express this

→ We changed "concentrations" for parameters (Page 20 Line 20)

Page 20, Line 13: you should look into non-reducing dissolution of lithogenic material. You are missing out on a big topic! Radic et al., 2011; Labatut et al., 2014; Abadie et al., 2017

→ We included this topic in the text as suggested.

Page 21 Lines 14-31: This observation challenges the traditional view of Fe oxidation with oxygen, either abiotically or microbially induced. Indeed, remineralisation can lower sediment oxygen concentrations, promoting reductive dissolution of PFe oxyhydroxides to DFe that can then diffuse across the sediment water interface as DFe(II) colloids (Homoky et al., 2011). Such processes will inevitably lead to rapid Fe removal through precipitation of nanoparticulate or colloidal Fe (oxyhydr)oxides, followed by aggregation or scavenging by larger particles (Boyd and Ellwood, 2010; Lohan and Bruland, 2008) unless complexion with Fe-binding organic ligands occurs (Batchelli et al., 2010; Gerringa et al., 2008). Previous work (e.g. Liu and Millero, 2002) has stated that it is only when sufficient organic matter and more specifically organic ligands are present in solution, that this sediment-derived DFe could remain in solution in excess of its inorganic solubility through organic complexation (Kondo and Moffett, 2015; Noble et al., 2012) or in suspension as colloids or nanoparticles (Raiswell and Canfield, 2012). There exist, however, another process that is favoured in oxic benthic boundary layers (BBL) with low organic matter degradation and/or low Fe oxides, which implies the dissolution of particles after resuspension, namely the non-reductive dissolution of sediment (Homoky et al., 2013; Radic et al., 2011). These higher oxygenated samples were located within DSOW, which mainly originate (75% of the overflow) from the Nordic Seas and the Arctic Ocean (Tanhua et al., 2005), in which the ultimate source of Fe was reported by Klunder et al. (2012) to come from Eurasian river waters. The major Arctic rivers were highlighted by Slagter et al. (2017) to be a source of Fe-binding organic ligands that are then further transported via the TPD across the Denmark Strait. It is thus more likely that the enhanced DFe concentrations measured within DSOW result from Fe-binding organic ligand complexation that were transported to the deep ocean as DSOW formed rather than the non-reductive dissolution of sediment.

Page 20, Line 14: "lead"

→ We changed the text accordingly (Page 21 Line 17).

Page 20, Line 16: instead of "these" use "this sediment-derived. . ."

➔ We have changed the text accordingly.

Page 20, Line 17: do not state this as facts... these are assumptions

➔ We have changed the text as suggested.

Page 20 line 34, Page 21 Lines 1-2: Previous work (e.g. Liu and Millero, 2002) has stated that it is only when sufficient organic matter and more specifically organic ligands are present in solution, that this sediment-derived DFe could remain in solution in excess of its inorganic solubility through organic complexation (Kondo and Moffett, 2015; Noble et al., 2012) or in suspension as colloids or nanoparticles (Raiswell and Canfield, 2012).

Page 20, Line 25: very poor sum-up, please improve.

Page 20, Line 26: You should also compare surface dFe data to AOU to look at dFe released from remineralisation. You have done this for >500 m depth, but it will be worth looking at this more closely below the surface mixed layer, where remineralisation occurs (below 100 m depth).

➔ We have included this just before Fe* section.

Page 23 Lines 9-31: The remineralisation of organic matter is a major source of macro and micronutrients in subsurface waters (from 50 to 250 m depth). Remineralisation is associated with the consumption of oxygen and therefore, Apparent Oxygen Utilization (AOU), can provide a quantitative estimate of the amount of material that has been remineralised. While no relationship was observed below 50 m depth for $NO_3^-$ or DFe and AOU considering all the stations, a significant correlation was found in the Subpolar gyre when removing the influence of margins (stations 29-49, 56, 60, 63-77) (AOU = 3.88 $NO_3^-$ − 39.32, $R^2$=0.79, n=69, p-value < 0.001). This correlation indicates that remineralisation of PON greatly translates into DIN and that $NO_3^-$ can be used as a good tracer for remineralisation in the studied area. Within these Subpolar gyre waters, there was a significant correlation between DFe and AOU (AOU = 22.6 DFe, $R^2$=0.34, n=53, p-value < 0.001). The open-ocean stations from Subpolar gyre also exhibited a good linear correlation between DFe and $NO_3^-$ ($R^2$=0.42, n=51, p-value < 0.05). The slope of the relationship, representing the typical remineralisation ratio, was $R_{Fe:N}$ = 0.07 ± 0.01 mmol mol$^{-1}$. The intercept of the regression line was -0.4 ± 0.2 nmol L$^{-1}$, reflecting possible excess of preformed $NO_3^-$ compare to DFe in these water masses. These significant correlations allow us to use the Fe* tracer to assess where DFe concentrations potentially limit phytoplankton growth by subtracting the contribution of organic matter remineralisation from the dissolved Fe pool, as defined by Rijkenberg et al. (2014) and Parekh et al. (2005) for $PO_4^{3-}$, and modified here for $NO_3^-$ as follow:

$$Fe^* = [DFe] - R_{Fe:N} \times [NO_3^-] \quad (eq. 4)$$

where $R_{Fe:N}$ refers to the average biological uptake ratio Fe over nitrogen, and [$NO_3^-$] refers to nitrate concentrations in seawater. Although in the following, we imposed a fixed biological $R_{Fe:N}$ of 0.05 mmol mol$^{-1}$, it is important to note that the biological uptake ratio of DFe:$NO_3^-$ is not likely to be constant. Indeed, this ratio has been found to range from 0.05 to 0.9 mmol mol$^{-1}$ depending on species (Ho et al., 2003; Sunda and Huntsman, 1995; Twining et al., 2004). The ratio we choose is thus less drastic to assess potential Fe limitation and more representative of the average biological uptake of DFe over $NO_3^-$ calculated for this study (i.e. $R_{Fe:N}$ = 0.07 ± 0.01 mmol mol$^{-1}$, for Subpolar waters). Negative values of Fe* indicate the removal of DFe that is faster than the input through remineralisation or external sources and positive values suggest input of DFe from external sources (Fig. 13).

Page 20, Line 30: change the ending of this sentence, not properly expressed

➜ We have split and changed the sentence.

Page 21 Lines 31-32, Page 22 Lines 1-2: Overall, almost all the stations from the GA01 voyage displayed DFe minima in surface water associated with maxima of TChl-*a* (Fig. 3). In the following section, we specifically addresses the question of whether DFe concentrations potentially limit phytoplankton growth. Note that macronutrients and DFe limitations relative to phytoplankton functional classes are dealt in Tonnard et al. (in prep.).

Page 20, Line 31: First explain why you talk about Fe:nitrate ratios. THis comes out of the blue! Also cite Fig 12.

➜ We have changed the text as suggested.

Page 22 Lines 18-20: A key determinant for assessing the significance of a DFe source is the magnitude of the DFe:macronutrient ratio supplied, since this term determines to which extent DFe will be utilised. The DFe:$NO_3^-$ ratios in surface waters varied from 0.02 (station 36) to 38.6 (station 61) mmol:mol with an average of 5 ± 10 mmol:mol (Fig. 12).

Page 21, Line 3: what do you mean by influence of the river, and the currents? specify what you mean

➜ We have changed the text for clarification.

Page 22 Lines 21-26: Although, the low nitrate concentrations observed at the eastern and western Greenland and Newfoundland Margins reflected a strong phytoplankton bloom which had reduced the concentrations as highlighted by the elevated integrated TChl-*a* concentrations ranging from 129.6 (station 78) to 398.3 (station 61) mg m$^{-2}$, at the Iberian Margin they likely reflected the influence of the N-limited Tagus River (stations 1, 2 and 4) with its low TChl-*a* integrated concentrations that ranged from 31.2 (station 1) to 46.4 (station 4) mg m$^{-2}$.

Page 21, Line 6: Can you provide a different kind of plot to help visualise this gradient you are talking about. In figure 12 this is impossible

➜ We have included a surface map of DFe:NO3- ratios.

[Figure]

DFe:NO3 [mmol:mol] @ depth=first

Page 21, Line 18: "disequilibrium" sounds a bit odd, better use the word "ranges"

➔ We have changed the text as suggested (Page 23 Line 8).

Page 21, Line 23: do you assume that all the nitrate in seawater comes from remineralisation? Better explain what assumptions this equation relies on

➔ We have modified the text accordingly (see above the response to your comment "Page 20, Line 26")

Page 21, Line 26: Rather, negative values of Fe* indicate the removal of dFe that is faster than the input through remineralisation or external sources and positive values suggest input of dFe from external sources

➔ We have changed the text as suggested.

Page 21, Line 27: remove "out"

➔ We have changed the text as suggested (Page 22 Line 31).

Page 21, Line 28: you talk about surface waters here but the calculations are done below 100 m depth. I would keep discussion on the external sources of dFe and then link to inputs of dFe rich water masses to surface waters above

➔ We have reorganised the section according to your comments and Reviewer#1 comments. Therefore we only focus the discussion on the top 200 m depth of the section.

[revised manuscript text omitted]

Page 22, Line 5: what has the low Fe supply to do with the "inefficient" carbon pump? If you want to talk about the carbon pump, and its inefficiency, you need to support with adequate statements/findings. I do not think this is a finding of your study

➔ We have removed this part (see above).

Page 22, Line 7: this comes a little out of the blue. Explain a little more the high remineralised carbon fluxes and how they were measured

➔ We replaced in context this sentence and the precision on the measurement of remineralised carbon fluxes is now precised earlier in the MS.

Page 22, Line 12: poor ending. What about other sources? Margins? Rivers?

➔ We have changed the text (see above).

Page 22, Line 15: first sentence superfluous

➔ We have removed this sentence as suggested (Page 22 Line 31).

Page 22, Line 23: depletion of nitrate? That doesn't make sense

➔ We have changed the sentence for clarification.

Page 23, Lines 21-24: Indeed, the intense wind-forcing of deep convection occurring in the Irminger Sea enables the LSW with its enhanced DFe concentrations to reach surface waters, thus initially sustaining intense phytoplankton growth during spring, but which will potentially limit the biological activity later on in the season due to its relative depletion compared to $NO_3^-$ as indicated by Fe*.

Page 22, Line 27: "entrained it to the deep. . ."

➔ We have changed the text as suggested (Page 23 Line 28).

Page 22, Line 29: "in the deep ocean"

➔ We have changed the text as suggested (Page 23 Line 30).

Page 22, Line 30: do you mean particles? Sediments are on the seafloor. Same for line 32

➔ Yes you are right, we have changed "sediments" for "particles".

Page 23, Line 3: conclusions need to be rewritten after the discussion is reworked.

➔ We have modified the conclusion to be more specific

Page 24 Lines 11-32, Page 25 Lines 1-7: The DFe concentrations measured during this study were in good agreement with previous studies that spanned the West European Basin. However, within the Irminger Basin the DFe concentrations measured during this study were up to 3 times higher than the ones measured by Rijkenberg et al. (2014) in deep waters (> 1000 m depth) that was likely explained by the different water masses encountered (i.e. the Polar Intermediate Water, ~ 2800 m depth) and by a stronger signal of the Iceland Scotland Overflow Water (ISOW) from 1200 to 2300 m depth. This corresponded to the most striking feature of the whole section with DFe concentrations reaching up to 2.5 nmol L$^{-1}$ within the ISOW, Denmark Strait Overflow Water (DSOW) and Labrador Sea Water (LSW), three water masses that are part of the Deep Western Boundary Current and was likely the result of a lateral advection of particles in the Irminger However, as these water masses reached the Labrador Sea, lower DFe levels were measured. These differences could be explained by different processes occurring within the benthic nepheloid layers, where DFe was sometimes trapped onto particles due to Mn-sediment within the Labrador Sea (Gourain et al., subm.) and sometimes released from the sediment potentially as a result of interactions with dissolved organic matter. Such Fe-binding organic ligands could have also be produced locally due to the intense remineralisation rate reported by Lemaître et al. (2017) of biogenic particles (Boyd et al., 2010; Gourain et al., subm.). The LSW exhibited increasing DFe concentrations along its flow path, likely resulting from sediment inputs at the Newfoundland Margin. Although DFe inputs through hydrothermal activity were expected at the slow spreading Reykjanes Ridge (Baker and German, 2004b; German et al., 1994),

our data did not evidence this specific source as previously pointed by Achterberg et al. (2018) further north (~60°N) from our section.

[revised manuscript text omitted]

---

## Author Comment (AC3) · 19 Sep 2018

Dear Christian Schlosser and Editor, First of all thank you so much for your feedback that greatly improved the manuscript. Please note that the good version of our answer to your comments is the one below that refers to the new version of the manuscript. All the best, Tonnard et al.

Please also note the supplement to this comment:
https://www.biogeosciences-discuss.net/bg-2018-147/bg-2018-147-AC3-supplement.pdf

[Figure]

**Supplement:**

**Christian Schlosser reviews**

Dear Christian Schlosser and Editor,

The reviewers are thanked for their insightful comments; these have helped to improve the manuscript considerably. Please see our detailed answers to the referees' comments below. Line numbers refer to the new version.

All the answers are attached as a supplementary file.

Best regards,

Tonnard et al.,

**Abstract**

Please, note that we added François Lacan as a co-author.

Line 29ff: Air-sea interactions responsible for deep winter convection – Did you mean special cooling!

➔ We have changed the sentence "Air-sea interactions were suspected to be responsible for the increase in DFe concentrations within subsurface waters of the Irminger Sea due to deep convection occurring the previous winter…," by "Enhanced air-sea interactions were suspected to be responsible for the increase in DFe concentrations within subsurface waters of the Irminger Sea due to deep convection occurring the previous winter…"

**Introduction**

Page 2 Line 4: I would also include oxygen, the whole ventilation and redox state of the deep ocean depends on deep water formation in the North Atlantic and Weddell Sea

➔ We removed this sentence.

Page 2 Line 6: "stores" is maybe the wrong term; I would rather go with "accumulates"

➔ We have modified this part based on your comment and Reviewer#2 comment.

[revised manuscript text omitted]

**MM**

Page 3 Line 11: Remove "the" from ". . .aboard the N/O. . ."

➔ We have modified this part as suggested

Page 3 Line 24ff: Two different filtration techniques were applied, 0.2 and 0.45 um. Did you test that both approaches deliver the same result? I know water is restricted and sometimes sampling techniques need to be changed, however, please indicate why you did this and that swapping between both filtration techniques did not cause problems (offset, etc.)!

➔ We have added precision as suggested. Please note that there was no station where both filtration techniques were used.

Page 4 Lines 10-14: Samples were either taken from the filtrate of particulate samples (collected on polyethersulfone filters, 0.45 µm supor®, see Gourain et al., this issue) or after filtration using 0.2 µm filter cartridges (Sartorius SARTOBRAN® 300) due to water budget restriction (Table 1). No significant difference was observed between DFe values filtered through 0.2 µm and 0.45 µm filters (p-value > 0.2, Wilcoxon test) for most stations. Differences were only observed between profiles of stations 11 and 13 and, 13 and 15.

Page 3 Line 26: exchange "on" by "using". By the way, did you apply pressured air to the Go-Flo's to filter your samples. If so, please state that!

➔ We have added precision as suggested

Page 3 Line 30: You did you use 0.2% HCl to acidify your water, or? It reads like that! I assume you used concentrated HCl and the dilution with the seawater was than 0.2%.

➔ We have modified the text for clarification

Page 4 Lines 17-18: Samples were then acidified to ~ pH 1.7 with HCl (Ultrapur® Merck, 2 ‰ v/v) under a class 100 laminar flow hood inside the clean container.

Page 4 Line 2: The first sentence does not fit here; first you preconcentrated your sample using a SeaFAST system. Than the eluent was introduced via a PFA nebulizer and cyclonic spray chamber into your instrument (please indicate what kind of instrument you used, Element?). Please clarify!

➔ We have modified this part as suggested

Page 4 Lines 27-30: Seawater samples were preconcentrated using a Sea*FAST*-pico™ (ESI, Elemental Scientific, USA) and the eluent was directly introduced via a PFA-ST nebulizer and a cyclonic spray chamber in an Element XR Sector Field Inductively Coupled Plasma Mass Spectrometry (Element XR SF-ICP-MS, Thermo Fisher Scientific Inc., Omaha, NE), following the protocol of Lagerström et al. (2013).

**Note that we have also changed part of the reagent text, as we found out there were mistakes.**

Page 4 Lines 31-33, Page 5 Lines 1-4: High-purity grade solutions and water (Milli-Q) were used to prepare the following reagents: the acetic acid-ammonium acetate buffer ($CH3COO^-$ and $NH4^+$) was made of 140 mL acetic acid (> 99% NORMATOM® - VWR chemicals) and ammonium hydroxide (25%, Merck Suprapur®) in 500 mL PTFE bottles and was adjusted to pH 6.0 ± 0.2 for the on-line pH adjustment of the samples. The eluent was made of 1.4 M nitric acid ($HNO_3$, Merck Ultrapur®) in Milli-Q water by a 10-fold dilution and spiked with 1 µg $L^{-1}$ $^{115}$In (SCP Science calibration standards) to allow for drift correction. Autosampler and column rinsing solutions were made of $HNO_3$ 2.5% (v/v) (Merck Suprapur®) in Milli-Q water. The carrier solution driven by the syringe pumps to move the sample and buffer through the flow injection system was made in the same way.

Page 4 Line 11: gravimetrically is perhaps not the right word, you used a balance, right!

➔ We did not changed our sentence here since gravimetrically means by weighting the standards.

Page 4 Line 13ff: please include "..in-house standard seawater..", was this seawater acidified in the same way?

➔ We have modified this part as suggested and added at the end of section 2.1 the precision on the sampling and acidification methods of the in-house standard seawater.

In section 2.1:

Page 4 lines 20-25: Large volumes of seawater sample (referred hereafter as the in-house standard seawater) were also collected using a towed fish at around 2-3 m deep and filtered in-line inside a clean container through a 0.2 µm pore size filter capsule (Sartorius SARTOBRAN® 300) and was stored unacidified in 20-30 L LDPE carboys (Nalgene™). All the carboys were cleaned following the guidelines of the GEOTRACES Cookbook (Cutter et al., 2017). This in-house standard seawater was used for calibration on the Sea*FAST*-pico™ - SF-ICP-MS (see Section 2.2) and was acidified to ~ pH 1.7 with HCl (Ultrapur® Merck, 2 ‰ v/v) at least 24h prior to analysis.

Page 4 Line 16ff: Please include the analytical precision, the blank, detection limit of the analytical method. And how many samples did you run normally and how much samples were between each calibration curve? Please also include, how you calculated your errors, standard deviation of the three slopes? Or just the s.d. of the Element?

➔ We have added precision as suggested.

Page 5 Lines 11-20: Data were blank-corrected by subtracting an average acidified Milli-Q blank that was pre-concentrated on the Sea*FAST*-pico™ in the same way as the samples and seawater

**Moved down [1]:** And how many samples did you run normally and how much samples were between each calibration curve?

**Moved (insertion) [1]**

standards. Each analytical session consisted of about fifty samples and two calibrations, one at the beginning and another one at the end of each analytical session. The errors associated to each sample were calculated as the standard deviation for five measurements of low-Fe seawater samples. The mean Milli-Q blank was equal to $0.08 \pm 0.09$ nmol $L^{-1}$ (n = 17). The detection limit, calculated for a given run as 3 times the standard deviation of the Milli-Q blanks, was on average $0.05 \pm 0.05$ nmol $L^{-1}$ (n = 17). Reproducibility was assessed through the standard deviation of replicate samples (every 10th sample was a replicate) and the average of the in-house standard seawater, and was equal to 17% (n = 84). Accuracy was determined from the analysis of consensus (SAFe S, GSP) and certified (NASS-7) seawater matrices (see Table 2) and in-house standard seawater (DFe = $0.42 \pm 0.07$ nmol $L^{-1}$, n = 84).

Page 4 Line 21: The CTD sensors were deployed on a stainless steel rosette. Correct? Please indicate and correct throw-out the rest of the text.

➔ We have modified this part as suggested

Page 6 Lines 11-15: Potential temperature (θ), Salinity (S), dissolved oxygen ($O_2$) and beam attenuation data were retrieved from the CTD sensors (CTD SBE911 equipped with a SBE-43) that were deployed on a stainless steel rosette. Nutrient and pigment samples were obtained from the stainless steel rosette casts and analysed according to Aminot and Kerouel (2007) and Ras et al. (2008), respectively. We used the data from the stainless steel rosette casts that were deployed immediately before or after our TMR casts. All these data are/will be available on the LEFE/CYBER database (http://www.obs-vlfr.fr/proof/php/geovide/geovide.php).

Page 4 Line 28: Name the parameter Δσt

▪ We have added this precision (Page 6 Line 18): "…where $Z_m$ is defined as an absolute change in **the density of seawater at a given temperature** ($\Delta\sigma_\theta \geq 0.125$ kg m$^{-3}$)…"

Page 4 Line 30: What do you mean with perturbation, at which depth, please indicate in Table 1, for which station this was the case.

➔ We have changed the word "perturbation" by "disturbance" for clarification. In addition, we have reported in Table 1 the precision on whether temperature and salinity profiles were uniform or disturbed with an asterisk symbol next to stations where profiles were not uniform and we added the following sentence in the legend of Table 1: "Note that the asterisk next to station numbers refers to disturbed temperature and salinity profiles as opposed to uniform profiles."

Page 5 Line 2ff: Please indicate for which data you applied statists on?

➔ We have modified the text as suggested.

Page 6 Lines (23-24): All statistical approaches, namely the comparison between the pore size used for filtration, correlations and Principal Component Analysis (PCA), were performed using the R statistical software (R development Core Team 2012).

Page 5 Line 3: You did not measure the p-value, you maybe determined or calculated the value.

➔ We have modified this part as suggested (Page 6 Line 25)

Page 5 Line 15: Include ". . .540 data points. . .

➔ We have modified this part as suggested (Page 7 Lines 8-9)

Page 5 Line 19: Exchange "The complete relational database. . ." by " The complete data set. . ."

➔ We have modified this part as suggested (Page 7 Line 12)

**Results**

Page 5 Line 27: I would swap the two sentences "For a schematic of water masses, currents and pathways, see Daniault et al. (2016)." and "Hereafter we summarise the main features (Fig. 1 and 2)."

➔ We have modified this part as suggested (Page 7 Lines 19-20)

Page 6 Line 1: Give a depth range of the "Upper waters (0 – 800 m)" or so! Please also include this to the Intermediate and Deep waters.

➔ We have added precisions as suggested

Page 6 Line 5: Did you mean with central water the Subarctic intermediate water (SAIW). Please clarify! Please also increase the letter size in Fig. 2. It is really hard to see on a normal A4 print out! There are no currents in Fig. 2, either you somehow include them or remove the caption.

➔ By central waters, we meant ENACW as defined in the first sentence of the paragraph and therefore changed "Cnetral Waters" by "ENACW" . We have removed the currents from the figure caption in Fig. 2 and we have increased the font size.

Page 6 Line 18: Please rewrite "..Labrador Sea Water (LSW).

➔ We have modified this part as suggested (Page 8 Line 9)

Page 6 Line 29ff: I do not understand the sentence, starting with "During GA01,. . ."

➔ We have rewritten this part

Page 8 Lines 19-20: During GEOVIDE, LSW formed by deep convection the previous winter was found at several stations from the Labrador Sea (68, 69, 71 and 77).

Page 6 Line 30ff, I am not sure about, explaining the different flow paths, It is really hard to follow without any drawing. Other question, is it really important, since you are just interested in water masses and their DFe signal, and not about currents! I would remove that!

➔ We have modified this part to make it shorter.

Page 8 Lines 20-22: After convecting, LSW splits into three main branches with two main cores separated by the Reykjanes Ridge (stations 1-32, West European and Iceland Basins; stations 40-60, Irminger Sea), and the last one entering the West European Basin (Zunino et al., 2017).

Page 7 Line 8: I do not see any silicic acid and nitrate data, please indicate concentrations and where they can be found.

➔ We have added the averages and SD for silicic acid and nitrate concentrations (Page 8, Line 26) and the reference where data can be found.

Page 8 Lines 24-27: North East Atlantic Deep Water (NEADW, 1.98 < θ < 2.50°C, 34.895 < S < 34.940) was the dominant water mass in the West European Basin at stations 1-29 from 2000 m depth to the
* * *
bottom and is characterized by high silicic acid ($42 \pm 4$ µmol $L^{-1}$), nitrate ($21.9 \pm 1.5$ µmol $L^{-1}$) concentrations and lower oxygen concentration ($O_2 \approx 252$ µmol $kg^{-1}$) (see Sarthou et al., 2018).

Page 7 Line 9ff: It is hard to understand what you mean with "PIW is in contact with the atmosphere once a year (?) during the time of winter convection.." All together there is a lot of water mass information, that can be found elsewhere in the special issue, I would rather shorten that part of the result section.

➔ We have modified this part to make it shorter

Page 8 Lines 28-32: Polar Intermediate Water (PIW, $\theta \approx 0°C$, $S \approx 34.65$) is a ventilated, dense, low-salinity water intrusion to the deep overflows within the Irminger and Labrador Seas that is formed at the Greenland shelf. PIW represents only a small contribution to the whole water mass pool (up to 27%) and was observed over the Greenland slope at stations 53 and 61 as well as in surface waters from station 63 (from 0 to ~ 200 m depth), in intermediate waters of stations 49, 60 and 63 (from ~ 500 to ~ 1500 m depth) and in bottom waters of stations 44, 68, 69, 71 and 77 with a contribution higher than 10%.

Page 7 Line 30: Cannot check if this is correct! No nitrate data available.

➔ The location of these data was already precised "Sarthou et al. (this issue)" . However, we have changed this reference by the accurate one and added the reference of the SEANOE data base and associated paper: García-Ibáñez et al., 2018; Pérez et al., 2018; Sarthou et al., 2018 Please note that in this manuscript, Nitrate data are changed for RFe/N data, therefore we did not added the nitrate data.

Page 8 Line 11-12: This is school book knowledge, that is why we are using sensors! Remove the two sentences! However this entire section 3.2.2 needs an overhaul.

➔ We have removed this part as suggested.

Page 9 Lines 24-28: Overall, most of the phytoplankton biomass was localised above 100 m depth with lower total chlorophyll-*a* (TChl-*a*) concentrations South of the Subarctic Front and higher at higher latitudes (Fig. 3). While comparing TChl-*a* maxima considering all stations, the lowest value (0.35 mg $m^{-3}$) was measured within the West European Basin (station 19, 50 m depth) while the highest values were measured at the Greenland (up to 4.9 mg $m^{-3}$, 30 m depth, station 53 and up to 6.6 mg $m^{-3}$, 23 m depth, station 61) and Newfoundland (up to 9.6 mg $m^{-3}$, 30 m depth, station 78) margins.

Page 8 Line 20ff: You can delete the first three sentences, they do not contain any important data!

➔ We have removed this part as suggested

Page 8 Line 29: Also station 61 and 78 are high, at least this is shown by your plot!

➔ We were talking about enhanced DFe at the surface compared to deeper DFe values, and this is only the case for stations 2, 4 and 56.

And replace ". . .were around. . ." by ". . .ranged from. . ."

➔ We have modified the text as suggested.

Generally, I would merge section 3.3, 3.3.1 and 3.3.2.

➔ We have modified the text as suggested.

Figure 4: Are you sure that single elevated values at site 40 (1500m) and at site 44 (500m) are correct. They just seem like outliers to me! Do we really need Fig.5 and 6, we see everything already in Fig. 4.

➔ These enhanced DFe concentrations are consistent with high ligand concentrations measured during this study. In addition, these samples were analysed during 3 separated analytical sessions on the seaFAST SF-ICP-MS. Regarding Figs 5 and 6, we agree that Fig. 6 is repetitive and therefore we removed it from the ms and included it into the supplementary material. However, we think Fig. 5 is helpful to understand the section 4.2.2 on high latitudes meteoric water and sea-ice processes.

Page 9 Line 1ff: rewrite sentence, hard to read!

➔ We have rewritten the sentence

Page 10 Lines 7-9: Considering the four oceanic basins, mean vertical profiles (supplementary material Fig. S2) showed increasing DFe concentrations down to 3000 m depth followed by decreasing DFe concentrations down to the bottom. Among deep-water masses, the lowest DFe concentrations were measured in the West European Basin.

Page 9 Line 6ff: Please provide numbers for surface waters.

➔ We have added this precision.

Page 10 Lines 12-14: Overall, surface DFe concentrations were higher (0.36 ± 0.18 nmol L$^{-1}$) in the North Atlantic Subpolar gyre (above 52°N) than in the North Atlantic Subtropical gyre (0.17 ± 0.05 nmol L$^{-1}$).

Page Line 9ff: But also at station 21 the DFe value is high. I do not think they are significantelly different from the others, s.d. is ± 20% and higher.

➔ We agree with you and removed this sentence.

Page 9 Line 17: NEADW was very similar to the median GEOVIDE voyage but compared to test of deep waters lower, please rewrite! But the DSOW in the Labrador Sea was similar.

➔ We have modified the text accordingly

Page 11 Lines 8-11: The DFe concentrations in the NEADW were relatively similar to the DFe median value of the GEOVIDE voyage (median DFe = 0.75 nmol L$^{-1}$, Figs. 4 and 7) with an average value of 0.74 ± 0.16 nmol L$^{-1}$ (n=18) and presented relatively low median DFe concentrations (median DFe = 0.71 nmol L$^{-1}$) compared to other deep water masses.

I am not sure Fig. 7 is really required. It just comprises what we already sea in Fig. 4. And apart from some outliers (hydrothermal? Any Mn data), surface waters, NADW and waters from the Labrador Sea, concentrations are around 1nM. And as numerous times shown, it is impossible to fingerprint water masses with DFe.

➔ We agree with this suggestion and Figure 7 was removed from the MS and added to supplementary material.

Page 10 Line 9: Others showed also elevated concentrations, for instance, station 44. However I understand why the authors decided to explain both station! For myself station 1 is not a problem, it

is very close to the continental margin and influenced by lateral water mass transport than the other stations farther off-shore. However, site 17 is a bit more tricky. Did you reanalyze that station, that would confirm that the analysis was alright and you do not face just a strange offset. Anyway, I would discuss station 17, but please rephrase some sentences, it was really hard to grasp the issue you wanted to bring across. From the first sentence it should be clear what the issue is, than explain (eg. Concentrations are irregularly high).

➔ We have modified the text as suggested

Page 11 Lines 20-28: Considering the entire section, two stations (stations 1 and 17) showed irregularly high DFe concentrations (> 1 nmol L$^{-1}$) throughout the water column, thus suggesting analytical issues. However, these two stations were analysed twice and provided similar results, therefore discarding any analytical issues. This means that these high values originated either from genuine processes or from contamination issues. If there had been contamination issues, one would expect a more random distribution of DFe concentrations and less consistence throughout the water column. It thus appears that contamination issues were unlikely to happen. Similarly, the influence of water masses to explain these distributions was discarded as the observed high homogenized DFe concentrations were restricted to these two stations. Station 1, located at the continental shelf-break of the Iberian Margin, also showed enhanced PFe concentrations from lithogenic origin suggesting a margin source (Gourain et al., 2018).

Page 10 Line 23ff: Please provide the numbers from the other studies. Would it be possible to plot the surface DFe concentration and put the graph in the sup material. Than you can relate to that!

➔ We have provided the numbers from other studies, updated Table 3 with the DFe values from Achterberg et al., 2018 and we added the following plot to the supplementary material:

[Figure]

Figure S4: Surface layer of DFe concentrations, new measurements are shown in red dots (GEOVIDE voyage), while previous studies are displayed in black. (Achterberg et al., 2018; Bergquist et al., 2007; Blain et al., 2004; Boye et al., 2006, 2003; de Jong et al., 2007; Gledhill et al., 1998; Hatta et al., 2015; Klunder et al., 2012; Laës et al., 2003; Martin et al., 1993; Measures et al., 2008; Mills et al., 2008; Mohamed et al., 2011; Nédélec et al., 2007; Nielsdóttir et al., 2009; Pohl et al., 2011; Rijkenberg et al., 2014; Sarthou et al., 2007, 2003; Sedwick et al., 2005; Ussher et al., 2013; Witter and Luther III, 1998; Wu and Boyle, 2002; Wu and Luther III, 1996, 1994; Wu et al., 2001).

Field Code Changed

➔ We also changed the text as we made a mistake in this section. Indeed, low DFe concentrations were previously measured in the central Irminger Sea. When we first wrote this part we considered stations that were closed to land likely impacted by sea-ice melting.

Page 12 Lines 3-7: Among the four distinct basins described in this paper, the Irminger Sea exhibited the highest DFe concentrations within the surface waters (from 0 to 250 m depth) with values ranging from 0.23 to 1.3 nmol $L^{-1}$ for open-ocean stations. Conversely, low DFe concentrations were previously reported in the central Irminger Sea by Rijkenberg et al. (2014) (April-May, 2010) and Achterberg et al. (2018) (April-May and July-August, 2010) with DFe concentrations ranging from 0.11 to 0.15 and from ~ 0 to 0.14 nmol $L^{-1}$, respectively (see supplementary material Fig. S4 and Table S2).

Page 10 Line 29: Please include an opening sentence, what you think is the reason (something similar to the last sentence). It is quite a step from Fe distribution to the original of water mass mixing.

➔ We have added an opening sentence

Page 12 Lines 9-11: Indeed, enhanced surface DFe concentrations measured during GEOVIDE in the Irminger Sea could be due to intense wind forcing events that would deepen the winter $Z_m$ down to the core of the Fe-rich LSW.

Page 11 Line 5: Explain what tip jets are!

➔ We have added a definition

Page 12 Lines 19-22: Moore (2003) and Piron et al. (2016) described low-level westerly jets centred northeast of Cape Farewell, over the Irminger Sea, known as tip jet events, whose structure depends upon the splitting occurring as the flow encounter the orographic features from Cape Farewell, and that are strong enough to induce deep convective mixing (Bacon et al., 2003; Pickart et al., 2003).

Page 11 Line 10ff: This process is called winter entrainment (Tagliabue et al. 2014). Rephrase sentence and delete the last one (You just repeat yourself.

➔ We have modified the text as suggested

Page 12 Lines 26-28: Such winter entrainment was likely the process involved in the vertical supply of DFe within surface waters fuelling the spring phytoplankton bloom with DFe values close to those found in LSW.

Page 11 Line 16: Also contaminated waters are introduced!

➔ Yes, we completely agree. However, since this section is dedicated to atmospheric deposition we did not specify this.

Page 11 Line 18: What is a stratification period? Be preciss!

➔ We have changed the text

Page 12 Line 32: "During the summer, when thermal stratification occurs, …"

Page 11 Line 218ff: You can not compare the Mediterranean surface waters with MOW. Rewrite! DAl and DFe behave entirely different in the water column (residence time, organic complexation,

concentrations , etc.), but both of them are likely to be scavenged from particles. So when a dust storm hits, both elements should decrease, do they actualy do this in the water column of the Mediterranean sea. However, I am not too much surprised to see no DFe signal in the MOW. However, I suggest you have a look for DFe literature values from deep Mediterranean waters (GA04 is not available, a pitty).

➔ Yes, we agree that DAl and DFe behave differently in the water column depending on organic complexation and that they are both likely to be scavenged from particles. However, DAl and DFe originating from dust deposition are not scavenged by the same type of particles. Indeed, Wuttig et al. (2013) reported that after a single dust deposition event DAl loss rates was highly affected by the concentrations of biogenic particles while DFe was removed by sinking dust particles. The same authors highlighted that the following dust deposition event were likely inducing the dissolution of Fe from dust particles depending on the amounts of Fe-binding organic ligands. Therefore, both elements should not necessarily decrease.

➔ We have changed the text as suggested

Page 13 Lines 2-14: After atmospheric deposition, the fate of Fe will depend on the nature of aerosols, vertical mixing, biological uptake and scavenging processes (Bonnet and Guieu, 2006; Wuttig et al., 2013). During GEOVIDE, MOW was observed from stations 1 to 29 between 1000 and 1200 m depth and associated with high dissolved aluminium (DAl, Menzel Barraqueta et al., 2018) concentrations (up to 38.7 nmol L$^{-1}$), confirming the high atmospheric deposition in the Mediterranean region. In contrast to Al, no DFe signature was associated with MOW (Figs. 2 and 3). This feature was also reported in some studies (Hatta et al., 2015; Thuróczy et al., 2010), while others measured higher DFe concentrations in MOW (Gerringa et al., 2017; Sarthou et al., 2007). However, MOW coincides with the maximum Apparent Oxygen Utilization (AOU) and it is not possible to distinguish the MOW signal from the remineralisation one (Sarthou et al., 2007). On the other hand, differences between studies are likely originating from the intensity of atmospheric deposition and the nature of aerosols. Indeed, Wagener et al. (2010) highlighted that large dust deposition events can accelerate the export of Fe from the water column through scavenging. As a result, in seawater with high DFe concentrations and where high dust deposition occurs, a strong individual dust deposition event could act as a sink for DFe. It thus becomes less evident to observe a systematic high DFe signature in MOW despite dust inputs.

Page 12 Line 1: The entire section 4.1.3 is highly speculative. I agree elevated DFe in the Irminger Basin needs to come from somewhere, however, just looking at your Chl a data it is a very productive site, so presumably PFe concentrations are elevated as well, if so you should mention that, than it is just elevated remineralization and intense deep mixing during winter time that is responsible. However, you need to rewrite that section, to make it less speculative, look for existing data!

➔ We have changed the text as suggested.

Page 13 Lines 16-33 and Page 14 Lines 1-16: As described in Section 3.1, the LSW exhibited increasing DFe concentrations from its source area, the Labrador Sea, toward the other basins with the highest DFe concentrations observed within the Irminger Sea, suggesting that the water mass was enriched in DFe either locally in each basin or during its flow path (Fig. 7). These DFe sources could originate from a combination of high export of PFe and its remineralisation in the mesopelagic area and/or the dissolution of sediment.

The Irminger and Labrador Seas exhibited the highest averaged integrated TChl-a concentrations (98 ± 32 mg m$^{-2}$ and 59 ± 42 mg m$^{-2}$) compared to the West European and Iceland Basins (39 ± 10 mg m$^{-2}$ and 53 ± 16 mg m$^{-2}$), when the influence of margins was discarded. Stations located in the Irminger (stations 40-56) and Labrador (stations 63-77) Seas, were largely dominated by diatoms (>50% of phytoplankton abundances) and displayed the highest chlorophillid-*a* concentrations, a tracer of senescent diatom cells, likely reflecting post-bloom condition (Tonnard et al., in prep.). This is in line with the highest POC export data reported by Lemaitre et al. (2018) in these two oceanic basins. This likely suggests that biogenic PFe export was also higher in the Labrador and Irminger Seas than in the West European and Iceland Basins. Although, Gourain et al. (2018) highlighted a higher biogenic contribution for particles located in the Irminger and Labrador Seas with relatively high PFe:PAl ratios (0.44 ± 0.12 mol:mol and 0.38 ± 0.10 mol:mol, respectively) compared to particles from the West European and Iceland Basins (0.22 ± 0.10 and 0.38 ± 0.14 mol:mol, respectively, see Fig. 6 in Gourain et al., 2018), they reported no difference in PFe concentrations between the four oceanic basins (see Fig. 12A in Gourain et al., 2018) when the influence of margins was discarded, which likely highlighted the remineralisation of PFe within the Irminger and Labrador Seas. Indeed, Lemaître et al. (2017) reported higher remineralisation rates within the Labrador (up to 13 mmol C m$^{-2}$ d$^{-1}$) and Irminger Seas (up to 10 mmol C m$^{-2}$ d$^{-1}$) using the excess barium proxy (Dehairs et al., 1997), compared to the West European and Iceland Basins (ranging from 4 to 6 mmol C m$^{-2}$ d$^{-1}$). Therefore, the intense remineralisation rates measured in the Irminger and Labrador Seas likely resulted in enhanced DFe concentrations within LSW.

Higher DFe concentrations were, however, measured in the Irminger Sea compared to the Labrador Sea and coincided with lower transmissometer values (i.e. 98.0-98.5% vs. >99%), thus suggesting a particle load of the LSW. This could be explained by the reductive dissolution of Newfoundland Margin sediments. Indeed, Lambelet et al. (2016) reported high dissolved neodymium (Nd) concentrations (up to 18.5 pmol.kg$^{-1}$) within the LSW at the edge of the Newfoundland Margin (45.73°W, 51.82°N) as well as slightly lower Nd isotopic ratio values relative to those observed in the Irminger Sea. They suggested that this water mass had been in contact with sediments approximately within the last 30 years (Charette et al., 2015). Similarly, during GA03, Hatta et al. (2015) attributed the high DFe concentrations in the LSW to continental margin sediments. Consequently, it is also possible that the elevated DFe concentrations from the three LSW branches which entered the West European and Iceland Basins and Irminger Sea was supplied through sediment dissolution (Measures et al., 2013) along the LSW pathway.

The enhanced DFe concentrations measured in the Irminger Sea and within the LSW were thus likely attributed to the combination of higher productivity, POC export and remineralisation as well as a DFe supply from reductive dissolution of Newfoundland sediments to the LSW along its flow path.

Page 12 Line 25: the elevated concentration on station 44, is not this just a single point?

➔ No, the elevated DFe concentrations at station 44 concerned three data points.

Page 12 Line 26ff: Replace "above" by "at", and what are i) sediment inputs (these are particles), and ii) intrusion of an Fe-rich water mass, please be more specific!

➔ We have changed the sentence as suggested and added precision.

Page 14 Lines 19-20: "… i) vertical diffusion from local sediment, ii) lateral advection of a water mass displaying enhanced DFe concentrations, and iii) local dissolution of Fe from particles."

Page 12 Line 33: How often have you analyzed the samples below 2.500 m at site 44. For me this is just one outlier, the two other samples from cast 44 in Fig 8A are not that out of the range.

→ The full station 44 was analysed during two separated analytical sessions on the seaFAST SF-ICP-MS from different sampling bottles with a good agreement between results. Therefore, we do not think that this data point is an outlier.

Page 13 Line 10ff: Your argument is based on four data points, I could aslo put a straight line through, with a similar R2. However this entire paragraph is highly speculative! In an earlier paragraph you mention that DFe do not fingerprint different water masses, and now they do? You should remove this section!

→ We agree on the fact that the polynomial fitting could also be a linear fitting. However, with either a polynomial or a linear fitting on the 5 data points, the conclusion would be the same with apparently, the dissolution of Fe-rich particles.
→ We reformulated this section as we considered it too speculative.

Page 15 Lines 2-20: However, considering the short residence time of DFe and the circulation of water masses in the Irminger Sea, it is possible that instead of being attributed to one specific water mass, these enhanced DFe concentrations resulted from lateral advection of the deep waters. Figure 8B) shows the concentrations of both DFe and PFe for the mixing line between DSOW/PIW and ISOW at station 44 and considering 100% contribution of ISOW for the shallowest sample (2218 m depth) and of DSOW/PIW for the deepest (2915 m depth), as these were the main water masses. This figure shows increasing DFe concentrations as DSOW/PIW mixed with ISOW. In addition, Le Roy et al. (2018) reported for the GEOVIDE voyage at station 44 a deviation from the conservative behaviour of $^{226}$Ra reflecting an input of this tracer centred at 2500 m depth, likely highlighting diffusion from deep-sea sediments and coinciding with the highest DFe concentrations measured at this station. Although the transmissometer values were lower at the sediment interface than at 2500 m depth, Deng et al. (2018) reported a stronger scavenged component of the $^{230}$Th at the same depth range, likely suggesting that the mixture of water masses were in contact with highly reactive particles. If there is evidence that the enhanced DFe concentrations observed at station 44 coincided with lateral advection of water masses that were in contact with particles, the difference of behaviour between DFe and $^{230}$Th remains unsolved. The only parameter that would explain without any ambiguity such differences of behaviour between DFe and $^{230}$Th would be the amounts of Fe-binding organic ligands for these samples. Indeed, although PFe concentrations decreased from the seafloor to the above seawater, this trend would likely be explained by a strong vertical diffusion alone and not necessarily from the dissolution of particles that were laterally advected.

Therefore, the high DFe concentrations observed might be inferred from local processes as ISOW mixes with both PIW and DSOW with a substantial load of Fe-rich particles that might have dissolved in solution due to Fe-binding organic ligands.

Page 13 Line 22: unpublished sources? You need to explain that! Did you look through your Mn and Pb data, when they are also high, we talk about a hydrothermal input of trace metals.

→ We have changed the text for clarification

Page 15 Lines 22-33 and page 16 Lines 1-12: Hydrothermal activity was assessed over the Mid Atlantic Ridge, namely the Reykjanes Ridge, from stations 36 to 42. Indeed, within the interridge database (http://www.interridge.org), the Reykjanes Ridge is reported to have active hydrothermal

sites that were either confirmed (Baker and German, 2004a; German et al., 1994; Olaffson et al., 1991; Palmer et al., 1995) close to Iceland or inferred (e.g. Chen, 2003; Crane et al., 1997; German et al., 1994; Sinha et al., 1997; Smallwood and White, 1998) closer to the GEOVIDE section as no plume was detected but a high backscatter was reported potentially corresponding to a lava flow. Therefore, hydrothermal activity at the sampling sites remains unclear with no elevated DFe concentrations or temperature anomaly above the ridge (station 38). However, enhanced DFe concentrations (up to $1.5 \pm 0.22$ nmol L$^{-1}$, station 36, 2200 m depth) were measured east of the Reykjanes Ridge (Fig. 4). This could be due to hydrothermal activity and resuspension of sunken particles at sites located North of the section and transported through the ISOW towards the section (Fig. 7). Indeed, Achterberg et al. (2018) highlighted at ~60°N and over the Reykjanes Ridge a southward lateral transport of an Fe plume of up to 250-300 km. In agreement with these observations, previous studies (e.g. Fagel et al., 1996; Fagel et al., 2001; Lackschewitz et al., 1996; Parra et al., 1985) reported marine sediment mineral clays in the Iceland Basin largely dominated by smectite (> 60%), a tracer of hydrothermal alteration of basaltic volcanic materials (Fagel et al., 2001; Tréguer and De La Rocha, 2013). Hence, the high DFe concentrations measured east of the Reykjanes Ridge could be due to a hydrothermal source and/or the resuspension of particles and their subsequent dissolution.

West of the Reykjanes Ridge, a DFe-enrichment was also observed in ISOW within the Irminger Sea (Figs. 4 and 7). The low transmissometer values within ISOW in the Irminger Sea compared to the Iceland Basin suggest a particle load. These particles could come from the Charlie Gibbs Fracture Zone (CGFZ, 52.67°N and 34.61°W) and potentially Bight Fracture Zone (BFZ, 56.91°N and 32.74°W) (Fig. 1) (Lackschewitz et al., 1996; Zou et al., 2017). Indeed, hydrographic sections of the northern valley of the CGFZ showed that below 2000 m depth the passage through the Mid-Atlantic Ridge was mainly filled with the ISOW (Kissel et al., 2009; Shor et al., 1980). Shor et al. (1980) highlighted a total westward transport across the sill, below 2000 m depth of about $2.4 \times 10^6$ m$^3$ s$^{-1}$ with ISOW carrying a significant load of suspended sediment (25 µg L$^{-1}$), including a 100-m-thick benthic nepheloid layer. It thus appears that the increase in DFe within ISOW likely came from sediment resuspension and dissolution as the ISOW flows across CGFZ and BFZ.

➔ Note that for Pb, no particular hydrothermal signal was observed during GEOVIDE (Zurbrick et al., 2018). For Mn, data are analysed but not yet processed.

Page 14 Line 3ff: Theer are no elevated DFe values farther east from the ridge!

➔ We have changed the text for clarification (see above). The DFe enrichment east of Reykjanes Ridge corresponded to the section on top of this sentence while further downstream corresponded to west of the Reykjanes Ridge.

Where is the CGFC and BFC. Questions over questions!

➔ These two features are now added to Fig. 1

[Figure]

**Figure 1: Map of the GEOTRACES GA01 voyage plotted on bathymetry as well as the major topographical features and main basins. Crossover station with GEOTRACES voyage (GA03) is shown as a red star. (Ocean Data View (ODV) software, version 4.7.6, R. Schlitzer, http://odv.awi.de, 2016). BFZ: Bight Fracture Zone, CGFZ: Charlie-Gibbs Fracture Zone.**

Page 14 Line 13ff: I am confused. Do we talk about station 40 and 1.75nM at 1500 m, this is a single high value for me, and not located in ISOW waters.

➔ Yes, we talk about station 40 (one point) and station 42 (three points) that are all located in the ISOW (see Fig. 7).

Page 14 Line 26ff: The DFe/DAl ratio in seawater can not compared with the Fe/Al ratio of dust particles. Both elements have different fractional solubility's. So the ratio is always different! Remove!

➔ We agree with the reviewer and have changed the text as suggested and have added some information

Page 16 lines 25-28: Our SML DFe inventories were about three times higher at station 1 (~ 1 nmol L$^{-1}$) than those calculated during the GA03 voyage (~ 0.3 nmol L$^{-1}$, station 1) during which atmospheric

deposition were about one order of magnitude higher (Shelley et al., 2018; Shelley et al., 2015), the atmospheric source seemed to be minor.

Page 15 Line 1: Remove most of them does not add to the story!

➔ We agree and have removed most of them.

Page 16 Line 32 and Page 17 Lines 1-2: Many types of industry (e.g. heavy metallurgy, ore processing, chemical industry) release metals including Fe, which therefore result in high levels recorded in surface sediments, suspended particulate matter, water and organisms in the lower estuary (Santos-Echeandia et al., 2010).

Page 15 Line 5: What do you mean with "..below ground biomass.." In general I do not understand, why you excluded sediments, that could be an additional source.

➔ We did not intend to exclude the sediment source and have change the text for clarification.

Page 16 lines 28-30 Consequently, the Tagus River appears to be the most likely source responsible for these enhanced DFe concentrations, either as direct input of DFe or indirectly through Fe-rich sediment carried by the Tagus River and their subsequent dissolution.

Page 15 Line 14ff: Fronts refer to temperature and salinity changes in surface waters, such as the Polar Front, not in the water column. Call it different; just use the term "fresh water lens". Why multi-year-sea ice?

➔ We have changed the text as suggested

Page 17 Lines 9-10: The presence of this freshwater lens suggests that sediment derived enrichment to these surface waters was unlikely.

➔ We talked about multiyear sea ice because of drainage processes and the release of brines (see below)

Page 15 Line 18ff: But glacial sources and land ice sheet is the same, just call, it " . . .freshwater induced by meteoric water and sea-ice melt." Than all is clear.

➔ We have made the correction as suggested (Page17 Lines 10-11)

Page 15 Line 27: Where do get the sea-ice fractions from, and explain how it works, include references! And what have brines to do with it, either ice forms or not! Brines are not part of your story, so far I can tell. Brines always from when sea-ice is formed, or in the desert by evaporation. And in line 31 you switch back to sea-ice formation, please stay with that term.

➔ We have included a section in the method on how these fractions were calculated.

Page 5 Lines 23-32, Page 6 Lines 1-9: We separated the mass contributions to samples from stations 53, 61 and 78 in Sea-Ice Melt (SIM) Meteoric Water (MW) and saline seawater inputs using the procedure and mass balance calculations that are fully described in Benetti et al. (2016) (Fig. 5D), E) and F)). Hereafter, we describe briefly the principle. We considered two types of seawater, namely the Atlantic Water (AW) and the Pacific Water (PW). After estimating the relative proportions of AW ($f_{AW}$) and PW ($f_{PW}$) and their respective salinity and $\delta^{18}$O affecting each samples, the contribution of SIM and MW can be determined using measured salinity ($S_m$) and $\delta^{18}$O ($\delta O_m^{18}$). The mass balance calculations are presented below:

$$f_{AW} + f_{PW} + f_{MW} + f_{SIM} = 1 \text{ (eq.1)}$$

$$f_{AW} \cdot S_{AW} + f_{PW} \cdot S_{PW} + f_{MW} \cdot S_{MW} + f_{SIM} \cdot S_{SIM} = S_m \text{ (eq.2)}$$

$$f_{AW} \cdot \delta O_{AW}^{18} + f_{PW} \cdot \delta O_{PW}^{18} + f_{MW} \cdot \delta O_{MW}^{18} + f_{SIM} \cdot \delta O_{SIM}^{18} = \delta O_m^{18} \text{ (eq.3)}$$

where $f_{AW}$, $f_{PW}$, $f_{MW}$, $f_{SIM}$ are the relative fraction of AW, PW, MW, and SIM. To calculate the relative fractions of AW, PW, MW and SIM we used the following end-members: $S_{AW}$ = 35, $\delta O_{AW}^{18}$ = +0.18‰ (Benetti et al., 2016); $S_{PW}$ = 32.5, $\delta O_{PW}^{18}$ = -1‰ (Cooper et al., 1997; Woodgate and Aagaard, 2005); $S_{MW}$ = 0, $\delta O_{MW}^{18}$ = -18.4‰ (Cooper et al., 2008); $S_{SIM}$ = 4, $\delta O_{SIM}^{18}$ = +0.5‰ (Melling and Moore, 1995).

In Figure 5 D), E) and F), negative sea-ice fractions indicated a net brine release while positive sea-ice fractions indicated a net sea-ice melting. Note that for stations over the Greenland Shelf, we assumed that the Pacific Water (PW) contribution was negligible for the calculations, supported by the very low PW fractions found at Cape Farewell in May 2014 (see Figure B1 in Benetti et al., 2017), while for station 78, located on the Newfoundland shelf, we used nutrient measurements to calculate the PW fractions, following the approach from Jones et al. (1998) (the data are published in Benetti et al., 2017).

➔ Regarding brines, they can originate from two different processes: either as a result of multiyear sea-ice melting or during sea-ice formation. Indeed, during the early melting season, multiyear sea-ice has a higher porosity and gravitational drainage of brine occur. These two processes of brine release might lead to different TM signatures in brine originating from sea-ice formation and brine originating from early melting of multiyear sea-ice (Petrich and Eicken, 2010; Wadhams, 2000).

Page 15 Line 33: But brines usually sink, because they are heavier than the surrounding water!!! It is really hard to follow your argumentation here.

We agree with the reviewer in the fact that brines sink due to higher density. However, after reaching neutral buoyancy, they will stop sinking.

Page 16 Line 11: You have to explain how you produced these numbers, a citation in an earlier paragraph is not enough!

➔ We have included a section in the method on how these fractions were calculated (see above)

Page 16 Line 15: How do you lose a sample! Generally fist you talk about the contribution of MW and then you switch to biological uptake of DFe, that in the same paragraph? You lose the reader here; this entire section needs an overhaul.

➔ We have reorganised this section.

Page 18 Lines 3-18: Surface waters (from 0 to ~ 100 m depth) from station 53 and 61 were characterized by high MW fractions (ranging from 8.3 to 7.4% and from 7.7 to 7.3% , respectively, from surface to ~100 m depth, Figs. 5D and E). These high MW fractions were both enriched in PFe and DFe (except station 53 for which no data was available close to the surface) compared to seawater located below 50 m depth, thus suggesting a MW source. These results are in line with previous observations, which highlighted strong inputs of DFe from a meteoric water melting source in Antarctica (Annett et al., 2015). Although the ability of MW from Greenland Ice Sheet and runoffs to deliver DFe and PFe to surrounding waters has previously been demonstrated (Bhatia et al., 2013; Hawkings et al., 2014; Schroth et al., 2014; Statham et al., 2008), both Fe fractions were lower at the sample closest to the surface, then reached a maximum at ~ 50 m depth and decreased at ~ 70 m depth, for station 61 (Fig. 4D). The surface DFe depletion was likely explained by phytoplankton uptake, as indicated by the high TChl-$a$ concentrations (up to 6.6 mg m$^{-3}$) measured from surface to

about 40 m depth, drastically decreasing at ~ 50 m depth to 3.9 mg m$^{-3}$ (Fig. 4D). Hence, it seemed that meteoric water inputs from the Greenland Margin likely fertilized surface waters with DFe, enabling the phytoplankton bloom to subsist. The profile of PFe can be explained by two opposite plausible hypotheses: 1) MW inputs did not released PFe, as if it was the case, one should expect higher PFe concentrations at the surface (~25 m depth) than the one measured at 50 m depth due to both the release from MW and the assimilation of DFe by phytoplankton 2) MW inputs can release PFe in a form that is directly accessible to phytoplankton with subsequent export of PFe as phytoplankton died. The latter solution explains the PFe maximum measured at ~ 50 m depth and is thus the most plausible.

Page 16 Line 32: ".. decreasing from surface to depth." Which depth, down to the bottom in 400 m depth? Be precise

➔ We have added this precision.

Page 18 Lines 21-22: Newfoundland shelf waters (station 78) were characterized by high MW fractions (up to 7%), decreasing from surface to 200 m depth (~2%).

Page 17 Line 15-25: What has the tropical and subtropical North Atlantic to do with your work! I assume very little, please delete or at least reduce the text.

➔ We agree with the reviewer and removed the part on tropical North Atlantic.

Page 18 Lines 32-33 and Page 19 Lines 1-4: On a regional scale, the North Atlantic basin receives the largest amount of atmospheric inputs due to its proximity to the Saharan Desert (Jickells et al., 2005), yet even in this region of high atmospheric deposition, inputs are not evenly distributed. Indeed, aerosol Fe loading measured during GEOVIDE (Shelley et al., 2017) were much lower (up to four orders of magnitude) than those measured during studies from lower latitudes in the North Atlantic (e.g. Baker et al., 2013; Buck et al., 2010; and for GA03, Shelley et al., 2015), but atmospheric inputs could still be an important source of Fe in areas far from land.

Page 17 Line 30: I would rather suggest to say: "Shelley et al. concluded that. . ." because without any trajectories here I can check, and more or less all this work was already published.

➔ We have removed this part.

Page 18 Line 13: Do you mean DOM? Or organic material OM. However, you talk about DOM for 7 lines, and then you don't have the data. Once sentence should be enough to point out the importance of DOM.

➔ We have removed this part.

Page 18 Line 16: This entire paragraph is very poor! It is interesting to compare elemental ratios of seawater with the soluble fraction of dust. But the reasoning here ". . .whether there was enough atmospheric input to sustain the SML DFe concentrations. . ." without any flux numbers, residence times is unscientific. Even more strange, at the end of the paragraph you don t even say, whether there is enough or not. Similar to the above, this needs serious work to make it worthwhile reading. There is too much hand waving, and too few data, sorry! I suggest you look up the actual flux numbers and then compare them with your data.

➔ We agree with the reviewer and removed this section to replace it by Turnover Times relative to Atmospheric Deposition (TTADs) as defined in Guieu et al. (2014).

Page 19 Lines 5-17:. In an attempt to estimate whether there was enough atmospheric input to sustain the SML DFe concentrations, we calculated Turnover Times relative to Atmospheric Deposition (TTADs, Guieu et al., 2014). To do so, we made the following assumptions: 1) the aerosol concentrations are a snapshot in time but are representative of the study region, 2) the aerosol solubility estimates based on two sequential leaches are an upper limit of the aerosol Fe in seawater and 3) the water column stratified just before the deposition of atmospheric inputs, so MLD DFe will reflect inputs from above. Thus, the TTADs were defined as the integrated DFe concentrations in the SML for each station divided by the contribution of soluble Fe contained in aerosols averaged per basin to the water volume of the SML. Although, TTADs were lower in the West European and Iceland Basins with an average of ~ 9 ± 3 months compared to other basins (7 ± 2 years and 5 ± 2 years for the Irminger and Labrador Seas, respectively) (Fig. 6) they were about three times higher than those reported for areas impacted by Saharan dust inputs (~ 3 months, Guieu et al., 2014). Therefore, the high TTADs measured in the Irminger and Labrador Seas and ranging from 2 to 15 years provided further evidence that atmospheric deposition were unlikely to supply Fe in sufficient quantity to be the main source of DFe (see Sections 4.2.1 and 4.3.2) while in the West European and Iceland Basins they played an additional source, perhaps the main source of Fe especially at station 36 which displayed TTAD of 3 months.

[Figure]

Figure 6: Plot of dissolved Fe (DFe) Turnover Times relative to Atmospheric Deposition (TTADs) calculated from soluble Fe contained in aerosols estimated from a two-stage sequential leach (UHP water, then 25% HAc, Shelley et al., this issue). Note that numbers on top of data points represent station numbers and that the colour coding refers to different region with in yellow, margin stations; in purple, the West European Basin; in blue, the Iceland Basin; in green, the Irminger Sea and in red,

the Labrador Sea. The numbers on top of the plot represent TTADs averaged for each oceanic basin and their standard deviation.

Page 19 Line 5ff: replace "on" by "in". And which similar pattern followed the station. Be precise! Sentence stating in Line 6 makes no sense, please rewrite!

➔ We have corrected the text and rephrased the next sentence as suggested.

Page 19 Lines 20-23: DFe concentration profiles from all coastal stations (stations 2, 4, 53, 56, 61 and 78) are reported in Figure 5. To avoid surface processes, only depths below 100 m depth will be considered in the following discussion. DFe and PFe followed a similar pattern at stations 2, 53, 56, and 78 with increasing concentrations towards the sediment, suggesting that either the sources of Fe supplied both Fe fractions (dissolved and particulate) or that PFe dissolution from sediments supplied DFe.

Page 19 Line 11: What has the composition of sediments to do with your PFe value? Nothing. . .

➔ We have changed the text for clarification.

Page 19 Lines 25-28: DFe:PFe ratios ranged from 0.01 (station 2, bottom sample) to 0.27 (station 4, ~ 400 m depth) mol:mol with an average value of $0.11 \pm 0.07$ mol:mol (n = 23, Table 4), highlighting a different behaviour of Fe between margins. This could be explained by the different nature of the sediments and/or different sediment conditions (e.g. redox, organic content).

Page 19 Line 15: "Intermediate behavior" of what? And then Chla? This paragraph is very hard to follow, what is the message you want to bring across, I can't tell!

➔ We removed this sentence for clarification.

Page 19 Lines 28-33: Based on particulate and dissolved Fe and dissolved Al data (Gourain et al., 2018; Menzel Barraqueta et al., 2018, Table 4), three main different types of margins were reported (Gourain et al., 2018) with the highest lithogenic contribution observed at the Iberian Margin (stations 2 and 4) and the highest biogenic contribution at the Newfoundland Margin (station 78). These observations are consistent with higher TChl-*a* concentrations measured at the Newfoundland Margin and to a lesser extent at the Greenland Margin and the predominance of diatoms relative to other functional phytoplankton classes at both margins (Tonnard et al., in prep.).

Page 19 Line 26: "respect"? I respect you as a person, but samples usually don't respect anything?

➔ We have changed the word "respected" by "followed".

Page 19 Line 30ff: How do you know its manganese oxide, just use particulate Mn. And why you do not include the transmissometer data. That is what you wanted to show, or not that resuspended sediments control you particulate fraction.

➔ We agree with the reviewer in the way that we are not sure these are manganese oxides as they were estimated as the fraction from the PMn that was not originating from a lithogenic fraction using Mn:Ti UCC ratio. Therefore, a biological source or a co-precipitation source without oxidation were not considered. We thus agree with the reviewer and we have changed the MnOx data towards PMn data in the PCA.
➔ We did not include the transmissometer data as we do not have true values for all samples and used the interpolated data.

Page 20 Line 3ff: You did not do a PCA for dFe, so how can you be sure that the dim1 controls DFe? I cannot follow.

→ Before performing the PCA, a huge number of variables were considered and we only kept the one that were correlated to DFe to build the PCA.
→ We have changed the text for clarification.

Page 20 Lines 6-16: Samples associated with high levels of particles (transmissometer < 99%) and below 500 m depth displayed a huge variability in DFe concentrations. From the entire dataset, 66 samples (~13% of the entire dataset) followed this criterion with 3 samples from the Iberian Margin (station 4), 14 samples from the West European Basin (station 1), 4 samples from the Iceland Basin (stations 29, 32, 36 and 38), 43 samples from the Irminger Sea (stations 40, 42, 44, 49 and 60) and 2 samples from the Labrador Sea (station 69). To determine which parameter was susceptible to explain the variation in DFe concentrations in these nepheloid layers, a Principal Component Analysis (PCA) on these samples. The input variables of the PCA were the particulate Fe, Al, and particulate manganese (PMn) (Gourain et al., 2018), the DAl (Menzel Barraqueta et al., 2018) and the Apparent Oxygen Utilization (AOU) and were all correlated to DFe concentrations explaining all together 93% of the subset variance (Fig. 11). The first dimension of the PCA was represented by the PAl, PFe and PMn concentrations and explained 59.5% of the variance, while the second dimension was represented by the DAl and the AOU parameters, explaining 33.2% of the variance. The two sets of variables were nearly at right angle from each other, indicating no correlation between them.

Page 20 Line 24: You did not show any evident information that would suggest that DFe is controlled by OM (you did not even show any data) and PMn. Like the others this paragraph needs more work!

→ We have reorganised the section and added complementary information.

Page 20 Lines 6-34, Page 21 lines 1-6:

*4.4.2 Nepheloid layers:*

[revised manuscript text omitted]

Page 20 Line 27ff: Include "some" in front of "maxima, Please tell me the difference between "the relationship between DFe and biological uptake" and "Did DFe concentrations potentially limit phytoplankton growth?" This sounds to me very connected with each other! Why not discussing that in the follow up paper?

We have corrected the first sentence as suggested and re-wrote the end of the paragraph for clarification. Note that we wanted to keep this discussion in this paper as it summarises the different processes discussed in this MS.

Page 21 Lines 8-11: Overall, almost all the stations from the GEOVIDE voyage displayed DFe minima in surface water associated with some maxima of TChl-*a* (Fig. 3). In the following section, we specifically address the question of whether DFe concentrations potentially limit phytoplankton growth. Note that macronutrients and DFe limitations relative to phytoplankton functional classes are dealt in Tonnard et al. (in prep.).

Page 20 Line 31: Include mean or average Plus standard deviation

➔ We have included the average and SD and corrected few mistakes.

Page 21 Lines 13-14: The DFe:$NO_3^-$ ratios in surface waters varied from 0.02 (station 36) to 38.6 (station 61) mmol:mol with an average of 5 ± 10 mmol:mol (see supplementary material Fig. S7).

Page 21 Line 4: Please include numbers! Following the text in the paragraph, it is very hard to follow, you jump between F:N ratios, water masses and Chl a. Try to keep it short and weed out unnecessary details. Otherwise you will lose the reader!

➔ We have changed the text accordingly and added a surface map of DFe:NO3- ratios.

[revised manuscript text omitted]

Page 21 Line 20ff: Can you explain to me why you calculate Fe* for the entire water body (Fig. 13) and explain DFe limitation of the phytoplankton community. They live in the first 100-200 m. Same fro Fig.14.

→ We calculated the Fe* for the entire water body as water mass circulation and/or processes such as deep convection/upwelling, … can homogenized deep water masses with surface water masses. Thus, looking at DFe:NO3- ratios in these water masses appeared for us to be as important as just looking at the surface where phytoplankton live.

→ However, we understand the reviewer's opinion and decided to restrict this section to the top 250 meters. Consequently, we did a new plot for Fig. 13 with only the upper water column and removed Fig. 14.

[Figure]

Reading the last sentence of the section on page 22 "However, atmospheric loading (and especially Fe) was higher within the subtropical gyre than elsewhere in the GEOVIDE section mainly due to the proximity to mineral dust source (i.e. the Sahara Desert)." I feel I am still stuck in the Atmospheric chapter. Please shorten the paragraph and just say what you can prove with data.

→ We have removed the last two sentences to shorten the paragraph.

Page 22 Line 14ff: The entire conclusion needs an overhaul!

➔ We have modified the conclusion to be more specific.

Page 23 Lines 5-31 Page 24 Lines 1-2: The DFe concentrations measured during this study were in good agreement with previous studies that spanned the West European Basin. However, within the Irminger Basin the DFe concentrations measured during this study were up to 3 times higher than the ones measured by Rijkenberg et al. (2014) in deep waters (> 1000 m depth) that was likely explained by the different water masses encountered (i.e. the Polar Intermediate Water, ~ 2800 m depth) and by a stronger signal of the Iceland Scotland Overflow Water (ISOW) from 1200 to 2300 m depth. This corresponded to the most striking feature of the whole section with DFe concentrations reaching up to 2.5 nmol $L^{-1}$ within the ISOW, Denmark Strait Overflow Water (DSOW) and Labrador Sea Water (LSW), three water masses that are part of the Deep Western Boundary Current and was likely the result of a lateral advection of particles in the Irminger  However, as these water masses reached the Labrador Sea, lower DFe levels were measured. These differences could be explained by different processes occurring within the benthic nepheloid layers, where DFe was sometimes trapped onto particles due to Mn-sediment within the Labrador Sea (Gourain et al., 2018) and sometimes released from the sediment potentially as a result of interactions with dissolved organic matter. Such Fe-binding organic ligands could have also be produced locally due to the intense remineralisation rate reported by Lemaître et al. (2017) of biogenic particles (Boyd et al., 2010; Gourain et al., 2018). The LSW exhibited increasing DFe concentrations along its flow path, likely resulting from sediment inputs at the Newfoundland Margin. Although DFe inputs through hydrothermal activity were expected at the slow spreading Reykjanes Ridge (Baker and German, 2004b; German et al., 1994), our data did not evidence this specific source as previously pointed by Achterberg et al. (2018) further north (~60°N) from our section.

In surface waters several sources of DFe were highlighted especially close to lands, with riverine inputs from the Tagus River at the Iberian margin (Menzel Barraqueta et al., 2018) and meteoric inputs (including coastal runoff and glacial meltwater) at the Newfoundland and Greenland margins (Benetti et al., 2016). Substantial sediment inputs were observed at all margins but with different intensity. The highest DFe sediment input was located at the Newfoundland margin, while the lowest was observed at the eastern Greenland margin. These differences could be explained by the different nature of particles with the most lithogenic located at the Iberian margin and the most biogenic, at the Newfoundland margin (Gourain et al., 2018). Although previous studies (e.g. Jickells et al., 2005; Shelley et al., 2015) reported that atmospheric inputs substantially fertilized surface waters from the West European Basin, in our study only stations located in the West European and Iceland Basins exhibited enhanced SML DFe inventories with lower TTADs. However, these TTADs were about three times higher that those reported for Saharan dust inputs and thus atmospheric deposition appeared to be a minor source of Fe at the sampling period. Finally, there was evidence of convective inputs of the LSW to surface seawater caused by long tip jet event (Piron et al., 2016) that deepened the winter mixed layer down to ~ 1200 m depth (Zunino et al., 2017), in which Fe was in excess of nitrate and where thus Fe was not limiting at the sampling period.

**Figures:**

Figure 1: great;

Figure 2: increase letter size, it is hard to read;

Figure 3: I am not sure that white contour lines for DFe help to understand Chl a. I would remove DFe and include this figure in the supplementary material.

Figure 4, 5 and 6(?): great

Figure 7, 8, 9, 10: can go in the supplementary material, maybe Fig. 9 you can keep

Figure 11 -14: in the sup mat.,

Maybe Figure 13 for the first 200 m can stay!

Table 3 and 4: belongs into the sup. Material

➔ As you suggested we only kept Figs 1, 2, 4, 5, 9, 10 (the new one) and 13 (with your suggestions), all other Figures are now in the supplementary material. Tables 3 is also in the supplementary material.

[revised manuscript text omitted]

---

## Author Comment (AC4) · 19 Sep 2018

[revised manuscript text omitted]

Commented [MT1]: I need information on how long before analysis was the calibration seawater acidified how many samples were run per day and how much samples were between each calibration curve
Also need the way the errors were calculated: standard deviation of the calibration slopes or SD of the Element ?

[revised manuscript text omitted]

---

## Author Comment (AC5) · 19 Sep 2018

**Supplementary material**

[revised manuscript text omitted]

---

## Author Comment (AC6) · 19 Sep 2018

Dear Referee#2 and Editor, First of all thank you so much for your feedback that greatly improved the manuscript. Please note that the good version of our answer to your comments is the one below that refers to the new version of the manuscript. All the best, Tonnard et al.

Please also note the supplement to this comment:
https://www.biogeosciences-discuss.net/bg-2018-147/bg-2018-147-AC6-supplement.pdf

[Figure]

**Supplement:**

**Anonymous Referee#2 reviews**

Dear Referee#2 and  Editor,

The reviewers are thanked for their insightful comments; these have helped to improve the manuscript considerably. Please see our detailed answers to the referees' comments below. Line numbers refer to the new version.

All the answers are attached as a supplementary file.

Best regards,

Tonnard et al.,

Please, note that we added François Lacan as a co-author.

Page 1, Line 35: "in the Denmark Straight. . ."

➔ We have changed the text accordingly.

Page 1, Line 35: explain what types of particles you are talking about and briefly explain the differences observed (which ones scavenge and which ones release dFe)

➔ We have added precision

Page 1 Lines 35-37: Finally, the nepheloid layers located in the different basins and at the Iberian Margin were found to act as either a source or a sink of DFe depending on the nature of particles with organic particles likely releasing DFe and Fe-Mn oxides scavenging DFe.

Page2, Line 4: The reasoning is not flowing properly here. You need to say that (1) high productivity leads to high atmospheric carbon capture and that (2) Deep water formation leads to sequestration of this carbon into deeper waters, where carbon is stored for longer. The last sentence comes a little out of the blue, needs to be better linked – instead close the paragraph highlighting why it is important to study trace metals in this area.

➔ We have reorganised the full introduction as suggested.

Page 2 Lines 1-33, Page 3 Lines 1-25:

**1 Introduction**

[revised manuscript text omitted]

Page 2, Line10: I can not follow the reasoning in this paragraph. A little bit of a muddle of all the phytoplankton limiting factors (light, nutrients, wind, temp) without a clear insight what factor limits where. Needs to be better explained

➔ We have reorganised the full introduction as suggested (see above).

Page 2, Line 15: the connection between light limitation and nutrient limitation is not clearly explained

➔ We have reorganised the full introduction as suggested (see above).

Page 2, Line26: what about soluble Fe? Is this not considered the most bioavailable?

➔ Yes, we absolutely agree, Therefore to avoid confusion we have decided to remove the following sentence "…and within its physical speciation, its dissolved form (DFe) is considered to be the most available form for phytoplankton (Morel, 2008; Morel et al., 2008).".

Page 3, Line5: Aims of this paper are a bit poor. Add better understanding of the biogeochemical cycling of dFe in the oceans - inform biogeochemical models – and why this is important, what you expect to achieve. . ..

➔ We have reorganised the full introduction as suggested (see above).

Page 3, Line 20: remove "national"

➔ Removed

Page 3, line 30: do you mean concentrated HCl?

➔ We have changed the text for clarification

Page 4 Lines 17-18: Samples were then acidified to ~ pH 1.7 with HCl (Ultrapur® Merck, 2 ‰ v/v) under a class 100 laminar flow hood inside the clean container.

Page 3, Line25: why different filtering methods? Have you compared the Fe concentrations in those fractions? i.e., have you collected the same sample with both filtration cut-offs and checked there is no significant difference?

➔ We added precision for clarification. Note that we did not collected the same samples with the two different techniques.

Page 4, Lines 10-14: Samples were either taken from the filtrate of particulate samples (collected on polyethersulfone filters, 0.45 μm supor®, see Gourain et al., this issue) or after filtration using 0.2 μm filter cartridges (Sartorius SARTOBRAN® 300) due to water budget restriction (Table 1). No significant difference was observed between DFe values filtered through 0.2 μm and 0.45 μm filters (p-value > 0.2, Wilcoxon test) for most stations. Differences were only observed between profiles of stations 11 and 13 and, 13 and 15.

Page 4, Line 4: remove "daily basis"

➔ We have changed the text "High-purity grade solutions and water (Milli-Q) were used on a daily basis to prepare the following reagents:" by "High-purity grade solutions and water (Milli-Q) were used to prepare the following reagents each day:" (Page 4 Line 31)

Page 4, Line 14: replace "run" with "analytical session"

➔ We have changed the text as suggested (Page 5 Line 10)

Page 4, line 19: replace "in nmol L-1" with "to nmol L-1"

➔ We have changed the text as suggested (Page 5 Line 21)

Page 4, Line 19: would it not be more correct to multiply by the specific density of each sample? If you do not have this data (normally this is a standard parameter obtained from temperature and salinity.

➔ Yes, we agree with the reviewer. However, the converting factor used for consensus materials has always been 1.025 kg L$^{-1}$, which is why we used this converting factor.

Page 4, Line 19: Please show a comparison of dFe data at the crossover-station with GA02, as an intercalibration exercise.

➔ We did not include an intercalibration exercise with the GA02 voyage as DFe was determined on board by FIA-CL during GA02, which thus means that some refractory DFe was not measured on board with only a short time of acidification (Chever et al., 2010).

➔ Just for your information, here after you will find the plot comparing both voyages;

[Figure]

Page 4, Line25: Why did you not use the CTD data from the trace metal casts? Please explain

→ We did not use the CTD data from the trace metal casts as the O2 data could not be calibrated. We therefore decided to use all the parameters from the same rosette, i.e. the stainless steel rosette.

Page 5, Line 12: awkward sentence, difficult to follow, please rewrite!

→ We have changed the text for clarification

Page 7, Lines 4-5: Using this water mass determination, DFe concentrations were considered as representative of a specific water mass only when the contribution of this specific water mass was higher than 60% of the total water mass pool.

Page 6, Line 5: which central waters? Names?

→ The central waters are Eastern North Atlantic Central Waters (ENACW). The acronym was added in the text there page 7 line 26.

Page 6, Line 9: to keep consistency, keep "stations 49 and 60" out of the parenthesis. Rephrase the end of the sentence to do so.

→ We have modified the text as suggested

Page 7 Lines 27-30, Page 8 Lines 1: West of the Subarctic Front, Iceland SubPolar Mode Waters (IcSPMW, $7.07 < \theta < 8°C$, $35.16 < S < 35.23$, $280 < O_2 < 289$ µmol $kg^{-1}$) was encountered from stations 34-40 (accounting for more than 45% of the water mass pool from 0 to ~ 800 m depth) and Irminger SubPolar Mode Waters (IrSPMW, $\theta \approx 5°C$, $S \approx 35.014$) from stations 42-44 (contributing to 40% of the water mass pool from 0 to ~ 250 m depth) and stations 49 and 60 (accounting for 40% of the water mass pool down to 1300 m depth).

Page 6, Line10: specify which stations

→ We have changed the text as suggested.

Page 8 Lines 1-2: The IcSPMW was also observed within the Subtropical gyre (stations 11-26), subducted below ENACW until ~ 1000 m depth.

Page 6, Line 14: remove "The" from start of sentence

→ We changed the text accordingly.

Page 6, Line 19: what is that contribution? 40 %? Please specify!

→ We have changed the text as suggested.

Page 8 Lines 9-10: The LSW was also observed in surface waters of station 44 with a similar contribution than IrSPMW (~ 40%).

Page 6, Line 28: ... "was" sourced from...

→ We have corrected the text accordingly.

Page 6, Line 30: and "in the" Labrador Sea

→ We have corrected the text accordingly.

Page 6, Line 34: I am getting a little lost with all those branches, not sure when you're talking about the same one and when you change talking about another one. Not clear, please rephrase this section.

➔ We have changed the text for clarification.

Page 8 Lines 20-22: After convecting, LSW splits into three main branches with two main cores separated by the Reykjanes Ridge (stations 1-32, West European and Iceland Basins; stations 40-60, Irminger Sea), and the last one entering the West European Basin (Zunino et al., 2017).

Page 7, Line 5: delete "The" from beginning of sentence. In this entire section please remove "the" in front of water masses. The text should be revised by one of the English speaking co-authors before submission.

➔ We have changed the text accordingly

Page 7, Line7: "lower" oxygen. . .

➔ We have changed the text accordingly.

Page 7, Line 4: what do you mean by dense shelf? Do you mean the water masses have higher density? Please rephrase

➔ We have changed the text for clarification

Page 8 Lines 28-32: Polar Intermediate Water (PIW, θ ≈ 0°C, S ≈ 34.65) is a ventilated, dense, low-salinity water intrusion to the deep overflows within the Irminger and Labrador Seas that is formed at the Greenland shelf. PIW represents only a small contribution to the whole water mass pool (up to 27%) and was observed over the Greenland slope at stations 53 and 61 as well as in surface waters from station 63 (from 0 to ~ 200 m depth), in intermediate waters of stations 49, 60 and 63 (from ~ 500 to ~ 1500 m depth) and in bottom waters of stations 44, 68, 69, 71 and 77 with a contribution higher than 10%.

Page 7, Line 16: "the" Charlie Gibbs. . .

➔ We have changed the text as suggested.

Page 7, Line 19: mixing with... remove "of the overflow"

➔ These lines have been removed according to Referee #1 comments

Page 7, Line 21: Which stations?

➔ The stations numbers were added for the Iceland Basin.

Page 9 Lines 4-6: ISOW was observed from 1500 m depth to the bottom of the entire Iceland Basin (stations 29-38) and from 1800 to 3000 m depth within the Irminger Sea (stations 40-60)

Page 7, Line 30: At least put a nitrate section figure in the supplementary file; otherwise text hard to follow. Data not yet available on the site you referenced

➔ Data are now available on the Sarthou et al., 2018 paper. However, we have changed this reference by the accurate one and added the reference of the SEANOE data base and associated paper: García-Ibáñez et al., 2018; Pérez et al., 2018; Sarthou et al., 2018
Please note that in this manuscript, Nitrate data are changed for RFe/N data, therefore we did not added the nitrate data.

Page 8, Line 4: how do you define the "most open ocean station" for the transect? Deepest? Furthest away from land masses? Are you sure this is st 23?

➔ We have change the text for clarification and corrected the station number.

**Formatted**

**Formatted**

Page 9, Lines 16-17: The low surface $NO_3^-$ concentrations (lower than 6 µmol $L^{-1}$) in the West European Basin extended from station 2 (closest station to continental land mass) to station 25 (open ocean station) with concentrations ranging from 0.02 (station 11) to 3.9 (station 25) µmol $L^{-1}$.

Page 8, Line 4: it is unclear when you switch to talk about non-surface nitrate concentrations. Please rephrase this section to make this clearer.

➔ We have changed the section for clarification as suggested

Page 9, Lines 12-22: Surface nitrate ($NO_3^-$) concentrations (García-Ibáñez et al., 2018; Pérez et al., 2018; Sarthou et al., 2018) ranged from 0.01 to 10.1 µmol $L^{-1}$ (stations 53 and 63, respectively). There was considerable spatial variability in $NO_3^-$ surface distributions with high concentrations found in the Iceland Basin and Irminger Sea (higher than 6 µmol $L^{-1}$), as well as at stations 63 (10.1 µmol $L^{-1}$) and 64 (5.1 µmol $L^{-1}$), and low concentrations observed in the West European Basin, in the Labrador Sea and above continental margins. The low surface concentrations in the West European Basin ranged from 0.02 (station 11) to 3.9 (station 25) µmol $L^{-1}$. Station 26 delineating the extreme western boundary of the West European Basin exhibited enhanced $NO_3^-$ concentrations as a result of mixing between ENACW and IcSPMW, although these surface waters were dominated by ENACW. In the Labrador Sea (stations 68-78) low surface concentrations were observed with values ranging from 0.04 (station 68) to 1.8 (station 71) µmol $L^{-1}$. At depth, the lowest concentrations (lower than 15.9 µmol $L^{-1}$) were measured in ENACW (~ 0 - 800 m depth) and DSOW (> 1400 m depth), while the highest concentrations were measured within NEADW (up to 23.5 µmol $L^{-1}$), and in the mesopelagic zone of the West European and Iceland Basins (higher than 18.4 µmol $L^{-1}$).

Page 8, Line 6: which depths are you talking about?

➔ We added precision as suggested (see above)

Page 8, Line 12: isn't the fluorometer calibrated with the Chl-a measurements?

➔ We removed this part as suggested by Referee#1. However, the fluorometer was calibrated separately from the Chl-*a* measurements. To do so, 6 stations (including 6 different water masses) at 6 depths (surface, chlorophyll-max down to the base of the euphotic zone) were sampled. These samples included early morning, late evening and daytime samples to account for non-photochemical quenching, which causes a decrease of fluorescence signal at the surface during day-time. Therefore, all profiles were calibrated using night-time dependency and corrected day-time surface data for non-photochemical quenching.

Page 8, Line 13: Specify which depth range you are considering for looking at min/max Chl-a concentrations. Evidently, minimum Chl-a concentrations are found in the deep ocean

➔ As you said, the minimum Chl-*a* concentrations are found in the deep ocean and therefore we only reported min and max values for the maximum Chl-*a* concentrations considering all stations. We have changed the text for clarification

Page 9, Lines 24-28: Overall, most of the phytoplankton biomass was localised above 100 m depth with lower total chlorophyll-*a* (TChl-*a*) concentrations South of the Subarctic Front and higher at higher latitudes (Fig. 3). While comparing TChl-*a* maxima considering all stations, the lowest value (0.35 mg $m^{-3}$) was measured within the West European Basin (station 19, 50 m depth) while the highest values were measured at the Greenland (up to 4.9 mg $m^{-3}$, 30 m depth, station 53 and up to 6.6 mg $m^{-3}$, 23 m depth, station 61) and Newfoundland (up to 9.6 mg $m^{-3}$, 30 m depth, station 78) margins.

Page 8, Line 19: remind the reader here here that all the dFe data can be found in the supplementary file.

- → We have added this precision as suggested. Note that Referee#1 suggested to gather subsections 3.3.1 and 3.3.2, to remove the first lines from section 3.3 (Page 8 Lines 20-25) and to change section 3.3.3 for section 3.4, which is what we did.

Page 10 Lines 2-3: Dissolved Fe concentrations (see supplementary material) ranged from $0.09 \pm 0.01$ nmol L$^{-1}$ (station 19, 20 m depth) to $7.8 \pm 0.5$ nmol L$^{-1}$ (station 78, 371 m depth) (see Fig. 3).

Page 9, Line 13: when describing the regions, go in same order as in Figure, otherwise confusing (start Labrador Sea and end WEB)

- → We have made the corrections accordingly.

Page 10 Lines 19-32 and Page 11 Lines 1-11: In the Labrador Sea, IrSPMW exhibited an average DFe concentration of $0.61 \pm 0.21$ nmol L$^{-1}$ (n=14). DFe concentrations in the LSW were the lowest in this basin, with an average value of $0.71 \pm 0.27$ nmol L$^{-1}$ (n=53) (Fig. 7). Deeper, ISOW displayed slightly higher average DFe concentrations ($0.82 \pm 0.05$ nmol L$^{-1}$, n=2). Finally, DSOW had the lowest average ($0.68 \pm 0.06$ nmol L$^{-1}$, n=3, Fig. 7) and median (0.65 nmol L$^{-1}$) DFe values for intermediate and deep waters.

In the Irminger Sea, surface waters were composed of SAIW ($0.56 \pm 0.24$ nmol L$^{-1}$, n=4) and IrSPMW ($0.72 \pm 0.32$ nmol L$^{-1}$, n=34). The highest open-ocean DFe concentrations (up to $2.5 \pm 0.3$ nmol L$^{-1}$, station 44, 2600 m depth) were measured within this basin. In the upper intermediate water LSW was identified only at stations 40 to 44, and had the highest DFe values with an average of $1.2 \pm 0.3$ nmol L$^{-1}$ (n=14). ISOW showed higher DFe concentrations than in the Iceland Basin ($1.3 \pm 0.2$ nmol L$^{-1}$, n=4). At the bottom, DSOW was mainly located at stations 42 and 44 and presented the highest average DFe values ($1.4 \pm 0.4$ nmol L$^{-1}$, n=5) as well as the highest variability from all the water masses presented in this section (Fig. 7).

In the Iceland Basin, SAIW and IcSPMW displayed similar averaged DFe concentrations ($0.67 \pm 0.30$ nmol L$^{-1}$, n=7 and $0.55 \pm 0.34$ nmol L$^{-1}$, n=22, respectively). Averaged DFe concentrations were similar in both LSW and ISOW, and higher than in SAIW and IcSPMW ($0.96 \pm 0.22$ nmol L$^{-1}$, n=21 and $1.0 \pm 0.3$ nmol L$^{-1}$, n=10, respectively, Fig. 7).

Finally, in the West European Basin, DFe concentrations in ENACW were the lowest of the whole section with an average value of $0.30 \pm 0.16$ nmol L$^{-1}$ (n=64). MOW was present deeper in the water column but was not characterized by particularly high or low DFe concentrations relative to the surrounding Atlantic waters (Fig. 7). The median DFe value in MOW was very similar to the median value when considering all water masses (0.77 nmol L$^{-1}$, Figs. 4 and 7). LSW and IcSPMW displayed slightly elevated DFe concentrations compared to the overall median with mean values of $0.82 \pm 0.08$ (n=28) and $0.80 \pm 0.04$ (n=8) nmol L$^{-1}$, respectively. The DFe concentrations in NEADW were relatively similar to the DFe median value of the GEOVIDE voyage (median DFe = 0.75 nmol L$^{-1}$, Figs. 4 and 7) with an average value of $0.74 \pm 0.16$ nmol L$^{-1}$ (n=18) and presented relatively low median DFe concentrations (median DFe = 0.71 nmol L$^{-1}$) compared to other deep water masses.

Page 9, Line 13: remove "the" before MOW and in front of LSW (line 15), SAIW (line 24) and IrSPMW (line 24).

- → We have removed "the" in front of each water masses throughout the section

Page 9, Line 20: this is confusing, specify that you mean similar averages and not ranges (the range is larger for IcSPMW)

➔ We have added this precision as suggested (see above)

Page 9, Line 22: LSW and ISOW averages are also similar, combine sentences.

➔ We have changed the text accordingly (see above).

Page 9: Line 24: "composed of" instead of "characterised by"

➔ We have changed the text as suggested (see above).

Page 9, Line 32: Delete "compared to other ones"

➔ We have changed as suggested (see above)

Page 10, Line 2: "lowest average dFe value" (DSOW also shows the highest deep water dFe concentrations)

➔ We have changed the text as suggested (see above).

Page 10, Line 9: it is a little odd to start explaining what cannot be included in any of your sub-sections. You start introducing the general structure of your discussion and then you go into much detail explaining dFe trends all the sudden. This is totally out of place here. You should add this paragraph to the end of the discussion or in a new section.

➔ We have added a new section for this paragraph but we kept this new section there in the MS, as we think it would be odd to let the reader know at the end of the MS what we did not include.

Page 11 Lines 13-31:

**4 Discussion**

In the following sections, we will first discuss the high DFe concentrations observed throughout the water column of stations 1 and 17 located in the West European Basin (Section 4.1), then, the relationship between water masses and the DFe concentrations (Section 4.2) in intermediate (Section 4.2.2 and 4.2.3) and deep (Section 4.2.4 and 4.2.5) waters. We will also discuss the role of wind (Section 4.2.1), rivers (Section 4.3.1), meteoric water and sea-ice processes (Section 4.3.2), atmospheric deposition (Section 4.3.3) and sediments (Section 4.4) in delivering DFe. Finally, we will discuss the potential Fe limitation using DFe:$NO_3^-$ ratios (Section 4.5).

**4.1 High DFe concentrations at station 1 and 17**

Considering the entire section, two stations (stations 1 and 17) showed irregularly high DFe concentrations (> 1 nmol L$^{-1}$) throughout the water column, thus suggesting analytical issues. However, these two stations were analysed twice and provided similar results, therefore discarding any analytical issues. This means that these high values originated either from genuine processes or from contamination issues. If there had been contamination issues, one would expect a more random distribution of DFe concentrations and less consistence throughout the water column. It thus appears that contamination issues were unlikely to happen. Similarly, the influence of water masses to explain these distributions was discarded as the observed high homogenized DFe concentrations were restricted to these two stations. Station 1, located at the continental shelf-break of the Iberian Margin, also showed enhanced PFe concentrations from lithogenic origin suggesting a margin source (Gourain et al., 2018). Conversely, no relationship was observed between DFe and PFe nor transmissometry for station 17. However, Ferron et al. (2016) reported a

strong dissipation rate at the Azores-Biscay Rise (station 17) due to internal waves. The associated vertical energy fluxes could explain the homogenized profile of DFe at station 17, although such waves are not clearly evidenced in the velocity profiles. Consequently, the elevated DFe concentrations observed at station 17 remain unsolved.

Page 10, Line 20: I don't understand the aim of discussing dFe with water masses? I'd rather focus on the sources and sinks of dFe along the section. Discuss then the role of water masses in distributing the dFe signals. This means completely changing the focus of this section

➔ We decided to discuss DFe with the main water masses, i.e. LSW, ISOW and DSOW as they all constitute the Deep Western Boundary Current (DWBC).

Page 11, Line 2: Flow of thoughts not clear, reasons of deep winter mixing scattered; bits of information thrown in a little randomly. You should start off by saying that deep winter mixing is an important mechanism supplying nutrients to the surface ocean in the North Atlantic Ocean; then say how this deep winter mixing is produced (from what I can understand in your text are you trying to say this is due to the effects of wind + convective mixing+ subduction/upwelling; am I right? This was not clear); then say what the specific conditions were in the year you sampled. I am still giving you corrections on the section, which you can incorporate in your rewritten discussion if fitting. I think you can recycle some parts of your discussion.

➔ We have modified the text as suggested for clarification.

Page 12 Lines 12-22: In the North Atlantic Ocean, the warm and salty water masses of the upper limb of the MOC are progressively cooled and become denser, and subduct into the abyssal ocean. In some areas of the SubPolar North Atlantic, deep convective winter mixing provides a rare connection between surface and deep waters of the MOC thus constituting an important mechanism in supplying nutrients to the surface ocean (de Jong et al., 2012; Louanchi and Najjar, 2001). Deep convective winter mixing is triggered by the effect of wind and a pre-conditioning of the ocean in such a way that the inherent stability of the ocean is minimal. Pickart et al. (2003) demonstrated that these conditions are satisfied in the Irminger Sea with the presence of weakly stratified surface water, a close cyclonic circulation, which leads to the shoaling of the thermocline and intense winter air-sea buoyancy fluxes (Marshall and Schott, 1999). Moore (2003) and Piron et al. (2016) described low-level westerly jets centred northeast of Cape Farewell, over the Irminger Sea, known as tip jet events, whose structure depends upon the splitting occurring as the flow encounter the orographic features from Cape Farewell, and that are strong enough to induce deep convective mixing (Bacon et al., 2003; Pickart et al., 2003).

Page 11, Line 7: "events" and "with a positive NAO"

➔ We have corrected as suggested.

Page 11, Line 8: "The winter mixed layer depth"

➔ We have corrected as suggested.

Page 11, Line 9: instead of " and were" use "which was"

➔ We have corrected as suggested.

Page 11, Line 11: "close to those found in LSW"

➔ We have corrected as suggested.

Page 11, Line 13: sentence incomprehensive, please rephrase

→ This sentence has been removed according to Referee#1 comments and gather with previous sentence.

Page 12 Lines 26-27: Such winter entrainment was likely the process involved in the vertical supply of DFe within surface waters fuelling the spring phytoplankton bloom with DFe values close to those found in LSW.

Page 11, Line 24: need to improve a little the flow of thoughts in this paragraph. Start by saying why you see no MOW dFe signal, then support/contradict that argument(s) by what has been seen in other studies.

→ We have modified this section according to your comments and Referee#1 comments.

Page 12 Lines 29-32 and Page 13 Lines 1-14: On its northern shores, the Mediterranean Sea is bordered by industrialized European countries, which act as a continuous source of anthropogenic derived constituents into the atmosphere, and on the southern shores by the arid and desert regions of north African and Arabian Desert belts, which act as sources of crustal material in the form of dust pulses (Chester et al., 1993; Guerzoni et al., 1999; Martin et al., 1989). During the summer, when thermal stratification occurs, DFe concentrations in the SML can increase over the whole Mediterranean Sea by 1.6-5.3 nmol $L^{-1}$ in response to the accumulation of atmospheric Fe from both anthropogenic and natural origins (Bonnet and Guieu, 2004; Guieu et al., 2010; Sarthou and Jeandel, 2001). After atmospheric deposition, the fate of Fe will depend on the nature of aerosols, vertical mixing, biological uptake and scavenging processes (Bonnet and Guieu, 2006; Wuttig et al., 2013). During GEOVIDE, MOW was observed from stations 1 to 29 between 1000 and 1200 m depth and associated with high dissolved aluminium (DAl, Menzel Barraqueta et al., 2018) concentrations (up to 38.7 nmol $L^{-1}$), confirming the high atmospheric deposition in the Mediterranean region. In contrast to Al, no DFe signature was associated with MOW (Figs. 2 and 4). This feature was also reported in some studies (Hatta et al., 2015; Thuróczy et al., 2010), while others measured higher DFe concentrations in MOW (Gerringa et al., 2017; Sarthou et al., 2007). However, MOW coincides with the maximum Apparent Oxygen Utilization (AOU) and it is not possible to distinguish the MOW signal from the remineralisation one (Sarthou et al., 2007). On the other hand, differences between studies are likely originating from the intensity of atmospheric deposition and the nature of aerosols. Indeed, Wagener et al. (2010) highlighted that large dust deposition events can accelerate the export of Fe from the water column through scavenging. As a result, in seawater with high DFe concentrations and where high dust deposition occurs, a strong individual dust deposition event could act as a sink for DFe. It thus becomes less evident to observe a systematic high DFe signature in MOW despite dust inputs.

Page 12, Line 3: "suggesting that the water mass is enriched in dFe during its flow path"

→ We have changed the text as suggested taking into account Referee#1 comments.

Page 13 Lines 16-18: As described in Section 3.1, the LSW exhibited increasing DFe concentrations from its source area, the Labrador Sea, toward the other basins with the highest DFe concentrations observed within the Irminger Sea, suggesting that the water mass was enriched in DFe either locally in each basin or during its flow path (see supplementary material Fig. S3).

Page 12, Line 4: start by saying what those sources are and then support with the available literature.

→ We have reorganised this section based on your comments and Reviewer#1 comments.

Page 13 Lines 16-33, Page 14 Lines 1-16: As described in Section 3.1, the LSW exhibited increasing DFe concentrations from its source area, the Labrador Sea, toward the other basins with the highest

DFe concentrations observed within the Irminger Sea, suggesting that the water mass was enriched in DFe either locally in each basin or during its flow path (Fig. 7). These DFe sources could originate from a combination of high export of PFe and its remineralisation in the mesopelagic area and/or the dissolution of sediment.

The Irminger and Labrador Seas exhibited the highest averaged integrated TChl-a concentrations (98 $\pm$ 32 mg m$^{-2}$ and 59 $\pm$ 42 mg m$^{-2}$) compared to the West European and Iceland Basins (39 $\pm$ 10 mg m$^{-2}$ and 53 $\pm$ 16 mg m$^{-2}$), when the influence of margins was discarded. Stations located in the Irminger (stations 40-56) and Labrador (stations 63-77) Seas, were largely dominated by diatoms (>50% of phytoplankton abundances) and displayed the highest chlorophillid-*a* concentrations, a tracer of senescent diatom cells, likely reflecting post-bloom condition (Tonnard et al., in prep.). This is in line with the highest POC export data reported by Lemaitre et al. (2018) in these two oceanic basins. This likely suggests that biogenic PFe export was also higher in the Labrador and Irminger Seas than in the West European and Iceland Basins. Although, Gourain et al. (2018) highlighted a higher biogenic contribution for particles located in the Irminger and Labrador Seas with relatively high PFe:PAl ratios (0.44 $\pm$ 0.12 mol:mol and 0.38 $\pm$ 0.10 mol:mol, respectively) compared to particles from the West European and Iceland Basins (0.22 $\pm$ 0.10 and 0.38 $\pm$ 0.14 mol:mol, respectively, see Fig. 6 in Gourain et al., 2018), they reported no difference in PFe concentrations between the four oceanic basins (see Fig. 12A in Gourain et al., 2018) when the influence of margins was discarded, which likely highlighted the remineralisation of PFe within the Irminger and Labrador Seas. Indeed, Lemaître et al. (2017) reported higher remineralisation rates within the Labrador (up to 13 mmol C m$^{-2}$ d$^{-1}$) and Irminger Seas (up to 10 mmol C m$^{-2}$ d$^{-1}$) using the excess barium proxy (Dehairs et al., 1997), compared to the West European and Iceland Basins (ranging from 4 to 6 mmol C m$^{-2}$ d$^{-1}$). Therefore, the intense remineralisation rates measured in the Irminger and Labrador Seas likely resulted in enhanced DFe concentrations within LSW.

Higher DFe concentrations were, however, measured in the Irminger Sea compared to the Labrador Sea and coincided with lower transmissometry values (i.e. 98.0-98.5% vs. >99%), thus suggesting a particle load of the LSW. This could be explained by the reductive dissolution of Newfoundland Margin sediments. Indeed, Lambelet et al. (2016) reported high dissolved neodymium (Nd) concentrations (up to 18.5 pmol.kg$^{-1}$) within the LSW at the edge of the Newfoundland Margin (45.73°W, 51.82°N) as well as slightly lower Nd isotopic ratio values relative to those observed in the Irminger Sea. They suggested that this water mass had been in contact with sediments approximately within the last 30 years (Charette et al., 2015). Similarly, during GA03, Hatta et al. (2015) attributed the high DFe concentrations in the LSW to continental margin sediments. Consequently, it is also possible that the elevated DFe concentrations from the three LSW branches which entered the West European and Iceland Basins and Irminger Sea was supplied through sediment dissolution (Measures et al., 2013) along the LSW pathway.

The enhanced DFe concentrations measured in the Irminger Sea and within the LSW were thus likely attributed to the combination of higher productivity, POC export and remineralisation as well as a DFe supply from reductive dissolution of Newfoundland sediments to the LSW along its flow path.

Page 12, Line 6: change "the ones" for "those"

➔ We have corrected as suggested (see above).

Page 12, Line 12: Provide a brief description of how the remineralisation rates were measured...

➔ We have added this precision (see above).

Page 12, Line 18: You are repeating yourself!

→ We removed this sentence.

Page 12, Line 19: conspitious? clearly visible? change this word

→ We have changed the text for clarification (see above).

Page 12, Line 21: confusing between remineralisation and bacteria mediated ligand production. Please be clear.

→ We have reformulated this sentence (see above).

Page 12, Line 27: "Hereafter"

→ We have corrected the text.

Page 13, Line 2: you also need to consider vertical/lateral inputs (think in 3D), so sedimentary Fe could be coming from further north, for example, within nepheloid layers. You can't discard sedimentary inputs just by looking at vertical gradients.

→ We have reorganised the idea of this paragraph and wehave considered your suggestions (see below).

Page 13, Line 3: remove "the" before "Polar Intermediate Water" and "PIW" (line 6)

→ We have corrected the text accordingly.

Page 13, Line 9: instead of thinking that one water mass carries a certain dFe concentration (think of the short Fe residence times), this water mass might have "picked up" some dFe from, e.g., the sediments, on its pathway or It might have picked up particles in suspension which dissolve over time, etc!

→ We have reorganised the idea of this paragraph and we have considered your suggestions.

Page 15 Lines 2-20: However, considering the short residence time of DFe and the circulation of water masses in the Irminger Sea, it is possible that instead of being attributed to one specific water mass, these enhanced DFe concentrations resulted from lateral advection of the deep waters. Figure 8B) shows the concentrations of both DFe and PFe for the mixing line between DSOW/PIW and ISOW at station 44 and considering 100% contribution of ISOW for the shallowest sample (2218 m depth) and of DSOW/PIW for the deepest (2915 m depth), as these were the main water masses. This figure shows increasing DFe concentrations as DSOW/PIW mixed with ISOW. In addition, Le Roy et al. (2018) reported for the GEOVIDE voyage at station 44 a deviation from the conservative behaviour of $^{226}$Ra reflecting an input of this tracer centred at 2500 m depth, likely highlighting diffusion from deep-sea sediments and coinciding with the highest DFe concentrations measured at this station. Although the transmissometer values were lower at the sediment interface than at 2500 m depth, Deng et al. (2018) reported a stronger scavenged component of the $^{230}$Th at the same depth range, likely suggesting that the mixture of water masses were in contact with highly reactive particles. If there is evidence that the enhanced DFe concentrations observed at station 44 coincided with lateral advection of water masses that were in contact with particles, the difference of behaviour between DFe and $^{230}$Th remains unsolved. The only parameter that would explain without any ambiguity such differences of behaviour between DFe and $^{230}$Th would be the amounts of Fe-binding organic ligands for these samples. Indeed, although PFe concentrations decreased from the seafloor to the above seawater, this trend would likely be explained by a strong vertical diffusion alone and not necessarily from the dissolution of particles that were laterally advected.

Therefore, the high DFe concentrations observed might be inferred from local processes as ISOW mixes with both PIW and DSOW with a substantial load of Fe-rich particles that might have dissolved in solution due to Fe-binding organic ligands.

Page 13, Line 11: instead of "seawater" use "water masses"

→ We have reformulated this sentence (see above)

Page 13, Line 11: specify what you mean; 100 % contribution of which water mass at which depth

→ We have changed the text as suggested (see above).

Page 13, Line 18: watch out, this leads to miss-understanding as the reader thinks you are saying that particulate Fe is sustained by organic ligands. I don't think that is what you want to say.

→ We have changed the text as suggested (see above).

Page 13, Line 19: restructure this paragraph, say what you think the reasons are behind the high dFe concentrations, then support that by correlations, graphs etc and then compare to literature to confirm or dispute your theory. end the section with a stronger statement of what you think is happening in these deep water masses

→ We have changed the text as suggested (see above).

Page 13, Line 26: your explanation is a little long winded. Say that Mid Ocean Ridges can be a source of dFe but that this has not been found in the Reykjanes Ridge so far

→ We have changed the text according to your comments and referee#1 comments.

Page 15 Lines 22-30: Hydrothermal activity was assessed over the Mid Atlantic Ridge, namely the Reykjanes Ridge, from stations 36 to 42. Indeed, within the interridge database (http://www.interridge.org), the Reykjanes Ridge is reported to have active hydrothermal sites that were either confirmed (Baker and German, 2004a; German et al., 1994; Olaffson et al., 1991; Palmer et al., 1995) close to Iceland or inferred (e.g. Chen, 2003; Crane et al., 1997; German et al., 1994; Sinha et al., 1997; Smallwood and White, 1998) closer to the GEOVIDE section as no plume was detected but a high backscatter was reported potentially corresponding to a lava flow. Therefore, hydrothermal activity at the sampling sites remains unclear with no elevated DFe concentrations or temperature anomaly above the ridge (station 38).

Page 13, Line 30: you are repeating you pattern again; please first explain the signals you see and then compare to the literature

→ We have changed the text according to your comments and Referee#1 comments.

Page 15 Lines 28-33, Page 16 Lines 1-3: However, enhanced DFe concentrations (up to $1.5 \pm 0.22$ nmol L$^{-1}$, station 36, 2200 m depth) were measured east of the Reykjanes Ridge (Fig. 3). This could be due to hydrothermal activity and resuspension of sunken particles at sites located North of the section and transported through the ISOW towards the section (see supplementary material Fig. S3). Indeed, Achterberg et al. (2018) highlighted at ~60°N and over the Reykjanes Ridge a southward lateral transport of an Fe plume of up to 250-300 km. In agreement with these observations, previous studies (e.g. Fagel et al., 1996; Fagel et al., 2001; Lackschewitz et al., 1996; Parra et al., 1985) reported marine sediment mineral clays in the Iceland Basin largely dominated by smectite (> 60%), a tracer of hydrothermal alteration of basaltic volcanic materials (Fagel et al., 2001; Tréguer and De La Rocha, 2013). Hence, the high DFe concentrations measured east of the Reykjanes Ridge

could be due to a hydrothermal source and/or the resuspension of particles and their subsequent dissolution.

Page 14, Line 4-6: are these located on your transect? If not, you need to give their locations (coordinates) and explain where they are relative to your section. Are these GEOVIDE hydrographic sections (I don't think so) or from other cruises? Specify which ones. Cite appropriate Figures, and supporting literature.

➔ We have changed the text as suggested. Note that we also added the location of CGFZ and BFZ on Figure 1.

Page 16 Lines 4-12: West of the Reykjanes Ridge, a DFe-enrichment was also observed in ISOW within the Irminger Sea (Figs. 4 and 7). The low transmissometer values within ISOW in the Irminger Sea compared to the Iceland Basin suggest a particle load. These particles could come from the Charlie Gibbs Fracture Zone (CGFZ, 52.67°N and 34.61°W) and potentially Bight Fracture Zone (BFZ, 56.91°N and 32.74°W) (Fig. 1) (Lackschewitz et al., 1996; Zou et al., 2017). Indeed, hydrographic sections of the northern valley of the CGFZ showed that below 2000 m depth the passage through the Mid-Atlantic Ridge was mainly filled with the ISOW (Kissel et al., 2009; Shor et al., 1980). Shor et al. (1980) highlighted a total westward transport across the sill, below 2000 m depth of about 2.4 x $10^6$ m$^3$ s$^{-1}$ with ISOW carrying a significant load of suspended sediment (25 µg L$^{-1}$), including a 100-m-thick benthic nepheloid layer. It thus appears that the increase in DFe within ISOW likely came from sediment resuspension and dissolution as the ISOW flows across CGFZ and BFZ.

Page 14, Line 11: complete this section with more recent references. There have been previous studies in this area

➔ We have removed the following sentence "The seabed of this area has been identified as a depositional environment with patches of ripples and rock fragments surrounded by moat." Based on Referee#1 comments and have changed the text (see above).

Page 14, Line 20: restructure, again start by saying what you see, explaining the reason of these signals showing correlations, and then supporting literature

➔ We have changed the text as suggested.

Page 16 Lines 19-32, Page 17 Lines 1-2: Enhanced DFe surface concentrations (up to 1.07 ± 0.12 nmol L$^{-1}$) were measured over the Iberian Margin (stations 1-4) and coincided with salinity minima (~ <35) and enhanced DAl concentrations (up to 31.8 nmol L$^{-1}$, Menzel Barraqueta et al., 2018). DFe and DAl concentrations were both significantly negatively correlated with salinity (R$^2$ = ~1 and 0.94, respectively) from stations 1 to 13 (Fig. 5). Salinity profiles from station 1 to 4 showed evidence of a freshwater source with surface salinity ranging from 34.95 (station 1) to 35.03 (station 4). Within this area, only two freshwater sources were possible: 1) wet atmospheric deposition (4 rain events, Shelley, pers. comm.) and 2) the Tagus River, since the ship SADCP data revealed a northward circulation (P. Lherminier and P. Zunino, Ifremer Brest, pers. comm.). Our SML DFe inventories were about three times higher at station 1 (~ 1 nmol L$^{-1}$) than those calculated during the GA03 voyage (~ 0.3 nmol L$^{-1}$, station 1) during which atmospheric deposition were about one order of magnitude higher (Shelley et al., 2018; Shelley et al., 2015), the atmospheric source seemed to be minor. Consequently, the Tagus River appears as the most likely source responsible for these enhanced DFe concentrations, either as direct input of DFe or indirectly through Fe-rich sediment carried by the Tagus River and their subsequent dissolution. The Tagus estuary is the largest in the western European coast and very industrialized (Canário et al., 2003; de Barros, 1986; Figueres et al., 1985; Gaudencio et al., 1991; Mil-Homens et al., 2009), extends through an area of 320 km$^2$ and is

characterized by a large water flow of 15.5 $10^9$ m$^3$ y$^{-1}$ (Fiuza, 1984). Many types of industry (e.g. heavy metallurgy, ore processing, chemical industry) release metals including Fe, which therefore result in high levels recorded in surface sediments, suspended particulate matter, water and organisms in the lower estuary (Santos-Echeandia et al., 2010).

Page 14, Line 26: You can't say "et al." in a personal communication. All people should be mentioned. Also State first name and affiliation in a personal communication.

➔ We have modified the text to fulfil this comment (P. Lherminier and P. Zunino, Ifremer Brest, pers. comm.)

Page 14, Line 26: Very difficult to follow; why "however"? Start by explaining the typical ratios of different sources before you give the observed ratios in your study. Then discuss the What dFe:dAl ratio is expected from a river source? And from an atmospheric source?

➔ We have removed this sentence according to your comment and Referee#1 comment and we have changed the text for clarification (see above).

Page 14, Line 29: Difficult to understand; you need to present each option that could lead to the enhanced dFe signal, discuss and then accept or discard

➔ We have changed the text as suggested (see above).

Page 15, Line 18: I don't understand this "extended as close as 200 km from our Greenland stations". Please rephrase. Also avoid parenthesis next to parenthesis, and explain clearly what can be found in the link and what info you got from there.

➔ We reword the sentence for clarification.

Page 17 Lines 10-17: The most plausible sources would be freshwater induced by meteoric water and sea-ice melt. Conversely, deeper in the water column, brine signals were calculated at stations 53 (100 m depth, Fig. 5D) 61 (100 m depth, Fig. 5E) and 78 (30 m depth, Fig. 5F). The release of brines could originate from two different processes: the sea-ice formation or the early melting of multiyear sea ice due to gravitational drainage and subsequent brine release (Petrich and Eicken, 2010; Wadhams, 2000). Indeed, during the winter preceding the GEOVIDE voyage, multiyear sea ice extended 200 km far from our Greenland stations (http://nsidc.org/arcticseaicenews/).

Page 15, Line 20: how were these calculated? give equations

➔ We have included a section in the method on how these fractions were calculated.

Page 5 Lines 23-32, Page 6 Lines 1-9: We separated the mass contributions to samples from stations 53, 61 and 78 in Sea-Ice Melt (SIM) Meteoric Water (MW) and saline seawater inputs using the procedure and mass balance calculations that are fully described in Benetti et al. (2016) (Fig. 5D), E) and F)). Hereafter, we describe briefly the principle. We considered two types of seawater, namely the Atlantic Water (AW) and the Pacific Water (PW). After estimating the relative proportions of AW ($f_{AW}$) and PW ($f_{PW}$) and their respective salinity and $\delta^{18}$O affecting each samples, the contribution of SIM and MW can be determined using measured salinity ($S_m$) and $\delta^{18}$O ($\delta O_m^{18}$). The mass balance calculations are presented below:

$$f_{AW} + f_{PW} + f_{MW} + f_{SIM} = 1 \text{ (eq.1)}$$

$$f_{AW}.S_{AW} + f_{PW}.S_{PW} + f_{MW}.S_{MW} + f_{SIM}.S_{SIM} = S_m \text{ (eq.2)}$$

$$f_{AW}.\delta O_{AW}^{18} + f_{PW}.\delta O_{PW}^{18} + f_{MW}.\delta O_{MW}^{18} + f_{SIM}.\delta O_{SIM}^{18} = \delta O_m^{18} \text{ (eq.3)}$$

where $f_{AW}$, $f_{PW}$, $f_{MW}$, $f_{SIM}$ are the relative fraction of AW, PW, MW, and SIM. To calculate the relative fractions of AW, PW, MW and SIM we used the following end-members: $S_{AW} = 35$, $\delta O^{18}_{AW} = +0.18‰$ (Benetti et al., 2016); $S_{PW} = 32.5$, $\delta O^{18}_{PW} = -1‰$ (Cooper et al., 1997; Woodgate and Aagaard, 2005); $S_{MW} = 0$, $\delta O^{18}_{MW} = -18.4‰$ (Cooper et al., 2008); $S_{SIM} = 4$, $\delta O^{18}_{SIM} = +0.5‰$ (Melling and Moore, 1995).

Negative sea-ice fractions indicated a net brine release while positive sea-ice fractions indicated a net sea-ice melting. Note that for stations over the Greenland Shelf, we assumed that the Pacific Water (PW) contribution was negligible for the calculations, supported by the very low PW fractions found at Cape Farewell in May 2014 (see Figure B1 in Benetti et al., 2017), while for station 78, located on the Newfoundland shelf, we used nutrient measurements to calculate the PW fractions, following the approach from Jones et al. (1998) (the data are published in Benetti et al., 2017).

Page 15, Line 27: I suggest instead of describing the Figures, you should use them to support your discussion. So try to avoid starting sentences and paragraphs describing Figures! You do this very often

➔ We have changed the text as suggested.

Page 17 Lines 20-33, Page 18 Lines 1-2: Considering the sampling period at stations 53 (16 June 2014) and 61 (19 June 2014), sea-ice formation unlikely explained the brine signals calculated as this period coincides with summer melting in both the Central Arctic and East Greenland (Markus et al., 2009). However, it is possible that the brines observed in our study could originates from sea-ice formation, which occurred during the previous winter(s) at 66°N (and/or higher latitudes). The residence time can vary from days (von Appen et al., 2014) to 6-9 months (Sutherland et al., 2009). Due to our observed strong brine signal at station 61 we suggest that the residence time was potentially longer than average. Given that the brine signal was higher at station 61 than at station 53 (which was located upstream in the EGC), we suggest that station 53 was exhibiting a freshening as a result of the transition between the freezing period toward the melting period. This would result in a dilution of the brine signal at the upstream station. Consequently, the salinity of this brine signal may reflect sea ice formation versus melting which may have an effect on the trace metal concentration within this water (Hunke et al., 2011). The associated brine water at station 61 (100 m depth) was slightly depleted in both DFe and PFe, which may be attributed to sea ice formation processes. Indeed, Janssens et al. (2016) highlighted that as soon as sea ice forms, sea salts are efficiently flushed out of the ice while PFe is trapped within the crystal matrix and DFe accumulates, leading to an enrichment factor of these two Fe fractions compared to underlying seawater. Conversely, the brine signal observed at station 53 (100 m depth) showed slight enrichment in DFe, which may be attributed to brine release during early sea ice melting and the associated release of DFe into the underlying water column as the brine sinks until reaching neutral buoyancy due to higher density.

Page 15, Line 27: how is sea-ice fraction calculated? please provide equations (if you calculated it yourself) or references where this data is published. Provide clear info so the reader can understand

➔ We have included a section in the method on how these fractions were calculated (see above)

Page 15, Line 30: explain why you compare to dAl in these profiles.

➔ We have removed DAl distribution.

Page 15, Line 33: "originates"

➔ We have corrected accordingly.

Page 16, Line 7: I can't believe that sea-ice can "uptake" Fe! The concentration of Fe in the newly formed ice and in the remaining water should stay the same. You need to find another explanation!

➜ We reword the sentence and add a reference for clarification (see above).

Page 16, Line 10: you mean sea-ice formation? Hence release of brine? Also explain that the brine sinks because it is denser (this is why it is observed below the surface)

➜ We have changed the text as suggested (see above).

➜ Regarding brines, they can originate from two different processes: either as a result of multiyear sea-ice melting or during sea-ice formation. Indeed, during the early melting season, multiyear sea-ice has a higher porosity and gravitational drainage of brine occur. These two processes of brine release might lead to different TM signatures in brine originating from sea-ice formation and brine originating from early melting of multiyear sea-ice (Petrich and Eicken, 2010; Wadhams, 2000).

Page 16, Line 10: "release of dFe"

➜ We have corrected accordingly.

Page 16, Line 10: "underlying water column"

➜ We have corrected accordingly.

Page 16, Line 11: split this sentence, it is too long

➜ We have changed the text as suggested.

Page 16 Lines 29-32: Surface waters (from 0 to ~ 50 m depth) from station 53 and 61 were characterized by high MW fractions (ranging from 8.3 to 7.4% and from 7.7 to 7.4% , respectively, from surface to ~50 m depth). Within these surface waters, station 53 exhibited substantial sea-ice melting contribution (1.5%, 4 m depth) while station 61 exhibited low contribution (0.6%) from brine release that was linearly increasing with depth (1.3% at 50 m depth and 2.2 % at 100 m depth).

Page 16, Line 15: instead of describing the data (this is more appropriate for results section) you should say what correlates with what (e.g., low Fe with low MW). difficult to follow flow of thoughts

➜ We have changed the text as suggested.

Page 18 Lines 3-18: Surface waters (from 0 to ~ 100 m depth) from station 53 and 61 were characterized by high MW fractions (ranging from 8.3 to 7.4% and from 7.7 to 7.3% , respectively, from surface to ~100 m depth, Figs. 5D and E). These high MW fractions were both enriched in PFe and DFe (except station 53 for which no data was available close to the surface) compared to seawater located below 50 m depth, thus suggesting a MW source. These results are in line with previous observations, which highlighted strong inputs of DFe from a meteoric water melting source in Antarctica (Annett et al., 2015). Although the ability of MW from Greenland Ice Sheet and runoffs to deliver DFe and PFe to surrounding waters has previously been demonstrated (Bhatia et al., 2013; Hawkings et al., 2014; Schroth et al., 2014; Statham et al., 2008), both Fe fractions were lower at the sample closest to the surface, then reached a maximum at ~ 50 m depth and decreased at ~ 70 m depth, for station 61 (Fig. 4D). The surface DFe depletion was likely explained by phytoplankton uptake, as indicated by the high TChl-$a$ concentrations (up to 6.6 mg m$^{-3}$) measured from surface to about 40 m depth, drastically decreasing at ~ 50 m depth to 3.9 mg m$^{-3}$ (Fig. 4D). Hence, it seemed that meteoric water inputs from the Greenland Margin likely fertilized surface waters with DFe, enabling the phytoplankton bloom to subsist. The profile of PFe can be explained by two opposite

plausible hypotheses: 1) MW inputs did not released PFe, as if it was the case, one should expect higher PFe concentrations at the surface (~25 m depth) than the one measured at 50 m depth due to both the release from MW and the assimilation of DFe by phytoplankton 2) MW inputs can release PFe in a form that is directly accessible to phytoplankton with subsequent export of PFe as phytoplankton died. The latter solution explains the PFe maximum measured at ~ 50 m depth and is thus the most plausible.

Page 16, Line 17: remind the reader in which figure information can be found (Chl-a, ect)

➔ We added these precisions throughout this section.

Page 16, Line 24: results in agreement with the capacity of. . .? weird sentence, please change

➔ We have changed the text as suggested.

Page 18 Lines 7-11: Although the ability of MW from Greenland Ice Sheet and runoffs to deliver DFe and PFe to surrounding waters have previously been demonstrated (Bhatia et al., 2013; Hawkings et al., 2014; Schroth et al., 2014; Statham et al., 2008), both Fe fractions were lower at the sample closest to the surface, then reached a maximum at ~ 50 m depth and decreased at ~ 70 m depth, for station 61 (Fig. 4D).

Page 16, Line 27: so far you have not talked about dFe:dAl ratio in meteoric water. Explain if this ratio is used to trace MW inputs and what you see in your profiles. should explain at the start.

➔ We have added a sentence at the beginning of the section to fulfil this comment and removed the DAl data.

Page 17 Lines 15-17: In the following sections, we discuss the potential for meteoric water supply, sea-ice formation and sea-ice melting to affect DFe distribution.

Page 16, Line 28: change "noting that" for "although"

➔ We changed the text as suggested.

Page 17, Line 6: or maybe the brine conditions (pH, salinity etc) make the Fe more bioavailable; or maybe this peak is not Fe related, but related to the release of other TM, or a phytoplankton group that thrives in brine and does not require much Fe? Since this shelf is further south, the environmental conditions may be more favourable for phytoplankton to grow and hence consuming all the dFe more rapidly. You should explore all the possibilities

➔ We have changed the text as suggested.

Page 18 Lines 28-30: This either suggests that the brine likely contained important amounts of Fe (dissolved and/or particulate Fe) that were readily available for phytoplankton and consumed at the sampling period by potentially sea-ice algae themselves (Riebesell et al., 1991) or that another nutrient was triggering the phytoplankton bloom.

Page 17, Line 19: You repeat the word "dust" too much

➔ We have changed the text according to your comment and Referee#1 comment.

Page 18 Lines 1-2 Page 19 Lines 1-4: On a regional scale, the North Atlantic basin receives the largest amount of atmospheric inputs due to its proximity to the Saharan Desert (Jickells et al., 2005), yet even in this region of high atmospheric deposition, inputs are not evenly distributed. Indeed, aerosol Fe loading measured during GEOVIDE (Shelley et al., 2017) were much lower (up to

four orders of magnitude) than those measured during studies from lower latitudes in the North Atlantic (e.g. Baker et al., 2013; Buck et al., 2010; and for GA03, Shelley et al., 2015), but atmospheric inputs could still be an important source of Fe in areas far from land.

Page 17, Line 20, "proportions"

➔ We have changed the text (see above).

Page 18, Line 3: you are going into too much detail here about aerosols, which is part of the Shelley papers

➔ We have removed this part.

Page 18, Line 6: Meskhidze et al. (2017) is not in your reference list

➔ We added the reference to the reference list.

Page 18, Line 10: information is a little randomly thrown in... what is your point? This is not a review paper

➔ We have removed this part.

Page 18, Line 13: What does OM mean? write abbreviations out first time. Do you mean organic matter?

➔ Yes, we meant organic matter but we have removed this part.

Page 18, Line 15: So all this background information to lastly say that you don't comment on this? Rephrase, make some assumptions or delete some detail

➔ We have removed this part.

Page 18, Line 26: of those stations, station 40 is most similar to total aerosol dFe:dAl ratios. station 26 is closer to the soluble than the total composition, so I don't understand what you are saying

➔ We removed all this part and changed it for turnover times relative to atmospheric deposition as defined in Guieu et al., 2014.

Page 19 Lines 5-17: In an attempt to estimate whether there was enough atmospheric input to sustain the SML DFe concentrations, we calculated Turnover Times relative to Atmospheric Deposition (TTADs, Guieu et al., 2014). To do so, we made the following assumptions: 1) the aerosol concentrations are a snapshot in time but are representative of the study region, 2) the aerosol solubility estimates based on two sequential leaches are an upper limit of the aerosol Fe in seawater and 3) the water column stratified just before the deposition of atmospheric inputs, so MLD DFe will reflect inputs from above. Thus, the TTADs were defined as the integrated DFe concentrations in the SML for each station divided by the contribution of soluble Fe contained in aerosols averaged per basin to the water volume of the SML. Although, TTADs were lower in the West European and Iceland Basins with an average of ~ 9 ± 3 months compared to other basins (7 ± 2 years and 5 ± 2 years for the Irminger and Labrador Seas, respectively) (Fig. 6) they were about three times higher than those reported for areas impacted by Saharan dust inputs (~ 3 months, Guieu et al., 2014). Therefore, the high TTADs measured in the Irminger and Labrador Seas and ranging from 2 to 15 years provided further evidence that atmospheric deposition were unlikely to supply Fe in sufficient quantity to be the main source of DFe (see Sections 4.2.1 and 4.3.2) while in the West European and Iceland Basins they played an additional source, perhaps the main source of Fe especially at station 36 which displayed TTAD of 3 months.

[Figure]

Figure 6: Plot of dissolved Fe (DFe) Turnover Times relative to Atmospheric Deposition (TTADs) calculated from soluble Fe contained in aerosols estimated from a two-stage sequential leach (UHP water, then 25% HAc, Shelley et al., this issue). Note that numbers on top of data points represent station numbers and that the colour coding refers to different region with in yellow, margin stations; in purple, the West European Basin; in blue, the Iceland Basin; in green, the Irminger Sea and in red, the Labrador Sea. The numbers on top of the plot represent TTADs averaged for each oceanic basin and their standard deviation.

Page 18, Line 28: And what about all the other stations where the ratios are different? I would say this is more of a "coincidence" that these data fall onto the black line. The multiple reactions occurring as Fe enters the ocean change this Fe:Al ratio rapidly

➔ Same as above, this part has been removed.

Page 18, Line 30: remove "time"

➔ This part has been removed (see above)

Page 19, Line 2: I don't understand.... these station points on your figure 10 are similar to those of other stations. What makes you think they have a higher atmospheric influence??

➔ This part has also been removed (see above)

Page 19, Line 4: in this section you should mention that bottom water dFe concentrations were significantly higher on the Newfoundland margin than on the Greenland margins

➔ We have included this precision.

Page 19 Lines 20-24: DFe concentration profiles from all coastal stations (stations 2, 4, 53, 56, 61 and 78) are reported on Figure 5. To avoid surface processes, only depths below 100 m depth will be considered in the following discussion. Stations where DFe and PFe followed a similar pattern are stations 2, 53, 56, and 78, suggesting that either the sources of Fe supplied both Fe fractions (dissolved and particulate) or that PFe dissolution from sediments supplied DFe. Among the different margins, the Newfoundland Margin exhibited the highest deep-water DFe concentrations.

Page 19, Line 10: "mol:mol" to stay consistent

➔ We have changed the text as suggested.

Page 19, Line 10: what is the average useful for? Show a plot with dFe:pFe ratios or a table

➔ Because the SD is relatively high thus conferring distinct signature throughout the water column (below 100 m depth) of each station. This allows the comparison station by station instead of comparing each samples. These ratios were added to Table 3.

Page 19, Line 11: as well as different sediment compositions, this could be also due to different supply mechanisms? Different sediment conditions (redox, organic content, temp, etc)

➔ We added this precision in the text.

Page 19 Lines 25-28: DFe:PFe ratios ranged from 0.01 (station 2, bottom sample) to 0.27 (station 4, ~ 400 m depth) mol:mol with an average value of 0.11 ± 0.07 mol:mol (n = 23, Table 3), highlighting a different behaviour of Fe between margins. This could be explained by the different nature of the sediments and/or different sediment conditions (e.g. redox, organic content).

Page 19, Line 17: Where is the predominance of diatoms?

➔ We changed the sentence for clarification.

Page 19 Lines 31-33: These observations are consistent with the high TChl-*a* concentrations measured at the Newfoundland Margin and to a lesser extent at the Greenland Margin and the predominance of diatoms relative to other functional phytoplankton classes at both margins (Tonnard et al., in prep.).

Page 19, Line 21: I think this section is great, but you need to organise the ideas clearly. Now difficult to follow. Also name this section "nepheloid layers"

➔ We changed the section title as suggested and reorganised the flow of ideas.

Page 19, Line 26: explain the criterion first. Explain briefly the PCA and the results you show in figure 11. Information is thrown in a little randomly. Please organise you paragraphs

➔ We reorganised the paragraph as suggested.

Page 20 Lines 6-16: Samples associated with high levels of particles (transmissometer < 99%) and below 500 m depth displayed a huge variability in DFe concentrations. From the entire dataset, 66 samples (~13% of the entire dataset) followed this criterion with 3 samples from the Iberian Margin (station 4), 14 samples from the West European Basin (station 1), 4 samples from the Iceland Basin (stations 29, 32, 36 and 38), 43 samples from the Irminger Sea (stations 40, 42, 44, 49 and 60) and 2 samples from the Labrador Sea (station 69). To determine which parameter was susceptible to explain the variation in DFe concentrations in these nepheloid layers, a Principal Component Analysis (PCA) on these samples. The input variables of the PCA were the particulate Fe, Al, and particulate manganese (PMn) (Gourain et al., 2018), the DAl (Menzel Barragueta et al., 2018) and

the Apparent Oxygen Utilization (AOU) and were all correlated to DFe concentrations explaining all together 93% of the subset variance (see supplementary material Fig. S6). The first dimension of the PCA was represented by the PAl, PFe and PMn concentrations and explained 59.5% of the variance, while the second dimension was represented by the DAl and the AOU parameters, explaining 33.2% of the variance. The two sets of variables were nearly at right angle from each other, indicating no correlation between them.

Page 20, Line 1: I would not call it AOU "concentrations" find another way to express this

➔ We changed "concentrations" for parameters (Page 20 Line 20)

Page 20, Line 13: you should look into non-reducing dissolution of lithogenic material. You are missing out on a big topic! Radic et al., 2011; Labatut et al., 2014; Abadie et al., 2017

➔ We included this topic in the text as suggested.

Page 20 Lines 26-34, Page 21 Lines 1-6: This observation challenges the traditional view of Fe oxidation with oxygen, either abiotically or microbially induced. Indeed, remineralisation can decrease sediment oxygen concentrations, promoting reductive dissolution of PFe oxyhydroxides to DFe that can then diffuse across the sediment water interface as DFe(II) colloids (Homoky et al., 2011). Such processes will inevitably lead to rapid Fe removal through precipitation of nanoparticulate or colloidal Fe (oxyhydr)oxides, followed by aggregation or scavenging by larger particles (Boyd and Ellwood, 2010; Lohan and Bruland, 2008) unless complexion with Fe-binding organic ligands occurs (Batchelli et al., 2010; Gerringa et al., 2008). There exist, however, another process that is favoured in oxic benthic boundary layers (BBL) with low organic matter degradation and/or low Fe oxides, which implies the dissolution of particles after resuspension, namely the non-reductive dissolution of sediment (Homoky et al., 2013; Radic et al., 2011). In addition, these higher oxygenated samples were located within DSOW, which mainly originate (75% of the overflow) from the Nordic Seas and the Arctic Ocean (Tanhua et al., 2005), in which the ultimate source of Fe was reported by Klunder et al. (2012) to come from Eurasian river waters. The major Arctic rivers were highlighted by Slagter et al. (2017) to be a source of Fe-binding organic ligands that are then further transported via the TPD across the Denmark Strait. Hence, the enhanced DFe concentrations measured within DSOW might result from Fe-binding organic ligand complexation that were transported to the deep ocean as DSOW formed rather than the non-reductive dissolution of sediment.

Page 20, Line 14: "lead"

➔ We changed the text accordingly.

Page 20, Line 16: instead of "these" use "this sediment-derived. . ."

➔ We have changed the text accordingly.

Page 20, Line 17: do not state this as facts... these are assumptions

➔ We have changed the text as suggested(see above).

Page 20, Line 25: very poor sum-up, please improve

➔ We have changed the full section according to your comment (see above).

Page 20, Line 26: You should also compare surface dFe data to AOU to look at dFe released from remineralisation. You have done this for >500 m depth, but it will be worth looking at this more closely below the surface mixed layer, where remineralisation occurs (below 100 m depth).

→ We have included this just before Fe* section.

[revised manuscript text omitted]

Page 20, Line 30: change the ending of this sentence, not properly expressed

➔ We have split and changed the sentence (see above).

Page 20, Line 31: First explain why you talk about Fe:nitrate ratios. THis comes out of the blue! Also cite Fig 12.

➔ We have changed the text as suggested (see above).

Page 21, Line 3: what do you mean by influence of the river, and the currents? specify what you mean

➔ We have changed the text for clarification (see above).

Page 21, Line 6: Can you provide a different kind of plot to help visualise this gradient you are talking about. In figure 12 this is impossible

➔ We have included a surface map of DFe:NO3- ratios.

[Figure]

Page 21, Line 18: "disequilibrium" sounds a bit odd, better use the word "ranges"

➔ We have changed the text as suggested (see above).

Page 21, Line 23: do you assume that all the nitrate in seawater comes from remineralisation? Better explain what assumptions this equation relies on

➔ We have modified the text accordingly (see above the response to your comment "Page 20, Line 26")

Page 21, Line 26: Rather, negative values of Fe* indicate the removal of dFe that is faster than the input through remineralisation or external sources and positive values suggest input of dFe from external sources

➔ We have changed the text as suggested.

Page 21, Line 27: remove "out"

➔ We have changed the text as suggested.

Page 21, Line 28: you talk about surface waters here but the calculations are done below 100 m depth. I would keep discussion on the external sources of dFe and then link to inputs of dFe rich water masses to surface waters above

➔ We have reorganised the section according to your comments and Reviewer#1 comments. Therefore we only focus the discussion on the top 200 m depth of the section.

[revised manuscript text omitted]

Page 22, Line 5: what has the low Fe supply to do with the "inefficient" carbon pump? If you want to talk about the carbon pump, and its inefficiency, you need to support with adequate statements/findings. I do not think this is a finding of your study.

➔ We have removed this part (see above).

Page 22, Line 7: this comes a little out of the blue. Explain a little more the high remineralised carbon fluxes and how they were measured

➔ We replaced in context this sentence and the precision on the measurement of remineralised carbon fluxes is now precised earlier in the MS.

Page 22, Line 12: poor ending. What about other sources? Margins? Rivers?

➔ We have changed the text (see above).

Page 22, Line 15: first sentence superfluous

➔ We have removed this sentence as suggested.

Page 22, Line 23: depletion of nitrate? That doesn't make sense

➔ We have changed the sentence for clarification (see above).

Page 22, Line 27: "entrained it to the deep. . ."

➔ We have changed the text as suggested.

Page 22, Line 29: "in the deep ocean"

➔ We have changed the text as suggested.

Page 22, Line 30: do you mean particles? Sediments are on the seafloor. Same for line 32

➔ Yes you are right, we have changed "sediments" for "particles".

Page 23, Line 3: conclusions need to be rewritten after the discussion is reworked.

➔ We have modified the conclusion to be more specific

[revised manuscript text omitted]

---

## Author Comment (AC8) · 19 Sep 2018

Dear Referee#2 and Editor, You will find on the link below the new version of the supplementary material based on all feedback. All the best, Tonnard et al.

Please also note the supplement to this comment:
https://www.biogeosciences-discuss.net/bg-2018-147/bg-2018-147-AC8-supplement.pdf
* * *

---

## Author Response (AR2)

**Response to reviewer#2**

We are grateful to reviewer#2 for his/her detailed review which helped a lot improving the manuscript.
Changes has been made in the text and our responses to the reviewer's comments are written below (in bold blue). The page and line numbers correspond to the marked-up manuscript.
We also wanted to apologize for sending our revisions with such a long delay. Manon Tonnard (first author of the article) has now left science after her PhD and it took us long time to get the hardrive with all the figures, versions, and data. You will find at the end of these responses the modified manuscript with track changes.

Review comments of the resubmitted version of Tonnard et al.

To start I would like to comment on some practicalities. Continuous line numbers throughout the text (and not restarting at every page) would have made life much easier for the reviewers. In addition, responses to reviewers' comments in underlined red very hard to read in this pdf-converted tracked changes file! I am aware this is the first experience publishing but please consider giving responses in a well-organised way next time. You can highlight text in other smoother colours or in bold that are easier to read than red underlined text with hundreds of text boxes attached on the side. You should also consider that when giving response to reviews it is important to include the new line numbers of the new text. This was done sometimes, but most of the time (for the short responses) the reviewer had to find its way around the text. Things like this considerably increase the revision time, especially for such long papers.

**We are very sorry for this and we appreciate the reviewer's time. This new version of the manuscript has continuous line numbers and we provide new line numbers for each modification. We hope that the review process will be smoother.**

I appreciate the amount of effort that has gone into making the requested changes. Even though the paper has considerably improved, it still needs a few changes before publication. Apart from the minor corrections below, there are a few more important points below:
1) some of your discussion is based on the expectation to see "DFe fingerprints" in specific water masses. Realistically, I do not think that the DFe residence time is long enough to persist along the flow path of a water mass, and I therefore would like to motivate you to compare the residence time of DFe in a certain region to the time it takes for this DFe to travel from the "source" region (i.e., surface ocean, sediments, hydrothermal ridge, river) along the flow path of the water mass in question considering its flow velocity. I believe you will find that these times are not compatible to see a DFe fingerprint and that internal processes along these flow paths are much more important. I encourage you to include these calculations in your paper when you discuss DFe signatures in water masses.

Given the wide uncertainties in iron input fluxes in the ocean, residence times (RTs) of dissolved iron (DFe) in the water column are poorly constrained. In the surface waters, estimations of DFe RTs generally range from weeks to months in the surface waters and from tens to hundreds of years in deep waters (de Baar and de Jong, 2001; Sarthou et al., 2003; Croot et al., 2004; Bergquist and Boyle 2006; Gerringa et al 2015; Tagliabue et al., 2016).

To compare with these DFe RTs, we investigated the transit time of the water masses (MSW, LSW, DSOW, and ISOW) that we considered in our discussion on the section concerning the DFe signature in water masses.

For the MOW, the translation velocity was calculated to be $\sim$ 3-8 cm s$^{-1}$ during the GEOVIDE cruise, using L-ADCP data. Our farest station influenced by MOW at $\sim$ 60% (station 13) was located $\sim$ 2000 km far from the origin of the MOW in the Mediterranean Sea, which would mean a transit time of $\sim$ 1-2 years. This transit time would allow Fe signal to be preserved, which would confirm our hypothesis of scavenging of DFe on particles. References to the transit time of MOW was added to the text (P. 18, lines 522-528).

For the LSW, we suggested (P. 20, lines 581-585) that "The enhanced DFe concentrations measured in the Irminger Sea and within the LSW were thus likely attributed to the combination of higher productivity, POC export and remineralisation as well as a DFe supply from reductive dissolution of Newfoundland sediments to the LSW along its flow path." Using temperature and salinity anomalies, Yashayaev et al. (2007) showed that the LSW reaches the Irminger Sea and the Iceland Basin in 1-2 years and 4-5 years, respectively, after its formation in the Labrador Sea. This transit time would allow the Fe signal to be preserved, when DFe residence times range from weeks to months in the surface waters and from tens to hundreds of years in deep waters (de Baar and de Jong, 2001; Sarthou et al., 2003; Croot et al., 2004; Bergquist and Boyle 2006; Gerringa et al 2015; Tagliabue et al., 2016). This sentence was added in the text, (P. 20 Lines 581-585), with the references cited above (P. 18, lines 527-528).

For the high concentrations of DFe in the Irminger Sea, we tried to simplify this section and invoque a new hypothesis on the exchanges between the DSOW and the slope. See P. 21, Lines 607-616.

Concerning the transport of DFe from hydrothermal origin East and West of the Reykjanes Ridge (RR), the transit time of the ISOW is now taken into account. Kanzow and Zenk (2014) investigated the fluctuations of the ISOW plume around the RR. The transit time, west of RR, between 61°N and the Bight Fracture Zone (BFZ) was around 5 months, with additional $\sim$ 3 months to reach our station West of RR. This information

**is now added in the text and the text has been simplified to clarify the message (p. 22, lines 635-637 and 644-645).**

**For the Tagus river inputs, we also estimated the transit time of the waters from the estuary to our stations 1 and 2. It is equal to ~ 15 days, which is compatible with the DFe RT in the surface waters. This was added to the text (p. 23, lines 6662-665)**

2) Include some important descriptions in your methods section: Chl-a sampling and measurement techniques, fluorometer measurements, how the fluorometer is calibrated with the Chl-a measurements; describe why you used the sensor data from the stainless steel CTD; include the interclibration plots, even though they are different they are important for the GEOTRACES community, and explain the likely reasons why they are different; more detail on the different filtration techniques, including information on which samples are filtered through which size fraction (table S1); use the correct density values to make the conversion of units

**Information regarding pigment data acquisition and in particular Chl-a data are now provided in P. 869, lines 232-2251.**

**GEOTRACES intercalibration: we could potentially compare our data (Station 44) with two stations sampled nearby during GA02 section (Stations 5 & 6). The comparison is presented in Figure 1 below:**

[Figure]

**Figure 1. Temperature (°C), Salinity, and DFe (nmol L$^{-1}$) vertical profiles at station 44 (GA01), 5& 6 (GA02), with approximate location of water masses.**

DFe vertical concentrations measured at stations 5&6 of GA02 section are similar, yet they are very different from the concentrations measured at station 44 sampled during GA01. Such differences could be explained by: (i) the different water mass contributions - indeed, variable contribution of water masses such as LSW, DSOW, and ISOW between cruises led to different temperature and salinity profiles. Especially at depths below 2500m, there was an intrusion of PIW during GA01 and not during GA02. So below this depth, we cannot really compare datasets. Overall, DFe concentrations measured during GA01 are higher than those measured during GA02 in LSW and ISOW water masses. (ii) Difference in analytical techniques and acidification time employed during the two cruises: During the GA02 voyage, DFe concentrations were directly measured on-board, with a short acidification time, using Flow Injection Analysis with chemiluminescent detection (FIA-CL), while during GEOVIDE, the acidification time was much longer (more than a year) and samples were analysed using a SeaFAST-picoTM coupled to an Element XR (see section 2.2). This would suggest that some refractory DFe was not measured with the on-board measurements during GA02 voyage.

**Information on which samples are filtered through which size fraction are now provided in Tables 1 and S1.**

**We followed the reviewer's recommendation and used the correct density values to make the conversion from nmol kg$^{-1}$ to nmol L$^{-1}$. However, the resulting concentrations differ by the former ones (calculated with a fixed density of 1.025) at the third digit only, so these differences do not appear in the Table and are not visible on the figures.**

3) Figures should be "cited" much more often in the text to support the discussion. Some figures are missing to explain correlations (see below). Some Figure do not correspond to what is said in the text (see below). Please revise the entire text on how it is supported by the Figures and the supplementary material. Some of the figures in the supplementary material are not even mentioned in the main text. Are they really necessary?

**All figures and their reference in the text have been carefully checked. All of them are now correctly cited.**

Specific comments:

Page1, Line 31: remove cause of "enhanced se-air interactions". Suggestion: Deep winter convection occurring the previous winter provided iron-to-nitrate ratios sufficient to sustain phytoplankton growth and lead to relatively elevated DFe concentrations within subsurface waters of the Irminger Sea.

**Done**

Page 2, Line 4: delete "the production of sinking biogenic particles". POC is biogenic particles, so you don't repeat. Rather say, they are exported through "sinking and ocean currents".

**Done**

Page 2, line 10: "to be able to".

**Done**

Page 2, line 10: I would move the sentence mentioning specifically Fe to the paragraph starting line 30, since this is where you start properly talking about the role of trace metals in the ocean; the rest of the nutrient discussion is good.

**Done**

Page 3, Line 10: "throughout" instead of "along".

**Done**

Page 4, line 12: You need to say that a filtration techniques were not directly compared. You can not say that you didn't observe significant differences since you have not directly compare these techniques.

**We are now stating this in P. 5, lines 139-143.**

Change sentence: "Fe concentration differences between stations 11 and 13, and 13 and 15 were most likely due to different filtration techniques".

**Done**

Page 4, Line 29: "Spectrometer"

**Done**

Page 5, line 13: you've already covered calibrations beforehand so please delete the repetition.

**Done**

Page 5, line 16: "considering all analytical sessions".

**Done**

Page 5, Line 22: I still believe it is more correct to use the actual density of each seawater sample to make this conversion. The universal 1.025 value is used for conversions when actual density is not available.

**Following this comment, we used the correct density values to make the conversion from nmol kg$^{-1}$ to nmol L$^{-1}$. However, the resulting concentrations differ by the former ones (calculated with a fixed density of 1.025) at the third digit only, so these differences do not appear in the Table and are not visible on the figures.**

Page 5, Line 22: I think it is important to include the intercalibration plot you show in your response document in the paper in this section, with an explanation of why these profiles are so different. Just because they don't coincide, doesn't mean you should discard them. Intercalibration is important for the GEOTRACES community.

**See answer and figure above. We decided to include this information in this document and not in the manuscript since this intercalibration was not performed on a true "cross-over" station, resulting in sampling slightly different water masses. Note that these data and comparison will also be submitted to the GEOTRACES S&I committee for future inclusion in the next IDP.**

Page 5, line 24: "from Sea-Ice Melt".

**The whole sentence has been modified (P. 7, lines 196-198).**

Page 5, line 26: "we briefly describe the principle".

**Done**

Page 5, line 27: remove "the" in front of water masses and check throughout the text.

**Done**

Page 5, line 27: explain in the text how you estimated the proportions of AW and PW in each sample. **Done in P. 7-8, Lines 199-208.**

Page 6, line 10: missing description of Chl-a sampling and measurements. Please include the description you have given in the reviewers response document here in this section.  Also add which casts Chl-a measurements were made on and which casts fluorometer measurements were made on. **The description of the analytical method is now provided together with information on casts (classical CTD cast for the Chl-a) in P. 8-9, lines 232-251.**

Page 6, line 11: and was fluorescence not also measured with a sensor?

**Fluorescence was indeed measured using a sensor on the CTD, but as we did not directly use these data for our discussion, we do not mention it in the text.**

Page 6, line 12: please briefly explain here why you use the data from the stainless casts and not from your trace metal rosette casts. Please include in the paper what you described in the reviewers response, that the O2 data was not calibrated on the TM CTD, etc

**We used data from the classical rosette because the O2 data could not be calibrated on the trace metal clean rosette due to a too long sampling time. This information is now added in the text together with information on salinity calibration (P. 8, lines 228-232).**

Page 7, line 11: put into brackets "the outliers are flagged with number 3"

**Done**

Page 7, line 24: "representing 60% of the…"

**Done**

Page 7, line 26: "ENACW was also…" (remove "the")

**Done**

Page 7, line 28 and line 30: "SubPolar Mode Water"

**Done**

Page 7, line 24: "representing 40% of the…"

**Done**

Page 8, line 2: "below ENACW up to…"

**Done**

Page 8, line 4: briefly say why sea-surface salinity is lower. Ice melt?

**This is likely due to ice melting and meteoric water inputs. This has been added P. 11, Line 316**

Page 8, line 18: remove "the" in front of water masses

**Done**

Page 9, line 23: Which rosette is the Chl-a data coming from? Stainless steel or TMR? A brief description has to be included in the methods section

**See above and in P. 8-9 Lines 232-251**

Page 10, line 14: delete "upper" since you already say "surface"

**Done**

Page 10, line 18: Fingerprinting water masses? I do not see a correlation of water masses and Fe concentrations in figure 3. I would call this section "DFe signatures in water masses" or "DFe concentrations in water masses"

**The title of the section has been changed to "DFe signatures in water masses"**

Page 12, line 19: instead of low-level say low-altitude to avoid confusion with wind force.

**Done**

Page 13, line 3: the fate of atmospheric Fe also depends on Fe-ligand availability.

**We agree and added "Fe-ligand binding capacity"**

Page 13, line 10: "remineralisation signal"

**Done**

Page 13, line 14: If you insist on discussing the DFe concentrations in relation to water masses it is VERY IMPORTANT to compare residence times of DFe with the time it takes for each water mass to travel from the "source" region. I.e., for Mediterranean water how long does it take for water masses to travel from the surface Mediterranean to where there is still 60% of this water mass - station 29?; for deep water masses, how long does it take for this water mass to travel from a sediment contact region to the place where there is still 60 % of this water mass present; for intermediate water masses, how long does it take for these water masses from the moment they are in contact with the atmosphere to travel to the place where there is still 60 % of this water mass, etc. Also note that DFe residence times vary in different regions and at different concentrations.

**We agree with the reviewer, but given the wide uncertainties in iron input fluxes, residence times (RTs) of dissolved iron (DFe) in the water column are poorly constrained. Generally, estimations of DFe RT range from weeks to months in the surface waters and from tens to hundreds of years in the deep waters. We are now considering the transit time of water masses in our discussion. See our responses above.**

Page 13, line 24: chlorophyllide-a

**Done**

Page 14, line 21: remove "to occur"

**Done**

Page 14, line 27: Why figure S4? I only see surface values in this figure but you talk about dFe at 2500 m depth!

**Indeed we made a mistake when referring to Supp Fig. 4 and Table S2. Corrected information is now provided.**

Page 14, line 28: sediment inputs can also happen diagonally, horizontally etc, doesn't have to be strictly vertically. Please mention that

**We agree and we changed "underneath sediments" to "local sediments".**

Page 14, line 30, "during GEOVIDE"

**Done**

Page 15, line 3: what is its residence time? Look for values in your study area in the published bibliography.

**Concerning the RTs of DFe, see our reply above.**

Page 15, line 2-4: this sentence makes no sense. Not from water masses but from lateral advection of deep waters? Are deep waters not water masses?

**We agree that this sentence was confusing and removed it.**

Page 15, line 11: "scavenging component". Page 15, line 11: remove "the" in front of 230-Th. Page 15, line 14: briefly explain what the differences are in behaviour of Fe and Th. Page 15, line 15: "would be the presence of Fe-binding organic ligands in these samples…". Page 15, line 16: "from the seafloor to the overlying deep waters… ". Page 15, line 17: diffusion of particles? Diffusion is a term applied for dissolved substances… Page 15, line 20: "due to the presence of Fe-binding…" Page 15, line 30: "transported within ISOW…" (remove "the").

**The whole section was likely confusing. We tried to simplify it and now suggest a new hypothesis on exchanges between the DSOW and the slope. See text P. 21, Lines 607-616.**

Page 16, line 3: you could put into brackets "basaltic" since at the MOR volcanic particles might be most present on the seafloor. "resuspension of (basaltic) particles…"

**Done**

Page 16, line 4: which station? 40? I can't see stn 40 in figure 4! why do you cite figure 7? what is Fe* telling you about hydrothermal vents? needs to be clarified. Figure 4 focuses on surface and subsurface waters, so how can this help interpreting deep hydrothermal sources? Please cite the correct figures.

**We were indeed talking about station 40 and did not cite the correct Figure (we should have cited Fig. 3). This was changed in the text P. 22 Lines 640-645.**

Page 16, line 4: low transmissiometer data at which station? Please at least show a transmissiometer profile.

**Stations are now specified in the text and we refer to Fig. 4A from Gourain et al. (2019) to avoid additional figures in the paper.**

Page 16, line 8, line 12: remove "the" in front of water masses

**This sentence has been removed (see above)**

Page 16, line 16: "DFe to surface waters…"

**Done**

Page 16, line 27: sentence confusing, where was the atmospheric deposition higher, during GA01 or GA03?

**We agree that our sentence was confusing and we changed it: "Atmospheric deposition were about one order of magnitude higher during GA03 than during to GA01 (Shelley et al., 2018; Shelley et al., 2015), thus the atmospheric source seemed to be minor during GA01."**

Page 16, line 31: ", that extends through an area…"

**Done**

Page 17, line 15: "extended 200 km off the Greenland stations…"

**Done**

Page 17, line 22: briefly mention in brackets above which salinity values are considered "brine"

**Here, we do not define brine release with salinity values but with the % of sea-ice melting (negative values). This information was added P. 23, Lines 688.**

Page 17, line 23: residence time of brines? You should mention that the calculated brine and sea-ice signals are mixed in the surrounding water since salinity anomalies are not associated to those signals. If you talk about residence times, do you talk of the pure brine or the seawater mixed brines? This is confusing and should be clarified in the main text

**No, we meant the residence time of water masses on the Greenland shelf. This section was likely confusing and too speculative, so we shortened it and tried to make it clearer (P. 24, Lines 698-716).**

Page 17, line 33: I can't see a salinity signal at 100 m depth.

**The brine signal was seen with the strongest negative %sea-ice melting (~ -2%). To avoid confusion, we added "strongest" (P. 24, Line 706).**

Page 18, line 2: what does a correlation plot show dFe vs MW or Sea-ice fraction? A little hard to see correlations on vertical profiles

**We do not mention any correlation here. In the depth range 50-160 m, the brine signal increases (%sea-ice melting decreases) while DFe and PFe concentrations increase. In the text, we now mention depth rage instead of just one depth to make it clearer (P. 24, Lines 702 and 707).**

Page 18, line 5: "for which no DFe data was available…"

**Done**

Page 18, line 11: "the surface DFe depletion is likely explained…"

**Done**

Page 19, line 24: "DFe and PFe concentrations…"

**Done**

Page 19, line 25: "in the samples closest to the seafloor…"

**Done**

Page 19, line 27: station 2 and 4 are the same margin, so delete "different behaviour of Fe among different margins"

**Done**

Page 20, line 1: "the more lithogenic sediments…"

**This has been changed as well as "the more biogenic sediment (Newfoundland margin)"**

Page 20, line 11: remove "the" in front of particulate Fe and in front of Dal and in front of AOU

**Done**

Page 20, line 20: "enrichment of Fe in the particulate phase…" (since the Fe will be adsorbed, and not "within" the particle)

**Done**

Page 20, line 22: "and Mn from the dissolved phase…"

**Done**

Page 21, line 11: "classes are dealt with in Tonnard…"

**Done**

Page 21, line 22: "from the Iceland Basin and the Irminger Sea. "

**Done**

Page 21, line 33: "stations from the subpolar gyre…"

**Done**

Page 22, line 22: it is hard to follow these correlations, slope and intercept values without the corresponding figures. Please add the figures

**The figures were added as supplementary figures and text has been changed accordingly.**

Page 22, line 3: "compared"

**Done**

Page 22, line 6: "follows"

**Done**

Page 22, line 13: add the reference where you got this ratio from

**This ratio is from our data (see text P. 29, line 864 and Sup. Fig S8)**

Page 22, line 21: how do you know the Fe:N requirement of the phytoplankton in that area? please add a reference

**The ratio mentioned here is the one we measured and the lower limit of the literature values (Ho et al., 2003; Sunda and Huntsman, 1995; Twining et al., 2004, P. 29, lines 874-875). As this was likely confusing, we changed the sentence: "Figure 7 (see also supplementary material S7, S8, S9 and S10) exhibits Fe:N ratios lower than 0.05 mmol mol$^{-1}$, suggesting that Fe could also limit the low–Fe requirement phytoplankton class (RFe:N = 0.05 mmol mol$^{-1}$) within the Iceland Basin, the Irminger, and the Labrador Seas." P. 30, Lines 886-889.**

Page 22, line 22: "deposition to IcSPMW" (remove "the")

**Done, as well as "the" before LSW.**

Page 22, line 24-27: I can see lowest Fe* in the Iceland Basin, not the West European Basin

**The low Fe* in the West European Basin was found at 200 m and this can be seen on Fig. S9 and S10. Reference to these Figures is now added to the text P. 30, Line 894-895.**

Page 23, line 10: remove "the" in front of ISOW.

**Done**

Page 23, line 12    : "in the Irminger Sea"

**Done**

Page 23, line 31: "tip jet events"

**Done**

Page 36, line 36: MOW not labelled in the plots

**Done**

Page 38, line 4: do you mean sea-ice or sea-ice-melt?

**We mean "sea-ice melting" and add it to the fig. caption.**

Page 39: you forgot to add the station numbers to the graph

**Done**

Supplementary material: add page numbers and add the paper info at the top of this document (title, authors, affiliations, etc).

**Done**

Page3: why did you not consider station 1 and 17?

**The explanation is provided in the text in section 4.1 P. 16, Lines 455-470. This was added to the Figure caption.**

Please briefly explain. what do the black dots represent?

**This diagram is a classical whisker diagram, with black dots representing outliers.**

Describe the red line in the legend. What is the value of DFe meadian? Correct error on y axis title

**Done**

Page 8: specify the sampling depth for these surface values

**Depths varied from 5 to 30 m. This was added to the fig. caption.**

Table S1: add a column informing the filtration size for each sample (since you have not directly compared these methods on natural samples)

**Done**

[revised manuscript text omitted]